# Learning and Generalization in RNNs

**Abhishek Panigrahi** *
Department of Computer Science
Princeton University
ap34@princeton.edu

**Navin Goyal**
Microsoft Research India
navingo@microsoft.com

## Abstract

Simple recurrent neural networks (RNNs) and their more advanced cousins LSTMs etc. have been very successful in sequence modeling. Their theoretical understanding, however, is lacking and has not kept pace with the progress for feedforward networks, where a reasonably complete understanding in the special case of highly overparameterized one-hidden-layer networks has emerged. In this paper, we make progress towards remedying this situation by proving that RNNs can learn functions of sequences. In contrast to the previous work that could only deal with functions of sequences that are sums of functions of individual tokens in the sequence, we allow general functions. Conceptually and technically, we introduce new ideas which enable us to extract information from the hidden state of the RNN in our proofs—addressing a crucial weakness in previous work. We illustrate our results on some regular language recognition problems.

## 1 Introduction

Simple Recurrent Neural Networks [1] also known as Elman RNNs or vanilla RNNs (just RNNs henceforth) along with their more advanced versions such as LSTMs [2] and GRU [3] are among the most successful models for processing sequential data, finding wide-ranging applications including natural language processing, audio processing [4] and time series classification [5]. Feedforward networks (FFNs) model functions on inputs of fixed length, such as vectors in $\mathbb{R}^d$. In contrast, RNNs model functions whose input consists of sequences of tokens $\mathbf{x}^{(1)}, \mathbf{x}^{(2)}, \ldots$, where $\mathbf{x}^{(i)} \in \mathbb{R}^d$ for each $i$. RNNs have a notion of memory; formally it is given by the hidden state vector which is denoted by $\mathbf{h}^{(t)}$ after processing the $t$-th token. RNNs apply a fixed function to $\mathbf{h}^{(t)}$ and $\mathbf{x}^{(t+1)}$ to compute $\mathbf{h}^{(t+1)}$ and the output. This fixed function is modeled by a neural networks with one hidden-layer. Compared to FFNs, new challenges arise in the analysis of RNNs: for example, the use of memory and the same function at each step introduces dependencies across time and RNN training suffers from vanishing and exploding gradients [6].

Studies aimed at understanding the effectiveness of RNNs have been conducted since their introduction; for some of the early work, see, e.g., [7, 8]. These works take the form of experimental probing of the inner workings of these models as well as theoretical studies. The theoretical studies are often focused on expressibility, training and generalization questions in isolation rather than all together—the latter needs to be addressed to approach full understanding of RNNs and appears to be far more challenging. While experimental probing has continued apace, e.g., [9, 10], progress on theoretical front has been slow. It is only recently that training and generalization are starting to be addressed in the wake of progress on the relatively easier case of FFNs as discussed next.

RNNs are closely related to deterministic finite automata [11, 9] as well as to dynamical systems. With finite precision and ReLU activation, they are equivalent to finite automata [11] in computational power. In the last few years progress was made on theoretical analysis of overparameterized FFNs

---

*Work done as a research fellow in Microsoft Research India

35th Conference on Neural Information Processing Systems (NeurIPS 2021).

with one-hidden-layer, e.g., [12, 13, 14, 15, 16, 17, 18]. Building upon these techniques, [19] proved that RNNs trained with SGD (stochastic gradient descent) achieve small training loss if the number of neurons is sufficiently large polynomial in the number of training datapoints and the maximum sequence length.

But the gap between our understanding of RNNs and FFNs remains large. [20, 21] provide generalization bounds on RNNs in terms of certain norms of the parameters. While interesting, these bounds shed light on only a part of the picture as they do not consider the training of the networks nor do not preclude the possibility that the norms of the parameters for the trained networks are large leading to poor generalization guarantees. RNNs can be viewed as dynamical systems and many works have used this viewpoint to study RNNs, e.g., [22, 23, 24, 25]. Other related work includes relation to kernel methods, e.g., [26, 27, 28], linear RNNs [29], saturated RNNs [30, 31, 32], and echo state networks [33, 34]. Several other works talk about the expressive power of the novel sequence to sequence models Transformers [35, 36]. Due to a large number of works in this area it is not possible to be exhaustive: apart from the references directly relevant to our work we have only been able to include a small subset.

[37] gave the first "end-to-end" result for RNNs. Very informally, their result is: if the concept class consists of functions that are sums of functions of tokens then overparameterized RNNs trained using SGD with sufficiently small learning rate can learn such a concept class. They introduce new technical ideas, most notably what they call re-randomization which allows one to tackle the dependencies that arise because the same weights are used in RNN across time. However, an important shortcoming of their result is limited expressive power of their concept class: while this class can be surprisingly useful as noted there, it cannot capture problems where the RNN needs to make use of the information in the past tokens when processing a token (in their terminolgy, their concept class can *adapt* to time but not to tokens). Indeed, a key step in their proof shows that RNNs can learn to ignore the hidden state $\mathbf{h}^{(t)}$. (The above concept class comes up because it can be learnt even if $\mathbf{h}^{(t)}$ is ignored.) But the hidden state $\mathbf{h}^{(t)}$ is the hallmark of RNNs and is the source of information about the past tokens—in general, not something to be ignored. Thus, it is an important question to theoretically analyze RNNs' performance on general concept classes and it was also raised in [37]. This question is addressed in the present paper. As in previous work, we work with sequences of bounded length $L$. Without loss of generality, we work with token sequences $\mathbf{x}^{(1)}, \ldots, \mathbf{x}^{(L)}$ of fixed length as opposed to sequences of length up to $L$. Informally, our result is:

> Overparameterized RNNs can efficiently learn concept classes consisting of one-hidden-layer neural networks that take the entire sequence of tokens as input. The training algorithm used is SGD with sufficiently small step size.

By the universality theorem for one-hidden-layer networks, such RNNs can approximate all continuous functions of $\mathbf{x}^{(1)}, \ldots, \mathbf{x}^{(L)}$—though naturally the more complex the functions in the class the larger the network size required. We note that the above result applies to all three aspects mentioned above: expressive power, training and generalization. We illustrate the power of our result by showing that some regular languages such as PARITY can be recognized efficiently by RNNs.

In a concurrent work [38], the authors also study related problems. The results there are not directly comparable: On the one hand, they do not normalize the input unlike our work. On the other hand, the concept classes treated in their work appear to be weak. The authors show that RNNs can learn concept classes that include functions of the form $f(\mathbf{x}^{(\ell_1)}, \mathbf{x}^{(\ell_2)}, \ldots, \mathbf{x}^{(\ell_N)})$, where $\ell_1, \ldots, \ell_N \in [L]$. However, the number of SGD iterations and training samples necessary depend exponentially on the minimum of the two parameters, $N$ and $\ell_0 = \max(\ell_1, \ldots, \ell_N) - \min(\ell_1, \ldots, \ell_N)$.

## 2 Preliminaries

Let $\mathbb{S}^{d-1} := \{\mathbf{x} \in \mathbb{R}^d \mid \|\mathbf{x}\|_2 = 1\}$ be the unit sphere in $\mathbb{R}^d$. For positive integer $n$ define $[n] := \{1, 2, \ldots, n\}$. Given a vector $\mathbf{v}$, by $v_i$ we denote its $i$-th component. Given two vectors $\mathbf{a} \in \mathbb{R}^{d_1}$ and $\mathbf{b} \in \mathbb{R}^{d_2}$, $[\mathbf{a}, \mathbf{b}] \in \mathbb{R}^{d_1+d_2}$ denotes the concatenation of the two vectors. $\langle \cdot, \cdot \rangle$ denotes the standard dot product. Given a matrix $\mathbf{M}$, we will denote its $i$-th row as $\mathbf{m}_i$ and the element in row $i$ and column $j$ as $m_{ij}$. Given two matrices $\mathbf{A} \in \mathbb{R}^{a_1 \times a_2}$ and $\mathbf{B} \in \mathbb{R}^{b_1 \times b_2}$ with $a_1 = b_1$ let

$[\mathbf{A}, \mathbf{B}]_r \in \mathbb{R}^{a_1 \times (a_2+b_2)}$ denote the matrix whose rows are obtained by concatenating the respective rows of $\mathbf{A}$ and $\mathbf{B}$. Similarly, $[\mathbf{A}, \mathbf{B}]_c \in \mathbb{R}^{(a_1+b_1) \times a_2}$ (assuming $a_2 = b_2$) denotes the matrix whose columns are obtained by concatenating the columns of $\mathbf{A}$ and $\mathbf{B}$.

$O(\cdot)$ and $\Omega(\cdot)$ hide absolute constants. Similarly, $\mathrm{poly}(\cdot)$ denotes a polynomial in its arguments with degree and coefficients bounded by absolute constants; different instances of $\mathrm{poly}(\cdot)$ may refer to different polynomials. Writing out explicit constants would lead to unwieldy formulas without any new insights.

Let $\sigma : \mathbb{R} \to \mathbb{R}$, given by $\sigma(x) := \max\{x, 0\} = x\, \mathbb{I}_{x \geq 0}$, be ReLU activation function. ReLU can be extended to act on vectors by coordinate-wise application: $\sigma((x_1, \ldots, x_d)) := (\sigma(x_1), \ldots, \sigma(x_d))$. Note that ReLU is a positive homogenous function of degree 1, that is to say $\sigma(\lambda x) = \lambda\, \sigma(x)$ for all $x$ and all $\lambda \geq 0$.

To be learnable efficiently, the functions in the concept class need to be not too complex. We will quantify this with the following two complexity measures which are weighted norms of the Taylor expansion and intuitively can be thought of as quantifying network size and sample complexities, resp., needed to learn $\phi$ up to error $\epsilon$.

**Definition 2.1** (Function complexity [15]). Suppose that function $\phi : \mathbb{R} \to \mathbb{R}$ has Taylor expansion $\phi(z) = \sum_{i=0}^{\infty} c_i z^i$. For $R, \epsilon > 0$, define

$$\mathfrak{C}_\varepsilon(\phi, R) := \sum_{i=0}^{\infty} \left( (C^* R)^i + \left( \frac{\sqrt{\log(1/\varepsilon)}}{\sqrt{i}} C^* R \right)^i \right) |c_i|,$$

$$\mathfrak{C}_\mathfrak{s}(\phi, R) := C^* \sum_{i=0}^{\infty} (i+1)^{1.75} R^i |c_i|,$$

where $C^* = 10^4$. As an example, if $\phi(z) = z^d$ for positive integer $d$, then $\mathfrak{C}_\mathfrak{s}(\phi, R) = O(R^d)$ and $\mathfrak{C}_\varepsilon(\phi, R) = O(R^d \log^{d/2}(\frac{1}{\varepsilon}))$. For $\phi(z) = \sin z, \cos z, e^z$, we have $\mathfrak{C}_\mathfrak{s}(\phi, R) = O(1)$ and $\mathfrak{C}_\varepsilon(\phi, R) = \mathrm{poly}(1/\varepsilon)$. We have $\mathfrak{C}_s(\phi, R) \leq \mathfrak{C}_\varepsilon(\phi, R) \leq \mathfrak{C}_s(\phi, O(R)) \times \mathrm{poly}(1/\varepsilon)$ for all $\phi$ and for $\phi(z) = \sin z, e^z$ or constant degree polynomials, they only differ by $o(1/\varepsilon)$. See [15] for details. Note that $\phi$ itself is not a member of our concept class but functions like it will be used to construct members of our concept class.

## 3 Problem Formulation

In our set-up, RNNs output a label after processing the whole input sequence.[2] The data are generated from an unknown distribution $\mathcal{D}$ over $((\overline{\mathbf{x}}^{(2)}, \ldots, \overline{\mathbf{x}}^{(L-1)}), \mathbf{y}^*) \in ((\mathbb{S}^{d-2})^{L-2}, \mathcal{Y})$, for some label set $\mathcal{Y} \subset \mathbb{R}^{d_{\mathrm{out}}}$ for some positive integer $d_{\mathrm{out}}$. We call $\overline{\mathbf{x}} := (\overline{\mathbf{x}}^{(2)}, \ldots, \overline{\mathbf{x}}^{(L-1)})$ the *true sequence* and $\mathbf{y}^*$ the *true label*. Denote by $\mathcal{Z}$ the training dataset containing $N$ i.i.d. samples from $\mathcal{D}$. We preprocess the true sequence to *normalize* it:

**Definition 3.1** (Normalized Input sequence). Let $\overline{\mathbf{x}} = (\overline{\mathbf{x}}^{(2)}, \ldots, \overline{\mathbf{x}}^{(L-1)})$ be a given true input sequence of length $L - 2$, s.t. $\overline{\mathbf{x}}^{(i)} \in \mathbb{S}^{d-2}$ and $\overline{x}_{d-1}^{(i)} = \frac{1}{2}$, for all $i \in [2, L-1]$. The normalized input sequence $\mathbf{x} := (\mathbf{x}^{(1)}, \ldots, \mathbf{x}^{(L)})$ is given by

$$\mathbf{x}^{(1)} := (\mathbf{0}^{d-1}, 1), \quad \mathbf{x}^{(\ell)} := (\varepsilon_x \overline{\mathbf{x}}^{(\ell)}, 0), \quad \forall \ell \in [2, L-1], \quad \mathbf{x}^{(L)} := (\mathbf{0}^{d-1}, 1),$$

where we set $\varepsilon_x > 0$ later in Theorem 3.1.

We use normalized sequence in place of the true sequence as input to RNNs, as it helps in proofs, e.g., with bounds on the changes in the activation patterns at each RNN cell, when the input sequences change and also with inversion of RNNs (defined later). Our method can be applied without normalization too, but in that case our error bound has exponential dependence on the length of the input sequence. The extra dimension in the normalized sequence serves as bias which we do not use explicitly to simplify notation.

### 3.1 RNNs

**Definition 3.2** (Recurrent Neural Networks). We assume that the input sequences are of length $L$ for some given $L > 0$ and are of the form $\mathbf{x}^{(1)}, \ldots, \mathbf{x}^{(L)}$ with $\mathbf{x}^{(\ell)} \in \mathbb{R}^d$ for all $\ell \in [L]$. An RNN is

---

[2]While our set-up has similarity to previous work [37], there are also important differences.

specified by three matrices $\mathbf{W}_{\mathrm{rnn}} \in \mathbb{R}^{m \times m}$, $\mathbf{A}_{\mathrm{rnn}} \in \mathbb{R}^{m \times d}$ and $\mathbf{B}_{\mathrm{rnn}} \in \mathbb{R}^{d_{\mathrm{out}} \times m}$, where $m$ is the dimension of the hidden state and $d_{\mathrm{out}}$ is the dimension of the output. The hidden states of the RNN are given by $\mathbf{h}_{\mathrm{rnn}}^{(0)} = \mathbf{0} \in \mathbb{R}^m$ and

$$\mathbf{h}_{\mathrm{rnn}}^{(\ell)} := \sigma(\mathbf{A}_{\mathrm{rnn}}\mathbf{x}^{(\ell)} + \mathbf{W}_{\mathrm{rnn}}\mathbf{h}_{\mathrm{rnn}}^{(\ell-1)}) \quad \text{for } \ell \in [L]. \tag{1}$$

The output at each step $\ell \in [L]$ is given by $\mathbf{y}_{\mathrm{rnn}}^{(\ell)} = \mathbf{B}_{\mathrm{rnn}}\mathbf{h}_{\mathrm{rnn}}^{(\ell)}$. By *RNN cell* we mean the underlying FFN in (1). The $m$ rows of $\mathbf{W}_{\mathrm{rnn}}$ and $\mathbf{A}_{\mathrm{rnn}}$ correspond to the $m$ neurons in the RNN.

Pick the matrices $\mathbf{W} \in \mathbb{R}^{m \times m}$ and $\mathbf{A} \in \mathbb{R}^{m \times d}$ by sampling entries i.i.d. from $N(0, \frac{2}{m})$, and pick $\mathbf{B}$ by sampling entries i.i.d. from $N(0, \frac{2}{d_{\mathrm{out}}})$. When $\mathbf{W}_{\mathrm{rnn}} = \mathbf{W}$ and $\mathbf{A}_{\mathrm{rnn}} = \mathbf{A}$, the RNN is said to be at random initialization. We will denote the parameters of an RNN at initialization by dropping the subscript "rnn", thus the hidden states are $\{\mathbf{h}^{(\ell)}\}_{\ell \in [L]}$. In the following theorems, we will keep $\mathbf{B}_{\mathrm{rnn}}$ at initialization $\mathbf{B}$ and train only $\mathbf{A}_{\mathrm{rnn}}$ and $\mathbf{W}_{\mathrm{rnn}}$.

We write $F_{\mathrm{rnn}}^{(\ell)}(\mathbf{x}; \mathbf{W}_{\mathrm{rnn}}, \mathbf{A}_{\mathrm{rnn}}) = \mathbf{y}_{\mathrm{rnn}}^{(\ell)} = \mathbf{B}\mathbf{h}_{\mathrm{rnn}}^{(\ell)}$ for the output of the $\ell$-th step. Our goal is to use $\mathbf{y}_{\mathrm{rnn}}^{(L)} \in \mathbb{R}^{d_{\mathrm{out}}}$ to fit the true label $\mathbf{y}^* \in \mathcal{Y}$ using some loss function $G : \mathbb{R}^{d_{\mathrm{out}}} \times \mathcal{Y} \to \mathbb{R}$. We assume that for every $\mathbf{y}^* \in \mathcal{Y}, G\left(0^k, \mathbf{y}^*\right) \in [-1, 1]$ is bounded, and $G(\cdot, \mathbf{y}^*)$ is convex and 1-Lipschitz continuous in its first variable. This includes, for instance, the cross-entropy loss and $\ell_2$-regression loss (for bounded arguments).

## 3.2 Concept Class

We now define our target concept class, which we will show to be learnable by RNNs using SGD.

**Definition 3.3** (Concept Class). Our concept class consists of functions $F : \mathbb{R}^{(L-2)\cdot(d-1)} \to \mathbb{R}^{d_{\mathrm{out}}}$ defined as follows. Let $\Phi$ denote a set of smooth functions with Taylor expansions with finite complexity as in Def. 2.1. To define a function $F$, we choose a subset $\{\Phi_{r,s} : \mathbb{R} \to \mathbb{R}\}_{r \in [p], s \in [d_{\mathrm{out}}]}$ from $\Phi$, $\{\mathbf{w}_{r,s}^{\dagger} \in \mathbb{S}^{(L-2)(d-1)-1}\}_{r \in [p], s \in [d_{\mathrm{out}}]}$, a set of weight vectors, and $\{b_{r,s}^{\dagger} \in \mathbb{R}\}_{r \in [p], s \in [d_{\mathrm{out}}]}$, a set of output coefficients with $|b_{r,s}^{\dagger}| \leq 1$. Then, we define $F : \mathbb{R}^{(L-2)\cdot(d-1)} \to \mathbb{R}^{d_{\mathrm{out}}}$, where for each output dimension $s \in [d_{\mathrm{out}}]$ we define the $s$-th coordinate $F_s$ of $F = (F_1, \ldots, F_{d_{\mathrm{out}}})$ by

$$F_s(\overline{\mathbf{x}}) := \sum_{r \in [p]} b_{r,s}^{\dagger}\Phi_{r,s}\left(\langle\mathbf{w}_{r,s}^{\dagger}, [\overline{\mathbf{x}}^{(2)}, \ldots, \overline{\mathbf{x}}^{(L-1)}]\rangle\right). \tag{2}$$

To simplify formulas, we assume $\Phi_{r,s}(0) = 0$ for all $r$ and $s$. We denote the complexity of the concept class by

$$\mathfrak{C}_{\varepsilon}(\Phi, R) := \max_{\phi \in \Phi}\{\mathfrak{C}_{\varepsilon}(\phi, R)\}, \quad \mathfrak{C}_{\mathfrak{s}}(\Phi, R) := \max_{\phi \in \Phi}\{\mathfrak{C}_{\mathfrak{s}}(\phi, R)\}.$$

Let $F^*$ be a function in the concept class with smallest possible population loss which we denote by OPT. Hence, we are in an agnostic learning setting where our aim is to learn a function with population objective $\mathrm{OPT} + \varepsilon$. As one can observe, functions in the concept class are given by a one hidden layer network with $p$ neurons and smooth activations. We will show that the complexity of the functions $\Phi_{r,s}$ determines the number of neurons and the number of training samples necessary to train the recurrent neural network that has $\mathrm{OPT} + \varepsilon$ population loss.

While we have defined $F^*$ as a function of $\overline{\mathbf{x}}$, since there's a one-to-one correspondence between $\overline{\mathbf{x}}$ and $\mathbf{x}$, it will occasionally be convenient to talk about $F^*$ as being a function of $\mathbf{x}$—and this should cause no confusion. And similarly for other functions like $F_{\mathrm{rnn}}^{(\ell)}(\mathbf{x}; \mathbf{W}, \mathbf{A})$.

## 3.3 Objective Function and the Learning Algorithm

We assume that there exists a function $F^*$ in the concept class that can achieve a population loss OPT, i.e. $\underset{(\overline{\mathbf{x}}, \mathbf{y}^*) \sim \mathcal{D}}{\mathbb{E}} G(F^*(\overline{\mathbf{x}}), \mathbf{y}^*) \leq \mathrm{OPT}$. The following loss function is used for gradient descent:

$$\mathrm{Obj}(\mathbf{W}', \mathbf{A}') = \underset{(\overline{\mathbf{x}}, \mathbf{y}^*) \sim \mathcal{Z}}{\mathbb{E}} \mathrm{Obj}(\overline{\mathbf{x}}, \mathbf{y}^*; \mathbf{W}', \mathbf{A}'), \quad \text{where}$$

$$\mathrm{Obj}(\overline{\mathbf{x}}, \mathbf{y}^*; \mathbf{W}', \mathbf{A}') = G(\lambda F_{\mathrm{rnn}}^{(L)}(\mathbf{x}; \mathbf{W} + \mathbf{W}', \mathbf{A} + \mathbf{A}'), \mathbf{y}^*).$$

Parameter $\lambda$ whose value is set in the main Theorem 3.1 is a scaling factor needed for technical reasons discussed later. We consider vanilla stochastic gradient updates with $\mathbf{W}_t, \mathbf{A}_t$ denoting the matrices after $t$-steps of sgd. $\mathbf{W}_t$ and $\mathbf{A}_t$ are given by

$$\mathbf{W}_t = \mathbf{W}_{t-1} - \eta \, \nabla_{\mathbf{W}_{t-1}} \mathrm{Obj}(\overline{\mathbf{x}}, \mathbf{y}^*; \mathbf{W}_{t-1}, \mathbf{A}_{t-1}),$$
$$\mathbf{A}_t = \mathbf{A}_{t-1} - \eta \, \nabla_{\mathbf{A}_{t-1}} \mathrm{Obj}(\overline{\mathbf{x}}, \mathbf{y}^*; \mathbf{W}_{t-1}, \mathbf{A}_{t-1}),$$

where $(\overline{\mathbf{x}}, \mathbf{y}^*)$ is a random sample from $\mathcal{Z}$ and $\mathbf{x}$ is its normalized form. It should be noted that [37] train only $\mathbf{W}$.

*Remark.* We made two assumptions in our set-up: (1) input sequences are of fixed length, and (2) the output is only considered at the last step. These assumptions are without loss of generality and allow us to keep already quite complex formulas manageable without affecting the essential ideas. The main change needed to drop these assumptions is a change in the objective function, which will now include terms not just for how well the output fits the target at step $L$ but also for the earlier steps. The objective function for each step behaves in the same way as that for step $L$, and so the sum can be analyzed similarly. Intuitively speaking, considering the output at the end is the "hardest" training regime for RNNs as it uses the "minimal" amount of label information.

### 3.4  RNNs learn the concept class

We are now ready to state our main theorem. We use $\rho := 100 L d_{\text{out}} \log m$ in the following. Recall that a set $\Phi$ of smooth functions induces a concept class as in Def. 3.3.

**Theorem 3.1** (Main, restated in the appendix as Theorem D.5)**.** *Let $\Phi$ be a set of smooth functions. For $\epsilon_x := \frac{1}{\mathrm{poly}(\rho)}$ and $\varepsilon \in \left(0, \frac{1}{p \cdot \mathrm{poly}(\rho) \cdot \mathfrak{C}_s(\Phi, \mathcal{O}(\epsilon_x^{-1}))}\right)$, define complexity $C := \mathfrak{C}_\varepsilon(\Phi, \mathcal{O}(\epsilon_x^{-1}))$ and $\lambda := \frac{\varepsilon}{10 L \rho}$. Assume that the number of neurons $m \geq \mathrm{poly}\left(C, p, L, d_{\text{out}}, \varepsilon^{-1}\right)$ and the number of samples $N \geq \mathrm{poly}\left(C, p, L, d_{\text{out}}, \varepsilon^{-1}\right)$. Then with parameter choices $\eta := \Theta\left(\frac{1}{\varepsilon \rho^2 m}\right)$ and $T := \Theta(p^2 C^2 \, \mathrm{poly}(\rho) \varepsilon^{-2})$ with probability at least $1 - e^{-\Omega(\rho^2)}$ over the random initialization, SGD satisfies*

$$\underset{\mathrm{sgd}}{\mathbb{E}} \left[ \frac{1}{T} \sum_{t=0}^{T-1} \underset{(\overline{\mathbf{x}}, \mathbf{y}^*) \sim \mathcal{D}}{\mathbb{E}} \mathrm{Obj}\left(\overline{\mathbf{x}}, \mathbf{y}^*; \mathbf{W}_t, \mathbf{A}_t\right) \right] \leq \mathrm{OPT} + \varepsilon + 1/\mathrm{poly}(\rho). \tag{3}$$

Informally, the above theorem states that by SGD training of overparameterized RNNs with sufficiently small learning rate and appropriate preprocessing of the input sequence, we can efficiently find an RNN that has population objective nearly as small as OPT as $\varepsilon + 1/\mathrm{poly}(\rho)$ is small. The required number of neurons and the number of training samples have polynomial dependence on the function complexity of the concept class, the length of the input sequence, the output dimension, and the additional prediction error $\varepsilon$.

## 4  Proof Sketch

While the full proof is highly technical, in this section we will sketch the proof focusing on the conceptual aspects while minimizing the technical aspects to the essentials; full proofs are in the appendix. The high-level outline of our proof is as follows.

1. *Overparameterization simplifies the neural network behavior.* The function $F_{\text{rnn}}^{(L)}(\mathbf{x}; \mathbf{W} + \mathbf{W}', \mathbf{A} + \mathbf{A}')$ computed by the RNN is a function of the parameters $\mathbf{W}', \mathbf{A}'$ as well as of the input $\overline{\mathbf{x}}$. It is a highly non-linear and non-convex function in both the parameters and in the input. The objective function inherits these properties and its direct analysis is difficult. However, it has been realized in the last few years—predominantly for the FFN setting—that when the network is overparameterized (i.e., as the number of neurons $m$ becomes large compared to other paramters of the problem such as the complexity of the concept class), the network behavior simplifies in a certain sense. The general idea carries over to RNNs as well: in (4) below we write the first-order Taylor approximation of $F_{\text{rnn}}^{(L)}(\mathbf{x}; \mathbf{W} + \mathbf{W}', \mathbf{A} + \mathbf{A}')$ at $\mathbf{W}$ and $\mathbf{A}$ as a linear function of $\mathbf{W}'$ and $\mathbf{A}'$; it is still a non-linear function of the input

sequence. As in [37] we call this function *pseudo-network*, though our notion is more general as we vary both the parameters $\mathbf{W}'$ and $\mathbf{A}'$. Pseudo-network is a good approximation of the target network as a function of $\overline{\mathbf{x}}$ for all $\overline{\mathbf{x}}$.

2. *Existence of a good RNN.* In order to show that the RNN training successfully learns, we first show that there are parameters values for RNN so that as a function of $\overline{\mathbf{x}}$ it is a good approximation of $F^*$. Instead of doing this directly, we show that the pseudo-network can approximate $F^*$; this suffices as we know that the RNN and the pseudo-network remain close. This is done by constructing paramters $\mathbf{W}^*$ and $\mathbf{A}^*$ so that the resulting pseudo-network approximates the target function in the concept class (Section 4.2) for all $\overline{\mathbf{x}}$.

3. *Optimization.* SGD makes progress because the loss function is convex in terms of the pseudo-network which stays close to the RNN as a function of $\mathbf{x}$. Thus, SGD finds parameters with training loss close to that achieved by $\mathbf{W}^*, \mathbf{A}^*$.

4. *Generalization.* Apply a Rademacher complexity-based argument to show that SGD has low population loss.

Step 2 is the main novel contribution of our paper and we will give more details of this step in the rest of this section.[3]

## 4.1 Pseudo-network

We define the pseudo-network here. Suppose $\mathbf{W}, \mathbf{A}, \mathbf{B}$ are at random initialization. The linear term in the first-order Taylor approximation is given by the pseudo-network

$$F^{(L)}(\mathbf{x}; \mathbf{W}', \mathbf{A}') := \sum_{i=1}^{L} \mathbf{Back}_{i \to L} \mathbf{D}^{(i)} \left( \mathbf{W}' \mathbf{h}^{(i-1)} + \mathbf{A}' \mathbf{x}^{(i)} \right) \tag{4}$$

$$\approx F_{\mathrm{rnn}}^{(L)}(\mathbf{x}; \mathbf{W} + \mathbf{W}', \mathbf{A} + \mathbf{A}') - F_{\mathrm{rnn}}^{(L)}(\mathbf{x}; \mathbf{W}, \mathbf{A}). \qquad \text{(using Lemma G.3)}$$

This function approximates the change in the output of the RNN, when $(\mathbf{W}, \mathbf{A})$ changes to $(\mathbf{W} + \mathbf{W}', \mathbf{A} + \mathbf{A}')$. The parameter $\lambda$, that we defined in the objective function, will be used to make the contribution of $F_{\mathrm{rnn}}^{(L)}$ at initialization small thus making pseudo-network a good approximation of RNN. Hence, we can observe that the pseudo network is a good approximation of the RNN, provided the weights stay close to the initialization.

To complete the above definition of pseudo-network we define the two new notations in the above formula. For each $\ell \in [L]$, define $\mathbf{D}^{(\ell)} \in \mathbb{R}^{m \times m}$ as a diagonal matrix, with diagonal entries

$$d_{rr}^{(\ell)} := \mathbb{I}[\mathbf{w}_r^\top \mathbf{h}^{(\ell-1)} + \mathbf{a}_r^\top \mathbf{x}^{(\ell)} \geq 0], \quad \forall r \in [m]. \tag{5}$$

In words, the diagonal of matrix $\mathbf{D}^{(\ell)}$ represents the activation pattern for the RNN cell at step $\ell$ at initialization.

Define $\mathbf{Back}_{i \to j} \in \mathbb{R}^{d_{\mathrm{out}} \times m}$ for each $1 \leq i \leq j \leq L$ by

$$\mathbf{Back}_{i \to j} := \mathbf{B} \mathbf{D}^{(j)} \mathbf{W} \ldots \mathbf{D}^{(i+1)} \mathbf{W},$$

with $\mathbf{Back}_{i \to i} := \mathbf{B}$ for each $i \in [L]$. Matrices $\mathbf{Back}_{i \to j}$ in Eq. (4) arise naturally in the computation of the first-order Taylor approximation (equivalently, gradients w.r.t. the parameters) using standard matrix calculus. Very roughly, one can think of $\mathbf{Back}_{i \to j}$ as related to the backpropagation signal from the output at step $j$ to the parameters at step $i$.

## 4.2 Existence of good pseudo-network

Our goal is to construct $\mathbf{W}^*$ and $\mathbf{A}^*$ such that for any true input sequence $\overline{\mathbf{x}} = (\overline{\mathbf{x}}^{(2)}, \ldots, \overline{\mathbf{x}}^{(L-1)})$, if we define the normalized sequence $\mathbf{x} = (\mathbf{x}^{(1)}, \ldots, \mathbf{x}^{(L)})$, then with high probability we have

$$F^{(L)}(\mathbf{x}; \mathbf{W}^*, \mathbf{A}^*) \approx F^*(\overline{\mathbf{x}}). \tag{6}$$

---

[3]The above outline is similar to prior work, e.g., [37]. Details can be quite different though, e.g., they only train $\mathbf{W}$ and keep $\mathbf{A}$ fixed to its initial value. Their contribution was also mainly in Step 2 and the other steps were similar to prior work.

To simplify the presentation, in this sketch we will assume that $p$, the number of neurons in the concept class, and the output dimension $d_{\text{out}}$ are both equal to 1. Also, let the output weight $b^\dagger := 1$. These assumptions retain the main proof ideas while simplifying equations. Overall, we assume that the target function $F^* : \mathbb{R}^{(L-2)\cdot(d-1)} \to \mathbb{R}$ on a given sequence is given by

$$F^*(\overline{\mathbf{x}}) = \Phi^*(\langle \mathbf{w}^\dagger, [\overline{\mathbf{x}}^{(2)}, \cdots, \overline{\mathbf{x}}^{(L-1)}]\rangle), \tag{7}$$

where $\Phi^* : \mathbb{R} \to \mathbb{R}$ is a smooth function and $\mathbf{w}^\dagger \in \mathbb{S}^{(L-2)\cdot(d-1)-1}$.

First, we state Lemma 6.2 in [15], which is useful for our construction of the matrices $\mathbf{W}^*$ and $\mathbf{A}^*$. Consider a smooth function $\phi : [-1, 1] \to \mathbb{R}$. It can be approximated as a linear combination of step functions (derivatives of ReLU) for all $u \in (-1, 1)$, i.e., there exists a "weight function" $H : \mathbb{R}^2 \to \mathbb{R}$ such that $\phi(u) \approx \mathbb{E}_{\alpha_1,\beta_1,b_0}[H(\alpha_1, b_0) \mathbb{I}_{\alpha_1 u + \beta_1 \sqrt{1-u^2} + b_0 \geq 0}]$ where $\alpha_1, \beta_1 \sim \mathcal{N}(0,1)$ and $b_0 \sim \mathcal{N}(0,1)$ are independent random variables (we omitted some technical details).

The above statement can be straightforwardly extended to the following slightly more general version:

**Lemma 4.1.** *For every smooth function $\phi$, any $\overline{\mathbf{w}} \in \mathbb{S}^{d-1}$, and any $\varepsilon \in \left(0, \frac{1}{\mathfrak{C}_s(\phi,1)}\right)$ there exists a $H : \mathbb{R}^2 \to \left(-\mathfrak{C}_\varepsilon(\phi, 1), \mathfrak{C}_\varepsilon(\phi, 1)\right)$, which is $\mathfrak{C}_\varepsilon(\phi, 1)$-Lipschitz continuous and for all $\mathbf{u} \in \mathbb{S}^{d-1}$, we have*

$$\left| \phi(\overline{\mathbf{w}}^\top \mathbf{u}) - \mathbb{E}_{\mathbf{w}\sim\mathcal{N}(\mathbf{0},\mathbf{I}),b_0\sim\mathcal{N}(0,1)}[H(\mathbf{w}^\top \overline{\mathbf{w}}, b_0) \mathbb{I}_{\mathbf{w}^\top \mathbf{u} + b_0 \geq 0}] \right| \leq \varepsilon.$$

Very informally, this lemma states that the activation pattern of a one-layer ReLU network (given by $\mathbb{I}_{\mathbf{w}^\top \mathbf{u} \geq 0}$) at initialization can be used to express a smooth function of the dot product of the input vector with a fixed vector. While the above statement involves an expectation, one can easily replace it by an empirical average with slight increase in error. This statement formed the basis for FFN and RNN results in [15, 37]. Can we use it for RNNs for our general concept class? An attempt to do so is the following lemma showing that the pseudo-network can express any smooth function of the hidden state $\mathbf{h}^{(L-1)}$ and $\mathbf{x}^{(L)}$.

**Lemma 4.2** (Informal). *For a given smooth function $\phi$, a vector $\overline{\mathbf{w}} \in \mathbb{S}^{(m+d-1)}$, and any $\varepsilon \in \left(0, \frac{1}{\mathfrak{C}_s(\phi,1)}\right)$, there exist matrices $\mathbf{W}^*$ and $\mathbf{A}^*$ such that for every normalized input sequence $\mathbf{x} = (\mathbf{x}^{(1)}, \ldots, \mathbf{x}^{(L)})$ formed from a sequence $\overline{\mathbf{x}}$, we have with high probability,*

$$\left| F^{(L)}(\mathbf{x}; \mathbf{W}^*, \mathbf{A}^*) - \phi(\langle \overline{\mathbf{w}}, [\mathbf{h}^{(L-1)}, \mathbf{x}_{:d-1}^{(L)}]\rangle) \right| \leq \varepsilon,$$

*provided $m = \text{poly}(\frac{1}{\varepsilon}, L, \mathfrak{C}_\epsilon(\phi, \mathcal{O}(1)))$. Vector $\mathbf{x}_{:d-1}^{(L)}$ is $\mathbf{x}^{(L)}$ without the last coordinate, the bias term appended to each input.*

The reason $\mathbf{h}^{(L-1)}$ and $\mathbf{x}^{(L)}$ come up is because they serve as inputs to the RNN cell when processing the $L$-th input. The proof sketched below uses the fact that RNNs are one-layer FFNs unrolled over time. Hence, we could try to apply the result of Lemma 4.1 to the RNN cell at step $L$. However, a difficulty arises in carrying out this plan because the contributions of previous times steps also come up (as seen in the equations below) and it can be difficult to disentangle the contribution of step $L$. This is addressed in the proof:

*Proof.* Recall that $\mathbf{W}, \mathbf{A} \sim \mathcal{N}(\mathbf{0}, \frac{2}{m}\mathbf{I})$. Also, recall that we have assumed for simplicity $d_{\text{out}} = 1$. Hence, $\mathbf{B}$ and $\mathbf{Back}_{i\to L}$ are row and column vectors respectively. For typographical simplicty, denote by $b_r$ and $\mathbf{Back}_{i\to L,r}$ the respective $r$-th components of these vectors.

We set $\mathbf{W}^* := \mathbf{0}$ and for every $r \in [m]$, $\mathbf{a}_r^* := \frac{1}{m} b_r H(\sqrt{m/2}(\langle[\mathbf{w}_r, \mathbf{a}_{r,:d-1}], \overline{\mathbf{w}}\rangle), \sqrt{m/2}a_{r,d})\mathbf{e}_d$, for a function $H$ that we will describe below. With these choices we have

$$F^{(L)}(\mathbf{x}; \mathbf{W}^*, \mathbf{A}^*) = \sum_{i=1}^{L} \mathbf{Back}_{i\to L} \mathbf{D}^{(i)} \left(\mathbf{W}^* \mathbf{h}^{(i-1)} + \mathbf{A}^* \mathbf{x}^{(i)}\right)$$

$$= \frac{1}{m} \sum_{i=1}^{L} \sum_{r \in [m]} b_r \mathbf{Back}_{i\to L,r} H(\sqrt{m/2}(\langle[\mathbf{w}_r, \mathbf{a}_{r,:d-1}], \overline{\mathbf{w}}\rangle), \sqrt{m/2}a_{r,d}) \cdot \mathbb{I}_{\mathbf{w}_r^\top \mathbf{h}^{(i-1)} + \mathbf{a}_r^\top \mathbf{x}^{(i)} \geq 0}.$$

In the last step, we have simplified the formula using sum over neurons. The first $L - 1$ summands in the outer sum above nearly vanish due to small correlation between $\mathbf{B}$ and $\mathbf{Back}_{i \to L}$ for $i < L$ (see Lemma F.11). Recall that $\mathbf{Back}_{L \to L} = \mathbf{B}$ and thus the correlation is not small for $i = L$. This gives

$$F^{(L)}(\mathbf{x}; \mathbf{W}^*, \mathbf{A}^*) \approx \frac{1}{m} \sum_{r \in [m]} b_r^2 H(\sqrt{m/2}(\langle [\mathbf{w}_r, \mathbf{a}_{r,:d-1}], \overline{\mathbf{w}} \rangle), \sqrt{m/2} a_{r,d}) \cdot \mathbb{I}_{\mathbf{w}_r^\top \mathbf{h}^{(i-1)} + \mathbf{a}_r^\top \mathbf{x}^{(i)} \geq 0},$$

Now, this resembles a discretized version of Lemma 4.1. We can substitute $\mathbf{u}$ as $[\mathbf{h}^{(L-1)}, \mathbf{x}^{(L)}]$ in Lemma 4.1 and use concentration bounds with respect to the randomness of weights $\mathbf{W}$ and $\mathbf{A}$ to complete the proof. □

More generally, with much more technical work, it might be possible to prove an extension of the above lemma asserting the existence of a pseudo-network approximating a sum of functions of type $\sum_{i \in [L]} \phi_i(\langle \overline{\mathbf{w}}_i, [\mathbf{h}^{(i-1)}, \mathbf{x}^{(i)}] \rangle)$. However, even so it is not at all clear what class of functions of $\overline{\mathbf{x}}$ this represents because of the presence of the hidden state vectors.

Thus, the major challenge in constructing $\mathbf{W}^*$ and $\mathbf{A}^*$ to express the functions from the desired concept class is to use the information contained in $\mathbf{h}^{(\ell)}$. The construction of $\mathbf{W}^*$ in [37] is not able to use this information and ignores it by treating it as noise (which is also non-trivial). The idea underlying our construction is that $\mathbf{h}^{(\ell)}$ in fact contains information about all the inputs $\mathbf{x}^{(1)}, \ldots, \mathbf{x}^{(\ell)}$ up until step $\ell$. Furthermore and crucially, this information can be recovered approximately by a linear transformation (Theorem 4.5 below). This enables us to show:

**Theorem 4.3** (Existence of pseudo-network approximation for target function; abridged statement of Theorem D.2 in the appendix). *For every target function $F^*$ of the form Eq. (7), there exist matrices $\mathbf{W}^*$ and $\mathbf{A}^*$ such that with probability at least $1 - e^{-\Omega(\rho^2)}$ over $\mathbf{W}, \mathbf{A}, \mathbf{B}$, we have for every normalized input sequence $\mathbf{x} = (\mathbf{x}^{(1)}, \ldots, \mathbf{x}^{(L)})$ formed from a true sequence $\overline{\mathbf{x}}$,*

$$\left| F^{(L)}(\mathbf{x}; \mathbf{W}^*, \mathbf{A}^*) - \Phi^* \left( \langle \mathbf{w}^\dagger, [\overline{\mathbf{x}}^{(2)}, \ldots, \overline{\mathbf{x}}^{(L-2)}] \rangle \right) \right| \leq \varepsilon + \frac{1}{\text{poly}(\rho)},$$

*provided $m \geq \text{poly}(\rho, L, \varepsilon^{-1}, \mathfrak{C}_\varepsilon(\Phi, \mathcal{O}(\varepsilon_x^{-1})))$ and $\epsilon_x \leq \frac{1}{\text{poly}(\rho)}$.*

*Proof sketch.* By Theorem 4.5 there exists a matrix $\overline{\mathbf{W}}^{[L]}$ such that $\overline{\mathbf{W}}^{[L]\top} \mathbf{h}^{(L-1)} \approx [\mathbf{x}^{(1)}, \ldots, \mathbf{x}^{(L)}]$ for all input sequences $[\mathbf{x}^{(1)}, \ldots, \mathbf{x}^{(L)}]$. We can modify $\overline{\mathbf{W}}^{[L]}$ to get $[\overline{\mathbf{x}}^{(2)}, \ldots, \overline{\mathbf{x}}^{(L-1)}]$. Hence, by using $[\overline{\mathbf{W}}^{[L]} \mathbf{w}^\dagger, \mathbf{0}]$ as $\overline{\mathbf{w}}$ and $\Phi^*$ as $\phi$ in Lemma 4.2, we can have $F^{(L)}(\mathbf{x}; \mathbf{W}^*, \mathbf{A}^*) \approx \Phi^*(\langle \mathbf{w}^\dagger, \overline{\mathbf{W}}^{[L]\top} \mathbf{h}^{(L-1)} \rangle) \approx F^*(\overline{\mathbf{x}})$, implying $F^{(L)}$ and $F^*$ are close. Accounting for all the errors in inversion and approximation of function, we get the final bound. □

*Re-randomization.* In the proof sketches of Lemmas 4.2 and Theorem 4.3 above we swept a technical but critical consideration under the rug: the random variables $\{\mathbf{w}_r, \mathbf{a}_r\}_{r \in [m]}$, $\overline{\mathbf{W}}^{(L)}$, $\{\mathbf{Back}_{i \to L}\}_{i \in [L]}$ and $\{\mathbf{h}^{(i)}\}_{i \in [L]}$ are not independent. This invalidates application of standard concentration inequalities w.r.t. the randomness of $\mathbf{W}$ and $\mathbf{A}$—this application is required in the proofs. Here our new variation of the re-randomization technique from [37] comes in handy. The basic idea is the following: whenever we want to apply concentration bounds w.r.t. the randomness of $\mathbf{W}$ and $\mathbf{A}$, we divide the set of rows into disjoint sets of equal sizes. For each set, we will re-randomize the rows of the matrix $[\mathbf{W}, \mathbf{A}]_r$, show that the matrices $\overline{\mathbf{W}}^{[L]}$, $\{\mathbf{Back}_{i \to L}\}_{i \in [L]}$ and $\{\mathbf{h}^{(i)}\}_{i \in [L]}$ don't change a lot and then apply concentration bounds w.r.t. the new weights in the set. Finally, we account for the error from each set.

## 4.3 The rest of the proof

Having shown that there exists a pseudo-network approximation of the RNN that can also approximate the concept class, we will complete the proof by showing that SGD can find matrices with performance similar to $\mathbf{W}^*$ and $\mathbf{A}^*$ on the population objective $\text{Obj}(\cdot)$. Lemma D.3 shows that the training loss decreases with time. The basic idea is to use the fact that within small radius of perturbation, overparameterized RNNs behave as a linear network and hence the training can be

analyzed via convex optimization. Then, we show using Lemma D.4 that the Rademacher complexity for overparameterized RNNs is bounded. Again, the main idea here is that overparameterized RNNs behave as pseudo-networks in our overparameterized regime and hence their Rademacher complexity can be approximated by the Rademacher complexity of pseudo-networks. Finally, using generalization bounds on the Rademacher complexity, we get the final population-level objective in Theorem D.5.

### 4.4 Invertibility of RNNs at initialization

In this section, we describe how to get back $\mathbf{x}^{(1)}, \ldots, \mathbf{x}^{(L)}$ from the hidden state $\mathbf{h}^{(L)}$. The following lemma states that any linear function can be represented by a one-hidden layer FFN with activation function ReLU,with a small approximation error of the order $\frac{1}{\sqrt{m}}$:

**Lemma 4.4.** *[a simpler continuous version can be found in Lemma C.1 in the appendix] For any* $\mathbf{v} \in \mathbb{R}^d$, *the linear function taking* $\mathbf{x}$ *to* $\mathbf{v}^\top \mathbf{x}$ *for* $\mathbf{x} \in \mathbb{R}^d$, *can be represented as*

$$\left| \mathbf{v}^\top \mathbf{x} - \mathbf{p}^\top \sigma(\mathbf{T} \mathbf{x}) \right| \leq \frac{\|\mathbf{v}\| \cdot \|\mathbf{x}\|}{\sqrt{m}}, \tag{8}$$

*with* $\mathbf{p} = 2\mathbf{T}\mathbf{v}$, *where* $\mathbf{T} \in \mathbb{R}^{m \times d}$ *is a matrix with elements i.i.d. sampled from* $\mathcal{N}(0, 1)$.

Using the above lemma,[4] we will show that the hidden state $\mathbf{h}^{(L)}$ can be inverted using a matrix $\overline{\mathbf{W}}^{[L]}$ to get back the input sequence $\mathbf{x}^{(1)}, \ldots, \mathbf{x}^{(L)}$.

**Theorem 4.5.** *[informal version of Theorem D.1] There exists a set of matrices* $\{\overline{\mathbf{W}}^{[\ell]}\}_{\ell \in [L]}$, *which can possibly depend on* $\mathbf{W}$ *and* $\mathbf{A}$, *such that for any* $\varepsilon_x < \frac{1}{L}$ *and any given normalized sequence* $\mathbf{x}^{(1)}, \ldots, \mathbf{x}^{(L)}$, *with probability at least* $1 - e^{-\Omega(\rho^2)}$ *we have*

$$\left\| [\mathbf{x}^{(1)}, \ldots, \mathbf{x}^{(L)}] - \overline{\mathbf{W}}^{[L]\top} \mathbf{h}^{(L)} \right\|_\infty \leq \mathrm{poly}(L, \rho, m^{-1}, \varepsilon_x).$$

Very roughly, the above result is obtained by repeated application of Lemma 4.4 to go from $\mathbf{h}^{(\ell)}$ to $(\mathbf{h}^{(\ell-1)}, \mathbf{x}^{(\ell)})$ starting with $\ell = L$. This uses the fact that the RNN cell is a one-hidden layer neural network and hence Lemma 4.4 is applicable. Several difficulties need to be overcome to carry out this plan.

One difficulty is that a naive application of Lemma 4.4 results in exponential blowup of error with $L$. The reason is as follows: in Lemma 4.4, the factor 2 in the linear transformation $\mathbf{p}$ appears because of the activation function ReLU. A naive application will result in defining $\overline{\mathbf{W}}^{[\ell]}$ inductively as $2[\mathbf{W}\overline{\mathbf{W}}^{[\ell-1]}, \mathbf{A}]_r$ for $2 \leq \ell \leq L$, with $\mathbf{W}^{[1]} = \mathbf{A}$. This will lead to the approximation error exploding exponentially with $L$ in induction, since the approximation error at each step will depend on the norm of $\overline{\mathbf{W}}^{[\ell-1]}$.

However, one can observe from the proof of Lemma 4.4 that it suffices to use the following linear transformation: $p(\mathbf{w}) = \mathbf{w}^\top \mathbf{v} \mathbb{I}_{\mathbf{w}^\top \mathbf{x} \geq 0}$, which helps in removing the factor of 2. However, this requires knowing the indicator function $\mathbb{I}_{\mathbf{w}^\top \mathbf{x} \geq 0}$. This implies for RNNs, we need to know the indicator matrix $\mathbf{D}^{(\ell)}$ at each step.

To do so, we define a base sequence $\mathbf{x}_{(0)}$, whose indicator matrices $\{\mathbf{D}_{(0)}^{(\ell)}\}_{\ell \in [L]}$ will be used to approximate the indicator matrix $\mathbf{D}^{(\ell)}$ at each step $\ell$ for any sequence $\mathbf{x}$. That is, $\overline{\mathbf{W}}^{[\ell]}$ will be defined inductively as $[\mathbf{D}_{(0)}^{(\ell)} \mathbf{W}\overline{\mathbf{W}}^{[\ell-1]}, \mathbf{D}_{(0)}^{(\ell)}\mathbf{A}]_r$ for $2 \leq \ell \leq L$, with $\mathbf{W}^{[1]} = \mathbf{D}_{(0)}^{(1)}\mathbf{A}$. We then show that the approximation error of the indicator matrix builds up to give an error with polynomial dependence on $\varepsilon_x$ and $L$. We defer the technical details of this resolution to the full proof in the appendix.

Secondly, we apply re-randomization to tackle the dependence between $\mathbf{W}$, $\mathbf{A}$, $\{\mathbf{h}^{(\ell)}\}_{\ell \in [L]}$ and $\{\overline{\mathbf{W}}^{[\ell]}\}_{\ell \in [L]}$. We performed few toy experiments on the ability of invertibility for RNNs at initializa-

---

[4]This lemma is from a companion paper (forthcoming) where it is used to invert feedforward networks; apart from the above lemma, this work is very different from the present paper. We have reproduced the proof in full in the appendix.

tion (Sec. I). We observed, as predicted by our theorem above, that the error involved in inversion decreases with the number of neurons and increases with the length of the sequence (Fig. 4).

## 5  On concept classes

It is apparent that our concept class is very general as it allows arbitrary dependencies across tokens. To concretely illustrate the generality of our concept class, and to compare with previous work, we show that our result implies that RNNs can recognize a simple formal language $D_{L_1}$. Here we are working in the discrete setting where each input token comes from $\{0, 1\}$ possibly represented as a vector when fed to the RNN. For a sequence $\mathbf{z} \in \{0, 1\}^L$, we define $D_{L_1}(\mathbf{z})$ to be 1 if the number of 1's in $\mathbf{z}$ is exactly 1 and define it to be 0 otherwise. We can show that $D_{L_1}$ is not representable in the concept class of [37] (see Theorem H.1 in the appendix). However, we can show that the language $D_{L_1}$ can be recognized with a one-layer FFN with one neuron and quadratic activation. The idea is that we can simply calculate the number of 1's in the input string, which is doable using a single neuron. This implies that our concept class can represent language $D_{L_1}$ with low complexity.

More generally, we can show that our concept class can efficiently represent pattern matching problems, where strings belong to a language only if they contain given strings as substrings. In general, we can show that our concept class can express general regular languages. However, the complexity of the concept class may depend super-polynomially on the length of the input sequence, depending on the regular language (more discussion in sec. H). Some regular languages allow special treatment though. For example, consider the language PARITY. PARITY is the language over alphabet $\{0, 1\}$ with a string $w = (w_1, \ldots, w_j) \in$ PARITY iff $w_1 + \ldots + w_j = 1 \bmod 2$, for $j \geq 1$. We can show in sec. H that PARITY is easily expressible by our concept class with small complexity. RNNs perform well on regular language recognition task in our experiments in Sec. I. Figuring out which regular languages can be efficiently expressed by our concept class remains an interesting open problem.

## 6  Limitations and Conclusions

We proved the first result on the training and generalization of RNNs when the functions in the concept class are allowed to be essentially arbitrary continuous functions of the token sequence. Conceptually the main new idea was to show that the hidden state of the RNN contains information about the whole input sequence and this can be recovered via a linear transformation. We believe our techniques can be used to prove similar results for echo state networks.

Two main limitations of the present work are: (1) Our overparameterized setting requires the number of neurons to be large in terms of the problem parameters including the sequence length—and it is often qualitatively different from the practical setting. Theoretical analysis of practical parameter setting remains an outstanding challenge—even for one-hidden layer FFNs. (2) We did not consider generalization to sequences longer than those in the training data. Such a result would be very interesting but it appears that it would require stronger assumptions than our very general assumptions about the data distribution. Our techniques might be a useful starting point to that end: for example, if we knew that the distributions of the hidden states are similar at different times steps and the output is the same as the hidden state (i.e. $\mathbf{B}$ is the identity) then our results might easily generalize to higher lengths. We note that to our knowledge the limitation noted here holds for all works dealing with generalization for RNNs. (3) Understanding LSTMs remains open.

Our work addresses theoretical analysis of RNNs and is not directly concerned with applications. We do not anticipate any immediate societal impact of our work. In the long run, it may help improve RNNs and translate to impact on the applications.

## 7  Acknowledgements

We thank Sanjeev Arora, Raghav Somani and Abhishek Shetty for their valuable suggestions on improving the presentation of the paper. We also thank the reviewers, the area chair and the senior area chair for their valuable suggestions regarding this work.

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
