# Contents

The appendix has been structured as follows. We discuss few more notations and basic facts in section A. We prove few basic properties of the recurrent neural network at initialization in section B. In section C, we prove in lemma C.3 that any linear function on $[\mathbf{h}^{(\ell-1)}, \mathbf{x}^{(\ell)}]$ at RNN cell $\ell$ can be expressed as a linear transformation of the hidden state $\mathbf{h}^{(\ell)}$. In section D.1, we use the above lemma to show in theorem D.1 that from hidden state $\mathbf{h}^{[L]}$, one can get $[\mathbf{x}^{(1)}, \cdots, \mathbf{x}^{[L]}]$ using a linear transformation. We show in section D.2 that a linear approximation of the recurrent neural networks exist at initialization that can approximate the target function in our concept class. We finish the proof in section D.3, where we show that RNNs can find a network with population risk close to the target function. We discuss about the experiments in section I.

# A  Further Preliminaries

## A.1  Notations

Let $\mathbb{B}^d := \{\mathbf{x} \in \mathbb{R}^d \mid \|\mathbf{x}\|_2 \leq 1\}$ be the unit $L_2$-ball in $\mathbb{R}^d$, and let $\mathbb{S}^{d-1} := \{\mathbf{x} \in \mathbb{R}^d \mid \|\mathbf{x}\|_2 = 1\}$ be the unit $L_2$-sphere in $\mathbb{R}^d$. Let $V_d := \frac{\pi^{d/2}}{\Gamma((d+1)/2)}$ be the $d$-dimensional volume of $\mathbb{B}^d$ and let $\omega_{d-1} := \frac{2\pi^{d/2}}{\Gamma(d/2)}$ be the surface area (i.e. the $(d-1)$-dimensional volume) of $\mathbb{S}^{d-1}$. Given a matrix $\mathbf{T} \in \mathbb{R}^{d_1 \times d_2}$ and a set $S \subset [d_1]$, we denote $\mathbf{T}_S$ as the matrix $\mathbb{R}^{|S| \times d_2}$ that contains the rows of $\mathbf{T}$ whose indices are in the set $S$. We will denote a diagonal matrix $\mathbf{D}_S$ for a given set $S \subset [n]$ as $d_{ii} = 1$ for $i \in S$ and is 0 elsewhere.

For positive integer $n$ define $[n] := \{1, 2, \ldots, n\}$. For a matrix $\mathbf{M} \in \mathbb{R}^{m \times n}$, set $\|\mathbf{M}\|_{2,\infty} := \big\| (\|\mathbf{m}_1\|_2, \ldots \|\mathbf{m}_m\|_2) \big\|_\infty$, where $\mathbf{m}_1^T, \mathbf{m}_2^T, \ldots$ are the rows of $\mathbf{M}$. Let $\mu_d^\beta$ denote the Gaussian measure on $\mathbb{R}^d$ associated with the Gaussian probability distribution $\mathcal{N}(\mathbf{0}, \beta^2 \mathbf{I})$. Let $\mu_d := \mu_d^1$ denote the standard Gaussian measure on $\mathbb{R}^d$.

For simplicity of notation, we will use
$$\varrho := \frac{100 L d_{\text{out}} p \cdot \mathfrak{C}_\varepsilon(\Phi, \mathcal{O}(\varepsilon_x^{-1})) \cdot \log m}{\varepsilon}.$$
$$\rho := 100 L d_{\text{out}} \log m.$$

## A.2  Extra set of Notations for RNNs

We denote by $\mathbf{A}_{[d-1]} \in \mathbb{R}^{m \times (d-1)}$ the matrix containing the first $d-1$ columns of the matrix $\mathbf{A}$. Then, we define an alternate fixed sequence as follows: $\mathbf{x}_{(0)} := (\mathbf{x}_{(0)}^{(1)}, \ldots, \mathbf{x}_{(0)}^{(L)})$, where
$$\mathbf{x}_{(0)}^{(1)} = (\mathbf{0}^{d-1}, 1), \quad \mathbf{x}_{(0)}^{(\ell)} = (\mathbf{0}^{d-1}, \varepsilon_x), \quad \forall \ell \in [2, L-1], \quad \mathbf{x}_{(0)}^{(L)} = (\mathbf{0}^{d-1}, 1).$$
We will heavily use this fixed sequence to build our model later on. There is a small difference in our definition of normalized sequence and the definition in [37]. The difference is in the definition of $\mathbf{x}^{(L)}$; our choice gives a better and simpler error bound. This difference leads to only minor changes in the theorems that we take from [37] and we will account for those changes.

We re-introduce two more notations here for RNNs in def. 3.2. For each $\ell \in [L]$, define $\mathbf{D}^{(\ell)} \in \mathbb{R}^{m \times m}$ as a diagonal matrix, with diagonal entries
$$d_{rr}^{(\ell)} := \mathbb{I}[\mathbf{w}_r^\top \mathbf{h}^{(\ell-1)} + \mathbf{a}_r^\top \mathbf{x}^{(\ell)} \geq 0], \quad \forall r \in [m]. \tag{9}$$
Hence, $\mathbf{h}^{(\ell)} = \mathbf{D}^{(\ell)}(\mathbf{W}\mathbf{h}^{(\ell-1)} + \mathbf{A}\mathbf{x}^{(\ell)})$. Also, define $\mathbf{Back}_{i \to j} \in \mathbb{R}^{d_{\text{out}} \times m}$ for each $1 \leq i \leq j \leq L$ by
$$\mathbf{Back}_{i \to j} := \mathbf{B}\mathbf{D}^{(j)}\mathbf{W} \ldots \mathbf{D}^{(i+1)}\mathbf{W},$$
with $\mathbf{Back}_{i \to i} := \mathbf{B}$. Matrices $\mathbf{Back}_{i \to j}$ arise naturally in Eq. (4) in the first-order Taylor approximation in terms of the parameters of the function $F_{\text{rnn}}^{(\ell)}(\overline{\mathbf{x}}; \mathbf{W}, \mathbf{A})$. Very roughly, one can interpret $\mathbf{Back}_{i \to j}$ as related to the backpropagation signal from the output at step $j$ to the parameters at step $i$.

For the fixed base sequence $\mathbf{x}_{(0)}^{(1)}, \ldots, \mathbf{x}_{(0)}^{(L)}$, we will denote the hidden states by $\mathbf{h}_{(0)}^{(\ell)}$ and the diagonal matrices by $\mathbf{D}_{(0)}^{(\ell)}$ for $\ell \leq L$.

## A.3 Redefine Concept Class

In this section, we re-define the concept class introduced in the main paper. We introduce additional symbols related to the lipschitz constant and the absolute bounds over the functions, that are necessary in the proof of the main theorem.

**Definition A.1** (Concept Class). Our concept class consists of functions $F : \mathbb{R}^{(L-2)\cdot(d-1)} \to \mathbb{R}^{d_{\text{out}}}$ defined as follows. Let $\Phi$ denote a set of smooth functions with Taylor expansions with finite complexity as in Def. 2.1. To define a function $F$, we choose a subset $\{\Phi_{r,s} : \mathbb{R} \to \mathbb{R}\}_{r\in[p], s\in[d_{\text{out}}]}$ from $\Phi$, $\{\mathbf{w}_{r,s}^\dagger \in \mathbb{S}^{(L-2)(d-1)-1}\}_{r\in[p], s\in[d_{\text{out}}]}$, a set of weight vectors, and $\{b_{r,s}^\dagger \in \mathbb{R}\}_{r\in[p], s\in[d_{\text{out}}]}$, a set of output coefficients with $|b_{r,s}^\dagger| \leq 1$. Then, we define $F : \mathbb{R}^{(L-2)\cdot(d-1)} \to \mathbb{R}^{d_{\text{out}}}$, where for each output dimension $s \in [d_{\text{out}}]$ we define the $s$-th coordinate $F_s$ of $F = (F_1, \ldots, F_{d_{\text{out}}})$ by

$$F_s(\overline{\mathbf{x}}) := \sum_{r\in[p]} b_{r,s}^\dagger \Phi_{r,s}\left(\langle \mathbf{w}_{r,s}^\dagger, [\overline{\mathbf{x}}^{(2)}, \ldots, \overline{\mathbf{x}}^{(L-1)}]\rangle\right). \tag{10}$$

To simplify formulas, we assume $\Phi_{r,s}(0) = 0$ for all $r$ and $s$. We denote the complexity of the concept class by

$$\mathfrak{C}_\varepsilon(\Phi, R) := \max_{\phi\in\Phi}\{\mathfrak{C}_\varepsilon(\phi, R)\}, \quad \mathfrak{C}_{\mathfrak{s}}(\Phi, R) := \max_{\phi\in\Phi}\{\mathfrak{C}_{\mathfrak{s}}(\phi, R)\}.$$

Let $L_\phi$ denote the Lipschitz constant of function $\phi$ in the range $(-\sqrt{L}, \sqrt{L})$ and let $L_\Phi := \max_{\phi\in\Phi} L_\phi$. Also, $C_\phi$ denote the upper bound on the absolute value of $\phi$ in the range $(-\sqrt{L}, \sqrt{L})$ and let $C_\Phi := \max_{\phi\in\Phi} C_\phi$. We only focus on the properties of the functions in the above range, since the argument to the functions $\langle \mathbf{w}_{r,s}^\dagger, [\overline{\mathbf{x}}^{(2)}, \ldots, \overline{\mathbf{x}}^{(L-1)}]\rangle$ can be shown to lie in the above range. Using the definition of $\mathfrak{C}$ from def. 2.1, one can show that $C_\Phi, L_\Phi \leq \mathfrak{C}_s(\Phi, \sqrt{2L})$. We assume that there exists some function $F^*$ in the concept class that achieves population loss OPT. Hence, our aim is to learn a function with population loss $\text{OPT} + \varepsilon$.

## A.4 Important facts

We will need the following well-known results.

**Fact A.1** (e.g. Cor. 5.35 in [39]). *Let $\mathbf{A}$ be an $N \times n$ matrix whose entries are independent standard normal random variables. Then for every $t \geq 0$, with probability at least $1 - 2\exp\left(-t^2/2\right)$ one has*

$$\sqrt{N} - \sqrt{n} - t \leq s_{\min}(\mathbf{A}) \leq s_{\max}(\mathbf{A}) \leq \sqrt{N} + \sqrt{n} + t.$$

**Fact A.2** (e.g. Thm. 1.1 in [40]). *Let $\varepsilon \in (0, 1)$ and let $m, n, N$ be positive integers. Consider a random $m \times n$ matrix $\mathbf{W} = \mathbf{BA}$, where $\mathbf{A}$ is an $N \times n$ random matrix whose entries are independent random variables with mean zero and $(4 + \varepsilon)$-th moment bounded by 1, and $\mathbf{B}$ is an $m \times N$ non-random matrix. Then w.p. exceeding $1 - 2\exp\left(-t^2/2\right)$*

$$\|\mathbf{W}\| \leq C(\varepsilon)\|\mathbf{B}\|\left(\sqrt{n} + \sqrt{m} + t\right), \text{ and}$$

$$s_{\min}(\mathbf{W}) \geq C(\varepsilon)\|\mathbf{B}\|\left(\sqrt{m} - \sqrt{n-1} - t\right),$$

*where $C(\varepsilon)$ is a constant that depends only on $\varepsilon$.*

**Fact A.3** (Example 2.11 in [41]). *Let $Z_1, Z_2, \ldots$ be i.i.d. one-dimensional standard Gaussian random variables. Then*

$$\Pr\left[\left|\frac{1}{n}\sum_{i=1}^{n} Z_i^2 - 1\right| \geq t\right] \leq 2e^{-nt^2/8}, \quad \text{for all } t \in (0, 1).$$

**Fact A.4** (Maximum of Gaussians, see e.g. [42].). *Let $x_1, x_2, \ldots, x_n$ be $n$ Gaussians following $\mathcal{N}\left(0, \sigma^2\right)$. Then for any $\rho > 0$*

$$\Pr\left\{\max_{i\in[n]} |x_i| \leq \sqrt{2}\rho\sigma\right\} \geq 1 - 2ne^{-\rho^2}.$$

**Fact A.5** (Hoeffding's inequality). *Let $x_1, \cdots, x_n$ be $n$ independent random variables, with each $x_i$ strictly bounded in the interval $[a_i, b_i]$. Let $\overline{x} = \frac{1}{n}\sum_i x_i$. Then for any $\rho > 0$,*

$$\Pr\left[|\overline{x} - \mathbb{E}_x\overline{x}| \geq \frac{\rho}{n}\sqrt{\sum_i (a_i - b_i)^2}\right] \leq e^{-2\rho^2},$$

**Definition A.2** ($\epsilon$-net on the sphere). A set $\mathcal{N} \subset \mathbb{S}^{d-1}$ is called an $\epsilon$-net of $\mathbb{S}^{d-1}$ if every point in $\mathbb{S}^{d-1}$ is within Euclidean distance $\epsilon$ of some point in $\mathcal{N}$. In other words, for every $\mathbf{x} \in \mathbb{S}^{d-1}$ there is a point $\tilde{\mathbf{x}} \in \mathcal{N}$ such that $\|\mathbf{x} - \tilde{\mathbf{x}}\| \leq \epsilon$.

**Fact A.6** (see the proof of Cor. 4.2.13 in [43]). *$\mathbb{S}^{d-1}$ has an $\epsilon$-net of size at most $(3/\epsilon)^d$.*

Let $\mathcal{F} : \mathbb{R}^d \to \mathbb{R}$ be a class of function and $\mathcal{Z} = (\mathbf{x}_1, \ldots, \mathbf{x}_N)$ be a set of training examples in $\mathbb{R}^d$. The empirical rademacher complexity is given by

$$\hat{R}(\mathcal{F}, \mathcal{Z}) := \sup_{f \in \mathcal{F}} \mathbb{E}_{\zeta \in \{\pm 1\}^N}\left[\frac{1}{N}\sum_{q \in [N]} \zeta_q f(\mathbf{x}_q)\right]$$

**Fact A.7** (Generalization through rademacher complexity, [44]). *If for every function $f \in \mathcal{F}, |f| \leq b$, then with probability at least $1 - \delta$ for any $\delta \geq 0$,*

$$\sup_{f \in \mathcal{F}}\left[\mathbb{E}_{\mathbf{x}\in\mathcal{D}}f(\mathbf{x}) - \mathbb{E}_{\mathbf{x}\in\mathcal{Z}}f(\mathbf{x})\right] \leq 2\hat{R}(\mathcal{F}, \mathcal{Z}) + \mathcal{O}(\frac{b\sqrt{\log(1/\delta)}}{\sqrt{N}}).$$

**Fact A.8** (Rademacher complexity of linear networks, [44]). *Suppose $\|\mathbf{x}\|_2 = 1$ for all $\mathbf{x} \in \mathcal{X}$. The class $\mathcal{F} = \{\mathbf{x} \mapsto \langle \mathbf{w}, \mathbf{x}\rangle \mid \|\mathbf{w}\|_2 < B\}$ has rademacher complexity*

$$\widehat{R}(\mathcal{F}, \mathcal{Z}) \leq O\left(\frac{B}{\sqrt{N}}\right).$$

## B  Some basic properties of recurrent neural networks at initialization

The following lemma shows some basic properties of the recurrent neural network at initialization. They are a result of the concentration bounds that can be applied for gaussian weight matrices $\mathbf{W}$ and $\mathbf{A}$.

**Lemma B.1.** *For any $\epsilon_x \in (0, \frac{1}{L})$ and any normalized input sequence $(\mathbf{x}^{(1)}, \mathbf{x}^{(2)}, ..., \mathbf{x}^{(L)})$, for all $\ell \in [L]$ with probability at least $1 - e^{-\Omega(\rho^2)}$ we have*

1. $\left|\|\mathbf{h}^{(\ell)}\| - \sqrt{2 + (\ell - 2)\epsilon_x^2}\right| \leq \frac{\rho^2}{\sqrt{m}}$.

2. $\left\|\mathbf{W}\mathbf{h}^{(\ell)}\right\|_\infty, \left\|\mathbf{A}\mathbf{x}^{(\ell)}\right\|_\infty \leq \mathcal{O}(\frac{\rho}{\sqrt{m}})$, *for all $1 \leq \ell \leq L$.*

3. $\left\|\mathbf{h}^{(\ell)} - \mathbf{h}^{(\ell)}_{(0)}\right\| \leq \sqrt{L}\epsilon_x$, *for all $1 \leq \ell \leq L$.*

4. $\left\|\mathbf{W}(\mathbf{h}^{(\ell)} - \mathbf{h}^{(\ell)}_{(0)})\right\|_\infty, \left\|\mathbf{A}(\mathbf{x}^{(\ell)} - \mathbf{x}^{(\ell)}_{(0)})\right\|_\infty \leq \mathcal{O}(\frac{\rho\sqrt{L}\epsilon_x}{\sqrt{m}})$, *for all $1 \leq \ell \leq L$.*

5. $(1 - \frac{1}{100L})^{j-i+1}\|\mathbf{u}\| \leq \left\|\mathbf{D}^{(j)}\mathbf{W}\mathbf{D}^{(j-1)}\mathbf{W}\cdots\mathbf{D}^{(i+1)}\mathbf{W}\mathbf{D}^{(i)}\mathbf{W}\mathbf{u}\right\| \leq (1 + \frac{1}{100L})^{j-i+1}\|\mathbf{u}\|$ *for all $1 \leq i \leq j \leq L$ and any fixed vector $\mathbf{u}$.*

6. $(1 - \frac{1}{100L})^{j-i+1}\|\mathbf{u}\| \leq \left\|\mathbf{D}^{(j)}\mathbf{W}\mathbf{D}^{(j-1)}\mathbf{W}\cdots\mathbf{D}^{(i+2)}\mathbf{W}\mathbf{D}^{(i+1)}\mathbf{W}\mathbf{D}^{(i)}\mathbf{A}\mathbf{u}\right\| \leq (1 + \frac{1}{100L})^{j-i+1}\|\mathbf{u}\|$ *for all $1 \leq i \leq j \leq L$ and all vectors $\mathbf{u} \in \mathbb{R}^d$.*

7. $\left\|\mathbf{D}^{(j)}\mathbf{W}\mathbf{D}^{(j-1)}\mathbf{W}\cdots\mathbf{D}^{(i+1)}\mathbf{W}\mathbf{D}^{(i)}\mathbf{W}\right\| \leq \mathcal{O}(L^3)$ *for all $1 \leq i \leq j \leq L$.*

8. $\left\|\mathbf{D}^{(\ell)} - \mathbf{D}^{(\ell)}_{(0)}\right\|_0 \leq \mathcal{O}(L^{1/3}\epsilon_x^{2/3}m)$ *for all $1 \leq \ell \leq L$.*

9. $\left|\mathbf{u}^\top \mathbf{W}\mathbf{D}^{(j)}\mathbf{W}\mathbf{D}^{(j-1)}\mathbf{W}\cdots\mathbf{D}^{(i+1)}\mathbf{W}\mathbf{D}^{(i)}\mathbf{W}\mathbf{v}\right| \leq \mathcal{O}(\frac{\sqrt{s_1}\rho}{\sqrt{m}})\cdot\|\mathbf{u}\|\|\mathbf{v}\|$, *for all* $1 \leq i \leq j \leq$ $L$ *and for all* $s_1$*-sparse vectors* $\mathbf{u}$ *and* $s_2$*-sparse vectors* $\mathbf{v}$ *with* $s_1, s_2 \leq \frac{m}{\rho^3}$.

10. $\left|\mathbf{u}^\top \mathbf{W}\mathbf{D}^{(j)}\mathbf{W}\mathbf{D}^{(j-1)}\mathbf{W}\cdots\mathbf{D}^{(i+1)}\mathbf{W}\mathbf{D}^{(i)}\mathbf{W}\mathbf{v}\right| \leq \mathcal{O}(\frac{\sqrt{s_1}\rho}{\sqrt{m}})\cdot\|\mathbf{u}\|\|\mathbf{v}\|$, *for all* $1 \leq i \leq j \leq$ $L$ *and for all* $s_1$*-sparse vectors* $\mathbf{u}$*, with* $s_1 \leq \frac{m}{\rho^3}$*, and a fixed vector* $\mathbf{v}$.

11. $\left\|\mathbf{W}\mathbf{D}^{(j)}\mathbf{W}\mathbf{D}^{(j-1)}\mathbf{W}\cdots\mathbf{D}^{(i+1)}\mathbf{W}\mathbf{D}^{(i)}\mathbf{A}\right\|_{\infty,\infty} \leq \mathcal{O}(\frac{\rho}{\sqrt{m}})$, *for all* $1 \leq i \leq j \leq L$.

*Proof.* All of the properties except 4, 6, 9, 10 and 11 have been taken directly from Lemma B.1 and Lemma D.1 in [37].

4 The proof will follow from the proof of property 3. We outline the proof here. We have $\mathbf{W}(\mathbf{h}^{(\ell)} - \mathbf{h}^{(\ell)}_{(0)}) = \mathbf{W}\mathbf{U}\mathbf{U}^\top(\mathbf{h}^{(\ell)} - \mathbf{h}^{(\ell)}_{(0)})$ where $\mathbf{U} = GS\left(\mathbf{h}^{(1)}, \cdots, \mathbf{h}^{(L)}, \mathbf{h}^{(1)}_{(0)}, \cdots, \mathbf{h}^{(L)}_{(0)}\right)$. Each entry of $\mathbf{W}\mathbf{U}$ is i.i.d. from $\mathcal{N}\left(0, \frac{2}{m}\right)$. For any fixed $\mathbf{z}$ we have $\|\mathbf{W}\mathbf{U}\mathbf{z}\|_\infty \leq O(\sqrt{\rho}/\sqrt{m})$ with probability at least $1 - e^{-\Omega(\rho^2)}$ Taking $\epsilon$ -net over $\mathbf{z}$ and using $\left\|\mathbf{h}^{(\ell)} - \mathbf{h}^{(\ell)}_{(0)}\right\| \leq \sqrt{L}\epsilon_x$ from property 3 gives the desired bound. [5]

6 The proof will follow from property 5. We will give the brief outline here. For a fixed vector $\mathbf{u}$, property 5 shows that

$$(1-\frac{1}{100L})^{j-i}\left\|\mathbf{D}^{(i)}\mathbf{A}\mathbf{u}\right\| \leq \left\|\mathbf{D}^{(j)}\mathbf{W}\mathbf{D}^{(j-1)}\mathbf{W}\cdots\mathbf{D}^{(i+2)}\mathbf{W}\mathbf{D}^{(i+1)}\mathbf{W}\mathbf{D}^{(i)}\mathbf{A}\mathbf{u}\right\| \leq (1+\frac{1}{100L})^{j-i}\left\|\mathbf{D}^{(i)}\mathbf{A}\mathbf{u}\right\|.$$

Following the proof technique of Lemma 7.1 in [45], we can show that with probability $1 - e^{-\Omega(\rho^2)}$,

$$(1 - \mathcal{O}(\frac{\rho}{\sqrt{m}}))\|\mathbf{u}\| \leq \left\|\mathbf{D}^{(i)}\mathbf{A}\mathbf{u}\right\| \leq (1 + \mathcal{O}(\frac{\rho}{\sqrt{m}}))\|\mathbf{u}\|.$$

Thus, assuming $m \geq \mathcal{O}(\rho^2 L^2)$ so that $\frac{\rho}{\sqrt{m}} = \mathcal{O}(\frac{1}{L})$,

$$(1-\frac{1}{100L})^{j-i+1}\|\mathbf{u}\| \leq \left\|\mathbf{D}^{(j)}\mathbf{W}\mathbf{D}^{(j-1)}\mathbf{W}\cdots\mathbf{D}^{(i+2)}\mathbf{W}\mathbf{D}^{(i+1)}\mathbf{W}\mathbf{D}^{(i)}\mathbf{A}\mathbf{u}\right\| \leq (1+\frac{1}{100L})^{j-i+1}\|\mathbf{u}\|.$$

The proof will follow from using an $\epsilon$-net over $\mathbb{R}^d$ to quantify for all vectors $\mathbf{u}$.

9 The proof will follow from Lemma B.14 in [19]. We will give a brief overview here. Let $\mathbf{v}$ be a fixed $s_2$-sparse vector in $\mathbb{R}^m$. Then, letting $\mathbf{z} = \mathbf{D}^{(j)}\mathbf{W}\cdots\mathbf{D}^{(i)}\mathbf{W}\mathbf{v}$, we have w.p. $1 - e^{-\Omega(m/L^2)}$ from Lemma B.12 of [19], $(1 - \frac{1}{100L})^{j-i+1}\|\mathbf{v}\| \leq \|\mathbf{z}\| \leq (1 + \frac{1}{100L})^{j-i+1}\|\mathbf{v}\|$.

Let $\mathbf{z}^{(\ell)} = \mathbf{D}^{(\ell)}\mathbf{W}\cdots\mathbf{D}^{(i)}\mathbf{W}\mathbf{v}$ for $i \leq \ell \leq j$. Also let $\mathbf{U} = GS\left(\mathbf{h}^{(1)}, \cdots, \mathbf{h}^{(L)}, \mathbf{z}^{(1)}, \cdots, \mathbf{z}^{(L)}\right)$. Each entry of $\mathbf{W}\mathbf{U}$ is i.i.d. from $\mathcal{N}\left(0, \frac{2}{m}\right)$. The dimension of $\mathbf{W}\mathbf{U}$ is $(m, j - i + 1 + L)$. Using Fact A.3, we can show that for a $s_1$-sparse fixed vector $\mathbf{u}$, w.p. at least $1 - e^{-\Omega(Lt^2)}$,

$$\left\|(\mathbf{W}\mathbf{U})^\top \mathbf{u}\right\| \leq \mathcal{O}(\frac{\sqrt{L}t}{\sqrt{m}})\|\mathbf{u}\|.$$

---

[5]GS denotes Gram-schmidt orthonormalization.

Hence,

$$
\begin{aligned}
\left|\mathbf{u}^\top \mathbf{W}\mathbf{D}^{(j)}\mathbf{W}\mathbf{D}^{(j-1)}\mathbf{W}\cdots\mathbf{D}^{(i+1)}\mathbf{W}\mathbf{D}^{(i)}\mathbf{W}\mathbf{v}\right| &= \left|\mathbf{u}^\top \mathbf{W}\mathbf{U}\mathbf{U}^\top \mathbf{D}^{(j)}\mathbf{W}\mathbf{D}^{(j-1)}\mathbf{W}\cdots\mathbf{D}^{(i+1)}\mathbf{W}\mathbf{D}^{(i)}\mathbf{W}\mathbf{v}\right| \\
&\leq \left\|(\mathbf{W}\mathbf{U})^\top \mathbf{u}\right\|\left\|\mathbf{U}^\top \mathbf{D}^{(j)}\mathbf{W}\cdots\mathbf{D}^{(i)}\mathbf{W}\mathbf{v}\right\| \\
&= \left\|(\mathbf{W}\mathbf{U})^\top \mathbf{u}\right\|\left\|\mathbf{D}^{(j)}\mathbf{W}\cdots\mathbf{D}^{(i)}\mathbf{W}\mathbf{v}\right\| \\
&\leq \mathcal{O}(\frac{\sqrt{L}t}{\sqrt{m}})\|\mathbf{u}\|\|\mathbf{v}\| .
\end{aligned}
$$

The proof follows from setting $t = \rho\sqrt{s_1}$ and taking an $\epsilon$-net bound over all $s_2$-sparse vectors $\mathbf{v}$ and $s_1$-sparse vectors $\mathbf{u}$, that amounts to an error probability at least $1 - e^{\Omega(s_2 \log m)}e^{-\Omega(m/L^2)} - e^{\Omega(s_1 \log m)}e^{-\Omega(s_1 \rho^2)}$, which simplifies to atleast $1 - e^{-\Omega(\rho^2)}$, since $s_1, s_2 \leq \frac{m}{\rho^3}$.

10 The proof will follow the same technique used for property 9. The only difference is that $\mathbf{v}$ will be fixed and hence, no $\epsilon$-net is necessary for the vector $\mathbf{v}$.

11 The proof will follow the same technique used for property 9. $\mathbf{u}$ will be chosen from $\mathbf{e}_1, \cdots, \mathbf{e}_m$ and $\mathbf{v}$ will be chosen from the set of vectors $\left\{\mathbf{D}^{(i)}\mathbf{A}\mathbf{e}_1, \cdots, \mathbf{D}^{(i)}\mathbf{A}\mathbf{e}_d\right\}$. Thus, the union bound over $\mathbf{u}$ and $\mathbf{v}$ needs to consider only $m$ and $d$ vectors respectively, in place of the $\epsilon$-net over $s_1$ and $s_2$ sparse vectors.

$\square$

The following lemma shows that the hidden states at initialization are resilient to re-randomization of few rows of the gaussian matrices $\mathbf{W}$ and $\mathbf{A}$. The proof again follows from applying concentration bounds w.r.t. the new set of weights. This lemma is used multiple times later to break the correlations among different functions of $\mathbf{W}$ and $\mathbf{A}$.

**Lemma B.2** (Stability after re-randomization, Lemma E.1 in [37] ). *Consider a fixed set $\mathcal{K} \subseteq [m]$ with cardinality $N = |\mathcal{K}|$. Consider the following matrices.*

- $\widetilde{\mathbf{W}} \in \mathbb{R}^{m \times m}$ *where* $\widetilde{\mathbf{w}}_k = \mathbf{w}_k$ *for* $k \in [m]\backslash\mathcal{K}$ *but* $\widetilde{\mathbf{w}}_k \sim \mathcal{N}\left(0, \frac{2\mathbf{I}}{m}\right)$ *is i.i.d. for* $k \in \mathcal{K}$

- $\widetilde{\mathbf{A}} \in \mathbb{R}^{m \times d}$ *where* $\widetilde{\mathbf{a}}_k = \mathbf{a}_k$ *for* $k \in [m]\backslash\mathcal{K}$ *but* $\widetilde{\mathbf{a}}_k \sim \mathcal{N}\left(0, \frac{2\mathbf{I}}{m}\right)$ *is i.i.d. for* $k \in \mathcal{K}$

*For any normalized input sequence $\mathbf{x}^{(1)}, \ldots, \mathbf{x}^{(L)} \in \mathbb{S}^{d-1}$, we consider the following two executions of ESNs under $\mathbf{W}$ and $\widetilde{\mathbf{W}}$ respectively:*

$$
\begin{aligned}
\mathbf{g}^{(0)} = \mathbf{h}^{(0)} = 0 \qquad\qquad & \mathbf{g}^{(0)\prime} = \mathbf{h}^{(0)\prime} = 0 \\
\mathbf{g}^{(\ell)} = \mathbf{W}\mathbf{h}^{(\ell-1)} + \mathbf{A}\mathbf{x}^{(\ell)} \qquad & \tilde{\mathbf{g}}^{(\ell)} = \mathbf{g}^{(\ell)} + \mathbf{g}^{(\ell)\prime} = \widetilde{\mathbf{W}}\left(\mathbf{h}^{(\ell-1)} + \mathbf{h}^{(\ell-1)\prime}\right) + \widetilde{\mathbf{A}}\mathbf{x}^{(\ell)} \\
\mathbf{h}^{(\ell)} = \sigma\left(\mathbf{W}\mathbf{h}^{(\ell-1)} + \mathbf{A}\mathbf{x}^{(\ell)}\right) \quad & \tilde{\mathbf{h}}^{(\ell)} = \mathbf{h}^{(\ell)} + \mathbf{h}^{(\ell)\prime} = \sigma\left(\widetilde{\mathbf{W}}\left(\mathbf{h}^{(\ell-1)} + \mathbf{h}^{(\ell-1)\prime}\right) + \widetilde{\mathbf{A}}\mathbf{x}^{(\ell)}\right) \quad \text{for } \ell \in [L]
\end{aligned}
$$

*and define diagonal sign matrices $\mathbf{D}^{(\ell)} \in \{0,1\}^{m \times m}$ and $\widetilde{\mathbf{D}}^{(\ell)} = \mathbf{D}^{(\ell)} + \mathbf{D}^{(\ell)\prime} \in \{0,1\}^{m \times m}$ by letting*

$$
d_{k,k}^{(\ell)} = \mathbb{I}_{g_k^{(\ell)} \geq 0} \text{ and } \widetilde{d}_{k,k}^{(\ell)} = \mathbb{I}_{\tilde{g}_k^{(\ell)} \geq 0}
$$

*Let $N = |\mathcal{K}| \leq m/\rho^{23}$. Fix any normalized input sequence $\mathbf{x}^{(1)}, \ldots, \mathbf{x}^{(L)}$. We have, with probability at least $1 - e^{-\Omega(\rho^2)}$ over the randomness of $\mathbf{W}, \widetilde{\mathbf{W}}, \mathbf{A}, \widetilde{\mathbf{A}}$,*

1. $\left\|\mathbf{g}^{(\ell)\prime}\right\|, \left\|\mathbf{h}^{(\ell)\prime}\right\| \leq \mathcal{O}\left(\rho^5\sqrt{N/m}\right)$ *for every $\ell \in [L]$.*

2. $\left|\left\langle \mathbf{w}_k, \mathbf{h}^{(\ell)\prime}\right\rangle\right| \leq \mathcal{O}\left(\rho^5 N^{2/3} m^{-2/3}\right)$ *for every $k \in [m], \ell \in [L]$.*

3. $\left\| \mathbf{W}_{\mathcal{K}} \mathbf{h}^{(\ell)\prime} \right\| \leq \mathcal{O}\left( \rho^5 N^{2/3} m^{-2/3} \right)$     *for every $\ell \in [L]$.*

4. $\left\| \mathbf{D}^{(\ell)} - \widetilde{\mathbf{D}}^{(\ell)} \right\|_0 \leq \mathcal{O}(\rho^4 N^{1/3} m^{2/3})$, *for all $\ell \in [L]$.*

5. $\left\| \left( \prod_{i \leq \ell' \leq j} \widetilde{\mathbf{D}}^{(k-\ell'+1)} \widetilde{\mathbf{W}} - \prod_{i \leq \ell' \leq j} \mathbf{D}^{(k-\ell'+1)} \mathbf{W} \right) \mathbf{v} \right\|_2 \leq \mathcal{O}(\rho^5 (N/m)^{1/6}) \|\mathbf{v}\|$, *for all $1 \leq i \leq j \leq L$ and for a fixed vector $\mathbf{v}$.*

6. $\left\| \mathbf{W}_{\mathcal{K}} \left( \prod_{i \leq \ell' \leq j} \widetilde{\mathbf{D}}^{(k-\ell'+1)} \widetilde{\mathbf{W}} - \prod_{i \leq \ell' \leq j} \mathbf{D}^{(k-\ell'+1)} \mathbf{W} \right) \mathbf{v} \right\|_2 \leq \mathcal{O}(\rho^6 (N/m)^{2/3})$, *for all $1 \leq i \leq j \leq L$ and for a fixed vector $\mathbf{v}$.*

*Proof.* All the properties except 3, 5 and 6 follow from Lemma E.1 in [37].

3   The proof will follow the same technique as used for property 2. We give a brief overview here. We follow the same technique to expand the desired term

$$\mathbf{W}_{\mathcal{K}} \mathbf{h}^{(\ell)\prime} = \mathbf{D}_{\mathcal{K}} \mathbf{W} \mathbf{D}^{(\ell)\prime} (\mathbf{g}^{(\ell)} + \mathbf{g}^{(\ell)\prime}) + \mathbf{D}_{\mathcal{K}} \mathbf{W} \mathbf{D}^{(\ell)} \mathbf{g}^{(\ell)\prime}$$

We bound both the terms using the same technique with the following difference: we use property 9 of Lemma B.1 to bound the terms $\left\| \mathbf{D}_{\mathcal{K}} \mathbf{W} \mathbf{D}^{(\ell)\prime} \right\|$ and $\left\| \mathbf{D}_{\mathcal{K}} \mathbf{W} \mathbf{D}^{(\ell)} \mathbf{g}^{(\ell)\prime} \right\|$.

5   The proof follows from the proof of Lemma E.1(4) in [37]. In the proof of lemma E.1(4), the term that has been bounded is

$$\left\| \left( \prod_{i \leq \ell' \leq j} \widetilde{\mathbf{D}}^{(k-\ell'+1)} \widetilde{\mathbf{W}} - \prod_{i \leq \ell' \leq j} \mathbf{D}^{(k-\ell'+1)} \mathbf{W} \right) \mathbf{e}_k \right\|_2 \text{, where } k \in [m].$$

The important property of the vectors $\mathbf{e}_k$ that is used to bound the term above is the 1-sparsity of the vectors, which is necessary for using a property similar to property 9 of lemma B.1. However, we can show that the same bound holds for a fixed vector $\mathbf{v}$ by bounding the terms that contain $\mathbf{v}$ using property 10 of lemma B.1.

6   The proof will follow the same technique as used for property 5. We give a brief overview here.

For property 5, the term under consideration, $\left( \prod_{i \leq \ell' \leq j} \widetilde{\mathbf{D}}^{(k-\ell'+1)} \widetilde{\mathbf{W}} - \prod_{i \leq \ell' \leq j} \mathbf{D}^{(k-\ell'+1)} \mathbf{W} \right) \mathbf{v}$, was expanded into all the (exponentially many) difference terms, which were bounded separately. Denote the difference terms as $\mathbf{T}_1, \mathbf{T}_2, \cdots$.

For each term $\mathbf{T}_i$, the product $\mathbf{W}_{\mathcal{K}} \mathbf{T}_i$ can be written as a product of $\mathbf{D}_{\mathcal{K}} \mathbf{W} (\prod_{\ell_1 \leq \ell \leq \ell_2} \mathbf{D}^{(\ell)} \mathbf{W}) \overline{\mathbf{D}}$ and a term $\overline{\mathbf{T}}_i$, for some $i \leq \ell_1, \ell_2 \leq j$ and $\overline{\mathbf{D}}$ is either $\mathbf{D}_{\mathcal{K}}$ or $\mathbf{D}^{(\ell)} - \mathbf{D}^{(\ell)}_{(0)}$. The term $\overline{\mathbf{T}}_i$ will be bounded in a similar manner as has been done in the proof of property 5. However, the extra term that appears will be the bound of the norm of $\mathbf{D}_{\mathcal{K}} \mathbf{W} (\prod_{\ell_1 \leq \ell \leq \ell_2} \mathbf{D}^{(\ell)} \mathbf{W}) \overline{\mathbf{D}}$, which is bounded by $\mathcal{O}(\rho \sqrt{N/m})$ using property 9 of lemma B.1.

We will give an example for a term $\mathbf{T}_i$. Few terms will be of the form

$$(\prod_{\ell_1 \leq \ell \leq \ell_2} \mathbf{D}^{(\ell)} \mathbf{W}) \cdot \mathbf{D}^{(\ell_2)\prime} \mathbf{W} \cdot (\prod_{\ell_2 < \ell \leq \ell_3} \mathbf{D}^{(\ell)} \mathbf{W}) \mathbf{v},$$

for some $\ell_1, \ell_2, \ell_3$. We break its product with $\mathbf{W}_{\mathcal{K}}$ as

$$\mathbf{W}_{\mathcal{K}} \cdot (\prod_{\ell_1 \leq \ell \leq \ell_2} \mathbf{D}^{(\ell)} \mathbf{W}) \cdot \mathbf{D}^{(\ell_2)\prime} \mathbf{W} \cdot (\prod_{\ell_2 < \ell \leq \ell_3} \mathbf{D}^{(\ell)} \mathbf{W}) \mathbf{v}$$

$$= \underbrace{\left( \mathcal{D}_{\mathcal{K}} \mathbf{W} \cdot (\prod_{\ell_1 \leq \ell \leq \ell_2} \mathbf{D}^{(\ell)} \mathbf{W}) \cdot \mathbf{D}^{(\ell_2)\prime} \right)}_{\text{Term 1}} \cdot \underbrace{\left( \mathbf{D}^{(\ell_2)\prime} \cdot (\prod_{\ell_2 < \ell \leq \ell_3} \mathbf{D}^{(\ell)} \mathbf{W}) \mathbf{v} \right)}_{\text{Term 2}}.$$

Term 2 appears in the proof of property 5. Term 1 is the extra term that needs to be bounded and we can use property 9 of lemma B.1 to bound its norm by $\mathcal{O}(\rho\sqrt{N/m})$.

$\square$

# C  Invertibility at a single step

The section has been structured as follows: we first prove that a linear transformation of a random ReLU network can give back a linear function of the input in lemma C.1. We then explain why a simple application of the above lemma doesn't give a similar lemma for random RNNs which is, we need to make sure we break the correlations among input, the output vector and the weight matrices. We show that such correlations can be broken using the arguments in Claims C.6, C.8, C.7 and C.9. This then helps us to prove lemma C.3 using an application of lemma C.1.

## C.1  Invertibility of one layer ReLU networks

The following lemma is from a companion paper (Anonymous Authors); we reproduce its proof here for completeness.

**Lemma C.1.** *For any* $\mathbf{v} \in \mathbb{R}^d$*, the linear function taking* $\mathbf{x}$ *to* $\mathbf{v}^\top \mathbf{x}$ *for* $\mathbf{x} \in \mathbb{R}^d$*, can be represented as*

$$\mathbf{v}^\top \mathbf{x} = \int_{\mathbb{R}^d} p(\mathbf{w})\, \sigma(\mathbf{w}^\top \mathbf{x})\, \mathrm{d}\mu_d(\mathbf{w}), \tag{11}$$

*with*

$$p(\mathbf{w}) = 2\,\mathbf{w}^\top \mathbf{v}.$$

*Remark.* A similar statement can be gleaned from the proof of Proposition 4 of [46] which gives a similar representation except that $\mathbf{w}$ is uniformly distributed on $\mathbb{S}^{d-1}$ instead of being Gaussian. The proof there makes use of spherical harmonics and does not seem to immediately apply to the Gaussian case. The proof below is elementary and can be easily adapted to any spherically-symmetric distribution.

*Proof.* In the following, we re-parametrize $\mathbf{w}$ as $r\overline{\mathbf{w}}$ for some $r \geq 0, \overline{\mathbf{w}} \in \mathbb{S}^{d-1}$.

$$\frac{1}{2}\int_{\mathbb{R}^d} p(\mathbf{w})\,\sigma(\mathbf{w}^\top \mathbf{x})\,\mathrm{d}\mu_d(\mathbf{w}) = \int_{\mathbf{w}\in\mathbb{R}^d} \mathbf{v}^\top \mathbf{w}\left(\mathbf{w}^\top \mathbf{x}\right)\mathbb{I}_{\left(\mathbf{w}^\top \mathbf{x}\right)\geq 0}\,\mathrm{d}\mu(\mathbf{w})$$

$$= \mathbf{v}^\top \left(\int_{\mathbf{w}\in\mathbb{R}^d} \mathbf{w}\left(\mathbf{w}^\top \mathbf{x}\right)\mathbb{I}_{\left(\mathbf{w}^\top \mathbf{x}\right)\geq 0}\,\mathrm{d}\mu(\mathbf{w})\right)$$

$$= \mathbf{v}^\top \left(\frac{1}{(\sqrt{2\pi})^d}\int_{\overline{\mathbf{w}}\in\mathbb{S}^{d-1}}\int_{r=0}^{\infty} r\overline{\mathbf{w}}\left(r\overline{\mathbf{w}}^\top \mathbf{x}\right)\mathbb{I}_{\left(r\overline{\mathbf{w}}^\top \mathbf{x}\right)\geq 0}r^{d-1}e^{-r^2/2}\,\mathrm{d}r\,\mathrm{d}\overline{\mathbf{w}}\right)$$

$$= \left(\frac{1}{(\sqrt{2\pi})^d}\int_{r=0}^{\infty} r^{d+1}e^{-r^2/2}\,\mathrm{d}r\right)\mathbf{v}^\top \left(\int_{\overline{\mathbf{w}}\in\mathbb{S}^{d-1}}\overline{\mathbf{w}}\left(\overline{\mathbf{w}}^\top \mathbf{x}\right)\mathbb{I}_{\left(\overline{\mathbf{w}}^\top \mathbf{x}\right)\geq 0}\,\mathrm{d}\overline{\mathbf{w}}\right)$$

$$= \left(\frac{1}{(\sqrt{2\pi})^d}2^{d/2}\,\Gamma(d/2+1)\right)\mathbf{v}^\top \left(\int_{\overline{\mathbf{w}}\in\mathbb{S}^{d-1}}\overline{\mathbf{w}}\left(\overline{\mathbf{w}}^\top \mathbf{x}\right)\mathbb{I}_{\left(\overline{\mathbf{w}}^\top \mathbf{x}\right)\geq 0}\,\mathrm{d}\overline{\mathbf{w}}\right)$$

$$= \frac{d}{|\mathbb{S}^{d-1}|}\mathbf{v}^\top \left(\int_{\overline{\mathbf{w}}\in\mathbb{S}^{d-1}}\overline{\mathbf{w}}\,\overline{\mathbf{w}}^\top \mathbb{I}_{\left(\overline{\mathbf{w}}^\top \mathbf{x}\right)\geq 0}\,\mathrm{d}\overline{\mathbf{w}}\right)\mathbf{x}$$

$$= \frac{d}{|\mathbb{S}^{d-1}|}\mathbf{v}^\top \mathbf{C_x x}, \tag{12}$$

where $\mathbf{C_x} := \int_{\overline{\mathbf{w}} \in \mathbb{S}^{d-1}} \overline{\mathbf{w}} \overline{\mathbf{w}}^\top \mathbb{I}_{(\overline{\mathbf{w}}^\top \mathbf{x}) \geq 0} \, d\overline{\mathbf{w}}$. Let the orthogonal matrix $\mathbf{U_x}$ be such that $\mathbf{U_x}^\top \mathbf{x} = \mathbf{e}_1$ (the choice is not unique; we choose one arbitrarily). Then

$$
\begin{aligned}
\mathbf{C_x} &= \int_{\overline{\mathbf{w}} \in \mathbb{S}^{d-1}} \overline{\mathbf{w}} \overline{\mathbf{w}}^\top \mathbb{I}_{(\overline{\mathbf{w}}^\top \mathbf{x}) \geq 0} \, d\overline{\mathbf{w}} \\
&= \int_{\overline{\mathbf{w}} \in \mathbb{S}^{d-1}} \mathbf{U_x} \overline{\mathbf{w}} (\mathbf{U_x} \overline{\mathbf{w}})^\top \mathbb{I}_{(\mathbf{U_x} \overline{\mathbf{w}})^\top \mathbf{x} \geq 0} \, d\overline{\mathbf{w}} \\
&= \mathbf{U_x} \left( \int_{\overline{\mathbf{w}} \in \mathbb{S}^{d-1}} \overline{\mathbf{w}} \overline{\mathbf{w}}^\top \mathbb{I}_{\overline{\mathbf{w}}^\top (\mathbf{U_x}^\top \mathbf{x}) \geq 0} \, d\overline{\mathbf{w}} \right) \mathbf{U_x}^\top \\
&= \mathbf{U_x} \left( \int_{\overline{\mathbf{w}} \in \mathbb{S}^{d-1}} \overline{\mathbf{w}} \overline{\mathbf{w}}^\top \mathbb{I}_{\overline{w}_1 \geq 0} \, d\overline{\mathbf{w}} \right) \mathbf{U_x}^\top \\
&= \mathbf{U_x} \mathbf{C}_{\mathbf{e}_1} \mathbf{U_x}^\top.
\end{aligned}
\tag{13}
$$

Using the symmetry of $\mathbb{S}^{d-1}$ we claim

**Claim C.2.** *We have* $\mathbf{C}_{\mathbf{e}_1} = K_d \mathbf{I}$*, for a constant* $K_d$ *(evaluated below).*

*Proof.* Let's first restate the claim:

$$
[\mathbf{C}_{\mathbf{e}_1}]_{i,j} = \int_{\overline{\mathbf{w}} \in \mathbb{S}^{d-1}} \overline{w}_i \overline{w}_j \, \mathbb{I}_{\overline{w}_1 \geq 0} \, d\overline{\mathbf{w}} = \begin{cases} 0, & \text{if } i \neq j, \\ K_d, & \text{otherwise.} \end{cases}
$$

To prove this, note that

$$
\int_{\overline{\mathbf{w}} \in \mathbb{S}^{d-1}} \overline{w}_1^2 \, \mathbb{I}_{\overline{w}_1 \geq 0} \, d\overline{\mathbf{w}} = \frac{1}{2} \int_{\overline{\mathbf{w}} \in \mathbb{S}^{d-1}} \overline{w}_1^2 \, d\overline{\mathbf{w}},
$$

because $\overline{w}_1^2$ takes on the same value on $(\overline{w}_1, \overline{w}_2, \overline{w}_3, \ldots)$ and on $(-\overline{w}_1, \overline{w}_2, \overline{w}_3, \ldots)$. Similarly

$$
\int_{\overline{\mathbf{w}} \in \mathbb{S}^{d-1}} \overline{w}_2^2 \, \mathbb{I}_{\overline{w}_1 \geq 0} \, d\overline{\mathbf{w}} = \frac{1}{2} \int_{\overline{\mathbf{w}} \in \mathbb{S}^{d-1}} \overline{w}_2^2 \, d\overline{\mathbf{w}},
$$

because $\overline{w}_2^2$ takes on the same value on $(\overline{w}_1, \overline{w}_2, \overline{w}_3, \ldots)$ and on $(-\overline{w}_1, \overline{w}_2, \overline{w}_3, \ldots)$. Now clearly

$$
\int_{\overline{\mathbf{w}} \in \mathbb{S}^{d-1}} \overline{w}_1^2 \, d\overline{\mathbf{w}} = \int_{\overline{\mathbf{w}} \in \mathbb{S}^{d-1}} \overline{w}_2^2 \, d\overline{\mathbf{w}}.
$$

Thus we have shown that $\int_{\overline{\mathbf{w}} \in \mathbb{S}^{d-1}} \overline{w}_i^2 \, \mathbb{I}_{\overline{w}_1 \geq 0} \, d\overline{\mathbf{w}}$ does not depend on $i$. Now notice that

$$
\int_{\overline{\mathbf{w}} \in \mathbb{S}^{d-1}} \overline{w}_i \overline{w}_j \, \mathbb{I}_{\overline{w}_1 \geq 0} \, d\overline{\mathbf{w}} = 0
$$

because for each point $(\overline{w}_1, \overline{w}_2, \overline{w}_3, \ldots)$ there's a corresponding point $(\overline{w}_1, -\overline{w}_2, \overline{w}_3, \ldots)$ with the integrands taking on opposite values (or both are 0). A similar argument shows more generally that for all $i \neq j$ we have $\int_{\overline{\mathbf{w}} \in \mathbb{S}^{d-1}} \overline{w}_i \overline{w}_j \, \mathbb{I}_{\overline{w}_1 \geq 0} \, d\overline{\mathbf{w}} = 0$. $\qquad \square$

Now we evaluate $K_d$. Using Claim C.2, we can write

$$
K_d = \frac{1}{d} \operatorname{tr} \mathbf{C}_{\mathbf{e}_1} = \frac{1}{d} \int_{\overline{\mathbf{w}} \in \mathbb{S}^{d-1}} (\operatorname{tr} \overline{\mathbf{w}} \overline{\mathbf{w}}^\top) \mathbb{I}_{\overline{w}_1 \geq 0} \, d\overline{\mathbf{w}} = \frac{1}{d} \int_{\overline{\mathbf{w}} \in \mathbb{S}^{d-1}} \mathbb{I}_{\overline{w}_1 \geq 0} \, d\overline{\mathbf{w}} = \frac{|\mathbb{S}^{d-1}|}{2d}.
$$

Using the orthogonality of $\mathbf{U_x}$ and Claim C.2 in (13) it follows that

$$
\mathbf{C_x} = \mathbf{U_x} \mathbf{C}_{\mathbf{e}_1} \mathbf{U_x}^\top = K_d \mathbf{U_x} \mathbf{I} \mathbf{U_x}^\top = K_d \mathbf{I}.
$$

Continuing from where we left off in (12) we have

$$
\frac{1}{2} \int_{\mathbb{R}^d} p(\mathbf{w}) \, \sigma(\mathbf{w}^\top \mathbf{x}) \, d\mu_d(\mathbf{w}) = \frac{d}{|\mathbb{S}^{d-1}|} \mathbf{v}^\top \mathbf{C_x} \mathbf{x} = \frac{d K_d}{|\mathbb{S}^{d-1}|} \mathbf{v}^\top \mathbf{x} = \frac{1}{2} \mathbf{v}^\top \mathbf{x}.
$$

Thus,

$$
p(\mathbf{w}) = 2 \, \mathbf{w}^\top \mathbf{v}.
$$

$\qquad \square$

Lemma C.1 can be extended to gaussian distribution over $\mathbf{w}$ with variance $\beta\mathbf{I}$, for any $\beta > 0$.

**Corollary C.2.1.** *For any $\mathbf{v} \in \mathbb{R}^d$, the linear function taking $\mathbf{x}$ to $\mathbf{v}^\top\mathbf{x}$ for $\mathbf{x} \in \mathbb{R}^d$, can be represented as*

$$\frac{\beta^2}{2}\mathbf{v}^\top\mathbf{x} = \int_{\mathbb{R}^d} p(\mathbf{w})\,\sigma(\mathbf{w}^\top\mathbf{x})\,\mathrm{d}\mu_d^\beta(\mathbf{w}), \tag{14}$$

*with*

$$p(\mathbf{w}) = \mathbf{w}^\top\mathbf{v}.$$

Lemma C.1 can be discretized so that instead of the integral in (11), we use an empirical average. This comes at the expense of making the resulting version of (11) approximate. Furthermore, we can generalize the lemma so that instead of taking us from $\mathbf{h}^{(1)} = \sigma(\mathbf{W}\mathbf{x}^{(1)})$ to $\mathbf{v}^\top\mathbf{x}^{(1)}$ it takes us from $\mathbf{h}^{(\ell)}$ to $\mathbf{v}^\top[\mathbf{h}^{(\ell-1)}, \mathbf{x}^{(\ell)}]$ for every $\ell \in [L]$. The following lemma does both of these.

## C.2 Invertibility at a single step of RNN

**Lemma C.3.** *We have an RNN at random initialization as defined in Def. 3.2. Fix any $\ell \in \{0, 1, \ldots, L- 1\}$ and $\zeta \in (0, 1)$. Let $g : \mathbb{R}^{m+d} \times \mathbb{R}^{m+d} \to \mathbb{R}$ be given by $g(\mathbf{v}, [\mathbf{h}, \mathbf{x}]) = \mathbf{v}^\top[\mathbf{h}, \mathbf{x}]$. Consider a vector $\mathbf{v} \in \mathbb{R}^{m+d}$ which is stable against re-randomization, as specified later in Assumption 1 with constants $(\kappa, \zeta)$. Let $f\left(\mathbf{v}, \mathbf{h}^{(\ell-1)}, \mathbf{x}^{(\ell)}\right) = \sum_{i=1}^m u_i\sigma\left(\mathbf{w}_i^\top\mathbf{h}^{(\ell-1)} + \mathbf{a}_i^\top\mathbf{x}^{(\ell)}\right)$, where*

$$u_i = [\mathbf{w}_i, \mathbf{a}_i]^\top\mathbf{v}.$$

*Then for a given normalized sequence $\mathbf{x}^{(1)}, \cdots, \mathbf{x}^{(L)}$, with $\mathbf{x}^{(\ell)} \in \mathbb{R}^d$ for each $\ell \in [L]$, and for any constant $\rho > 0$, we have*

$$|g(\mathbf{v}, [\mathbf{h}^{(\ell-1)}, \mathbf{x}^{(\ell)}]) - f(\mathbf{v}, \mathbf{h}^{(\ell-1)}, \mathbf{x}^{(\ell)})| \le \mathcal{O}\left(\rho^{5+\kappa}m^{-1/12} + \rho^{1+\kappa}m^{-\zeta/2} + \rho^{1+\kappa}m^{-1/4} + \rho^{5+\kappa}m^{-1/4}\right)\|\mathbf{v}\|,$$

*with probability at least $1 - e^{-\Omega(\rho^2)}$.*

*Proof.* The major issue in using a discrete version of lemma C.1 directly for input $[\mathbf{h}^{(\ell-1)}, \mathbf{x}]$ is that there is a coupling between the randomness of the weights $\mathbf{W}, \mathbf{A}$ and the hidden vector $\mathbf{h}^{(\ell-1)}$. This coupling can be understood as the dependence of $\mathbf{h}^{(\ell-1)}$ on the choice of weight vectors in $\mathbf{W}$ and $\mathbf{A}$. There may also be a coupling between the randomness of $\mathbf{v}$ and the weights $\mathbf{W}, \mathbf{A}$, for which we take some assumption later. To decouple this randomness, we use the fact that ESNs are stable to re-randomization of few rows of the weight matrices and follow the proof technique of Lemma G.3 [37].

Choose a random subset $\mathcal{K} \subset [m]$ of size $|\mathcal{K}| = N$. Define the function $f_\mathcal{K}$ as

$$f_\mathcal{K}(\mathbf{v}, \mathbf{h}^{(\ell-1)}, \mathbf{x}^{(\ell)}) = \sum_{k\in\mathcal{K}} u_k\sigma(\mathbf{w}_k^\top\mathbf{h}^{(\ell-1)} + \mathbf{a}_k^\top\mathbf{x}^{(\ell)}).$$

Replace the rows $\{\mathbf{w}_k, \mathbf{a}_k\}_{k\in\mathcal{K}}$ of $\mathbf{W}$ and $\mathbf{A}$ with freshly new i.i.d. samples $\widetilde{\mathbf{w}}_k, \widetilde{\mathbf{a}}_k \sim \mathcal{N}\left(0, \frac{2}{m}\mathbf{I}\right)$. to form new matrices $\widetilde{\mathbf{W}}$ and $\widetilde{\mathbf{A}}$. For the given sequence, we follow the notation of Lemma B.2 to denote the hidden states corresponding to the old and the new weight matrices. We will assume one property for $\mathbf{v}$. Let say $\mathbf{v}$ depends on the matrices $\mathbf{W}$ and $\mathbf{A}$ and becomes $\widetilde{\mathbf{v}}$, with the new matrices $\widetilde{\mathbf{W}}$ and $\widetilde{\mathbf{A}}$. Then, we assume that the norm difference of $\mathbf{v}$ and $\widetilde{\mathbf{v}}$ is small with high probability.

**Assumption 1.** *With probability at least $1 - e^{-\Omega(\rho^2)}$, there exists constants $\kappa \ge 0$ and $\zeta < 1$ such that*

$$\left\|\mathbf{v} - \widetilde{\mathbf{v}}\right\| \le \mathcal{O}(\rho^\kappa(N/m)^\zeta\|\mathbf{v}\|)$$

$$\left\|[\mathbf{W}_\mathcal{K}, \mathbf{A}_\mathcal{K}]_r(\mathbf{v} - \widetilde{\mathbf{v}})\right\| \le \mathcal{O}(\rho^\kappa(N/m)^{0.5+\zeta}\|\mathbf{v}\|), \quad \forall k \in [m].$$

We will show later that the vector $\mathbf{v}$ that we need for inversion satisfies the above assumption with constants $(\kappa, \zeta) = (6, 1/6)$. Also, if $\mathbf{v}$ is independent of $\mathbf{W}$ and $\mathbf{A}$, then the constants needed in the assumption are $(\kappa, \zeta) = (0, 0)$.

The following claim shows that under the assumption 1, function $f_\mathcal{K}$ and $\frac{N}{m}g$ are close to each other with high probability.

**Claim C.4.** *For the given sequence* $\mathbf{x}^{(1)}, \cdots, \mathbf{x}^{(L)}$,

$$\left| f_{\mathcal{K}}(\mathbf{v}, \mathbf{h}^{(\ell-1)}, \mathbf{x}^{(\ell)}) - \frac{N}{m} g(\mathbf{v}, [\mathbf{h}^{(\ell-1)}, \mathbf{x}^{(\ell)}]) \right|$$
$$\leq \mathcal{O}\left( \rho^{5+\kappa} N^{7/6} m^{-7/6} + \rho^{1+\kappa}(N/m)^{1+\zeta} + \rho^{1+\kappa} N^{1/2} m^{-1} + \rho^{5+\kappa}(N/m)^{3/2} \right) \cdot \|\mathbf{v}\|,$$

*with probability exceeding* $1 - e^{-\Omega(\rho^2)}$.

The above claim has been restated and proven in claim C.5.

To complete the proof, we divide the set of neurons into $m/N$ disjoint sets $\mathcal{K}_1, \cdots, \mathcal{K}_{m/N}$, each set is of size $N$. We apply the Claim C.4 to each subset $\mathcal{K}_i$ and then add up the errors from each subset. That is, with probability at least $1 - \frac{m}{N} e^{-\Omega(\rho^2)}$,

$$f(\mathbf{h}^{(\ell-1)}, \mathbf{x}^{(\ell)}) = \sum_{i=1}^{m/N} f_{\mathcal{K}_i}(\mathbf{h}^{(\ell-1)}, \mathbf{x}^{(\ell)})$$
$$= \sum_{i=1}^{m/N} \frac{N}{m} g([\mathbf{h}^{(\ell-1)}, \mathbf{x}^{(\ell)}]) + error_{\mathcal{K}_i}$$
$$= g([\mathbf{h}^{(\ell-1)}, \mathbf{x}^{(\ell)}]) + \sum_{i=1}^{m/N} error_{\mathcal{K}_i},$$

where by Claim C.4,

$$|error_{\mathcal{K}_i}| \leq \mathcal{O}\left( \rho^{5+\kappa} N^{7/6} m^{-7/6} + \rho^{1+\kappa}(N/m)^{1+\zeta} + \rho^{1+\kappa} N^{1/2} m^{-1} + \rho^{5+\kappa}(N/m)^{3/2} \right) \cdot \|\mathbf{v}\|.$$

Thus,

$$\left| f(\mathbf{h}^{(\ell-1)}, \mathbf{x}^{(\ell)}) - g([\mathbf{h}^{(\ell-1)}, \mathbf{x}^{(\ell)}]) \right| \leq \mathcal{O}\left( \rho^{6+\kappa} N^{1/6} m^{-1/6} + \rho^{2+\kappa}(N/m)^{\zeta} + \rho^{1+\kappa} N^{-1/2} + \rho^{5+\kappa}(N/m)^{1/2} \right) \cdot \|\mathbf{v}\|,$$

with probability at least $1 - \frac{m}{N} e^{-\Omega(\rho^2)}$.

Choosing $N = m^{1/2}$, we have

$$\left| f(\mathbf{h}^{(\ell-1)}, \mathbf{x}^{(\ell)}) - g([\mathbf{h}^{(\ell-1)}, \mathbf{x}^{(\ell)}]) \right| \leq \mathcal{O}\left( \rho^{5+\kappa} m^{-1/12} + \rho^{1+\kappa} m^{-\zeta/2} + \rho^{1+\kappa} m^{-1/4} + \rho^{5+\kappa} m^{-1/4} \right) \cdot \|\mathbf{v}\|,$$

with probability at least $1 - \sqrt{m} e^{-\Omega(\rho^2)} \geq 1 - e^{-\Omega(\rho^2)}$. For Lemma B.2 to hold true, we need $N \leq \frac{m}{\rho^{23}}$. Hence, we require $\sqrt{m} \leq \frac{m}{\rho^{23}}$, which translates to $m \geq \rho^{46}$.

$\square$

## C.3   Proof of Claim C.4

**Claim C.5** (Restating claim C.4). *For the given sequence* $\mathbf{x}^{(1)}, \cdots, \mathbf{x}^{(L)}$,

$$\left| f_{\mathcal{K}}(\mathbf{v}, \mathbf{h}^{(\ell-1)}, \mathbf{x}^{(\ell)}) - \frac{N}{m} g(\mathbf{v}, [\mathbf{h}^{(\ell-1)}, \mathbf{x}^{(\ell)}]) \right|$$
$$\leq \mathcal{O}\left( \rho^{5+\kappa} N^{7/6} m^{-7/6} + \rho^{1+\kappa}(N/m)^{1+\zeta} + \rho^{1+\kappa} N^{1/2} m^{-1} + \rho^{5+\kappa}(N/m)^{3/2} \right) \cdot \|\mathbf{v}\|,$$

*with probability exceeding* $1 - e^{-\Omega(\rho^2)}$.

*Proof.* We will need $\{\mathbf{w}_k\}_{k \in \mathcal{K}}$ to satisfy several conditions. We will lower-bound the probability of each of these events and finally lower bound the probability of their intersection via the union bound. For the sake of clarity we will explicitly label these events $E_1, E_2, E_3,$ and $E_4$.

The following claim shows that since $\mathbf{h}^{(\ell-1)}$ doesn't change much with re-randomization (from lemma B.2), the function $f$ doesn't change much if we change the argument from $\mathbf{h}^{(\ell-1)}$ to $\widetilde{\mathbf{h}}^{(\ell-1)}$.

**Claim C.6.**

$$\left| f_\mathcal{K}(\mathbf{v}, \mathbf{h}^{(\ell-1)}, \mathbf{x}^{(\ell)}) - f_\mathcal{K}(\mathbf{v}, \widetilde{\mathbf{h}}^{(\ell-1)}, \mathbf{x}^{(\ell)}) \right| \leq \mathcal{O}\left( \rho^{5+\kappa} N^{7/6} m^{-7/6} \|\mathbf{v}\| \right),$$

*with probability at least* $1 - e^{-\Omega(\rho^2)}$.

*Proof.* We have,

$$\left| f_\mathcal{K}(\mathbf{v}, \mathbf{h}^{(\ell-1)}, \mathbf{x}^{(\ell)}) - f_\mathcal{K}(\mathbf{v}, \widetilde{\mathbf{h}}^{(\ell-1)}, \mathbf{x}^{(\ell)}) \right| = \left| \sum_{k \in \mathcal{K}} u_k \left( \sigma(\mathbf{w}_k^\top \mathbf{h}^{(\ell-1)} + \mathbf{a}_k^\top \mathbf{x}^{(\ell)}) - \sigma(\mathbf{w}_k^\top \widetilde{\mathbf{h}}^{(\ell-1)} + \mathbf{a}_k^\top \mathbf{x}^{(\ell)}) \right) \right|$$

$$= \left| \sum_{k \in \mathcal{K}} u_k \left( \sigma(\mathbf{w}_k^\top \mathbf{h}^{(\ell-1)} + \mathbf{a}_k^\top \mathbf{x}^{(\ell)}) - \sigma(\mathbf{w}_k^\top \widetilde{\mathbf{h}}^{(\ell-1)} + \mathbf{a}_k^\top \mathbf{x}^{(\ell)}) \right) \right|$$

$$\leq \|\mathbf{u}_\mathcal{K}\| \sqrt{\sum_{k \in \mathcal{K}} \left( \sigma(\mathbf{w}_k^\top \mathbf{h}^{(\ell-1)} + \mathbf{a}_k^\top \mathbf{x}^{(\ell)}) - \sigma(\mathbf{w}_k^\top \widetilde{\mathbf{h}}^{(\ell-1)} + \mathbf{a}_k^\top \mathbf{x}^{(\ell)}) \right)^2}$$

$$\leq \|\mathbf{u}_\mathcal{K}\| \sqrt{\sum_{k \in \mathcal{K}} \left( \left( \mathbf{w}_k^\top \mathbf{h}^{(\ell-1)} + \mathbf{a}_k^\top \mathbf{x}^{(\ell)} \right) - \left( \mathbf{w}_k^\top \widetilde{\mathbf{h}}^{(\ell-1)} - \mathbf{a}_k^\top \mathbf{x}^{(\ell)} \right) \right)^2}$$

$$= \|\mathbf{u}_\mathcal{K}\| \sqrt{\sum_{k \in \mathcal{K}} (\mathbf{w}_k^\top (\mathbf{h}^{(\ell-1)} - \widetilde{\mathbf{h}}^{(\ell-1)}))^2},$$

where we use cauchy schwartz inequality in the third step and 1-lipschitzness of the activation function ReLU in the pre-final step. We will bound the two factors above separately:

We have,

$$\|\mathbf{u}_\mathcal{K}\| = \left\| [\mathbf{W}_\mathcal{K}, \mathbf{A}_\mathcal{K}]_r \mathbf{v} \right\| = \left\| [\mathbf{W}_\mathcal{K}, \mathbf{A}_\mathcal{K}]_r \widetilde{\mathbf{v}} + [\mathbf{W}_\mathcal{K}, \mathbf{A}_\mathcal{K}]_r (\mathbf{v} - \widetilde{\mathbf{v}}) \right\| \leq \left\| [\mathbf{W}_\mathcal{K}, \mathbf{A}_\mathcal{K}]_r \widetilde{\mathbf{v}} \right\| + \left\| [\mathbf{W}_\mathcal{K}, \mathbf{A}_\mathcal{K}]_r \cdot (\mathbf{v} - \widetilde{\mathbf{v}}) \right\|. \tag{15}$$

We break $\mathbf{u}_\mathcal{K}$ into two terms, since we need to handle the correlation between $[\mathbf{W}_\mathcal{K}, \mathbf{A}_\mathcal{K}]_r$ and $\mathbf{v}$ and we will do that using assumption 1.

Since, $[\mathbf{W}_\mathcal{K}, \mathbf{A}_\mathcal{K}]_r$ and $\widetilde{\mathbf{v}}$ are independent, we can use the concentration inequality for chi-squared distributions (Fact A.3) to get

$$\left\| [\mathbf{W}_\mathcal{K}, \mathbf{A}_\mathcal{K}]_r \widetilde{\mathbf{v}} \right\| \leq \sqrt{\frac{2N}{m} + \frac{2\rho\sqrt{8N}}{m}} \|\widetilde{\mathbf{v}}\|,$$

with probability at least $1 - 2e^{-\rho^2}$. It can be further simplified into

$$\left\| [\mathbf{W}_\mathcal{K}, \mathbf{A}_\mathcal{K}]_r \widetilde{\mathbf{v}} \right\| \leq \sqrt{\frac{2N}{m}} \|\widetilde{\mathbf{v}}\| + \frac{2\rho\sqrt{2N}}{m} \|\widetilde{\mathbf{v}}\|,$$

using the fact that $\sqrt{1+y} \leq 1 + y/2$ for any variable $y > 0$. Also, from assumption 1, we have with probability $1 - e^{-\Omega(\rho^2)}$,

$$\left\| [\mathbf{W}_\mathcal{K}, \mathbf{A}_\mathcal{K}]_r (\mathbf{v} - \widetilde{\mathbf{v}}) \right\| \leq \mathcal{O}(\rho^\kappa (N/m)^{0.5+\zeta} \|\mathbf{v}\|).$$

Hence, with probability $1 - e^{-\Omega(\rho^2)}$,

$$\left\| [\mathbf{W}_\mathcal{K}, \mathbf{A}_\mathcal{K}]_r \mathbf{v} \right\| \leq \sqrt{\frac{2N}{m}} \|\widetilde{\mathbf{v}}\| + \frac{2\rho\sqrt{2N}}{m} \|\widetilde{\mathbf{v}}\| + \mathcal{O}(\rho^\kappa (N/m)^{0.5+\zeta} \|v\|).$$

Finally going back to eq. 15, with repeated utilization of assumption 1, we have

$$\left\| [\mathbf{W}_\mathcal{K}, \mathbf{A}_\mathcal{K}]_r \mathbf{v} \right\| \leq \sqrt{\frac{2N}{m}} \|\widetilde{\mathbf{v}}\| + \frac{2\rho\sqrt{2N}}{m} \|\widetilde{\mathbf{v}}\| + \mathcal{O}(\rho^\kappa (N/m)^{0.5+\zeta} \|\mathbf{v}\|)$$

$$\leq \sqrt{\frac{2N}{m}}\|\mathbf{v}\| + \frac{2\rho\sqrt{2N}}{m}\|\mathbf{v}\| + \sqrt{\frac{2N}{m}}\|\widetilde{\mathbf{v}} - \mathbf{v}\| + \frac{2\rho\sqrt{2N}}{m}\|\widetilde{\mathbf{v}} - \mathbf{v}\| + \mathcal{O}(\rho^\kappa(N/m)^{0.5+\zeta}\|\mathbf{v}\|)$$

$$\leq \sqrt{\frac{2N}{m}}\|\mathbf{v}\| + \frac{2\rho\sqrt{2N}}{m}\|\mathbf{v}\| + \sqrt{\frac{2N}{m}}\|\widetilde{\mathbf{v}} - \mathbf{v}\| + \frac{2\rho\sqrt{2N}}{m}\|\widetilde{\mathbf{v}} - \mathbf{v}\| + \mathcal{O}(\rho^\kappa(N/m)^{0.5+\zeta}\|\mathbf{v}\|)$$

$$\leq \sqrt{\frac{2N}{m}}\|\mathbf{v}\| + \frac{2\rho\sqrt{2N}}{m}\|\mathbf{v}\| + \sqrt{\frac{2N}{m}}\|\widetilde{\mathbf{v}} - \mathbf{v}\| + \frac{2\rho\sqrt{2N}}{m}\|\widetilde{\mathbf{v}} - \mathbf{v}\| + \mathcal{O}(\rho^\kappa(N/m)^{0.5+\zeta}\|\mathbf{v}\|)$$

$$\leq \sqrt{\frac{2N}{m}}\|\mathbf{v}\| + \frac{2\rho\sqrt{2N}}{m}\|\mathbf{v}\| + \sqrt{\frac{2N}{m}}\cdot\mathcal{O}(\rho^\kappa(N/m)^\zeta\|\mathbf{v}\|)$$
$$+ \frac{2\rho\sqrt{2N}}{m}\cdot\mathcal{O}(\rho^\kappa(N/m)^\zeta\|\mathbf{v}\|) + \mathcal{O}(\rho^\kappa(N/m)^{0.5+\zeta}\|\mathbf{v}\|)$$

$$\leq \mathcal{O}\left(\rho^\kappa\sqrt{N/m}\cdot\|\mathbf{v}\|\right),$$

giving us a bound on norm of $\mathbf{u}_\mathcal{K}$. Let us call this event $E_1$.

Lemma B.2 shows that with probability at least $1 - e^{-\Omega(\rho^2)}$,

$$\left\|\mathbf{h}^{(\ell-1)} - \widetilde{\mathbf{h}}^{(\ell-1)}\right\| \leq \mathcal{O}\left(\rho^5 N^{1/2}m^{-1/2}\right) \tag{16}$$

$$\left\|\mathbf{W}_\mathcal{K}\left(\mathbf{h}^{(\ell-1)} - \widetilde{\mathbf{h}}^{(\ell-1)}\right)\right\| \leq \mathcal{O}\left(\rho^5 N^{2/3}m^{-2/3}\right), \quad \forall k \in [m] \tag{17}$$

Let us call this event $E_2$.

Combining the two bounds we get

$$\left|f_\mathcal{K}(\mathbf{v}, \mathbf{h}^{(\ell-1)}, \mathbf{x}^{(\ell)}) - f_\mathcal{K}(\mathbf{v}, \widetilde{\mathbf{h}}^{(\ell-1)}, \mathbf{x}^{(\ell)})\right| \leq \|\mathbf{u}_\mathcal{K}\|\sqrt{\sum_{k\in\mathcal{K}}(\mathbf{w}_k^\top(\mathbf{h}^{(\ell-1)} - \widetilde{\mathbf{h}}^{(\ell-1)}))^2}$$

$$\leq \left(\rho^\kappa\sqrt{N/m}\right)\|\mathbf{v}\|\cdot\sqrt{\sum_{k\in\mathcal{K}}(\mathbf{w}_k^\top(\mathbf{h}^{(\ell-1)} - \widetilde{\mathbf{h}}^{(\ell-1)}))^2}$$

$$= \left(\rho^\kappa\sqrt{N/m}\right)\|\mathbf{v}\|\cdot\left\|\mathbf{W}_\mathcal{K}\left(\mathbf{h}^{(\ell-1)} - \widetilde{\mathbf{h}}^{(\ell-1)}\right)\right\|$$

$$\leq \left(\rho^\kappa\sqrt{N/m}\right)\|\mathbf{v}\|\cdot\mathcal{O}\left(\rho^5 N^{2/3}m^{-2/3}\right)$$

$$\leq \mathcal{O}\left(\rho^{5+\kappa}N^{7/6}m^{-7/6}\|\mathbf{v}\|\right),$$

with probability at least $\Pr[E_1 \cap E_2] \geq 1 - e^{-\Omega(\rho^2)} - 2e^{-\rho^2} \geq 1 - e^{-\Omega(\rho^2)}$. $\qquad\square$

The following claim shows that since $\mathbf{v}$ doesn't change much with re-randomization (from assumption 1), the function $f$ doesn't change much if we change the argument from $\mathbf{v}$ to $\widetilde{\mathbf{v}}$.

**Claim C.7.**

$$\left|f_\mathcal{K}(\mathbf{v}, \widetilde{\mathbf{h}}^{(\ell-1)}, \mathbf{x}^{(\ell)}) - f_\mathcal{K}(\widetilde{\mathbf{v}}, \widetilde{\mathbf{h}}^{(\ell-1)}, \mathbf{x}^{(\ell)})\right| \leq \mathcal{O}(\rho^{1+\kappa}(N/m)^{1+\zeta}\|\mathbf{v}\|),$$

*with probability at least* $1 - e^{-\Omega(\rho^2)}$.

*Proof.* Let $\mathbf{u} = [\mathbf{W}, \mathbf{A}]_r \widetilde{\mathbf{v}}$. We have,

$$
\left| f_{\mathcal{K}}(\mathbf{v}, \widetilde{\mathbf{h}}^{(\ell-1)}, \mathbf{x}^{(\ell)}) - f_{\mathcal{K}}(\widetilde{\mathbf{v}}, \widetilde{\mathbf{h}}^{(\ell-1)}, \mathbf{x}^{(\ell)}) \right| = \left| \sum_{k \in \mathcal{K}} (u_k - \widetilde{u}_k) \, \sigma(\mathbf{w}_k^\top \widetilde{\mathbf{h}}^{(\ell-1)} + \mathbf{a}_k^\top \mathbf{x}^{(\ell)}) \right|
$$

$$
\leq \sqrt{\sum_{k \in \mathcal{K}} \sigma(\mathbf{w}_k^\top \widetilde{\mathbf{h}}^{(\ell-1)} + \mathbf{a}_k^\top \mathbf{x}^{(\ell)})^2} \sqrt{\sum_{k \in \mathcal{K}} (u_k - \widetilde{u}_k)^2}
$$

$$
\leq \sqrt{\sum_{k \in \mathcal{K}} (\mathbf{w}_k^\top \widetilde{\mathbf{h}}^{(\ell-1)} + \mathbf{a}_k^\top \mathbf{x}^{(\ell)})^2} \sqrt{\sum_{k \in \mathcal{K}} (u_k - \widetilde{u}_k)^2}
$$

$$
= \sqrt{\sum_{k \in \mathcal{K}} (\mathbf{w}_k^\top \widetilde{\mathbf{h}}^{(\ell-1)} + \mathbf{a}_k^\top \mathbf{x}^{(\ell)})^2} \sqrt{\sum_{k \in \mathcal{K}} \langle [\mathbf{w}_k, \mathbf{a}_k], (\mathbf{v} - \widetilde{\mathbf{v}}) \rangle^2},
$$

where we use cauchy schwartz inequality in the second step and 1-lipschitzness of ReLU in the pre-final step.

We will bound the two factors above separately: Since, $[\mathbf{W}_{\mathcal{K}}, \mathbf{A}_{\mathcal{K}}]_r$ and $\widetilde{\mathbf{h}}$ are independent, we can use the concentration inequality for chi-squared distributions (Fact A.3) to get

$$
\sqrt{\sum_{k \in \mathcal{K}} (\mathbf{w}_k^\top \widetilde{\mathbf{h}}^{(\ell-1)} + \mathbf{a}_k^\top \mathbf{x}^{(\ell)})^2} \leq \sqrt{\frac{2N}{m} + \frac{2\rho\sqrt{8N}}{m}} \left\| [\widetilde{\mathbf{h}}^{(\ell-1)}, \mathbf{x}^{(\ell)}] \right\|,
$$

with probability at least $1 - 2e^{-\rho^2}$. It can be further simplified into

$$
\sqrt{\sum_{k \in \mathcal{K}} (\mathbf{w}_k^\top \widetilde{\mathbf{h}}^{(\ell-1)} + \mathbf{a}_k^\top \mathbf{x}^{(\ell)})^2} \leq \sqrt{\frac{2N}{m}} \left\| [\widetilde{\mathbf{h}}^{(\ell-1)}, \mathbf{x}^{(\ell)}] \right\| + \frac{2\rho\sqrt{2N}}{m} \left\| [\widetilde{\mathbf{h}}^{(\ell-1)}, \mathbf{x}^{(\ell)}] \right\|,
$$

using the fact that $\sqrt{1+y} \leq 1 + y/2$ for any variable $y > 0$. Let's call this event $E_3$.

We assume that our deep random neural network satisfies $\|\widetilde{\mathbf{h}}^{(\ell-1)}\| \in (\sqrt{2 + (\ell-2)\epsilon_x^2} - \frac{\rho^2}{\sqrt{m}}, \sqrt{2 + (\ell-2)\epsilon_x^2} + \frac{\rho^2}{\sqrt{m}})$ for all $\ell \in [L]$. This happens with probability at least $1 - e^{-\Omega(\rho^2)}$ w.r.t. the matrices $\widetilde{\mathbf{W}}$ and $\widetilde{\mathbf{A}}$ from Lemma B.1. Thus, provided $m \geq \Omega(\rho^4)$ and $\epsilon_x \leq \frac{1}{L}$, $\|\widetilde{\mathbf{h}}^{(\ell-1)}\| \in (\sqrt{2}, \sqrt{3})$. Let's call this event $E_4$. Also, since the sequence is assumed to be input normalized, $\left\| \mathbf{x}^{(\ell)} \right\| \leq 1$. Hence,

$$
\sqrt{\sum_{k \in \mathcal{K}} (\mathbf{w}_k^\top \widetilde{\mathbf{h}}^{(\ell-1)} + \mathbf{a}_k^\top \mathbf{x}^{(\ell)})^2} \leq 2\sqrt{\frac{2N}{m}} + \frac{4\rho\sqrt{2N}}{m},
$$

Again, from assumption 1, we have with probability $1 - e^{-\Omega(\rho^2)}$,

$$
\left\| [\mathbf{W}_{\mathcal{K}}, \mathbf{A}_{\mathcal{K}}]_r (\mathbf{v} - \widetilde{\mathbf{v}}) \right\| \leq \mathcal{O}(\rho^\kappa (N/m)^{0.5+\zeta} \|\mathbf{v}\|).
$$

Let us call this event $E_5$.

Combining the two bounds we get

$$
\left| f_{\mathcal{K}}(\mathbf{v}, \widetilde{\mathbf{h}}^{(\ell-1)}, \mathbf{x}^{(\ell)}) - f_{\mathcal{K}}(\widetilde{\mathbf{v}}, \widetilde{\mathbf{h}}^{(\ell-1)}, \mathbf{x}^{(\ell)}) \right| \leq \sqrt{\sum_{k \in \mathcal{K}} (\mathbf{w}_k^\top \widetilde{\mathbf{h}}^{(\ell-1)} + \mathbf{a}_k^\top \mathbf{x}^{(\ell)})^2} \sqrt{\sum_{k \in \mathcal{K}} \langle [\mathbf{w}_k, \mathbf{a}_k], (\mathbf{v} - \widetilde{\mathbf{v}}) \rangle^2}
$$

$$
\leq \left( 2\sqrt{\frac{2N}{m}} + \frac{4\rho\sqrt{2N}}{m} \right) \cdot \mathcal{O}(\rho^\kappa (N/m)^{0.5+\zeta} \|\mathbf{v}\|),
$$

$$
\leq \mathcal{O}(\rho^{1+\kappa} (N/m)^{1+\zeta} \|\mathbf{v}\|),
$$

with probability at least $\Pr[E_3 \cap E_4 \cap E_5] \geq 1 - 3e^{-\Omega(\rho^2)} \geq 1 - e^{-\Omega(\rho^2)}$. $\qquad \square$

The next claim shows that the functions $\frac{N}{m}g$ and $f$ are close to each other, using concentration bounds w.r.t. $\{\mathbf{w}_r\}_{r\in\mathcal{K}}$ and $\{\mathbf{a}_r\}_{r\in\mathcal{K}}$.

**Claim C.8.**

$$\left| f_{\mathcal{K}}(\widetilde{\mathbf{v}}, \widetilde{\mathbf{h}}^{(\ell-1)}, \mathbf{x}^{(\ell)}) - \frac{N}{m}g(\widetilde{\mathbf{v}}, [\widetilde{\mathbf{h}}^{(\ell-1)}, \mathbf{x}^{(\ell)}]) \right| \leq \mathcal{O}(\rho^{1+\kappa}N^{1/2}m^{-1}\|\mathbf{v}\|),$$

*with probability at least* $1 - e^{-\Omega(\rho^2)}$.

*Proof.* Since $\widetilde{\mathbf{v}}$ and $\widetilde{\mathbf{h}}^{(\ell-1)}$ doesn't depend on $\{\mathbf{w}_k\}_{k\in\mathcal{K}}$, we can use corollary C.2.1 directly to get

$$\mathbb{E}_{\{\mathbf{w}_k\}_{k\in\mathcal{K}}} f_{\mathcal{K}}(\widetilde{\mathbf{h}}^{(\ell-1)}, \mathbf{x}^{(\ell)}) = \frac{1}{2} \cdot \frac{2}{m} \cdot g([\widetilde{\mathbf{h}}^{(\ell-1)}, \mathbf{x}^{(\ell)}]) = \frac{1}{m}g([\widetilde{\mathbf{h}}^{(\ell-1)}, \mathbf{x}^{(\ell)}]).$$

Let $\overline{\widetilde{\mathbf{v}}} := \widetilde{\mathbf{v}}/\|\widetilde{\mathbf{v}}\|$ and $\overline{[\widetilde{\mathbf{h}}^{(\ell-1)}, \mathbf{x}^{(\ell)}]} = \frac{[\widetilde{\mathbf{h}}^{(\ell-1)}, \mathbf{x}^{(\ell)}]}{\|[\widetilde{\mathbf{h}}^{(\ell-1)}, \mathbf{x}^{(\ell)}]\|}$.

For the given sequence $\mathbf{x}^{(1)}, \cdots, \mathbf{x}^{(\ell)}$, we have

$$\left| \frac{1}{N}f_{\mathcal{K}}(\widetilde{\mathbf{v}}, \widetilde{\mathbf{h}}^{(\ell-1)}, \mathbf{x}^{(\ell)}) - \frac{1}{m}g(\widetilde{\mathbf{v}}, [\widetilde{\mathbf{h}}^{(\ell-1)}, \mathbf{x}^{(\ell)}]) \right|$$

$$= \left| \frac{1}{N}\sum_{k\in\mathcal{K}} [\mathbf{w}_k, \mathbf{a}_k]^\top \widetilde{\mathbf{v}}\sigma(\mathbf{w}_k^\top \widetilde{\mathbf{h}}^{(\ell-1)} + \mathbf{a}_k^\top \mathbf{x}^{(\ell)}) - \mathbb{E}_{\mathbf{w}\sim\mathcal{N}(0,\frac{2}{m}\mathbf{I}_{m+d})} \mathbf{w}^\top \widetilde{\mathbf{v}}\sigma(\mathbf{w}^\top [\widetilde{\mathbf{h}}^{(\ell-1)}, \mathbf{x}^{(\ell)}]) \right|$$

$$= \left| \widetilde{\mathbf{v}}^\top \left( \frac{1}{N}\sum_{k\in\mathcal{K}} [\mathbf{w}_k, \mathbf{a}_k][\mathbf{w}_k, \mathbf{a}_k]^\top \mathbb{I}_{[\mathbf{w}_k,\mathbf{a}_k]^\top \widetilde{\mathbf{h}}^{(\ell-1)}, \mathbf{x}^{(\ell)}]\geq 0} - \mathbb{E}_{\mathbf{w}\sim\mathcal{N}(0,\frac{2}{m}\mathbf{I}_{m+d})} \mathbf{w}\mathbf{w}^\top \mathbb{I}_{\mathbf{w}^\top [\widetilde{\mathbf{h}}^{(\ell-1)}, \mathbf{x}^{(\ell)}]\geq 0} \right) [\widetilde{\mathbf{h}}^{(\ell-1)}, \mathbf{x}^{(\ell)}] \right|$$

$$= \|\widetilde{\mathbf{v}}\| \|[\widetilde{\mathbf{h}}^{(\ell-1)}, \mathbf{x}^{(\ell)}]\| \left| \overline{\widetilde{\mathbf{v}}}^\top \left( \frac{1}{N}\sum_{k\in\mathcal{K}} [\mathbf{w}_k, \mathbf{a}_k][\mathbf{w}_k, \mathbf{a}_k]^\top \mathbb{I}_{[\mathbf{w}_k,\mathbf{a}_k]^\top \overline{[\widetilde{\mathbf{h}}^{(\ell-1)}, \mathbf{x}^{(\ell)}]}\geq 0} \right.\right.$$

$$\left.\left. - \mathbb{E}_{\mathbf{w}\sim\mathcal{N}(0,\frac{2}{m}\mathbf{I}_{m+d})} \mathbf{w}\mathbf{w}^\top \mathbb{I}_{\mathbf{w}^\top \overline{[\widetilde{\mathbf{h}}^{(\ell-1)}, \mathbf{x}^{(\ell)}]}\geq 0} \right) \overline{[\widetilde{\mathbf{h}}^{(\ell-1)}, \mathbf{x}^{(\ell)}]} \right|$$

$$\leq \frac{\|\widetilde{\mathbf{v}}\| \|[\widetilde{\mathbf{h}}^{(\ell-1)}, \mathbf{x}^{(\ell)}]\|}{2} \left| \frac{1}{N}\sum_{k\in\mathcal{K}} \left( \left( \overline{[\widetilde{\mathbf{h}}^{(\ell-1)}, \mathbf{x}^{(\ell)}]} + \overline{\widetilde{\mathbf{v}}} \right)^\top [\mathbf{w}_k, \mathbf{a}_k]\mathbb{I}_{[\mathbf{w}_k,\mathbf{a}_k]^\top \overline{[\widetilde{\mathbf{h}}^{(\ell-1)}, \mathbf{x}^{(\ell)}]}\geq 0} \right)^2 \right.$$

$$\left. - \mathbb{E}_{\mathbf{w}\sim\mathcal{N}(0,\frac{2}{m}\mathbf{I})} \left( \left( \overline{[\widetilde{\mathbf{h}}^{(\ell-1)}, \mathbf{x}^{(\ell)}]} + \overline{\widetilde{\mathbf{v}}} \right)^\top \mathbf{w}\mathbb{I}_{\mathbf{w}^\top \overline{[\widetilde{\mathbf{h}}^{(\ell-1)}, \mathbf{x}^{(\ell)}]}\geq 0} \right)^2 \right| \tag{18}$$

$$+ \frac{\|\widetilde{\mathbf{v}}\| \|[\widetilde{\mathbf{h}}^{(\ell-1)}, \mathbf{x}^{(\ell)}]\|}{2} \left| \frac{1}{N}\sum_{k\in\mathcal{K}} \left( \overline{[\widetilde{\mathbf{h}}^{(\ell-1)}, \mathbf{x}^{(\ell)}]}^\top [\mathbf{w}_k, \mathbf{a}_k]\mathbb{I}_{[\mathbf{w}_k,\mathbf{a}_k]^\top \overline{[\widetilde{\mathbf{h}}^{(\ell-1)}, \mathbf{x}^{(\ell)}]}\geq 0} \right)^2 \right.$$

$$\left. - \mathbb{E}_{\mathbf{w}\sim\mathcal{N}(0,\frac{2}{m}\mathbf{I})} \left( \overline{[\widetilde{\mathbf{h}}^{(\ell-1)}, \mathbf{x}^{(\ell)}]}^\top \mathbf{w}\mathbb{I}_{\mathbf{w}^\top \overline{[\widetilde{\mathbf{h}}^{(\ell-1)}, \mathbf{x}^{(\ell)}]}\geq 0} \right)^2 \right| \tag{19}$$

$$+ \frac{\|\widetilde{\mathbf{v}}\| \|[\widetilde{\mathbf{h}}^{(\ell-1)}, \mathbf{x}^{(\ell)}]\|}{2} \left| \frac{1}{N}\sum_{k\in\mathcal{K}} \left( \overline{\widetilde{\mathbf{v}}}^\top [\mathbf{w}_k, \mathbf{a}_k]\mathbb{I}_{[\mathbf{w}_k,\mathbf{a}_k]^\top \overline{[\widetilde{\mathbf{h}}^{(\ell-1)}, \mathbf{x}^{(\ell)}]}\geq 0} \right)^2 - \mathbb{E}_{\mathbf{w}\sim\mathcal{N}(0,\frac{2}{m}\mathbf{I})} \left( \overline{\widetilde{\mathbf{v}}}^\top \mathbf{w}\mathbb{I}_{\mathbf{w}^\top \overline{[\widetilde{\mathbf{h}}^{(\ell-1)}, \mathbf{x}^{(\ell)}]}\geq 0} \right)^2 \right| \tag{20}$$

The three terms above in (18), (19) and (20) correspond to the large deviation bounds for random variables $\left( \overline{[\widetilde{\mathbf{h}}^{(\ell-1)}, \mathbf{x}^{(\ell)}]} + \overline{\widetilde{\mathbf{v}}} \right)^\top [\mathbf{w}_k, \mathbf{a}_k]\mathbb{I}_{[\mathbf{w}_k,\mathbf{a}_k]^\top \overline{[\widetilde{\mathbf{h}}^{(\ell-1)}, \mathbf{x}^{(\ell)}]}\geq 0}$, $\overline{[\widetilde{\mathbf{h}}^{(\ell-1)}, \mathbf{x}^{(\ell)}]}^\top [\mathbf{w}_k, \mathbf{a}_k]\mathbb{I}_{[\mathbf{w}_k,\mathbf{a}_k]^\top \overline{[\widetilde{\mathbf{h}}^{(\ell-1)}, \mathbf{x}^{(\ell)}]}\geq 0}$ and $\overline{\widetilde{\mathbf{v}}}^\top [\mathbf{w}_k, \mathbf{a}_k]\mathbb{I}_{[\mathbf{w}_k,\mathbf{a}_k]^\top \overline{[\widetilde{\mathbf{h}}^{(\ell-1)}, \mathbf{x}^{(\ell)}]}\geq 0}$ respectively. Each of these random variables is sub-exponential as it is bounded above by a squared Gaussian random variable using the fact that $\mathbf{w} \sim \mathcal{N}\left(0, \frac{2}{m}\mathbf{I}\right)$:

- $((\overline{[\widetilde{\mathbf{h}}^{(\ell-1)}, \mathbf{x}^{(\ell)}]} + \overline{\widetilde{\mathbf{v}}})^\top [\mathbf{w}_k, \mathbf{a}_k] \mathbb{I}_{[\mathbf{w}_k, \mathbf{a}_k]^\top \overline{[\widetilde{\mathbf{h}}^{(\ell-1)}, \mathbf{x}^{(\ell)}]} \geq 0})^2 \leq ((\overline{[\widetilde{\mathbf{h}}^{(\ell-1)}, \mathbf{x}^{(\ell)}]} + \overline{\widetilde{\mathbf{v}}})^\top [\mathbf{w}_k, \mathbf{a}_k])^2$
  and $(\overline{[\widetilde{\mathbf{h}}^{(\ell-1)}, \mathbf{x}^{(\ell)}]} + \overline{\widetilde{\mathbf{v}}})^\top \mathbf{w} \sim N(0, \frac{2}{m} \|\overline{[\widetilde{\mathbf{h}}^{(\ell-1)}, \mathbf{x}^{(\ell)}]} + \overline{\widetilde{\mathbf{v}}}\|^2)$,

- $((\overline{[\widetilde{\mathbf{h}}^{(\ell-1)}, \mathbf{x}^{(\ell)}]})^\top [\mathbf{w}_k, \mathbf{a}_k] \mathbb{I}_{[\mathbf{w}_k, \mathbf{a}_k]^\top \overline{[\widetilde{\mathbf{h}}^{(\ell-1)}, \mathbf{x}^{(\ell)}]} \geq 0})^2 \leq ((\overline{[\widetilde{\mathbf{h}}^{(\ell-1)}, \mathbf{x}^{(\ell)}]})^\top [\mathbf{w}_k, \mathbf{a}_k])^2$ and
  $(\overline{[\widetilde{\mathbf{h}}^{(\ell-1)}, \mathbf{x}^{(\ell)}]})^\top \mathbf{w} \sim N(0, \frac{2}{m} \|\overline{[\widetilde{\mathbf{h}}^{(\ell-1)}, \mathbf{x}^{(\ell)}]}\|^2)$

- $((\overline{\widetilde{\mathbf{v}}})^\top [\mathbf{w}_k, \mathbf{a}_k] \mathbb{I}_{[\mathbf{w}_k, \mathbf{a}_k]^\top \overline{[\widetilde{\mathbf{h}}^{(\ell-1)}, \mathbf{x}^{(\ell)}]} \geq 0})^2 \leq ((\overline{\widetilde{\mathbf{v}}})^\top [\mathbf{w}_k, \mathbf{a}_k])^2$ and $(\overline{\widetilde{\mathbf{v}}})^\top \mathbf{w} \sim N(0, \frac{2}{m} \|\overline{\widetilde{\mathbf{v}}}\|^2)$.

For typographical convenience, denote the expressions in (18), (19) and (20) by $P(\{\mathbf{x}^{(\ell)}\}), Q(\{\mathbf{x}^{(\ell)}\})$ and $R(\{\mathbf{x}^{(\ell)}\})$, respectively. We assume that our deep random neural network satisfies $\|\widetilde{\mathbf{h}}^{(\ell-1)}\| \in \left( \sqrt{2 + (\ell - 2)\epsilon_x^2} - \frac{\rho^2}{\sqrt{m}}, \sqrt{2 + (\ell - 2)\epsilon_x^2} + \frac{\rho^2}{\sqrt{m}} \right)$ for all $\ell \in [L]$, with probability at least $1 - e^{-\Omega(\rho^2)}$. Thus, provided $m \geq \Omega(\rho^4)$ and $\epsilon_x \leq \frac{1}{L}$, $\|\widetilde{\mathbf{h}}^{(\ell-1)}\| \in \left( \sqrt{2}, \sqrt{3} \right)$. This event was taken care before in event $E_4$. Using this and concentration of chi-squared random variables (Fact A.3) we get

- $\Pr\left[ \frac{2P(\{\mathbf{x}^{(\ell)}\})}{\|\widetilde{\mathbf{v}}\| \|[\widetilde{\mathbf{h}}^{(\ell-1)}, \mathbf{x}^{(\ell)}]\|} \leq \frac{2\sqrt{2}\rho}{\sqrt{N}} \cdot \frac{2}{m} \|\overline{[\widetilde{\mathbf{h}}^{(\ell-1)}, \mathbf{x}^{(\ell)}]} + \overline{\widetilde{\mathbf{v}}}\|^2 \right] \leq e^{-\rho^2}$,

- $\Pr\left[ \frac{2Q(\{\mathbf{x}^{(\ell)}\})}{\|\widetilde{\mathbf{v}}\| \|[\widetilde{\mathbf{h}}^{(\ell-1)}, \mathbf{x}^{(\ell)}]\|} \leq \frac{2\sqrt{2}\rho}{\sqrt{N}} \cdot \frac{2}{m} \|\overline{[\widetilde{\mathbf{h}}^{(\ell-1)}, \mathbf{x}^{(\ell)}]}\|^2 \right] \leq e^{-\rho^2}$,

- $\Pr\left[ \frac{2R(\{\mathbf{x}^{(\ell)}\})}{\|\widetilde{\mathbf{v}}\| \|[\widetilde{\mathbf{h}}^{(\ell-1)}, \mathbf{x}^{(\ell)}]\|} \leq \frac{2\sqrt{2}\rho}{\sqrt{N}} \cdot \frac{2}{m} \|\overline{\widetilde{\mathbf{v}}}\|^2 \right] \leq e^{-\rho^2}$.

Define the following event

$$
\begin{aligned}
E_6(\{\mathbf{x}^{(\ell)}\}) := & \left( \frac{2P(\{\mathbf{x}^{(\ell)}\})}{\|\widetilde{\mathbf{v}}\| \|[\widetilde{\mathbf{h}}^{(\ell-1)}, \mathbf{x}^{(\ell)}]\|} \leq \frac{2\sqrt{2}\rho}{\sqrt{N}} \cdot \frac{2}{m} \|\overline{[\widetilde{\mathbf{h}}^{(\ell-1)}, \mathbf{x}^{(\ell)}]} + \overline{\widetilde{\mathbf{v}}}\|^2 \right) \\
& \cap \left( \frac{2Q(\{\mathbf{x}^{(\ell)}\})}{\|\widetilde{\mathbf{v}}\| \|[\widetilde{\mathbf{h}}^{(\ell-1)}, \mathbf{x}^{(\ell)}]\|} \leq \frac{2\sqrt{2}\rho}{\sqrt{N}} \cdot \frac{2}{m} \|\overline{[\widetilde{\mathbf{h}}^{(\ell-1)}, \mathbf{x}^{(\ell)}]}\|^2 \right) \\
& \cap \left( \frac{2R(\{\mathbf{x}^{(\ell)}\})}{\|\widetilde{\mathbf{v}}\| \|[\widetilde{\mathbf{h}}^{(\ell-1)}, \mathbf{x}^{(\ell)}]\|} \leq \frac{2\sqrt{2}\rho}{\sqrt{N}} \cdot \frac{2}{m} \|\overline{\widetilde{\mathbf{v}}}\|^2 \right)
\end{aligned}
$$

We have thus shown that for the given sequence $\mathbf{x}^{(1)}, \mathbf{x}^{(2)}, \cdots, \mathbf{x}^{(L)}$ with probability at least $1 - 3e^{-\rho^2} - e^{-\Omega(\rho^2)}$ the event $E_6(\mathbf{x}) \cap E_3$ occurs, which implies

$$
\begin{aligned}
& |\frac{1}{N} f_{\mathcal{K}}(\widetilde{\mathbf{v}}, \widetilde{\mathbf{h}}^{(\ell-1)}, \mathbf{x}^{(\ell)}) - \frac{1}{m} g([\widetilde{\mathbf{v}}, \widetilde{\mathbf{h}}^{(\ell-1)}, \mathbf{x}^{(\ell)}])| \\
& \qquad \leq \frac{\|\widetilde{\mathbf{v}}\| \|[\widetilde{\mathbf{h}}^{(\ell-1)}, \mathbf{x}^{(\ell)}]\|}{2} \cdot \frac{2\sqrt{2}\rho}{\sqrt{N}} \cdot \frac{2}{m} (\|\overline{[\widetilde{\mathbf{h}}^{(\ell-1)}, \mathbf{x}^{(\ell)}]} + \overline{\widetilde{\mathbf{v}}}\|^2 + \|\overline{[\widetilde{\mathbf{h}}^{(\ell-1)}, \mathbf{x}^{(\ell)}]}\|^2 + \|\overline{\widetilde{\mathbf{v}}}\|^2) \\
& \qquad \qquad \qquad \qquad \qquad \qquad \qquad \qquad \qquad \qquad \qquad \qquad \qquad \qquad (21) \\
& \qquad \leq \frac{16\sqrt{2}\rho}{\sqrt{N}m} \|\widetilde{\mathbf{v}}\| \|[\widetilde{\mathbf{h}}^{(\ell-1)}, \mathbf{x}^{(\ell)}]\| \leq \frac{32\sqrt{2}\rho}{\sqrt{N}m} \|\widetilde{\mathbf{v}}\|.
\end{aligned}
$$

Hence with probabilty $1 - e^{-\Omega(\rho^2)}$,

$$|f_{\mathcal{K}}(\widetilde{\mathbf{v}}, \widetilde{\mathbf{h}}^{(\ell-1)}, \mathbf{x}^{(\ell)}) - \frac{N}{m} g(\widetilde{\mathbf{v}}, [\widetilde{\mathbf{h}}^{(\ell-1)}, \mathbf{x}^{(\ell)}])| \leq \frac{32\sqrt{2N}\rho}{m} \|\widetilde{\mathbf{v}}\|.$$

We further use assumption 1 to get

$$
\begin{aligned}
|f_{\mathcal{K}}(\widetilde{\mathbf{v}}, \widetilde{\mathbf{h}}^{(\ell-1)}, \mathbf{x}^{(\ell)}) - \frac{N}{m} g(\widetilde{\mathbf{v}}, [\widetilde{\mathbf{h}}^{(\ell-1)}, \mathbf{x}^{(\ell)}])| &\leq \frac{32\sqrt{2}\rho}{\sqrt{N}m} \|\widetilde{\mathbf{v}}\| \\
&\leq \frac{32\sqrt{2}\rho\sqrt{N}}{m} \left( \|\mathbf{v}\| + \|\mathbf{v} - \widetilde{\mathbf{v}}\| \right) \\
&\leq \frac{32\sqrt{2}\sqrt{N}\rho}{m} \left( \|\mathbf{v}\| + \mathcal{O}\left( \rho^{\kappa}(N/m)^{\zeta} \right) \|\mathbf{v}\| \right) \\
&\leq \mathcal{O}(\rho^{1+\kappa} N^{1/2} m^{-1} \|\mathbf{v}\|),
\end{aligned}
$$

with probability exceeding $1 - e^{-\Omega(\rho^2)}$. $\qquad\qquad\qquad\qquad\qquad\qquad\qquad\qquad\square$

The following claim again uses the property that $\mathbf{v}$ and $\mathbf{h}^{(\ell-1)}$ doesn't change much with re-randomization to show that function $g$ is also stable to re-randomization.

**Claim C.9.**

$$\left| \frac{N}{m} g([\widetilde{\mathbf{v}}, \widetilde{\mathbf{h}}^{(\ell-1)}, \mathbf{x}^{(\ell)}]) - \frac{N}{m} g(\mathbf{v}, [\mathbf{h}^{(\ell-1)}, \mathbf{x}^{(\ell)}]) \right| \leq \mathcal{O}(\rho^{5+\kappa}(N/m)^{3/2}) \cdot \|\mathbf{v}\|,$$

*with probability exceeding $1 - e^{-\Omega(\rho^2)}$.*

*Proof.*

$$
\begin{aligned}
&\left| g(\widetilde{\mathbf{v}}, [\widetilde{\mathbf{h}}^{(\ell-1)}, \mathbf{x}^{(\ell)}]) - g(\mathbf{v}, [\mathbf{h}^{(\ell-1)}, \mathbf{x}^{(\ell)}]) \right| \\
&= \left| \mathbf{v}^{\top}[\mathbf{h}^{(\ell-1)}, \mathbf{x}^{(\ell)}] - \widetilde{\mathbf{v}}^{\top}[\widetilde{\mathbf{h}}^{(\ell-1)}, \mathbf{x}^{(\ell)}] \right| \\
&= \left| \mathbf{v}^{\top}[\mathbf{h}^{(\ell-1)}, \mathbf{x}^{(\ell)}] - \widetilde{\mathbf{v}}^{\top}[\mathbf{h}^{(\ell-1)}, \mathbf{x}^{(\ell)}] + \widetilde{\mathbf{v}}^{\top}[\widetilde{\mathbf{h}}^{(\ell-1)}, \mathbf{x}^{(\ell)}] - \widetilde{\mathbf{v}}^{\top}[\mathbf{h}^{(\ell-1)}, \mathbf{x}^{(\ell)}] \right| \\
&\leq \left| (\mathbf{v} - \widetilde{\mathbf{v}})^{\top}[\mathbf{h}^{(\ell-1)}, \mathbf{x}^{(\ell)}] - \widetilde{\mathbf{v}}^{\top}[\mathbf{h}^{(\ell-1)}, \mathbf{x}^{(\ell)}] \right| + \left| \widetilde{\mathbf{v}}^{\top}[\widetilde{\mathbf{h}}^{(\ell-1)}, \mathbf{x}^{(\ell)}] - \widetilde{\mathbf{v}}^{\top}[\mathbf{h}^{(\ell-1)}, \mathbf{x}^{(\ell)}] \right| \\
&\leq \|\mathbf{v} - \widetilde{\mathbf{v}}\| \left\| [\mathbf{h}^{(\ell-1)}, \mathbf{x}^{(\ell)}] \right\| + \|\widetilde{\mathbf{v}}\| \left\| \widetilde{\mathbf{h}}^{(\ell-1)} - \mathbf{h}^{(\ell-1)} \right\| \\
&\leq \|\mathbf{v} - \widetilde{\mathbf{v}}\| \left\| [\mathbf{h}^{(\ell-1)}, \mathbf{x}^{(\ell)}] \right\| + \|\widetilde{\mathbf{v}} - \mathbf{v}\| \left\| \widetilde{\mathbf{h}}^{(\ell-1)} - \mathbf{h}^{(\ell-1)} \right\| + \|\mathbf{v}\| \left\| \widetilde{\mathbf{h}}^{(\ell-1)} - \mathbf{h}^{(\ell-1)} \right\|.
\end{aligned}
$$

We assume that our deep random neural network satisfies $\|\mathbf{h}^{(\ell-1)}\| \in (\sqrt{2 + (\ell-2)\epsilon_x^2} - \frac{\rho^2}{\sqrt{m}}, \sqrt{2 + (\ell-2)\epsilon_x^2} + \frac{\rho^2}{\sqrt{m}})$ for all $\ell \in [L]$. This happens with probability at least $1 - e^{-\Omega(\rho^2)}$ w.r.t. the matrices $\mathbf{W}$ and $\mathbf{A}$ from Lemma B.1. Thus, provided $m \geq \Omega(\rho^4)$ and $\epsilon_x \leq \frac{1}{L}$, $\|\mathbf{h}^{(\ell-1)}\| \in \left( \sqrt{2}, \sqrt{3} \right)$. Let's call this event $E_7$. Using assumption 1, event $E_2$ and event $E_7$, we have with probability $1 - e^{-\Omega(\rho^2)}$,

$$
\begin{aligned}
&\left| g(\widetilde{\mathbf{v}}, [\widetilde{\mathbf{h}}^{(\ell-1)}, \mathbf{x}^{(\ell)}]) - g(\mathbf{v}, [\mathbf{h}^{(\ell-1)}, \mathbf{x}^{(\ell)}]) \right| \\
&\leq \|\mathbf{v} - \widetilde{\mathbf{v}}\| \left\| [\mathbf{h}^{(\ell-1)}, \mathbf{x}^{(\ell)}] \right\| + \|\widetilde{\mathbf{v}} - \mathbf{v}\| \left\| \widetilde{\mathbf{h}}^{(\ell-1)} - \mathbf{h}^{(\ell-1)} \right\| + \|\mathbf{v}\| \left\| \widetilde{\mathbf{h}}^{(\ell-1)} - \mathbf{h}^{(\ell-1)} \right\| \\
&\leq \mathcal{O}(\rho^{\kappa}(N/m)^{\zeta}(2 + \mathcal{O}(\rho^5 (N/m)^{1/2}))) \cdot \|\mathbf{v}\| + \mathcal{O}(\rho^5 (N/m)^{1/2}) \cdot \|\mathbf{v}\| \\
&\leq \mathcal{O}(\rho^{5+\kappa}(N/m)^{1/2}) \cdot \|\mathbf{v}\|.
\end{aligned}
$$

$\qquad\qquad\qquad\qquad\qquad\qquad\qquad\qquad\qquad\qquad\qquad\qquad\qquad\qquad\qquad\qquad\qquad\square$

Claims C.6, C.8, C.7 and C.9 hold if the event $E_1 \cap E_2 \cap E_3 \cap E_4 \cap E_5 \cap E_6 \cap E_7$ occurs. This has probability at least $1 - e^{-\Omega(\rho^2)}$. Thus, we have

$$\left| f_{\mathcal{K}}(\mathbf{v}, \mathbf{h}^{(\ell-1)}, \mathbf{x}^{(\ell)}) - \frac{N}{m} g(\mathbf{v}, [\mathbf{h}^{(\ell-1)}, \mathbf{x}^{(\ell)}]) \right|$$

$$\leq \left| f_{\mathcal{K}}(\mathbf{v}, \mathbf{h}^{(\ell-1)}, \mathbf{x}^{(\ell)}) - f_{\mathcal{K}}(\mathbf{v}, \widetilde{\mathbf{h}}^{(\ell-1)}, \mathbf{x}^{(\ell)}) \right| + \left| f_{\mathcal{K}}(\mathbf{v}, \widetilde{\mathbf{h}}^{(\ell-1)}, \mathbf{x}^{(\ell)}) - f_{\mathcal{K}}(\widetilde{\mathbf{v}}, \widetilde{\mathbf{h}}^{(\ell-1)}, \mathbf{x}^{(\ell)}) \right|$$

$$+ \left| f_{\mathcal{K}}(\widetilde{\mathbf{v}}, \widetilde{\mathbf{h}}^{(\ell-1)}, \mathbf{x}^{(\ell)}) - \frac{N}{m} g(\widetilde{\mathbf{v}}, [\widetilde{\mathbf{h}}^{(\ell-1)}, \mathbf{x}^{(\ell)}]) \right| + \left| \frac{N}{m} g([\widetilde{\mathbf{v}}, \widetilde{\mathbf{h}}^{(\ell-1)}, \mathbf{x}^{(\ell)}]) - \frac{N}{m} g(\mathbf{v}, [\mathbf{h}^{(\ell-1)}, \mathbf{x}^{(\ell)}]) \right|$$

$$\leq \mathcal{O}\left( \rho^{5+\kappa} N^{5/3} m^{-7/6} + \rho^{1+\kappa} (N/m)^{1+\zeta} + \rho^{1+\kappa} N^{1/2} m^{-1} + \rho^{5+\kappa} (N/m)^{3/2} \right) \cdot \|\mathbf{v}\| .$$

$\square$

# D   Generalization bounds of Recurrent neural networks

The proof has been structured as follows: In section D.1, we prove thm. D.1 where we show that a linear transformation of $\mathbf{h}^{(L)}$ can give back $[\mathbf{x}^{(1)}, \ldots, \mathbf{x}^{(L)}]$. The proof follows from a direct application of lemma C.3. Claim E.10 shows that the linear matrix at each induction step satisfies a property of stability necessary for the inductive application of lemma C.3.

In section D.2, we first define a pseudo recurrent neural network that stays close to the over parameterized RNN at initialization throughout SGD. We then show in thm. D.2 that there exists a pseudo network which can approximate the target function in concept class. The proof involves breaking correlations among the hidden states and the weight matrices and then we show that the pseudo network concentrates on the desired signal. The above two steps have been divided among the four intermediate claims: F.3, F.6, F.7 and F.8.

In section D.3, we prove theorem D.5 which shows that RNNs can attain a population risk similar to the target function in the concept class using SGD. First, we show that the pseudo neural network stays close to RNN with small perturbation around initialization in lemmas G.3 and G.2. We then show that there exists a RNN close to random RNN that can approximate the target function in lemma G.4. We complete the argument by showing that the SGD can find matrices with training loss close to the optimal in lemma D.3 and then bounding the Rademacher complexity of RNNs with bounds on the movement in the weight matrices in lemma D.4.

## D.1   Invertibility of RNNs at initialization

Let $\mathbf{W}^{(k_b, k_e)} = \prod_{k_b \geq \ell \geq k_e} \mathbf{D}_{(0)}^{(\ell)} \mathbf{W}$, if $k_b \geq k_c$. Otherwise, $\mathbf{W}^{(k_b, k_e)} = \mathbf{I}$. Define $\overline{\mathbf{W}}^{[\ell]}$ inductively as follows:

$$\overline{\mathbf{W}}^{[\ell]} = \left[ \mathbf{D}_{(0)}^{(\ell)} \mathbf{W} \overline{\mathbf{W}}^{[\ell-1]}, \ \mathbf{D}_{(0)}^{(\ell)} \mathbf{A} \right]_r, \quad \text{for } 2 \leq \ell \leq L,$$

with $\overline{\mathbf{W}}^{[1]} = \mathbf{D}_{(0)}^{(1)} \mathbf{A}$. We can show that $\overline{\mathbf{W}}^{[\ell]} = [\mathbf{W}^{(\ell,2)} \mathbf{D}_{(0)}^{(1)} \mathbf{A}, \mathbf{W}^{(\ell,3)} \mathbf{D}_{(0)}^{(2)} \mathbf{A}, \cdots, \mathbf{D}_{(0)}^{(\ell)} \mathbf{A}]_r$ for $\ell \geq 2$, which will be helpful for presentation later on.

**Theorem D.1.** *For any $\epsilon_x < \frac{1}{L}$ and a given normalized sequence $\mathbf{x}^{(1)}, \cdots, \mathbf{x}^{(L)}$,*

$$\left\| [\mathbf{x}^{(1)}, \cdots, \mathbf{x}^{(L)}] - \overline{\mathbf{W}}^{[L]\top} \mathbf{h}^{(L)} \right\|_{\infty}$$

$$\leq \mathcal{O}\left( L^4 \cdot (\rho^{11} m^{-1/12} + \rho^7 m^{-1/12} + \rho^7 m^{-1/4} + \rho^{11} m^{-1/4}) \right) + \mathcal{O}(\rho^2 L^{11/6} \epsilon_x^{5/3})$$

*with probability at least $1 - e^{-\Omega(\rho^2)}$.*

*Proof.* The theorem has been restated and proven in theorem E.1. $\square$

**Corollary D.1.1.** *For a given normalized sequence $\mathbf{x}$ and any $\varepsilon_x < \frac{1}{L}$, with probability at least $1 - e^{-\Omega(\rho^2)}$ w.r.t. the weights $\mathbf{W}$ and $\mathbf{A}$,*

$$\left| \overline{\mathbf{W}}^{[L]\top} h^{(L-1)} - \varepsilon_x [\overline{\mathbf{x}}^{(2)}, \cdots, \overline{\mathbf{x}}^{(L-1)}] \right|$$

$$\leq \mathcal{O}\left( L^4 \cdot (\rho^{11} m^{-1/12} + \rho^7 m^{-1/12} + \rho^7 m^{-1/4} + \rho^{11} m^{-1/4}) \right) + \mathcal{O}(L^{4/3} \varepsilon_x^{2/3})$$

$$\leq \mathcal{O}(L^4 \rho^{11} m^{-1/12} + \rho^2 L^{11/6} \varepsilon_x^{5/3}),$$

*where $\overline{\mathbf{W}}^{[\ell]}$ is slightly redefined as*

$$\overline{\mathbf{W}}^{[\ell]} = [\mathbf{D}_{(0)}^{(\ell-1)} \mathbf{W} \overline{\mathbf{W}}^{[\ell-2]}, \mathbf{D}_{(0)}^{(\ell-1)} \mathbf{A}_{[d-1]}]_r, \quad \text{for all } \ell \geq 4,$$

*with $\overline{\mathbf{W}}^{[2]} = \mathbf{D}_{(0)}^{(2)} \mathbf{A}_{[d-1]}$ and $\mathbf{A}_{[d-1]} \in \mathbb{R}^{m \times (d-1)}$ denotes the matrix which contains the first $d-1$ columns of the matrix $\mathbf{A}$.*

*Proof.* The difference from Thm. D.1 is that here we attempt to get the first $d-1$ dimensions of the vectors $\mathbf{x}^{(2)}, \cdots, \mathbf{x}^{(L-1)}$. This leads to a small change in the inversion matrix. $\qquad\square$

Note that in the above corollary, $\overline{\mathbf{W}}^{[L]} = [\mathbf{W}^{(L,3)} \mathbf{D}_{(0)}^{(2)} \mathbf{A}_{[d-1]}, \mathbf{W}^{(L,4)} \mathbf{D}_{(0)}^{(3)} \mathbf{A}_{[d-1]}, \cdots, \mathbf{W}^{(L,L)} \mathbf{D}_{(0)}^{(L-1)} \mathbf{A}_{[d-1]}]_r$, where $\mathbf{W}^{(k_b, k_e)} = \prod_{k_b \geq \ell > k_e} \mathbf{D}_{(0)}^{(\ell)} \mathbf{W}$. We are going to use this definition in the following theorems.

### D.2 Existence of good pseudo network

We first define a pseudo RNN model, which is shown later to stay close to the RNN model during the gradient descent dynamics.

**Definition D.1** (Pseudo Recurrent Neural Network). Given two matrices $\mathbf{W}^* \in \mathbb{R}^{m \times m}$ and $\mathbf{A}^* \in \mathbb{R}^{m \times d}$, the output for a pseudo RNN with activation function ReLU for a given sequence $\mathbf{x}$ are given by

$$F_s^{(\ell)}(\mathbf{x}; \mathbf{W}^*, \mathbf{A}^*) = \sum_{i \leq \ell} \mathbf{Back}_{i \to \ell, s} \mathbf{D}^{(i)} \left( \mathbf{W}^* h^{(i-1)} + \mathbf{A}^* \mathbf{x}^{(i)} \right) \quad \forall 1 \leq \ell \leq L, s \in [d_{\text{out}}],$$

where $\mathbf{Back}_{i \to \ell, s} = \mathbf{b}_s^\top \mathbf{D}^{(\ell)} \mathbf{W} \cdots \mathbf{D}^{(i+1)} \mathbf{W}$. For typographical simplicity, we will denote $F_s^{(\ell)}(\mathbf{x}; \mathbf{W}^*, \mathbf{A}^*)$ as $F_s^{(\ell)}$.

Now, we show that there exist two matrices $\mathbf{W}^*$ and $\mathbf{A}^*$, defined below, such that the pseudo network is close to the concept class under consideration.

**Definition D.2.** Define $\mathbf{W}^*$ and $\mathbf{A}^*$ as follows.

$$\mathbf{W}^* = 0$$

$$\mathbf{a}_r^* = \frac{d_{\text{out}}}{m} \sum_{s \in [d_{\text{out}}]} \sum_{r' \in [p]} b_{r,s} b_{r',s}^\dagger H_{r',s} \left( \theta_{r',s} \left( \langle \mathbf{w}_r, \overline{\mathbf{W}}^{[L]} \mathbf{w}_{r',s}^\dagger \rangle \right), \sqrt{m/2} a_{r,d} \right) \mathbf{e}_d, \quad \forall r \in [m],$$

where

$$\theta_{r',s} = \frac{\sqrt{m/2}}{\left\| \overline{\mathbf{W}}^{[L]} \mathbf{w}_{r',s}^\dagger \right\|},$$

and $\overline{\mathbf{W}}^{[L]} = [\mathbf{W}^{(L,3)} \mathbf{D}_{(0)}^{(2)} \mathbf{A}_{[d-1]}, \mathbf{W}^{(L,3)} \mathbf{D}_{(0)}^{(2)} \mathbf{A}_{[d-1]}, \cdots, \mathbf{W}^{(L,L)} \mathbf{D}_{(0)}^{(L-1)} \mathbf{A}_{[d-1]}]_r$, where $\mathbf{W}^{(k_b, k_e)} = \prod_{k_b \geq \ell > k_e} \mathbf{D}_{(0)}^{(\ell)} \mathbf{W}$.

In the following theorem, we show that the pseudo RNN can approximate the target concept class, using the weight $\mathbf{W}^*$ and $\mathbf{A}^*$ define above.

**Theorem D.2** (Existence of Good Pseudo Network). *The construction of $\mathbf{W}^*$ and $\mathbf{A}^*$ in Definition D.2 satisfies the following. For every normalized input sequence $\mathbf{x}^{(1)}, \cdots, \mathbf{x}^{(L)}$, we have with probability at least $1 - e^{-\Omega(\rho^2)}$ over $\mathbf{W}, \mathbf{A}, \mathbf{B}$, it holds for every $s \in [d_{\mathrm{out}}]$.*

$$F_s^{(L)} \overset{def}{=} \sum_{i=1}^{L} \mathbf{e}_s^\top \mathbf{Back}_{i \to L} D^{(i)} \left( \mathbf{W}^* \mathbf{h}^{(i-1)} + \mathbf{A}^* \mathbf{x}^{(i)} \right)$$

$$= \sum_{r \in [p]} b_{r,s}^\dagger \Phi_{r,s} \left( \left\langle \mathbf{w}_{r,s}^\dagger, [\overline{\mathbf{x}}^{(2)}, \cdots, \overline{\mathbf{x}}^{(L-2)}] \right\rangle \right)$$

$$\pm \mathcal{O}(d_{\mathrm{out}} L p \rho^2 \varepsilon + d_{\mathrm{out}} L^{17/6} p \rho^4 L_\Phi \varepsilon_x^{2/3} + d_{\mathrm{out}}^{3/2} L^5 p \rho^{11} L_\Phi C_\Phi \mathfrak{C}_\varepsilon(\Phi, \mathcal{O}(\varepsilon_x^{-1})) m^{-1/30}).$$

*Proof.* The theorem has been restated and proven in theorem F.1. $\qquad\square$

### D.3 Optimization and Generalization

First, we show that the training loss decreases with gradient descent. The main component of the proof is to show that overparameterized RNN stays close to its pseudo network throughout training. Since, the pseudo network is a linear network, it is easier to check the trajectory of the pseudo network during gradient descent. Since we have already shown that there exists a pseudo network that can approximate the true function, we can show that gradient descent can find some pseudo network that performs as well as the constructed pseudo network.

**Lemma D.3** (Decrease in training loss). *For a constant $\varepsilon_x = \frac{1}{\mathrm{poly}(\rho)}$ and for every constant $\varepsilon \in \left( 0, \frac{1}{p \cdot \mathrm{poly}(\rho) \cdot \mathfrak{C}_s(\Phi, \mathcal{O}(\varepsilon_x^{-1}))} \right)$, there exists $C' = \mathfrak{C}_\varepsilon(\Phi, \mathcal{O}(\varepsilon_x^{-1}))$, $C_\Phi = \mathfrak{C}_s(\Phi, \sqrt{2L})$, and a parameter $\lambda = \Theta\left( \frac{\varepsilon}{L\rho} \right)$ so that, as long as $m \geq \mathrm{poly}\left( C', p, L, d_{\mathrm{out}}, \varepsilon^{-1} \right)$ and $N \geq \Omega\left( \frac{\rho^3 p C_\Phi^2}{\varepsilon^2} \right)$, setting learning rate $\eta = \Theta\left( \frac{1}{\varepsilon \rho^2 m} \right)$ and $T = \Theta\left( \frac{p^2 C'^2 \mathrm{poly}(\rho)}{\varepsilon^2} \right)$, we have*

$$\mathbb{E}_{\mathrm{sgd}} \left[ \frac{1}{T} \sum_{t=0}^{T-1} \mathbb{E}_{(\overline{\mathbf{x}}, \mathbf{y}^*) \sim \mathcal{Z}} \mathrm{Obj}(\overline{\mathbf{x}}, \mathbf{y}^*; \mathbf{W} + \mathbf{W}_t, \mathbf{A} + \mathbf{A}_t) \right] \leq \mathrm{OPT} + \frac{\varepsilon}{2} + \frac{1}{\mathrm{poly}(\rho)},$$

*and $\|W_t\|_F \leq \frac{\Delta}{\sqrt{m}}$ for $\Delta = \frac{C'^2 p^2 \mathrm{poly}(\rho)}{\varepsilon^2}$.*

*Proof.* The lemma has been restated and proven in lemma G.1. $\qquad\square$

Now, we bound the rademacher complexity of overparameterized RNNs. The main component of the proof is to use the fact that overparameterized RNN stays close to its pseudo network throughout training. Since, the pseudo network is a linear network, it is easier to compute the rademacher complexity of pseudo network.

**Lemma D.4** (Rademacher Complexity of RNNs). *For every $s \in [d_{\mathrm{out}}]$, we have*

$$\mathbb{E}_{\zeta \in \{\pm 1\}^N} \left[ \sup_{\|\mathbf{W}'\|_F, \|\mathbf{A}'\|_F \leq \frac{\Delta}{\sqrt{m}}} \frac{1}{N} \sum_{q=1}^{N} \zeta_q F_{\mathrm{rnn},s}^{(L)}(\overline{\mathbf{x}}_q; \mathbf{W} + \mathbf{W}', \mathbf{A} + \mathbf{A}') \right] \leq \mathcal{O}(\frac{\rho^7 \Delta^{4/3}}{m^{1/6}} + \frac{\rho^2 \Delta}{\sqrt{N}}),$$

*where $\overline{\mathbf{x}}_1, \ldots, \overline{\mathbf{x}}_N$ denote the training samples in $\mathcal{D}$.*

*Proof.* The proof follows the same outline as lemma 8.1 in [37]. We give the outline here for completeness. From lemma G.3, we have w.p. at least $1 - e^{-\Omega(\rho^2)}$ for all $q \in [N]$, $s \in [d_{\mathrm{out}}]$ and for any $\mathbf{W}', \mathbf{A}'$ with $|\mathbf{W}'|, |\mathbf{A}'| \leq \frac{\Delta}{\sqrt{m}}$,

$$\left| F_{\mathrm{rnn},s}^{(L)}(\overline{\mathbf{x}}_q; \mathbf{W} + \mathbf{W}', \mathbf{A} + \mathbf{A}') - F_s^{(L)}(\overline{\mathbf{x}}_q; \mathbf{W}', \mathbf{A}') \right| \leq \mathcal{O}(\rho^7 \Delta^{4/3} m^{-1/6}).$$

Hence, $F_{\mathrm{rnn}}^{(L)}$ is close to $F^{(L)}$ and thus, its rademacher complexity will be close to that of $F^{(L)}$. Since, $F^{(L)}$ is a linear network, we can apply the rademacher complexity for linear networks (fact A.8) to get the final bound. $\qquad\square$

Now, we can combine both the theorems above to get the following theorem.

**Theorem D.5.** *For a constant $\epsilon_x = \frac{1}{\mathrm{poly}(\rho)}$ and for every constant $\varepsilon \in \left(0, \frac{1}{p\cdot\mathrm{poly}(\rho)\cdot\mathfrak{C}_s(\Phi,\mathcal{O}(\epsilon_x^{-1}))}\right)$, define complexity $C = \mathfrak{C}_\varepsilon(\Phi,\mathcal{O}(\epsilon_x^{-1}))$ and $\lambda = \frac{\varepsilon}{10L\rho}$, if the number of neurons $m \geq \mathrm{poly}\left(C, p, L, d_{\mathrm{out}}, \varepsilon^{-1}\right)$ and the number of samples is $N \geq \mathrm{poly}\left(C, p, L, d_{\mathrm{out}}, \varepsilon^{-1}\right)$, then SGD with $\eta = \Theta\left(\frac{1}{\varepsilon\rho^2 m}\right)$ and $T = \Theta(p^2 C^2 \mathrm{poly}(\rho)\varepsilon^{-2})$ satisfies that, with probability at least $1 - e^{-\Omega(\rho^2)}$ over the random initialization*

$$\mathop{\mathbb{E}}_{\mathrm{sgd}}\left[\frac{1}{T}\sum_{t=0}^{T-1}\mathop{\mathbb{E}}_{(\overline{\mathbf{x}},y^*)\sim\mathcal{D}}\left[\mathrm{Obj}\left(\overline{\mathbf{x}}, y^*; \mathbf{W}_t, \mathbf{A}_t\right)\right]\right] \leq \mathrm{OPT} + \varepsilon + \frac{1}{\mathrm{poly}(\rho)}.$$

*Proof.* The proof follows from Fact A.7 using Lemma D.3 and Lemma D.4. □

# E Invertibility of RNNs at initialization: proofs

## E.1 Proof of theorem D.1

**Theorem E.1** (Restating theorem D.1)**.** *For any $\epsilon_x < \frac{1}{L}$ and a given sequence $\mathbf{x}^{(1)}, \cdots, \mathbf{x}^{(L)}$,*

$$\left\| [\mathbf{x}^{(1)}, \cdots, \mathbf{x}^{(L)}] - \overline{\mathbf{W}}^{[L]\top} \mathbf{h}^{(L)} \right\|_\infty$$
$$\leq \mathcal{O}\left( L^4 \cdot (\rho^{11} m^{-1/12} + \rho^7 m^{-1/12} + \rho^7 m^{-1/4} + \rho^{11} m^{-1/4}) \right) + \mathcal{O}(\rho^2 L^{11/6} \epsilon_x^{5/3})$$

*with probability at least $1 - e^{-\Omega(\rho^2)}$.*

*Proof.* Define $\mathbf{V}^{(\ell)}$ inductively as follows:

$$\mathbf{V}^{(1)} = [\mathbf{0}_{d\times m}, \ \mathbf{I}_{d\times d}]_r \ \ [\mathbf{W}, \ \mathbf{A}]_r^\top \ \mathbf{D}_{(0)}^{(1)}$$
$$\mathbf{V}^{(\ell)} = \left[ \ \left[\mathbf{V}^{(\ell-1)}, \ \mathbf{0}_{(\ell-1)d\times d}\right]_r, \ \ [\mathbf{0}_{d\times m}, \ \mathbf{I}_{d\times d}]_r \ \right]_c \ \ [\mathbf{W}, \ \mathbf{A}]_r^\top \ \mathbf{D}_{(0)}^{(\ell)}$$

Now, we show three claims that help us to get the desired inequality.

**Claim E.2.** *With probability at least $1 - e^{-\Omega(\rho^2)}$,*

$$\left\| [\mathbf{x}^{(1)}, \cdots, \mathbf{x}^{(\ell)}] - V^{(\ell)} \mathbf{h}^{(\ell)} \right\|_\infty$$
$$\leq \mathcal{O}\left( \rho^{11} m^{-1/12} + \rho^7 m^{-1/12} + \rho^7 m^{-1/4} + \rho^{11} m^{-1/4} \right) \cdot \sum_{k<\ell} \left\| V^{(k)} \right\|_{2,\infty} + \mathcal{O}(\ell \rho^2 L^{5/6} \epsilon_x^{5/3}),$$

*for all $\ell \in [L]$.*

**Claim E.3.**
$$\mathbf{V}^{(\ell)} = \overline{\mathbf{W}}^{[\ell]\top}, \quad \text{for all } \ell \in [L].$$

**Claim E.4.**
$$\left\| \overline{\mathbf{W}}^{[\ell]\top} \right\|_{2,\infty} \leq \mathcal{O}(L^3), \quad \text{for all } \ell \in [L],$$

*with probability at least $1 - 2e^{-\rho^2}$.*

The above claims have been restated and proven in claims E.6, E.7 and E.9 respectively. Hence, using claims E.3, E.2 and E.4, we have

$$\left\| \overline{\mathbf{W}}^{[L]\top} \mathbf{h}^{(L)} - [\mathbf{x}^{(1)}, \cdots, \mathbf{x}^{(L)}] \right\|_\infty = \left\| V^{(L)} \mathbf{h}^{(L)} - [\mathbf{x}^{(1)}, \cdots, \mathbf{x}^{(L)}] \right\|_\infty$$

$$\leq \mathcal{O}\left( \rho^{11} m^{-1/12} + \rho^7 m^{-1/12} + \rho^7 m^{-1/4} + \rho^{11} m^{-1/4} \right) \left( \sum_{\ell<L} \left\| V^{(\ell)} \right\|_{2,\infty} \right) + \mathcal{O}(\rho^2 L^{11/6} \epsilon_x^{5/3})$$

$$= \mathcal{O}\left( \rho^{11} m^{-1/12} + \rho^7 m^{-1/12} + \rho^7 m^{-1/4} + \rho^{11} m^{-1/4} \right) \left( \sum_{\ell<L} \left\| \overline{\mathbf{W}}^{[\ell]\top} \right\|_{2,\infty} \right) + \mathcal{O}(\rho^2 L^{11/6} \epsilon_x^{5/3})$$

$$\leq \mathcal{O}\left( \rho^{11} m^{-1/12} + \rho^7 m^{-1/12} + \rho^7 m^{-1/4} + \rho^{11} m^{-1/4} \right) \left( \sum_{\ell<L} \mathcal{O}(L^3) \right) + \mathcal{O}(\rho^2 L^{11/6} \epsilon_x^{5/3})$$

$$= \mathcal{O}\left( L^4 \cdot (\rho^{11} m^{-1/12} + \rho^7 m^{-1/12} + \rho^7 m^{-1/4} + \rho^{11} m^{-1/4}) \right) + \mathcal{O}(\rho^2 L^{11/6} \epsilon_x^{5/3}).$$

$\square$

## E.2 Proofs of the helping claims

The following restatement of Lemma C.3 in matrix notation will be useful in the sequel.

**Lemma E.5.** *Let $\mathbf{V} \in \mathbb{R}^{k \times (m+d)}$ for $k \geq 1$, such that each row of $\mathbf{V}$ satisfies assumption 1 with constants $(\kappa, \zeta)$. For all $\ell \in \{0, 1, \ldots, L-1\}$ and for a given sequence $\mathbf{x}^{(1)}, \cdots, \mathbf{x}^{(\ell)}$ we have*

$$\left\| \mathbf{V}[\mathbf{h}^{(\ell-1)}, \mathbf{x}^{(\ell)}] - \mathbf{V}([\mathbf{W}, \mathbf{A}])^\top \mathbf{h}^{(\ell)} \right\|_\infty$$
$$\leq \mathcal{O}\left( \rho^{5+\kappa} m^{-1/12} + \rho^{1+\kappa} m^{-\zeta/2} + \rho^{1+\kappa} m^{-1/4} + \rho^{5+\kappa} m^{-1/4} \right) \cdot \|\mathbf{V}\|_{2,\infty},$$

*with probability at least $1 - k e^{-\Omega(\rho^2)}$.*

**Claim E.6** (Restating claim E.2). *With probability at least $1 - e^{-\Omega(\rho^2)}$,*

$$\left\| [\mathbf{x}^{(1)}, \cdots, \mathbf{x}^{(\ell)}] - V^{(\ell)} \mathbf{h}^{(\ell)} \right\|_\infty$$
$$\leq \mathcal{O}\left( \rho^{11} m^{-1/12} + \rho^7 m^{-1/12} + \rho^7 m^{-1/4} + \rho^{11} m^{-1/4} \right) \cdot \sum_{k < \ell} \left\| \mathbf{V}^{(k)} \right\|_{2,\infty} + \mathcal{O}(\ell \rho^2 L^{5/6} \epsilon_x^{5/3}),$$

*for all $\ell \in [L]$.*

*Proof.* We prove the claim by induction. For $\ell = 1$, we have

$$\left\| \mathbf{x}^{(1)} - \mathbf{V}^{(1)} \mathbf{h}^{(1)} \right\|_\infty = \left\| [\mathbf{0}_{d \times m}, \mathbf{I}_{d \times d}]_r [\mathbf{h}^{(0)}, \mathbf{x}^{(1)}] - \mathbf{V}^{(1)} \mathbf{h}^{(1)} \right\|_\infty$$
$$= \left\| [\mathbf{0}_{d \times m}, \mathbf{I}_{d \times d}]_r [\mathbf{h}^{(0)}, \mathbf{x}^{(1)}] - [\mathbf{0}_{d \times m}, \mathbf{I}_{d \times d}]_r \ [\mathbf{W}, \mathbf{A}]_r^\top \ \mathbf{D}_{(0)}^{(1)} \mathbf{h}^{(1)} \right\|_\infty$$
$$\leq \left\| [\mathbf{0}_{d \times m}, \mathbf{I}_{d \times d}]_r [\mathbf{h}^{(0)}, \mathbf{x}^{(1)}] - [\mathbf{0}_{d \times m}, \mathbf{I}_{d \times d}]_r \ [\mathbf{W}, \mathbf{A}]_r^\top \ \mathbf{D}^{(1)} \mathbf{h}^{(1)} \right\|_\infty$$
$$+ \left\| [\mathbf{0}_{d \times m}, \mathbf{I}_{d \times d}]_r \ [\mathbf{W}, \mathbf{A}]_r^\top \ (\mathbf{D}_{(0)}^{(1)} - \mathbf{D}^{(1)}) \mathbf{h}^{(1)} \right\|_\infty.$$

Since by the definition of $\mathbf{D}^{(1)}$, $\mathbf{D}^{(1)} \mathbf{h}^{(1)} = \mathbf{h}^{(1)}$, we have

$$\left\| \mathbf{x}^{(1)} - \mathbf{V}^{(1)} \mathbf{h}^{(1)} \right\|_\infty \leq \left\| [\mathbf{0}_{d \times m}, \mathbf{I}_{d \times d}]_r [\mathbf{h}^{(0)}, \mathbf{x}^{(1)}] - [\mathbf{0}_{d \times m}, \mathbf{I}_{d \times d}]_r \ [\mathbf{W}, \mathbf{A}]_r^\top \ \mathbf{D}^{(1)} \mathbf{h}^{(1)} \right\|_\infty$$
$$+ \left\| [\mathbf{0}_{d \times m}, \mathbf{I}_{d \times d}]_r \ [\mathbf{W}, \mathbf{A}]_r^\top \ (\mathbf{D}_{(0)}^{(1)} - \mathbf{D}^{(1)}) \mathbf{h}^{(1)} \right\|_\infty$$
$$= \left\| [\mathbf{0}_{d \times m}, \mathbf{I}_{d \times d}]_r [\mathbf{h}^{(0)}, \mathbf{x}^{(1)}] - [\mathbf{0}_{d \times m}, \mathbf{I}_{d \times d}]_r \ [\mathbf{W}, \mathbf{A}]_r^\top \ \mathbf{h}^{(1)} \right\|_\infty \qquad (22)$$
$$+ \left\| [\mathbf{0}_{d \times m}, \mathbf{I}_{d \times d}]_r \ [\mathbf{W}, \mathbf{A}]_r^\top \ (\mathbf{D}_{(0)}^{(1)} - \mathbf{D}^{(1)}) \mathbf{h}^{(1)} \right\|_\infty. \qquad (23)$$

Now, using Lemma E.5, we can show that Eq. 22 is small, i.e.

$$\left\| [\mathbf{0}_{d \times m}, \mathbf{I}_{d \times d}]_r [\mathbf{h}^{(0)}, \mathbf{x}^{(1)}] - [\mathbf{0}_{d \times m}, \mathbf{I}_{d \times d}]_r \ [\mathbf{W}, \mathbf{A}]_r^\top \ \mathbf{h}^{(1)} \right\|_\infty$$
$$\leq \mathcal{O}\left( \rho^5 m^{-1/12} + \rho m^{-1/4} + \rho^5 m^{-1/4} \right) \cdot \left\| [\mathbf{0}_{d \times m}, \mathbf{I}_{d \times d}]_r \right\|_{2,\infty}$$
$$= \mathcal{O}\left( \rho^5 m^{-1/12} + \rho m^{-1/4} + \rho^5 m^{-1/4} \right),$$

where we have used the fact that $[\mathbf{0}_{d \times m}, \mathbf{I}_{d \times d}]_r$ doesn't depend on $\mathbf{W}$ and $\mathbf{A}$ and hence each row satisfies assumption 1 with $(\kappa, \zeta) = (0, 0)$. Now, we will show that Eq. 23 is small. Note that $\mathbf{x}^{(1)} = \mathbf{x}_{(0)}^{(1)}$ after input normalization. Also, $\mathbf{h}^{(0)}$ is 0 for any sequence. Thus,

$$\mathbf{D}_{(0)}^{(1)} = \mathbf{D}^{(1)}.$$

Hence,

$$\left\| [\mathbf{0}_{d \times m}, \mathbf{I}_{d \times d}]_r \ [\mathbf{W}, \mathbf{A}]_r^\top \ (\mathbf{D}_{(0)}^{(1)} - \mathbf{D}^{(1)}) \mathbf{h}^{(1)} \right\|_\infty = 0$$

Thus, continuing from Eq. 22 and Eq. 23, we have
$$\left\|\mathbf{x}^{(1)} - \mathbf{V}^{(1)}\mathbf{h}^{(1)}\right\|_\infty \leq \mathcal{O}\left(\rho^5 m^{-1/12} + \rho m^{-1/4} + \rho^5 m^{-1/4}\right).$$
Assuming the claim is true for all $\ell \leq \ell'$, we now try to prove for $\ell = \ell' + 1$. We have

$$\left\|\left[\left[\mathbf{V}^{(\ell')}, \mathbf{0}_{\ell'd\times d}\right]_r, \; [\mathbf{0}_{d\times m}, \mathbf{I}_{d\times d}]_r\right]_c [\mathbf{h}^{(\ell')}, \mathbf{x}^{(\ell'+1)}] - \mathbf{V}^{(\ell'+1)}\mathbf{h}^{(\ell'+1)}\right\|_\infty$$

$$= \left\|\left[\mathbf{V}^{(\ell')}\mathbf{h}^{(\ell')}, \mathbf{x}^{(\ell'+1)}\right] - \mathbf{V}^{(\ell'+1)}\mathbf{h}^{(\ell'+1)}\right\|_\infty$$

$$\geq \left\|\mathbf{V}^{(\ell'+1)}\mathbf{h}^{(\ell'+1)} - \left[[\mathbf{x}^{(1)}, \cdots, \mathbf{x}^{(\ell')}], \mathbf{x}^{(\ell'+1)}\right]\right\|_\infty$$

$$- \left\|\left[\mathbf{V}^{(\ell')}\mathbf{h}^{(\ell')}, \mathbf{x}^{(\ell'+1)}\right] - \left[[\mathbf{x}^{(1)}, \cdots, \mathbf{x}^{(\ell')}], \mathbf{x}^{(\ell'+1)}\right]\right\|_\infty$$

$$= \left\|\mathbf{V}^{(\ell'+1)}\mathbf{h}^{(\ell'+1)} - [\mathbf{x}^{(1)}, \cdots, \mathbf{x}^{(\ell'+1)}]\right\|_\infty$$

$$- \left\|\mathbf{V}^{(\ell')}\mathbf{h}^{(\ell')} - [\mathbf{x}^{(1)}, \cdots, \mathbf{x}^{(\ell')}]\right\|_\infty.$$

Thus,

$$\left\|\mathbf{V}^{(\ell'+1)}\mathbf{h}^{(\ell'+1)} - [\mathbf{x}^{(1)}, \cdots, \mathbf{x}^{(\ell'+1)}]\right\|_\infty$$

$$\leq \left\|\left[\left[\mathbf{V}^{(\ell')}, \mathbf{0}_{\ell'd\times d}\right]_r, \; [\mathbf{0}_{d\times m}, \mathbf{I}_{d\times d}]_r\right]_c [\mathbf{h}^{(\ell')}, \mathbf{x}^{(\ell'+1)}] - \mathbf{V}^{(\ell'+1)}\mathbf{h}^{(\ell'+1)}\right\|_\infty \tag{24}$$

$$+ \left\|\mathbf{V}^{(\ell')}\mathbf{h}^{(\ell')} - [\mathbf{x}^{(1)}, \cdots, \mathbf{x}^{(\ell')}]\right\|_\infty \tag{25}$$

Using induction, we have in Eq. 25,
$$\left\|\mathbf{V}^{(\ell')}[\mathbf{h}^{(\ell'-1)}, \mathbf{x}^{(\ell')}] - [\mathbf{x}^{(1)}, \cdots, \mathbf{x}^{(\ell')}]\right\|_\infty$$

$$\leq \mathcal{O}\left(\rho^{11} m^{-1/12} + \rho^7 m^{-1/12} + \rho^7 m^{-1/4} + \rho^{11} m^{-1/4}\right) \cdot \sum_{k<\ell'}\left\|\mathbf{V}^{(k)}\right\|_{2,\infty} + \mathcal{O}(\ell'\rho^2 L^{5/6}\epsilon_x^{5/3}).$$

Now, we show that Eq. 24 is small.
$$\mathbf{V}^{(\ell'+1)}\mathbf{h}^{(\ell'+1)}$$

$$= \left[\left[\mathbf{V}^{(\ell')}, \mathbf{0}_{\ell'd\times d}\right]_r, \; [\mathbf{0}_{d\times m}, \mathbf{I}_{d\times d}]_r\right]_c [\mathbf{W}, \mathbf{A}]_r^\top \mathbf{D}_{(0)}^{(\ell'+1)}\mathbf{h}^{(\ell'+1)}$$

$$= \left[\left[\mathbf{V}^{(\ell')}, \mathbf{0}_{\ell'd\times d}\right]_r, \; [\mathbf{0}_{d\times m}, \mathbf{I}_{d\times d}]_r\right]_c [\mathbf{W}, \mathbf{A}]_r^\top (\mathbf{D}_{(0)}^{(\ell'+1)} - \mathbf{D}^{(\ell'+1)})\mathbf{h}^{(\ell'+1)}$$

$$+ \left[\left[\mathbf{V}^{(\ell')}, \mathbf{0}_{\ell'd\times d}\right]_r, \; [\mathbf{0}_{d\times m}, \mathbf{I}_{d\times d}]_r\right]_c [\mathbf{W}, \mathbf{A}]_r^\top \mathbf{D}^{(\ell'+1)}\mathbf{h}^{(\ell'+1)}$$

$$= \left[\left[\mathbf{V}^{(\ell')}, \mathbf{0}_{\ell'd\times d}\right]_r, \; [\mathbf{0}_{d\times m}, \mathbf{I}_{d\times d}]_r\right]_c [\mathbf{W}, \mathbf{A}]_r^\top (\mathbf{D}_{(0)}^{(\ell'+1)} - \mathbf{D}^{(\ell'+1)})\mathbf{h}^{(\ell'+1)} \tag{26}$$

$$+ \left[\left[\mathbf{V}^{(\ell')}, \mathbf{0}_{\ell'd\times d}\right]_r, \; [\mathbf{0}_{d\times m}, \mathbf{I}_{d\times d}]_r\right]_c [\mathbf{W}, \mathbf{A}]_r^\top \mathbf{h}^{(\ell'+1)}, \tag{27}$$

where in the final step, we have used the definition of $\mathbf{D}^{(\ell'+1)}$ to get $\mathbf{h}^{(\ell'+1)} = \mathbf{D}^{(\ell'+1)}\mathbf{h}^{(\ell'+1)}$. First, we will focus on Eq. 27. Using Lemma E.5, we have

$$\left\|\left[\left[\mathbf{V}^{(\ell')}, \mathbf{0}_{\ell'd\times d}\right]_r, \; [\mathbf{0}_{d\times m}, \mathbf{I}_{d\times d}]_r\right]_c [\mathbf{W}, \mathbf{A}]_r^\top \mathbf{h}^{(\ell'+1)} - \left[\left[\mathbf{V}^{(\ell')}, \mathbf{0}_{\ell'd\times d}\right]_r, [\mathbf{0}_{d\times m}, \mathbf{I}_{d\times d}]_r\right]_c \left[\mathbf{h}^{(\ell')}, \mathbf{x}^{(\ell'+1)}\right]\right\|_{2,\infty}$$

$$\leq \mathcal{O}\left(\rho^{11} m^{-1/12} + \rho^7 m^{-1/12} + \rho^7 m^{-1/4} + \rho^{11} m^{-1/4}\right) \cdot \left\|\left[\left[\mathbf{V}^{(\ell')}, \mathbf{0}_{\ell'd\times d}\right]_r, \; [\mathbf{0}_{d\times m}, \mathbf{I}_{d\times d}]_r\right]_c\right\|_{2,\infty}$$

$$= \mathcal{O}\left(\rho^{11} m^{-1/12} + \rho^7 m^{-1/12} + \rho^7 m^{-1/4} + \rho^{11} m^{-1/4}\right) \cdot \left\|\mathbf{V}^{(\ell')}\right\|_{2,\infty}. \tag{28}$$

In the above steps, we have used Claim E.10 to show that the rows of the matrix $\left[\left[\mathbf{V}^{(\ell')},\ \mathbf{0}_{\ell'd\times d}\right]_r,\ \left[\mathbf{0}_{d\times m},\ \mathbf{I}_{d\times d}\right]_r\right]_c$ satisfies assumption 1 with $(\kappa,\zeta)=(6,\frac{1}{6})$. Next, we show that Eq. 26 is small, i.e.

$$\left\|\left[\left[\mathbf{V}^{(\ell')},\ \mathbf{0}_{\ell'd\times d}\right]_r,\ \left[\mathbf{0}_{d\times m},\ \mathbf{I}_{d\times d}\right]_r\right]_c\ [\mathbf{W},\ \mathbf{A}]_r^\top\ (\mathbf{D}_{(0)}^{(\ell'+1)}-\mathbf{D}^{(\ell'+1)})\mathbf{h}^{(\ell'+1)}\right\|_\infty$$

$$=\left\|\left[\left[\mathbf{V}^{(\ell')},\ \mathbf{0}_{\ell'd\times d}\right]_r,\ \left[\mathbf{0}_{d\times m},\ \mathbf{I}_{d\times d}\right]_r\right]_c\ [\mathbf{W},\ \mathbf{A}]_r^\top\ (\mathbf{D}_{(0)}^{(\ell'+1)}-\mathbf{D}^{(\ell'+1)})\ \mathbf{D}^{(\ell'+1)}\ [\mathbf{W},\ \mathbf{A}]_r\left[\mathbf{h}^{(\ell')},\mathbf{x}^{(\ell'+1)}\right]\right\|_\infty$$

$$=\max_{i\in[d]}\left|\left[\left[\mathbf{V}^{(\ell')},\ \mathbf{0}_{\ell'd\times d}\right]_r,\ \left[\mathbf{0}_{d\times m},\ \mathbf{I}_{d\times d}\right]_r\right]_{1,i}^\top\ [\mathbf{W},\ \mathbf{A}]_r^\top\ (\mathbf{D}_{(0)}^{(\ell'+1)}-\mathbf{I})\ \mathbf{D}^{(\ell'+1)}\ [\mathbf{W},\ \mathbf{A}]_r\left[\mathbf{h}^{(\ell')},\mathbf{x}^{(\ell'+1)}\right]\right|$$

$$=\max_{i\in[d]}\left|\sum_{k\in[m]}\left\langle[\mathbf{w}_k,\mathbf{a}_k],\ \left[\left[\mathbf{V}^{(\ell')},\ \mathbf{0}_{\ell'd\times d}\right]_r,\ \left[\mathbf{0}_{d\times m},\ \mathbf{I}_{d\times d}\right]_r\right]_{1,i}\right\rangle d_{kk}^{(\ell'+1)}(d_{(0),kk}^{(\ell'+1)}-1)\left\langle[\mathbf{w}_k,\mathbf{a}_k],\left[\mathbf{h}^{(\ell')},\mathbf{x}^{(\ell'+1)}\right]\right\rangle\right|$$

$$\leq\max_{i\in[d]}\left\|[\mathbf{W},\ \mathbf{A}]_r\left[\left[\mathbf{V}^{(\ell')},\ \mathbf{0}_{\ell'd\times d}\right]_r,\ \left[\mathbf{0}_{d\times m},\ \mathbf{I}_{d\times d}\right]_r\right]_{1,i}\right\|_\infty$$

$$\cdot\left(\sum_{k\in[m]}d_{kk}^{(\ell'+1)}(1-d_{(0),kk}^{(\ell'+1)})\left|\left\langle[\mathbf{w}_k,\mathbf{a}_k],\left[\mathbf{h}^{(\ell')},\mathbf{x}^{(\ell'+1)}\right]\right\rangle\right|\right),\tag{29}$$

where we use cauchy-schwartz inequality in the final step. First note that,

$$\max_{i\in[d]}\left\|[\mathbf{W},\ \mathbf{A}]_r\left[\left[\mathbf{V}^{(\ell')},\ \mathbf{0}_{\ell'd\times d}\right]_r,\ \left[\mathbf{0}_{d\times m},\ \mathbf{I}_{d\times d}\right]_r\right]_{1,i}\right\|_\infty$$

$$=\left\|[\mathbf{W},\ \mathbf{A}]_r\left[\left[\mathbf{V}^{(\ell')},\ \mathbf{0}_{\ell'd\times d}\right]_r,\ \left[\mathbf{0}_{d\times m},\ \mathbf{I}_{d\times d}\right]_r\right]_1^\top\right\|_{\infty,\infty}$$

$$=\left\|\left[\mathbf{W}\mathbf{V}^{(\ell')\top},\mathbf{A}\right]_2\right\|_{\infty,\infty}$$

$$\leq\max\left(\left\|\mathbf{W}\mathbf{V}^{(\ell')\top}\right\|_{\infty,\infty},\ \|\mathbf{A}\|_{\infty,\infty}\right).$$

First, note that using fact A.4, we have

$$\|\mathbf{A}\|_{\infty,\infty}\leq\frac{\rho}{\sqrt{m}},$$

with probability at least $1-m^2e^{\rho^2/2}$, which is equal to $1-e^{-\Omega(\rho^2)}$, since we are using $\rho=100Ld_{\text{out}}\log m$. Also, from claim E.8, we have

$$\left\|\mathbf{W}\mathbf{V}^{(\ell')\top}\right\|_{\infty,\infty}\leq\mathcal{O}(\frac{\rho}{\sqrt{m}}),$$

with probability at least $1-e^{-\Omega(\rho^2)}$. Hence,

$$\max_{i\in[d]}\left\|[\mathbf{W},\ \mathbf{A}]_r\left[\left[\mathbf{V}^{(\ell')},\ \mathbf{0}_{\ell'd\times d}\right]_r,\ \left[\mathbf{0}_{d\times m},\ \mathbf{I}_{d\times d}\right]_r\right]_{1,i}\right\|_\infty$$

$$\leq\left\|\left[\mathbf{W}\mathbf{V}^{(\ell')\top},\mathbf{A}\right]_2\right\|_{\infty,\infty}$$

$$\leq\max\left(\left\|\mathbf{W}\mathbf{V}^{(\ell')\top}\right\|_{\infty,\infty},\ \|\mathbf{A}\|_{\infty,\infty}\right)$$

$$\leq\mathcal{O}(\frac{\rho}{\sqrt{m}}).$$

By the definition of $\mathbf{D}^{(\ell'+1)}$ and $\mathbf{D}^{(\ell'+1)}_{(0)}$, we can see that

$$\sum_{k\in[m]} d^{(\ell'+1)}_{kk}(1-d^{(\ell'+1)}_{(0),kk}) = \left|\left\{ k\in[m]\ :\ d^{(\ell'+1)}_{kk}\ =\ 1\ \ \&\ \ d^{(\ell'+1)}_{(0),kk}\ =\ 0\right\}\right|$$

$$\leq \left|\left\{ k\in[m]\ :\ d^{(\ell'+1)}_{kk}\ \neq d^{(\ell'+1)}_{(0),kk}\right\}\right|$$

$$= \left\|\mathbf{D}^{(\ell'+1)} - \mathbf{D}^{(\ell'+1)}_{(0)}\right\|_0$$

$$\leq \mathcal{O}(L^{5/6}\epsilon_x^{5/3}m),$$

where we use Lemma B.1 in the final step. Now, we focus on

$$\max_{k\in[m]}\left|\left\langle [\mathbf{w}_k,\mathbf{a}_k],\left[\mathbf{h}^{(\ell')},\mathbf{x}^{(\ell'+1)}\right]\right\rangle\right| d^{(\ell'+1)}_{kk}(1-d^{(\ell'+1)}_{(0),kk}).$$

First, we note that

$$\left| k\in[m]\ :\ d^{(\ell'+1)}_{kk}(1-d^{(\ell'+1)}_{(0),kk}) = 1\right| = \left|\left\{ k\in[m]\ :\ d^{(\ell'+1)}_{kk}\ =\ 1\ \ \&\ \ d^{(\ell'+1)}_{(0),kk}\ =\ 0\right\}\right|.$$

Hence,

$$\left| k\in[m]\ :\ d^{(\ell'+1)}_{kk}(1-d^{(\ell'+1)}_{(0),kk}) = 1\right|$$

$$= \left|\left\{ k\in[m]\ :\ \left\langle [\mathbf{w}_k,\mathbf{a}_k],\left[\mathbf{h}^{(\ell')},\mathbf{x}^{(\ell'+1)}\right]\right\rangle \leq 0 \text{ and } \left\langle [\mathbf{w}_k,\mathbf{a}_k],\left[\mathbf{h}^{(\ell')}_{(0)},\mathbf{x}^{(\ell'+1)}_{(0)}\right]\right\rangle \geq 0\right\}\right|.$$

This implies that for the set $S_m = \left| k\in[m]\ :\ d^{(\ell'+1)}_{kk}(1-d^{(\ell'+1)}_{(0),kk}) = 1\right|$,

$$\left|\left\langle [\mathbf{w}_k,\mathbf{a}_k],\left[\mathbf{h}^{(\ell')},\mathbf{x}^{(\ell'+1)}\right]\right\rangle\right| \leq \left|\left\langle [\mathbf{w}_k,\mathbf{a}_k],\left[\mathbf{h}^{(\ell')} - \mathbf{h}^{(\ell')}_{(0)},\mathbf{x}^{(\ell'+1)} - \mathbf{x}^{(\ell'+1)}_{(0)}\right]\right\rangle\right|,\ \text{ for all } k\in S_m.$$

From lemma B.1, we have with probability $1 - e^{-\Omega(\rho^2)}$,

$$\max_{k\in[m]}\left|\left\langle [\mathbf{w}_k,\mathbf{a}_k],\left[\mathbf{h}^{(\ell')} - \mathbf{h}^{(\ell')}_{(0)},\mathbf{x}^{(\ell'+1)} - \mathbf{x}^{(\ell'+1)}_{(0)}\right]\right\rangle\right| \leq \mathcal{O}(\rho\sqrt{L}\epsilon_x m^{-1/2}).$$

Thus,

$$\sum_{k\in[m]} d^{(\ell'+1)}_{kk}(1-d^{(\ell'+1)}_{(0),kk})\left|\left\langle [\mathbf{w}_k,\mathbf{a}_k],\left[\mathbf{h}^{(\ell')},\mathbf{x}^{(\ell'+1)}\right]\right\rangle\right|$$

$$\leq \left(\sum_{k\in[m]} d^{(\ell'+1)}_{kk}(1-d^{(\ell'+1)}_{(0),kk})\right)\cdot\left(\max_{k\in[m]}\left|\left\langle [\mathbf{w}_k,\mathbf{a}_k],\left[\mathbf{h}^{(\ell')},\mathbf{x}^{(\ell'+1)}\right]\right\rangle\right| d^{(\ell'+1)}_{kk}(1-d^{(\ell'+1)}_{(0),kk})\right)$$

$$\leq \left(\sum_{k\in[m]} d^{(\ell'+1)}_{kk}(1-d^{(\ell'+1)}_{(0),kk})\right)\cdot\left(\max_{k\in S_m}\left|\left\langle [\mathbf{w}_k,\mathbf{a}_k],\left[\mathbf{h}^{(\ell')},\mathbf{x}^{(\ell'+1)}\right]\right\rangle\right|\right)$$

$$\leq \left(\sum_{k\in[m]} d^{(\ell'+1)}_{kk}(1-d^{(\ell'+1)}_{(0),kk})\right)\cdot\left(\max_{k\in S_m}\left|\left\langle [\mathbf{w}_k,\mathbf{a}_k],\left[\mathbf{h}^{(\ell')} - \mathbf{h}^{(\ell')}_{(0)},\mathbf{x}^{(\ell'+1)} - \mathbf{x}^{(\ell'+1)}_{(0)}\right]\right\rangle\right|\right)$$

$$\leq \mathcal{O}(L^{5/6}\epsilon_x^{2/3}m)\cdot\mathcal{O}(\rho\sqrt{L}\epsilon_x m^{-1/2}) = \mathcal{O}(\rho L^{5/6}\epsilon_x^{5/3}\sqrt{m}).$$

Thus, finally Eq. 29 boils down to

$$\left\|\left[\ \left[\mathbf{V}^{(\ell')},\ \mathbf{0}_{\ell'd\times d}\right]_r,\ \left[\mathbf{0}_{d\times m},\ \mathbf{I}_{d\times d}\right]_r\ \right]_c\ [\mathbf{W},\ \mathbf{A}]_r^\top\ (\mathbf{D}^{(\ell'+1)}_{(0)} - \mathbf{D}^{(\ell'+1)})\mathbf{h}^{(\ell'+1)}\right\|_\infty$$

$$\leq \max_{i \in [d]} \left\| [\mathbf{W}, \, \mathbf{A}]_r \left[ \, \left[ \mathbf{V}^{(\ell')}, \, \mathbf{0}_{\ell' d \times d} \right]_r, \; [\mathbf{0}_{d \times m}, \, \mathbf{I}_{d \times d}]_r \, \right]_{1,i} \right\|_\infty$$

$$\cdot \left( \sum_{k \in [m]} d_{kk}^{(\ell'+1)} (1 - d_{(0),kk}^{(\ell'+1)}) \left| \left\langle [\mathbf{w}_k, \mathbf{a}_k], \, \left[ \mathbf{h}^{(\ell')}, \mathbf{x}^{(\ell'+1)} \right] \right\rangle \right| \right)$$

$$\leq \mathcal{O}(\frac{\rho}{\sqrt{m}}) \cdot \mathcal{O}(\frac{\rho}{\sqrt{m}}) \cdot \mathcal{O}(\rho L^{5/6} \epsilon_x^{5/3} \sqrt{m})$$

$$= \mathcal{O}(\rho^2 L^{5/6} \epsilon_x^{5/3}).$$

Hence, continuing from Eq. 26 and Eq. 27, we have

$$\mathbf{V}^{(\ell'+1)} \mathbf{h}^{(\ell'+1)}$$

$$= \left[ \; \left[ \mathbf{V}^{(\ell')}, \, \mathbf{0}_{\ell' d \times d} \right]_r, \; [\mathbf{0}_{d \times m}, \, \mathbf{I}_{d \times d}]_r \; \right]_c [\mathbf{W}, \, \mathbf{A}]_r^\top (\mathbf{D}_{(0)}^{(\ell'+1)} - \mathbf{D}^{(\ell'+1)}) \mathbf{h}^{(\ell'+1)}$$

$$+ \left[ \; \left[ \mathbf{V}^{(\ell')}, \, \mathbf{0}_{\ell' d \times d} \right]_r, \; [\mathbf{0}_{d \times m}, \, \mathbf{I}_{d \times d}]_r \; \right]_c [\mathbf{W}, \, \mathbf{A}]_r^\top \mathbf{h}^{(\ell'+1)}$$

$$= \left[ \; \left[ \mathbf{V}^{(\ell')}, \, \mathbf{0}_{\ell' d \times d} \right]_r, [\mathbf{0}_{d \times m}, \, \mathbf{I}_{d \times d}]_r \; \right]_c \left[ \mathbf{h}^{(\ell')}, \mathbf{x}^{(\ell'+1)} \right]$$

$$\pm \mathcal{O} \left( \rho^{11} m^{-1/12} + \rho^7 m^{-1/12} + \rho^7 m^{-1/4} + \rho^{11} m^{-1/4} \right) \cdot \left\| \mathbf{V}^{(\ell')} \right\|_{2,\infty}$$

$$\pm \mathcal{O}(\rho^2 L^{5/6} \epsilon_x^{5/3}).$$

Hence, from Eq. 24 and Eq. 25, we have

$$\left\| \mathbf{V}^{(\ell'+1)} \mathbf{h}^{(\ell'+1)} - [\mathbf{x}^{(1)}, \cdots, \mathbf{x}^{(\ell'+1)}] \right\|_\infty$$

$$\leq \left\| \left[ \; \left[ \mathbf{V}^{(\ell')}, \, \mathbf{0}_{\ell' d \times d} \right]_r, \; [\mathbf{0}_{d \times m}, \, \mathbf{I}_{d \times d}]_r \; \right]_c [\mathbf{h}^{(\ell')}, \mathbf{x}^{(\ell'+1)}] - \mathbf{V}^{(\ell'+1)} \mathbf{h}^{(\ell'+1)} \right\|_\infty$$

$$+ \left\| \mathbf{V}^{(\ell')} \mathbf{h}^{(\ell')} - [\mathbf{x}^{(1)}, \cdots, \mathbf{x}^{(\ell')}] \right\|_\infty$$

$$\leq \mathcal{O} \left( \rho^{11} m^{-1/12} + \rho^7 m^{-1/12} + \rho^7 m^{-1/4} + \rho^{11} m^{-1/4} \right) \cdot \sum_{k < \ell'} \left\| \mathbf{V}^{(k)} \right\|_{2,\infty} + \mathcal{O}(\ell' \rho^2 L^{5/6} \epsilon_x^{5/3})$$

$$+ \mathcal{O} \left( \rho^{11} m^{-1/12} + \rho^7 m^{-1/12} + \rho^7 m^{-1/4} + \rho^{11} m^{-1/4} \right) \cdot \left\| \mathbf{V}^{(\ell')} \right\|_{2,\infty} + \mathcal{O}(L^{5/6} \rho^2 \epsilon_x^{5/3})$$

$$= \mathcal{O} \left( \rho^{11} m^{-1/12} + \rho^7 m^{-1/12} + \rho^7 m^{-1/4} + \rho^{11} m^{-1/4} \right) \cdot \sum_{k \leq \ell'} \left\| \mathbf{V}^{(k)} \right\|_{2,\infty} + \mathcal{O}((\ell' + 1) \rho^2 L^{5/6} \epsilon_x^{5/3}).$$

Thus, the claim follows by induction.

$$\square$$

**Claim E.7** (Restating claim E.3).

$$\mathbf{V}^{(\ell)} = \overline{\mathbf{W}}^{[\ell]\top}, \quad \textit{for all } \ell \in [L].$$

*Proof.* We prove the claim by induction. For $\ell = 1$,

$$\mathbf{V}^{(1)} = [\mathbf{0}_{d \times m}, \, \mathbf{I}_{d \times d}]_r \; [\mathbf{W}, \, \mathbf{A}]_r^\top \mathbf{D}_{(0)}^{(1)} = \mathbf{A}^\top \mathbf{D}_{(0)}^{(1)} := \overline{\mathbf{W}}^{[1]\top}.$$

Assuming the claim holds true for for all $\ell \leq \ell'$, we now prove for $\ell = \ell' + 1$.

$$
\begin{aligned}
\mathbf{V}^{(\ell'+1)} &= \left[ \ \left[ \mathbf{V}^{(\ell')}, \ \mathbf{0}_{(\ell')d \times d} \right]_r, \ \left[ \mathbf{0}_{d \times m}, \ \mathbf{I}_{d \times d} \right]_r \ \right]_c \ \left[ \mathbf{W}, \ \mathbf{A} \right]_r^\top \mathbf{D}_{(0)}^{(\ell'+1)} \\
&= \left[ \ \left[ \mathbf{V}^{(\ell')}, \ \mathbf{0}_{(\ell')d \times d} \right]_r \left[ \mathbf{W}, \ \mathbf{A} \right]_r^\top \mathbf{D}_{(0)}^{(\ell'+1)}, \ \left[ \mathbf{0}_{d \times m}, \ \mathbf{I}_{d \times d} \right]_r \ \left[ \mathbf{W}, \ \mathbf{A} \right]_r^\top \mathbf{D}_{(0)}^{(\ell'+1)} \right]_c \\
&= \left[ \ \mathbf{V}^{(\ell')} \mathbf{W}^\top \mathbf{D}_{(0)}^{(\ell'+1)}, \ \mathbf{A}^\top \mathbf{D}_{(0)}^{(\ell'+1)} \right]_c \\
&= \left[ \ \overline{\mathbf{W}}^{[\ell']\top} \mathbf{W}^\top \mathbf{D}_{(0)}^{(\ell'+1)}, \ \mathbf{A}^\top \mathbf{D}_{(0)}^{(\ell'+1)} \right]_c := \overline{\mathbf{W}}^{[\ell'+1]\top},
\end{aligned}
$$

where in the pre-final step, we use induction argument for $\ell = \ell'$. Hence, the claim follows from induction. $\qquad\square$

**Claim E.8.** *With probability* $1 - e^{-\Omega(\rho^2)}$, *for all* $\ell \in [L]$,

$$
\left\| \mathbf{W} \mathbf{V}^{(\ell)\top} \right\|_{\infty,\infty} \leq \mathcal{O}(\frac{\rho}{\sqrt{m}}).
$$

*Proof.* Fix an $\ell \in [L]$. From claim E.3, we have

$$
\mathbf{V}^{(\ell)\top} = \overline{\mathbf{W}}^{[\ell]}.
$$

Thus,

$$
\begin{aligned}
\left\| \mathbf{W} \mathbf{V}^{(\ell)\top} \right\|_{\infty,\infty} &= \left\| \mathbf{W} \overline{\mathbf{W}}^{[\ell]} \right\|_{\infty,\infty} \\
&\leq \max_{k \leq \ell} \left\| \mathbf{W} \mathbf{W}^{(\ell,k+1)} \right\|_{\infty,\infty} \\
&= \max_{k \leq \ell} \left\| \mathbf{W} \mathbf{D}^{(\ell)} \mathbf{W} \cdots \mathbf{D}^{(k+1)} \mathbf{W} \mathbf{D}^{(k)} \mathbf{A} \right\|_{\infty,\infty} \\
&\leq \mathcal{O}(\frac{\rho}{\sqrt{m}}),
\end{aligned}
$$

with probability at least $1 - e^{-\Omega(\rho^2)}$. Here, we use lemma B.1 in the final step that says

$$
\left\| \mathbf{W} \mathbf{D}^{(j)} \mathbf{W} \cdots \mathbf{D}^{(i+1)} \mathbf{W} \mathbf{D}^{(i)} \mathbf{A} \right\|_{\infty,\infty} \leq \mathcal{O}(\frac{\rho}{\sqrt{m}}),
$$

with high probability for any $1 \leq i \leq j \leq L$. $\qquad\square$

**Claim E.9** (Restating claim E.4)**.**

$$
\left\| \overline{\mathbf{W}}^{[\ell]\top} \right\|_{2,\infty} \leq \mathcal{O}(L^3), \quad \text{for all } \ell \in [L],
$$

*with probability at least* $1 - 2e^{-\rho^2}$.

*Proof.* Since $\overline{\mathbf{W}}^{[\ell]} = [\mathbf{W}^{(\ell)} \mathbf{A}, \mathbf{W}^{(\ell-1)} \mathbf{A}, \cdots, \mathbf{A}]_r$,

$$
\begin{aligned}
\left\| \overline{\mathbf{W}}^{[\ell]\top} \right\|_{2,\infty} &= \max_{k \leq \ell} \left\| \left( \mathbf{W}^{(\ell,k+1)} \mathbf{D}_{(0)}^{(k)} \mathbf{A} \right)^\top \right\|_{2,\infty} \\
&\leq \max_{k \leq \ell} \left\| \mathbf{W}^{(\ell,k+1)} \mathbf{D}_{(0)}^{(k)} \mathbf{A} \right\|_2 \\
&\leq \max_{k \leq \ell} \left\| \mathbf{W}^{(\ell,k+1)} \right\|_2 \left\| \mathbf{D}_{(0)}^{(k)} \right\|_2 \|\mathbf{A}\|_2 \\
&\leq \max_{k \leq \ell} \left\| \mathbf{W}^{(\ell,k+1)} \right\| \|\mathbf{A}\|_2.
\end{aligned}
$$

From Fact A.1, we can show that with probability exceeding $1 - e^{-\rho^2}$,
$$\|\mathbf{A}\| \leq \sqrt{2}(1 + \sqrt{dm^{-1}} + \sqrt{2}\rho m^{-0.5}) \leq 5,$$
provided $m \geq d\rho^2$. Also, from Lemma B.1, we have for any $k \leq L$, w.p. at least $1 - e^{-\Omega(\rho^2)}$,
$$\left\|\mathbf{W}^{(\ell,k+1)}\right\| = \left\|\mathbf{D}^{(\ell)}\mathbf{W}\mathbf{D}^{(\ell-1)}\mathbf{W}\cdots\mathbf{D}^{(k)}\mathbf{W}\right\| \leq \mathcal{O}(L^3).$$
Hence,
$$\left\|\overline{\mathbf{W}}^{[\ell]\top}\right\|_{2,\infty} \leq \max_{k\leq\ell}\left\|\mathbf{W}^{(\ell,k+1)}\right\|\|\mathbf{A}\|_2 \leq 5 \cdot \mathcal{O}(L^3) = \mathcal{O}(L^3).$$
$\square$

**Claim E.10.** *Choose a random subset $\mathcal{K} \subset [m]$ of size $|\mathcal{K}| = N$. Replace the rows $\{\mathbf{w}_k, \mathbf{a}_k\}_{k\in\mathcal{K}}$ of $\mathbf{W}$ and $\mathbf{A}$ with freshly new i.i.d. samples $\widetilde{\mathbf{w}}_k, \widetilde{\mathbf{a}}_k \sim \mathcal{N}\left(0, \frac{2}{m}\mathbf{I}\right)$ to form new matrices $\widetilde{\mathbf{W}}$ and $\widetilde{\mathbf{A}}$. Define $\widetilde{\mathbf{V}}^{(\ell)}$ inductively as follows:*
$$\widetilde{\mathbf{V}}^{(1)} = [\mathbf{0}_{d\times m}, \ \mathbf{I}_{d\times d}]_r \ \left[\widetilde{\mathbf{W}}, \ \widetilde{\mathbf{A}}\right]_r^\top \ \widetilde{\mathbf{D}}_{(0)}^{(1)}$$
$$\widetilde{\mathbf{V}}^{(\ell)} = \left[ \ \left[\widetilde{\mathbf{V}}^{(\ell-1)}, \ \mathbf{0}_{(\ell-1)d\times d}\right]_r, \ [\mathbf{0}_{d\times m}, \ \mathbf{I}_{d\times d}]_r \ \right]_c \ \left[\widetilde{\mathbf{W}}, \ \widetilde{\mathbf{A}}\right]_r^\top \ \widetilde{\mathbf{D}}_{(0)}^{(\ell)}$$
*Then, with probability $1 - e^{-\Omega(\rho^2)}$, for all $\ell \geq 2$,*

- $$\left\|\left[ \ \left[\widetilde{\mathbf{V}}^{(\ell)}, \ \mathbf{0}_{\ell d\times d}\right]_r, \ [\mathbf{0}_{d\times m}, \ \mathbf{I}_{d\times d}]_r \ \right]_c - \left[ \ \left[\mathbf{V}^{(\ell)}, \ \mathbf{0}_{\ell d\times d}\right]_r, \ [\mathbf{0}_{d\times m}, \ \mathbf{I}_{d\times d}]_r \ \right]_c\right\|_{2,\infty} \leq$$
$\mathcal{O}(\rho^6 (N/m)^{1/6})$.

- $$\left\|\left( \left[ \ \left[\widetilde{\mathbf{V}}^{(\ell)}, \ \mathbf{0}_{\ell d\times d}\right]_r, \ [\mathbf{0}_{d\times m}, \ \mathbf{I}_{d\times d}]_r \ \right]_c - \left[ \ \left[\mathbf{V}^{(\ell)}, \ \mathbf{0}_{\ell d\times d}\right]_r, \ [\mathbf{0}_{d\times m}, \ \mathbf{I}_{d\times d}]_r \ \right]_c\right)[\mathbf{W}_\mathcal{K}, \mathbf{A}_\mathcal{K}]_r^\top\right\|_{2,\infty} \leq$$
$\mathcal{O}(\rho^6 (N/m)^{2/3})$.

*Proof.* Let $\widetilde{\mathbf{W}}^{(k)} = \prod_{1\leq\ell\leq k+1} \widetilde{\mathbf{D}}_{(0)}^{(k-\ell+1)}\widetilde{\mathbf{W}}$. Let $\overline{\widetilde{\mathbf{W}}}^{[L]} = [\widetilde{\mathbf{W}}^{(\ell-1)}\widetilde{\mathbf{D}}_{(0)}^{(1)}\widetilde{\mathbf{A}}, \widetilde{\mathbf{W}}^{(\ell-2)}\widetilde{\mathbf{D}}_{(0)}^{(2)}\widetilde{\mathbf{A}}, \cdots, \widetilde{\mathbf{D}}_{(0)}^{(\ell)}\widetilde{\mathbf{A}}]_r$. Then, using the same induction technique as in Claim E.3, we can show that
$$\widetilde{\mathbf{V}}^{(\ell)} = \overline{\widetilde{\mathbf{W}}}^{[\ell]\top}, \quad \text{for all } \ell \in [L].$$
Hence,
$$\left\|\left[ \ \left[\widetilde{\mathbf{V}}^{(\ell)}, \ \mathbf{0}_{(\ell-1)d\times d}\right]_r, \ [\mathbf{0}_{d\times m}, \ \mathbf{I}_{d\times d}]_r \ \right]_c - \left[ \ \left[\mathbf{V}^{(\ell)}, \ \mathbf{0}_{(\ell-1)d\times d}\right]_r, \ [\mathbf{0}_{d\times m}, \ \mathbf{I}_{d\times d}]_r \ \right]_c\right\|_{2,\infty}$$

$$= \left\|\widetilde{\mathbf{V}}^{(\ell)} - \mathbf{V}^{(\ell)}\right\|_{2,\infty} = \left\|\overline{\mathbf{W}}^{[\ell]\top} - \overline{\widetilde{\mathbf{W}}}^{[\ell]\top}\right\|_{2,\infty}$$

$$\leq \max_{k\leq\ell}\left\|\left(\widetilde{\mathbf{W}}^{(\ell,k+1)}\widetilde{\mathbf{D}}_{(0)}^{(k)}\widetilde{\mathbf{A}} - \mathbf{W}^{(\ell,k+1)}\mathbf{D}_{(0)}^{(k)}\mathbf{A}\right)^\top\right\|_{2,\infty}$$

$$\leq \max_{k\leq\ell}\left\|\left(\widetilde{\mathbf{W}}^{(\ell,k+1)}\widetilde{\mathbf{D}}_{(0)}^{(k)}\widetilde{\mathbf{A}} - \mathbf{W}^{(\ell,k+1)}\mathbf{D}_{(0)}^{(k)}\mathbf{A}\right)^\top\right\|_{2,\infty}$$

$$= \max_{k\leq\ell}\left\|\left(\widetilde{\mathbf{W}}^{(\ell,k+1)}\widetilde{\mathbf{D}}_{(0)}^{(\ell,k+1)}\widetilde{\mathbf{A}} - \mathbf{W}^{(\ell,k+1)}\widetilde{\mathbf{D}}_{(0)}^{(k)}\widetilde{\mathbf{A}}\right)^\top + \left(\mathbf{W}^{(\ell,k+1)}\widetilde{\mathbf{D}}_{(0)}^{(k)}\widetilde{\mathbf{A}} - \mathbf{W}^{(\ell,k+1)}\mathbf{D}_{(0)}^{(k)}\mathbf{A}\right)^\top\right\|_{2,\infty}$$

$$\leq \max_{k\leq\ell}\left\|\left(\widetilde{\mathbf{W}}^{(\ell,k+1)}\widetilde{\mathbf{D}}_{(0)}^{(\ell,k+1)}\widetilde{\mathbf{A}} - \mathbf{W}^{(\ell,k+1)}\widetilde{\mathbf{D}}_{(0)}^{(k)}\widetilde{\mathbf{A}}\right)^\top\right\|_{2,\infty} + \left\|\left(\mathbf{W}^{(\ell,k+1)}\widetilde{\mathbf{D}}_{(0)}^{(k)}\widetilde{\mathbf{A}} - \mathbf{W}^{(\ell,k+1)}\mathbf{D}_{(0)}^{(k)}\mathbf{A}\right)^\top\right\|_{2,\infty}$$

$$\leq \max_{k\leq\ell}\left\|\left(\widetilde{\mathbf{W}}^{(\ell,k+1)}\widetilde{\mathbf{D}}_{(0)}^{(\ell,k+1)}\widetilde{\mathbf{A}} - \mathbf{W}^{(\ell,k+1)}\widetilde{\mathbf{D}}_{(0)}^{(k)}\widetilde{\mathbf{A}}\right)^\top\right\|_{2,\infty} + \left\|\left(\mathbf{W}^{(\ell,k+1)}\widetilde{\mathbf{D}}_{(0)}^{(k)}\widetilde{\mathbf{A}} - \mathbf{W}^{(\ell,k+1)}\mathbf{D}_{(0)}^{(k)}\mathbf{A}\right)^\top\right\|_2.$$

We will bound the two terms separately. From Fact A.1, we can show that with probability exceeding $1 - e^{-\rho^2}$,

$$\left\|\widetilde{\mathbf{A}}\right\| \leq \sqrt{2/m}(\sqrt{m} + \sqrt{d} + \sqrt{2}\rho) \leq 5, \tag{30}$$

provided $m \geq d\rho^2$. Using the above bound, we get the following for all $k \in [\ell]$:

$$
\begin{aligned}
\left\|\left(\widetilde{\mathbf{W}}^{(\ell,k+1)}\widetilde{\mathbf{D}}_{(0)}^{(k)}\widetilde{\mathbf{A}} - \mathbf{W}^{(\ell,k+1)}\widetilde{\mathbf{D}}_{(0)}^{(k)}\widetilde{\mathbf{A}}\right)^{\top}\right\|_{2,\infty} &\leq \max_{i\in[d]}\left\|\left(\widetilde{\mathbf{W}}^{(\ell,k+1)} - \mathbf{W}^{(\ell,k+1)}\right)\left(\widetilde{\mathbf{D}}_{(0)}^{(k)}\widetilde{\mathbf{A}}\right)_i^{\top}\right\|_2 \\
&\leq \max_{i\in[d]}\mathcal{O}(\rho^5(N/m)^{1/6})\cdot\left\|\left(\widetilde{\mathbf{D}}_{(0)}^{(k)}\widetilde{\mathbf{A}}\right)_i^{\top}\right\|_2 \\
&\leq \mathcal{O}(\rho^5(N/m)^{1/6}),
\end{aligned}
$$

where in the second-final step we have used Lemma B.2 to have

$$\left\|\left(\widetilde{\mathbf{W}}^{(\ell,k+1)} - \mathbf{W}^{(\ell,k+1)}\right)\mathbf{v}\right\|_2 = \left\|\left(\prod_{\ell\geq\ell'\geq k+1}\widetilde{\mathbf{D}}_{(0)}^{(\ell')}\widetilde{\mathbf{W}} - \prod_{\ell\geq\ell'\geq k+1}\mathbf{D}_{(0)}^{(\ell')}\mathbf{W}\right)\mathbf{v}\right\|_2 \leq \mathcal{O}(\rho^5(N/m)^{1/6})\|\mathbf{v}\|,$$

for the vectors $\mathbf{v} \in \left\{\left(\widetilde{\mathbf{D}}_{(0)}^{(k)}\widetilde{\mathbf{A}}\right)_i^{\top}\right\}_{i\in[d]}$. From Lemma B.1, we have for any $k < \ell$, w.p. at least $1 - e^{-\Omega(\rho^2)}$,

$$\left\|\mathbf{W}^{(\ell,k+1)}\right\| = \left\|\mathbf{D}^{(\ell)}\mathbf{W}\mathbf{D}^{(\ell-1)}\mathbf{W}\cdots\mathbf{D}^{(k+1)}\mathbf{W}\right\| \leq \mathcal{O}(L^3). \tag{31}$$

Also, we have used the following fact:

$$
\begin{aligned}
\max_{i\in[d]}\left\|\left(\widetilde{\mathbf{D}}_{(0)}^{(k)}\widetilde{\mathbf{A}}\right)_i^{\top}\right\|_2 &\leq \left\|\left(\widetilde{\mathbf{D}}_{(0)}^{(k)}\widetilde{\mathbf{A}}\right)^{\top}\right\|_2 \\
&\leq \left\|\left(\widetilde{\mathbf{D}}_{(0)}^{(k)}\widetilde{\mathbf{A}}\right)^{\top}\right\|_2
\end{aligned}
$$

Again since only $N$ rows of $\mathbf{A}$ and $\widetilde{\mathbf{A}}$ are different, we can show from Fact A.1 that with probability exceeding $1 - e^{-\Omega(\rho^2)}$,

$$\left\|\mathbf{A} - \widetilde{\mathbf{A}}\right\| \leq \sqrt{2/m}(\sqrt{N} + \sqrt{d} + \sqrt{2}\rho) \leq \mathcal{O}((\rho + \sqrt{d})\sqrt{N/m}) \leq \mathcal{O}(\rho\sqrt{N/m}), \tag{32}$$

since we are using $\rho = 100Ld\log m$. Also from Lemma B.2, we have that with probability exceeding $1 - e^{-\Omega(\rho^2)}$,

$$\left\|\widetilde{\mathbf{D}}_{(0)}^{(k)} - \mathbf{D}_{(0)}^{(k)}\right\|_0 \leq \mathcal{O}(\rho^4 N^{1/3}m^{2/3}).$$

Thus, again we can use Fact A.1 to show that with probability exceeding $1 - 2e^{-\rho^2}$,

$$
\begin{aligned}
\left\|\left(\widetilde{\mathbf{D}}_{(0)}^{(k)} - \mathbf{D}_{(0)}^{(k)}\right)\mathbf{A}\right\|_2 &\leq \sqrt{2/m}(\sqrt{d} + \sqrt{\left\|\widetilde{\mathbf{D}}_{(0)}^{(k)} - \mathbf{D}_{(0)}^{(k)}\right\|_0} + \sqrt{2}\rho) \\
&\leq \sqrt{2/m}(\sqrt{d} + \sqrt{\mathcal{O}(\rho^4 N^{1/3}m^{2/3})} + \sqrt{2}\rho) \\
&\leq \sqrt{2/m}\mathcal{O}((\sqrt{d} + \sqrt{2}\rho)\sqrt{\mathcal{O}(\rho^4 N^{1/3}m^{2/3})}) \\
&\leq \mathcal{O}(\rho^3 N^{1/6}m^{-1/6}), 
\end{aligned} \tag{33}
$$

where again we are using $\rho = 100Ld\log m$. Thus,

$$
\begin{aligned}
&\left\|\mathbf{W}^{(\ell,k+1)}\widetilde{\mathbf{D}}_{(0)}^{(k)}\widetilde{\mathbf{A}} - \mathbf{W}^{(\ell,k+1)}\mathbf{D}_{(0)}^{(k)}\mathbf{A}\right\|_2 \\
&= \left\|\mathbf{W}^{(\ell,k+1)}\right\|_2\left\|\widetilde{\mathbf{D}}_{(0)}^{(k)}\widetilde{\mathbf{A}} - \mathbf{D}_{(0)}^{(k)}\mathbf{A}\right\|_2
\end{aligned}
$$

$$
\begin{aligned}
&= \left\| \mathbf{W}^{(\ell,k+1)} \right\|_2 \left\| \widetilde{\mathbf{D}}^{(k)}_{(0)} \widetilde{\mathbf{A}} - \mathbf{D}^{(k)}_{(0)} \widetilde{\mathbf{A}} + \mathbf{D}^{(k)}_{(0)} \widetilde{\mathbf{A}} - \mathbf{D}^{(k)}_{(0)} \mathbf{A} \right\|_2 \\
&\leq \left\| \mathbf{W}^{(\ell,k+1)} \right\|_2 \left( \left\| \widetilde{\mathbf{D}}^{(k)}_{(0)} \widetilde{\mathbf{A}} - \mathbf{D}^{(k)}_{(0)} \widetilde{\mathbf{A}} \right\|_2 + \left\| \mathbf{D}^{(k)}_{(0)} \widetilde{\mathbf{A}} - \mathbf{D}^{(k)}_{(0)} \mathbf{A} \right\|_2 \right) \\
&\leq \left\| \mathbf{W}^{(\ell,k+1)} \right\|_2 \left( \left\| \widetilde{\mathbf{D}}^{(k)}_{(0)} \widetilde{\mathbf{A}} - \mathbf{D}^{(k)}_{(0)} \widetilde{\mathbf{A}} \right\|_2 + \left\| \widetilde{\mathbf{A}} - \mathbf{A} \right\|_2 \right) \\
&\leq \mathcal{O}(L^3) \cdot \left( \mathcal{O}(\rho^3 (N/m)^{1/6}) + \mathcal{O}(\rho (N/m)^{1/2}) \right) \leq \mathcal{O}(\rho^6 (N/m)^{1/6}), \qquad (34)
\end{aligned}
$$

since we are using $\rho = 100 L d \log m$. Hence, we have

$$
\begin{aligned}
\left\| \overline{\mathbf{W}}^{[\ell]\top} - \widetilde{\overline{\mathbf{W}}}^{[\ell]\top} \right\|_{2,\infty} &\leq \max_{k \leq \ell} \left\| \left( \widetilde{\mathbf{W}}^{(\ell,k+1)} \widetilde{\mathbf{D}}^{(k)}_{(0)} \widetilde{\mathbf{A}} - \mathbf{W}^{(\ell,k+1)} \widetilde{\mathbf{D}}^{(k)}_{(0)} \widetilde{\mathbf{A}} \right)^{\top} \right\|_{2,\infty} + \left\| \mathbf{W}^{(\ell,k+1)} \widetilde{\mathbf{D}}^{(k)}_{(0)} \widetilde{\mathbf{A}} - \mathbf{W}^{(\ell,k+1)} \mathbf{D}^{(k)}_{(0)} \mathbf{A} \right\|_2 \\
&\leq \mathcal{O}(\rho^6 (N/m)^{1/6}) + \mathcal{O}(\rho^5 N^{1/6} m^{-1/6}) = \mathcal{O}(\rho^6 (N/m)^{1/6}),
\end{aligned}
$$

which gives the first result. Now, we focus on

$$
\left\| \left( \left[ \; \left[ \widetilde{\mathbf{V}}^{(\ell)}, \; \mathbf{0}_{\ell d \times d} \right]_r, \; \left[ \mathbf{0}_{d \times m}, \; \mathbf{I}_{d \times d} \right]_r \; \right]_c - \left[ \; \left[ \mathbf{V}^{(\ell)}, \; \mathbf{0}_{\ell d \times d} \right]_r, \; \left[ \mathbf{0}_{d \times m}, \; \mathbf{I}_{d \times d} \right]_r \; \right]_c \right) [\mathbf{W}_{\mathcal{K}}, \mathbf{A}_{\mathcal{K}}]_r^{\top} \right\|_{2,\infty} .
$$

Note that the above is equivalent to

$$
\left\| \widetilde{\mathbf{W}}^{[\ell]\top} \mathbf{W}_{\mathcal{K}}^{\top} - \mathbf{W}^{[\ell]\top} \mathbf{W}_{\mathcal{K}}^{\top} \right\|_{2,\infty} ,
$$

using the relation between $\widetilde{\mathbf{V}}^{(\ell)}$ and $\widetilde{\mathbf{W}}^{[\ell]\top}$, and $\mathbf{V}^{(\ell)}$ and $\mathbf{W}^{[\ell]\top}$. Continuing

$$
\begin{aligned}
&\left\| \widetilde{\mathbf{W}}^{[\ell]\top} \mathbf{W}_{\mathcal{K}}^{\top} - \mathbf{W}^{[\ell]\top} \mathbf{W}_{\mathcal{K}}^{\top} \right\|_{2,\infty} \\
&\leq \max_{k \leq \ell} \left\| \left( \mathbf{W}_{\mathcal{K}} \widetilde{\mathbf{W}}^{(\ell,k+1)} \widetilde{\mathbf{D}}^{(k)}_{(0)} \widetilde{\mathbf{A}} - \mathbf{W}_{\mathcal{K}} \mathbf{W}^{(\ell,k+1)} \mathbf{D}^{(k)}_{(0)} \mathbf{A} \right)^{\top} \right\|_{2,\infty} \\
&= \max_{k \leq \ell} \left\| \left( \mathbf{W}_{\mathcal{K}} \widetilde{\mathbf{W}}^{(\ell,k+1)} \widetilde{\mathbf{D}}^{(k)}_{(0)} \widetilde{\mathbf{A}} - \mathbf{W}_{\mathcal{K}} \mathbf{W}^{(\ell,k+1)} \widetilde{\mathbf{D}}^{(k)}_{(0)} \widetilde{\mathbf{A}} + \mathbf{W}_{\mathcal{K}} \mathbf{W}^{(\ell,k+1)} \widetilde{\mathbf{D}}^{(k)}_{(0)} \widetilde{\mathbf{A}} - \mathbf{W}_{\mathcal{K}} \mathbf{W}^{(\ell,k+1)} \mathbf{D}^{(k)}_{(0)} \mathbf{A} \right)^{\top} \right\|_{2,\infty} \\
&\leq \max_{k \leq \ell} \left\| \left( \mathbf{W}_{\mathcal{K}} \widetilde{\mathbf{W}}^{(\ell,k+1)} \widetilde{\mathbf{D}}^{(k)}_{(0)} \widetilde{\mathbf{A}} - \mathbf{W}_{\mathcal{K}} \mathbf{W}^{(\ell,k+1)} \widetilde{\mathbf{D}}^{(k)}_{(0)} \widetilde{\mathbf{A}} \right)^{\top} \right\|_{2,\infty} + \left\| \left( \mathbf{W}_{\mathcal{K}} \mathbf{W}^{(\ell,k+1)} \widetilde{\mathbf{D}}^{(k)}_{(0)} \widetilde{\mathbf{A}} - \mathbf{W}_{\mathcal{K}} \mathbf{W}^{(\ell,k+1)} \mathbf{D}^{(k)}_{(0)} \mathbf{A} \right)^{\top} \right\|_{2,\infty} \\
&\leq \max_{k \leq \ell} \left\| \left( \mathbf{W}_{\mathcal{K}} \widetilde{\mathbf{W}}^{(\ell,k+1)} \widetilde{\mathbf{D}}^{(k)}_{(0)} \widetilde{\mathbf{A}} - \mathbf{W}_{\mathcal{K}} \mathbf{W}^{(\ell,k+1)} \widetilde{\mathbf{D}}^{(k)}_{(0)} \widetilde{\mathbf{A}} \right)^{\top} \right\|_{2,\infty} + \left\| \mathbf{W}_{\mathcal{K}} \mathbf{W}^{(\ell,k+1)} \left( \widetilde{\mathbf{D}}^{(k)}_{(0)} \widetilde{\mathbf{A}} - \mathbf{D}^{(k)}_{(0)} \mathbf{A} \right) \right\|_2
\end{aligned}
$$
$$(35)$$

Using lemma B.2, we have with probability $1 - e^{-\Omega(\rho^2)}$, we have for all $k \leq \ell$

$$
\left\| \left( \mathbf{W}_{\mathcal{K}} \widetilde{\mathbf{W}}^{(\ell,k+1)} - \mathbf{W}_{\mathcal{K}} \mathbf{W}^{(\ell,k+1)} \right) \mathbf{v} \right\|_2 = \left\| \mathbf{W}_{\mathcal{K}} \left( \prod_{\ell \geq \ell' \geq k+1} \widetilde{\mathbf{D}}^{(\ell')} \widetilde{\mathbf{W}} - \prod_{\ell \geq \ell' \geq k+1} \mathbf{D}^{(\ell')} \mathbf{W} \right) \mathbf{v} \right\|_2
$$
$$
\leq \mathcal{O}(\rho^6 (N/m)^{2/3}) \|\mathbf{v}\| ,
$$

for any fixed vector $\mathbf{v}$. We will use union bound to make sure the above property is satisfied for all vector $\mathbf{v}$ in the set $\left\{ \left( \widetilde{\mathbf{D}}^{(k)}_{(0)} \widetilde{\mathbf{A}} \right)^{\top}_i \right\}_{i \in [d]}$. From Eq. 30, with probability $1 - e^{-\Omega(\rho^2)}$

$$
\left\| \widetilde{\mathbf{A}} \right\| \leq \sqrt{2} (1 + \sqrt{d m^{-1}} + \sqrt{2} \rho m^{-0.5}) \leq 5,
$$

provided $m \geq d\rho^2$. Thus, the first term in Eq. 35 can be bounded as

$$\max_{k \leq \ell} \left\| \left( \mathbf{W}_{\mathcal{K}} \widetilde{\mathbf{W}}^{(\ell,k+1)} \widetilde{\mathbf{D}}_{(0)}^{(k)} \widetilde{\mathbf{A}} - \mathbf{W}_{\mathcal{K}} \mathbf{W}^{(\ell,k+1)} \widetilde{\mathbf{D}}_{(0)}^{(k)} \widetilde{\mathbf{A}} \right)^{\top} \right\|_{2,\infty}$$

$$= \max_{k \leq \ell} \max_{i \in [d]} \left\| \left( \mathbf{W}_{\mathcal{K}} \widetilde{\mathbf{W}}^{(\ell,k+1)} \widetilde{\mathbf{D}}_{(0)}^{(k)} - \mathbf{W}_{\mathcal{K}} \mathbf{W}^{(\ell,k+1)} \right) \left( \widetilde{\mathbf{D}}_{(0)}^{(k)} \widetilde{\mathbf{A}} \right)_{i} \right\|_{2}$$

$$\leq \max_{k \leq \ell} \max_{i \in [d]} \mathcal{O}(\rho^6 (N/m)^{2/3}) \cdot \left\| \left( \widetilde{\mathbf{D}}_{(0)}^{(k)} \widetilde{\mathbf{A}} \right)_{i} \right\|$$

$$\leq \max_{k \leq \ell} \mathcal{O}(\rho^6 (N/m)^{2/3}) \cdot \left\| \widetilde{\mathbf{D}}_{(0)}^{(k)} \widetilde{\mathbf{A}} \right\|_{2}$$

$$\leq \mathcal{O}(\rho^6 (N/m)^{2/3}).$$

Now, we focus on the second term in Eq. 35. We have for any $k \leq \ell$,

$$\left\| \mathbf{W}_{\mathcal{K}} \mathbf{W}^{(\ell,k+1)} \left( \widetilde{\mathbf{D}}_{(0)}^{(k)} \widetilde{\mathbf{A}} - \mathbf{D}_{(0)}^{(k)} \mathbf{A} \right) \right\|_{2}$$

$$= \left\| \mathbf{W}_{\mathcal{K}} \mathbf{W}^{(\ell,k+1)} \left( \widetilde{\mathbf{D}}_{(0)}^{(k)} \widetilde{\mathbf{A}} - \mathbf{D}_{(0)}^{(k)} \widetilde{\mathbf{A}} + \mathbf{D}_{(0)}^{(k)} \widetilde{\mathbf{A}} - \mathbf{D}_{(0)}^{(k)} \mathbf{A} \right) \right\|_{2}$$

$$\leq \left\| \mathbf{W}_{\mathcal{K}} \mathbf{W}^{(\ell,k+1)} \left( \widetilde{\mathbf{D}}_{(0)}^{(k)} \widetilde{\mathbf{A}} - \mathbf{D}_{(0)}^{(k)} \widetilde{\mathbf{A}} \right) \right\|_{2} + \left\| \mathbf{W}_{\mathcal{K}} \mathbf{W}^{(\ell,k+1)} \left( \mathbf{D}_{(0)}^{(k)} \widetilde{\mathbf{A}} - \mathbf{D}_{(0)}^{(k)} \mathbf{A} \right) \right\|_{2}$$

$$\leq \left\| \mathbf{W}_{\mathcal{K}} \mathbf{W}^{(\ell,k+1)} \left( \widetilde{\mathbf{D}}_{(0)}^{(k)} - \mathbf{D}_{(0)}^{(k)} \right) \right\|_{2} \left\| \left( \widetilde{\mathbf{D}}_{(0)}^{(k)} - \mathbf{D}_{(0)}^{(k)} \right) \widetilde{\mathbf{A}} \right\|_{2} + \left\| \mathbf{W}_{\mathcal{K}} \mathbf{W}^{(\ell,k+1)} \left( \mathbf{D}_{(0)}^{(k)} \widetilde{\mathbf{A}} - \mathbf{D}_{(0)}^{(k)} \mathbf{A} \right) \right\|_{2}$$

$$= \left\| \mathbf{D}_{\mathcal{K}} \mathbf{W} \mathbf{W}^{(\ell,k+1)} \left( \widetilde{\mathbf{D}}_{(0)}^{(k)} - \mathbf{D}_{(0)}^{(k)} \right) \right\|_{2} \left\| \left( \widetilde{\mathbf{D}}_{(0)}^{(k)} - \mathbf{D}_{(0)}^{(k)} \right) \widetilde{\mathbf{A}} \right\|_{2} + \left\| \mathbf{D}_{\mathcal{K}} \mathbf{W} \mathbf{W}^{(\ell,k+1)} \mathbf{D}_{(0)}^{(k)} \mathbf{D}_{\mathcal{K}} \left( \widetilde{\mathbf{A}} - \mathbf{A} \right) \right\|_{2}$$

$$\leq \left\| \mathbf{D}_{\mathcal{K}} \mathbf{W} \mathbf{W}^{(\ell,k+1)} \left( \widetilde{\mathbf{D}}_{(0)}^{(k)} - \mathbf{D}_{(0)}^{(k)} \right) \right\|_{2} \left\| \left( \widetilde{\mathbf{D}}_{(0)}^{(k)} - \mathbf{D}_{(0)}^{(k)} \right) \widetilde{\mathbf{A}} \right\|_{2} + \left\| \mathbf{D}_{\mathcal{K}} \mathbf{W} \mathbf{W}^{(\ell,k+1)} \mathbf{D}_{(0)}^{(k)} \mathbf{D}_{\mathcal{K}} \right\|_{2} \left\| \widetilde{\mathbf{A}} - \mathbf{A} \right\|_{2}$$

Since, $\|D_{\mathcal{K}}\|_0 = N$ and $\left\| \widetilde{\mathbf{D}}_{(0)}^{(k)} - \mathbf{D}_{(0)}^{(k)} \right\|_0 \leq \mathcal{O}(\rho^4 N^{1/3} m^{2/3})$ with probability exceeding $1 - e^{-\Omega(\rho^2)}$ from Lemma B.2, we can use Lemma B.1 to get

$$\left\| \mathbf{W}_{\mathcal{K}} \mathbf{W}^{(\ell,k+1)} \left( \widetilde{\mathbf{D}}_{(0)}^{(k)} \widetilde{\mathbf{A}} - \mathbf{D}_{(0)}^{(k)} \mathbf{A} \right) \right\|_{2}$$

$$\leq \left\| \mathbf{D}_{\mathcal{K}} \mathbf{W} \mathbf{W}^{(\ell,k+1)} \left( \widetilde{\mathbf{D}}_{(0)}^{(k)} - \mathbf{D}_{(0)}^{(k)} \right) \right\|_{2} \left\| \left( \widetilde{\mathbf{D}}_{(0)}^{(k)} - \mathbf{D}_{(0)}^{(k)} \right) \widetilde{\mathbf{A}} \right\|_{2} + \left\| \mathbf{D}_{\mathcal{K}} \mathbf{W} \mathbf{W}^{(\ell,k+1)} \mathbf{D}_{(0)}^{(k)} \mathbf{D}_{\mathcal{K}} \right\|_{2} \left\| \widetilde{\mathbf{A}} - \mathbf{A} \right\|_{2}$$

$$= \left\| \mathbf{D}_{\mathcal{K}} \mathbf{W} \mathbf{D}^{(\ell)} \mathbf{W} \cdots \mathbf{D}^{(k+1)} \mathbf{W} \left( \widetilde{\mathbf{D}}_{(0)}^{(k)} - \mathbf{D}_{(0)}^{(k)} \right) \right\|_{2} \left\| \left( \widetilde{\mathbf{D}}_{(0)}^{(k)} - \mathbf{D}_{(0)}^{(k)} \right) \widetilde{\mathbf{A}} \right\|_{2}$$

$$\quad + \left\| \mathbf{D}_{\mathcal{K}} \mathbf{W} \mathbf{W} \mathbf{D}^{(\ell)} \mathbf{W} \cdots \mathbf{D}^{(k+1)} \mathbf{W} \mathbf{D}_{(0)}^{(k)} \mathbf{D}_{\mathcal{K}} \right\|_{2} \left\| \widetilde{\mathbf{A}} - \mathbf{A} \right\|_{2}$$

$$\leq \mathcal{O}(\rho (N/m)^{1/2}) \cdot \left( \left\| \left( \widetilde{\mathbf{D}}_{(0)}^{(k)} - \mathbf{D}_{(0)}^{(k)} \right) \widetilde{\mathbf{A}} \right\|_{2} + \left\| \widetilde{\mathbf{A}} - \mathbf{A} \right\|_{2} \right).$$

Further using Eq. 33 and Eq. 32, we have

$$\left\| \mathbf{W}_{\mathcal{K}} \mathbf{W}^{(\ell,k+1)} \left( \widetilde{\mathbf{D}}_{(0)}^{(k)} \widetilde{\mathbf{A}} - \mathbf{D}_{(0)}^{(k)} \mathbf{A} \right) \right\|_{2}$$

$$\leq \mathcal{O}(\rho (N/m)^{1/2}) \cdot \left( \left\| \left( \widetilde{\mathbf{D}}_{(0)}^{(k)} - \mathbf{D}_{(0)}^{(k)} \right) \widetilde{\mathbf{A}} \right\|_{2} + \left\| \widetilde{\mathbf{A}} - \mathbf{A} \right\|_{2} \right)$$

$$\leq \mathcal{O}(\rho (N/m)^{1/2}) \cdot \left( \mathcal{O}(\rho^3 (N/m)^{1/6}) + \mathcal{O}(\rho (N/m)^{1/2}) \right) = \mathcal{O}(\rho^4 (N/m)^{2/3}).$$

Thus, connecting all the bounds above in Eq. 35, we get

$$\left\|\widetilde{\mathbf{W}}^{[\ell]\top}\mathbf{W}_{\mathcal{K}}^{\top} - \mathbf{W}^{[\ell]\top}\mathbf{W}_{\mathcal{K}}^{\top}\right\|_{2,\infty}$$

$$\leq \leq \max_{k\leq\ell}\left\|\left(\mathbf{W}_{\mathcal{K}}\widetilde{\mathbf{W}}^{(\ell,k+1)}\widetilde{\mathbf{D}}_{(0)}^{(k)}\widetilde{\mathbf{A}} - \mathbf{W}_{\mathcal{K}}\mathbf{W}^{(\ell,k+1)}\widetilde{\mathbf{D}}_{(0)}^{(k)}\widetilde{\mathbf{A}}\right)^{\top}\right\|_{2,\infty} + \left\|\mathbf{W}_{\mathcal{K}}\mathbf{W}^{(\ell,k+1)}\left(\widetilde{\mathbf{D}}_{(0)}^{(k)}\widetilde{\mathbf{A}} - \mathbf{D}_{(0)}^{(k)}\mathbf{A}\right)\right\|_{2}$$

$$\leq \mathcal{O}(\rho^6(N/m)^{2/3}) + \mathcal{O}(\rho^4(N/m)^{2/3}) = \mathcal{O}(\rho^6(N/m)^{2/3}). \tag{36}$$

□

# F  Existence of good pseudo network: proofs

## F.1  Proof of theorem D.2

**Definition F.1** (Restating defintion D.2). Define $\mathbf{W}^*$ and $\mathbf{A}^*$ as follows.

$\mathbf{W}^* = 0$

$$\mathbf{a}_r^* = \frac{d_{\text{out}}}{m}\sum_{s\in[d_{\text{out}}]}\sum_{r'\in[p]} b_{r,s}b_{r',s}^{\dagger}H_{r',s}\left(\theta_{r',s}\left(\langle\mathbf{w}_r, \overline{\mathbf{W}}^{[L]}\mathbf{w}_{r',s}^{\dagger}\rangle\right), \sqrt{m/2}a_{r,d}\right)\mathbf{e}_d, \quad \forall r\in[m],$$

where

$$\theta_{r',s} = \frac{\sqrt{m/2}}{\left\|\overline{\mathbf{W}}^{[L]}\mathbf{w}_{r',s}^{\dagger}\right\|},$$

and $\overline{\mathbf{W}}^{[L]} = [\mathbf{W}^{(L,3)}\mathbf{D}_{(0)}^{(2)}\mathbf{A}_{[d-1]}, \cdots, \mathbf{W}^{(L,L)}\mathbf{D}_{(0)}^{(L-1)}\mathbf{A}_{[d-1]}]_r$, where $\mathbf{W}^{(k_b,k_e)} = \prod_{k_b\geq\ell>k_e}\mathbf{D}_{(0)}^{(\ell)}\mathbf{W}$.

Using Lemma B.1, we have with probability at least $1 - e^{-\Omega(\rho^2)}$, for all $\ell\in[L]$ and any vector $\mathbf{u}\in\mathbb{R}^d$

$$\left(1 - \frac{1}{100L}\right)^{L}\|\mathbf{u}\| \leq \left\|\mathbf{W}^{(L,\ell+1)}\mathbf{D}_{(0)}^{(\ell)}\mathbf{A}\mathbf{u}\right\| = \left\|\prod_{L\geq\ell'\geq\ell+1}\mathbf{D}_{(0)}^{(\ell')}\mathbf{W}\mathbf{D}_{(0)}^{(\ell)}\mathbf{A}\mathbf{u}\right\| \leq \left(1 + \frac{1}{100L}\right)^{L}\|\mathbf{u}\|.$$

Since, for any vector $\mathbf{u}\in\mathbb{R}^{Ld}$,

$$\min_{\ell\in[L]}\frac{\left\|\mathbf{W}^{(L,\ell+1)}\mathbf{D}_{(0)}^{(\ell)}\mathbf{A}\mathbf{u}_{\ell d:(\ell+1)d}\right\|}{\left\|\mathbf{u}_{\ell d:(\ell+1)d}\right\|}\|\mathbf{u}\| \leq \left\|\mathbf{W}^{[L]}\mathbf{u}\right\| \leq \max_{\ell\in[L]}\frac{\left\|\mathbf{W}^{(L,\ell+1)}\mathbf{D}_{(0)}^{(\ell)}\mathbf{A}\mathbf{u}_{\ell d:(\ell+1)d}\right\|}{\left\|\mathbf{u}_{\ell d:(\ell+1)d}\right\|}\|\mathbf{u}\|,$$

where $\mathbf{u}_{\ell d:(\ell+1)d}\in\mathbb{R}^{Ld}$ refers to a vector that is equal to the vector $\mathbf{u}$ in the dimensions from $\ell d$ to $(\ell+1)d$ and 0 outside, we have

$$\left(1 - \frac{1}{100L}\right)^{L}\left\|\mathbf{w}_{r',s}^{\dagger}\right\| \leq \left\|\overline{\mathbf{W}}^{[L]}\mathbf{w}_{r',s}^{\dagger}\right\| \leq \left(1 + \frac{1}{100L}\right)^{L}\left\|\mathbf{w}_{r',s}^{\dagger}\right\|$$

and thus we have $\forall r'\in[p], s\in[d_{\text{out}}]$,

$$\left(1 + \frac{1}{100L}\right)^{-L} \leq \sqrt{2/m}\theta_{r',s} = \left(1 - \frac{1}{100L}\right)^{-L}. \tag{37}$$

**Theorem F.1** (Restating theorem D.2). *The construction of $\mathbf{W}^*$ and $\mathbf{A}^*$ in Definition D.2 satisfies the following. For every normalized input sequence $\mathbf{x}^{(1)}, \cdots, \mathbf{x}^{(L)}$, we have with probability at least $1 - e^{-\Omega(\rho^2)}$ over $\mathbf{W}, \mathbf{A}, \mathbf{B}$, it holds for every $s\in[d_{\text{out}}]$.*

$$F_s^{(L)} \stackrel{def}{=} \sum_{i=1}^{L}\mathbf{e}_s^{\top}\mathbf{Back}_{i\to L}D^{(i)}\left(\mathbf{W}^*\mathbf{h}^{(i-1)} + \mathbf{A}^*\mathbf{x}^{(i)}\right)$$

$$= \sum_{r\in[p]}b_{r,s}^{\dagger}\Phi_{r,s}\left(\left\langle\mathbf{w}_{r,s}^{\dagger}, [\overline{\mathbf{x}}^{(2)}, \cdots, \overline{\mathbf{x}}^{(L-2)}]\right\rangle\right)$$

$$\pm\mathcal{O}(d_{\text{out}}Lp\rho^2\varepsilon + d_{\text{out}}L^{17/6}p\rho^4 L_{\Phi}\varepsilon_x^{2/3} + d_{\text{out}}^{3/2}L^5 p\rho^{11}L_{\Phi}C_{\Phi}\mathfrak{C}_{\varepsilon}(\Phi, \mathcal{O}(\varepsilon_x^{-1}))m^{-1/30}).$$

*Proof.* We fix a given normalized sequence $\mathbf{x}^{(1)}, \cdots, \mathbf{x}^{(L)}$ and an index $s \in [d_{\text{out}}]$. The pseudo network for the fixed sequence is given by

$$F_s^{(L)} = \sum_{i=1}^{L} \mathbf{e}_s^\top \mathbf{Back}_{i \to L} D^{(i)} \left( \mathbf{W}^* \mathbf{h}^{(i-1)} + \mathbf{A}^* \mathbf{x}^{(i)} \right)$$

$$= \frac{d_{\text{out}}}{m} \sum_{i=1}^{L} \sum_{s' \in [d_{\text{out}}]} \sum_{r' \in [p]} \sum_{r \in [m]} b_{r,s'} b_{r',s'}^\dagger \mathbf{Back}_{i \to L, r, s}$$

$$H_{r',s'} \left( \theta_{r',s'} \langle \mathbf{w}_r, \overline{\mathbf{W}}^{[L]} \mathbf{w}_{r',s'}^\dagger \rangle, \sqrt{m/2} a_{r,d} \right) \mathbb{I}_{\mathbf{w}_r^\top \mathbf{h}^{(i-1)} + \mathbf{a}_r^\top \mathbf{x}^{(i)} \geq 0} \qquad (38)$$

First of all, we can't show that the above formulation concentrates on the required signal, because of the dependencies of randomness between $\mathbf{W}$, $\mathbf{A}$, $\mathbf{Back}$, $\overline{\mathbf{W}}^{[L]}$ and $\left\{ \mathbf{h}^{(\ell)} \right\}_{\ell \in [L]}$. To decouple this randomness, we use the fact that ESNs are stable to re-randomization of few rows of the weight matrices and follow the proof technique of Lemma G.3 in [37]. Choose a random subset $\mathcal{K} \subset [m]$ of size $|\mathcal{K}| = N$. Define the function $F_s^{(L), \mathcal{K}}$ as

$$F_s^{(L), \mathcal{K}} (\mathbf{h}^{(L-1)}, \mathbf{x}^{(L)}) \overset{\text{def}}{=} \frac{d_{\text{out}}}{m} \sum_{i=1}^{L} \sum_{s' \in [d_{\text{out}}]} \sum_{r' \in [p]} \sum_{r \in \mathcal{K}} b_{r,s'} b_{r',s'}^\dagger \mathbf{Back}_{i \to L, r, s}$$

$$H_{r',s'} \left( \theta_{r',s'} \langle \mathbf{w}_r, \overline{\mathbf{W}}^{[L]} \mathbf{w}_{r',s'}^\dagger \rangle, \sqrt{m/2} a_{r,d} \right) \mathbb{I}_{\mathbf{w}_r^\top \mathbf{h}^{(i-1)} + \mathbf{a}_r^\top \mathbf{x}^{(i)} \geq 0}.$$

We show the following claim.

**Claim F.2.** *With probability at least* $1 - e^{-\Omega(\rho^2)}$, *for any* $\varepsilon \in (0, \min_{r,s} \frac{1}{C_s(\Phi_{r,s}, \mathcal{O}(\varepsilon_x^{-1}))})$,

$$\left| F_s^{(L), \mathcal{K}} (\mathbf{h}^{(L-1)}, \mathbf{x}^{(L)}) - \frac{d_{\text{out}}}{m} \sum_{i=1}^{L} \sum_{s' \in [d_{\text{out}}]} \sum_{r' \in [p]} \sum_{r \in \mathcal{K}} b_{r,s'} b_{r',s'}^\dagger \mathbf{Back}_{i \to L, r, s} \Phi_{r',s} \left( \left\langle \mathbf{w}_{r',s}^\dagger, [\overline{\mathbf{x}}^{(1)}, \cdots, \overline{\mathbf{x}}^{(L)}] \right\rangle \right) \right|$$

$$\leq \mathcal{O}(d_{\text{out}} L p \rho^8 \mathfrak{C}_\varepsilon(\Phi, \mathcal{O}(\varepsilon_x^{-1})) N^{5/3} m^{-7/6}) + \frac{d_{\text{out}}}{m} \cdot \mathcal{O}(\mathfrak{C}_\varepsilon(\Phi_{r's}, \mathcal{O}(\varepsilon_x^{-1})) \rho^2 \sqrt{d_{\text{out}} L p N})$$

$$+ \frac{d_{\text{out}} L p N}{m} \rho^2 (\varepsilon + \mathcal{O}(L_\Phi \rho^5 (N/m)^{1/6}) + \mathcal{O}(\varepsilon_x^{-1} L_\Phi L^4 \rho^{11} m^{-1/12} + L_\Phi \rho^2 L^{11/6} \varepsilon_x^{2/3})) + \mathcal{O}(\rho^8 d_{\text{out}} L p N^{7/6} m^{-7/6}).$$

The above claim has been restated and proven in claim F.4. The above claim states that the function $F_s^{(L), \mathcal{K}} (\mathbf{h}^{(L-1)}, \mathbf{x}^{(L)})$ contains information about the true function.

To complete the proof, we divide the set of neurons into $m/N$ disjoint sets $\mathcal{K}_1, \cdots, \mathcal{K}_{m/N}$, each set is of size $N$. We apply the Claim F.2 to each subset $\mathcal{K}_i$ and then add up the errors from each subset. That is, with probability at least $1 - \frac{m}{N} e^{-\Omega(\rho^2)}$,

$$F_s^{(L)} (\mathbf{h}^{(\ell-1)}, \mathbf{x}^{(\ell)})$$

$$= \sum_{j=1}^{m/N} F_s^{(L), \mathcal{K}_i} (\mathbf{h}^{(\ell-1)}, \mathbf{x}^{(\ell)})$$

$$= \sum_{j=1}^{m/N} \frac{d_{\text{out}}}{m} \sum_{i=1}^{L} \sum_{s' \in [d_{\text{out}}]} \sum_{r' \in [p]} \sum_{r \in \mathcal{K}_j} b_{r,s'} b_{r',s'}^\dagger \mathbf{Back}_{i \to L, r, s} \Phi_{r',s} \left( \left\langle \mathbf{w}_{r',s}^\dagger, [\overline{\mathbf{x}}^{(2)}, \cdots, \overline{\mathbf{x}}^{(L-1)}] \right\rangle \right) + \sum_{j=1}^{m/N} error_{\mathcal{K}_j}$$

$$= \frac{d_{\text{out}}}{m} \sum_{i=1}^{L} \sum_{s' \in [d_{\text{out}}]} \sum_{r' \in [p]} \sum_{r \in [m]} b_{r,s'} b_{r',s'}^\dagger \mathbf{Back}_{i \to L, r, s} \Phi_{r',s} \left( \left\langle \mathbf{w}_{r',s}^\dagger, [\overline{\mathbf{x}}^{(2)}, \cdots, \overline{\mathbf{x}}^{(L-1)}] \right\rangle \right) + \sum_{j=1}^{m/N} error_{\mathcal{K}_j},$$

where by Claim F.2,

$$|error_{\mathcal{K}_i}| \leq \mathcal{O}(d_{\text{out}} L p \rho^8 \mathfrak{C}_\varepsilon(\Phi, \mathcal{O}(\varepsilon_x^{-1})) N^{5/3} m^{-7/6}) + \frac{d_{\text{out}}}{m} \cdot \mathcal{O}(\mathfrak{C}_\varepsilon(\Phi_{r's}, \mathcal{O}(\varepsilon_x^{-1})) \rho^2 \sqrt{d_{\text{out}} L p N}) +$$

$$+ \frac{d_{\text{out}} L p N}{m} \rho^2 (\varepsilon + \mathcal{O}(L_\Phi \rho^5 (N/m)^{1/6}) + \mathcal{O}(\varepsilon_x^{-1} L_\Phi L^4 \rho^{11} m^{-1/12} + L_\Phi L^{11/6} \rho^2 \varepsilon_x^{2/3})) +$$
$$\mathcal{O}(\rho^8 d_{\text{out}} L p N^{7/6} m^{-7/6}).$$

Thus,

$$\left| F_s^{(L)}(\mathbf{h}^{(\ell-1)}, \mathbf{x}^{(\ell)}) - \frac{d_{\text{out}}}{m} \sum_{i=1}^L \sum_{s' \in [d_{\text{out}}]} \sum_{r' \in [p]} \sum_{r \in [m]} b_{r,s'} b_{r',s'}^\dagger \mathbf{Back}_{i \to L, r, s} \Phi_{r',s} \left( \left\langle \mathbf{w}_{r',s}^\dagger, [\overline{\mathbf{x}}^{(2)}, \cdots, \overline{\mathbf{x}}^{(L-1)}] \right\rangle \right) \right|$$

$$\leq \mathcal{O}(d_{\text{out}} L p \rho^8 \mathfrak{C}_\varepsilon(\Phi, \mathcal{O}(\varepsilon_x^{-1})) N^{2/3} m^{-1/6}) + \frac{\sqrt{d_{\text{out}}^3 L p}}{\sqrt{N}} \cdot \mathcal{O}(\mathfrak{C}_\varepsilon(\Phi_{r's}, \mathcal{O}(\varepsilon_x^{-1})) \rho^2)$$

$$+ d_{\text{out}} L p \rho^2 (\varepsilon + \mathcal{O}(L_\Phi \rho^5 (N/m)^{1/6}) + \mathcal{O}(\varepsilon_x^{-1} L_\Phi L^4 \rho^{11} m^{-1/12} + L_\Phi L^{11/6} \rho^2 \varepsilon_x^{2/3})) + \mathcal{O}(\rho^8 d_{\text{out}} L p N^{1/6} m^{-1/6}),$$

with probability at least $1 - \frac{m}{N} e^{-\Omega(\rho^2)}$. Choosing $N = m^{0.2}$, we have

$$\left| F_s^{(L)}(\mathbf{h}^{(\ell-1)}, \mathbf{x}^{(\ell)}) - \frac{d_{\text{out}}}{m} \sum_{i=1}^L \sum_{s' \in [d_{\text{out}}]} \sum_{r' \in [p]} \sum_{r \in [m]} b_{r,s'} b_{r',s'}^\dagger \mathbf{Back}_{i \to L, r, s} \Phi_{r',s} \left( \left\langle \mathbf{w}_{r',s}^\dagger, [\overline{\mathbf{x}}^{(2)}, \cdots, \overline{\mathbf{x}}^{(L-1)}] \right\rangle \right) \right|$$

$$\leq \mathcal{O}(d_{\text{out}} L p \rho^8 \mathfrak{C}_\varepsilon(\Phi, \mathcal{O}(\varepsilon_x^{-1})) m^{-1/30}) + \mathcal{O}(\mathfrak{C}_\varepsilon(\Phi_{r's}, \mathcal{O}(\varepsilon_x^{-1})) \rho^2 \sqrt{d_{\text{out}}^3 L p} m^{-0.1}) \tag{39}$$

$$+ d_{\text{out}} L p \rho^2 (\varepsilon + \mathcal{O}(L_\Phi \rho^5 m^{-2/15}) + \mathcal{O}(\varepsilon_x^{-1} L_\Phi L^4 \rho^{11} m^{-1/12} + L_\Phi L^{11/6} \rho^2 \varepsilon_x^{2/3})) + \mathcal{O}(\rho^8 d_{\text{out}} L p m^{-2/15}), \tag{40}$$

with probability at least $1 - m^{0.8} e^{-\Omega(\rho^2)} \geq 1 - e^{-\Omega(\rho^2)}$.

Now, in the next claim, we show that the $f$ concentrates on the desired term.

**Claim F.3.** *With probability exceeding* $1 - e^{-\Omega(\rho^2)}$,

$$\left| b_{r',s}^\dagger \Phi_{r',s} \left( \left\langle \mathbf{w}_{r',s}^\dagger, [\overline{\mathbf{x}}^{(2)}, \cdots, \overline{\mathbf{x}}^{(L-1)}] \right\rangle \right) \right.$$

$$\left. - \frac{d_{\text{out}}}{m} \sum_{i=1}^L \sum_{s' \in [d_{\text{out}}]} \sum_{r \in [m]} b_{r,s'} b_{r',s'}^\dagger \mathbf{Back}_{i \to L, r, s} \Phi_{r',s} \left( \left\langle \mathbf{w}_{r',s}^\dagger, [\overline{\mathbf{x}}^{(2)}, \cdots, \overline{\mathbf{x}}^{(L-1)}] \right\rangle \right) \right|$$

$$\leq \mathcal{O}(L d_{\text{out}} \rho C_\Phi m^{-0.25}).$$

The claim is restated and proven in claim F.16.

Thus, introducing claim F.3 in eq. 40, we have

$$\left| F_s^{(L)}(\mathbf{h}^{(\ell-1)}, \mathbf{x}^{(\ell)}) - \sum_{r' \in [p]} b_{r',s}^\dagger \Phi_{r',s} \left( \left\langle \mathbf{w}_{r',s}^\dagger, [\overline{\mathbf{x}}^{(2)}, \cdots, \overline{\mathbf{x}}^{(L-1)}] \right\rangle \right) \right|$$

$$\leq \mathcal{O}(d_{\text{out}} L p \rho^8 \mathfrak{C}_\varepsilon(\Phi, \mathcal{O}(\varepsilon_x^{-1})) m^{-1/30}) + \mathcal{O}(\mathfrak{C}_\varepsilon(\Phi_{r's}, \mathcal{O}(\varepsilon_x^{-1})) \rho^2 \sqrt{d_{\text{out}}^3 L p} m^{-0.1})$$

$$+ d_{\text{out}} L p \rho^2 (\varepsilon + \mathcal{O}(L_\Phi \rho^5 m^{-2/15}) + \mathcal{O}(\varepsilon_x^{-1} L_\Phi L^4 \rho^{11} m^{-1/12} + L_\Phi L^{11/6} \rho^2 \varepsilon_x^{2/3}))$$

$$+ \mathcal{O}(\rho^8 d_{\text{out}} L p m^{-2/15}) + \mathcal{O}(L p d_{\text{out}} \rho C_\Phi m^{-0.25})$$

$$\leq \mathcal{O}(d_{\text{out}} L p \rho^2 \varepsilon + d_{\text{out}} L^{17/6} p \rho^4 L_\Phi \varepsilon_x^{2/3} + d_{\text{out}}^{3/2} L^5 p \rho^{11} L_\Phi C_\Phi \mathfrak{C}_\varepsilon(\Phi, \mathcal{O}(\varepsilon_x^{-1})) m^{-1/30}).$$

$\square$

## F.2   Proof of Claim F.2

**Claim F.4** (Restating claim F.2). *With probability at least* $1 - e^{-\Omega(\rho^2)}$, *for any* $\varepsilon \in$ $(0, \min_{r,s} \frac{1}{C_s(\Phi_{r,s}, \mathcal{O}(\varepsilon_x^{-1}))})$,

$$\left| F_s^{(L),\mathcal{K}}(\mathbf{h}^{(L-1)}, \mathbf{x}^{(L)}) - \frac{d_{\text{out}}}{m} \sum_{i=1}^{L} \sum_{s' \in [d_{\text{out}}]} \sum_{r' \in [p]} \sum_{r \in \mathcal{K}} b_{r,s'} b_{r',s'}^\dagger \mathbf{Back}_{i \to L, r, s} \Phi_{r',s} \left( \left\langle \mathbf{w}_{r',s}^\dagger, [\overline{\mathbf{x}}^{(1)}, \cdots, \overline{\mathbf{x}}^{(L)}] \right\rangle \right) \right|$$

$$\leq \mathcal{O}(d_{\text{out}} L p \rho^8 \mathfrak{C}_\varepsilon(\Phi, \mathcal{O}(\varepsilon_x^{-1})) N^{5/3} m^{-7/6}) + \frac{d_{\text{out}}}{m} \cdot \mathcal{O}(\mathfrak{C}_\varepsilon(\Phi_{r's}, \mathcal{O}(\varepsilon_x^{-1})) \rho^2 \sqrt{d_{\text{out}} L p N})$$

$$+ \frac{d_{\text{out}} L p N}{m} \rho^2 (\varepsilon + \mathcal{O}(L_\Phi \rho^5 (N/m)^{1/6}) + \mathcal{O}(\varepsilon_x^{-1} L_\Phi L^4 \rho^{11} m^{-1/12} + L_\Phi \rho^2 L^{11/6} \varepsilon_x^{2/3})) + \mathcal{O}(\rho^8 d_{\text{out}} L p N^{7/6} m^{-7/6}).$$

*Proof.* We will replace the rows $\{\mathbf{w}_k, \mathbf{a}_k\}_{k \in \mathcal{K}}$ of $\mathbf{W}$ and $\mathbf{A}$ with freshly new i.i.d. samples $\widetilde{\mathbf{w}}_k, \widetilde{\mathbf{a}}_k \sim \mathcal{N}\left(0, \frac{2}{m}\mathbf{I}\right)$. to form new matrices $\widetilde{\mathbf{W}}$ and $\widetilde{\mathbf{A}}$. For the given sequence, we follow the notation of Lemma B.2 to denote the hidden states corresponding to the old and the new weight matrices. Let $\widetilde{F}_s^{(L),\mathcal{K}}$ denote the following function:

$$\widetilde{F}_s^{(L),\mathcal{K}}(\widetilde{\mathbf{h}}^{(L-1)}, \mathbf{x}^{(L)}) \overset{\text{def}}{=} \frac{d_{\text{out}}}{m} \sum_{i=1}^{L} \sum_{s' \in [d_{\text{out}}]} \sum_{r' \in [p]} \sum_{r \in \mathcal{K}} b_{r,s'} b_{r',s'}^\dagger \widetilde{\mathbf{Back}}_{i \to L, r, s}$$

$$H_{r',s'}\left(\widetilde{\theta}_{r',s'} \langle \mathbf{w}_r, \widetilde{\widetilde{\mathbf{W}}}^{[L]} \mathbf{w}_{r',s'}^\dagger \rangle, \sqrt{m/2} a_{r,d}\right) \mathbb{I}_{\mathbf{w}_r^\top \widetilde{\mathbf{h}}^{(i-1)} + \mathbf{a}_r^\top \mathbf{x}^{(i)} \geq 0},$$

where

$$\widetilde{\theta}_{r',s} = \frac{\sqrt{m/2}}{\left\| \widetilde{\widetilde{\mathbf{W}}}^{[L]} \mathbf{w}_{r',s}^\dagger \right\|}.$$

Using similar technique used to find the bounds of $\theta_{r',s}$ in eq. 37, we ca show that $\forall r' \in [p], s \in [d_{\text{out}}]$, w.p. at least $1 - e^{-\Omega(\rho^2)}$ over $\widetilde{\mathbf{W}}, \widetilde{\mathbf{A}}$,

$$\left(1 + \frac{1}{100L}\right)^{-L} \leq \sqrt{2/m} \widetilde{\theta}_{r',s} \leq \left(1 - \frac{1}{100L}\right)^{-L}. \tag{41}$$

Again, there is one important relation between $\theta_{r',s}$ and $\widetilde{\theta}_{r',s}$ that we will require later on, which we prove in the next claim.

**Claim F.5.** *With probability at least* $1 - e^{-\Omega(\rho^2)}$, *for all* $r' \in [p], s \in [d_{\text{out}}]$,

$$\left| \widetilde{\theta}_{r',s} \theta_{r',s}^{-1} - 1 \right| \leq \mathcal{O}(\rho^5 (N/m)^{1/6}).$$

The claim has been restated and proven in claim F.12. A simple corollary of the above claim is given below.

**Corollary F.5.1.** *With probability at least* $1 - e^{-\Omega(\rho^2)}$, *for all* $r' \in [p], s \in [d_{\text{out}}]$,

$$\left| \widetilde{\theta}_{r',s} - \theta_{r',s} \right| \leq \mathcal{O}(\rho^5 (N/m)^{1/6}).$$

The above corollary follows from the bounds on $\theta_{r',s}$ from eq. 37.

We will first show that $\widetilde{F}_s^{(L),\mathcal{K}}(\widetilde{\mathbf{h}}^{(L-1)}, \mathbf{x}^{(L)})$ and $F_s^{(L),\mathcal{K}}(\mathbf{h}^{(L-1)}, \mathbf{x}^{(L)})$ are close. The claim has been restated and proven in claim F.13.

**Claim F.6.** *With probability at least* $1 - e^{-\Omega(\rho^2)}$,

$$\left| \widetilde{F}_s^{(L),\mathcal{K}}(\widetilde{\mathbf{h}}^{(L-1)}, \mathbf{x}^{(L)}) - F_s^{(L),\mathcal{K}}(\mathbf{h}^{(L-1)}, \mathbf{x}^{(L)}) \right| \leq \mathcal{O}(d_{\text{out}} L p \rho^8 \mathfrak{C}_\varepsilon(\Phi, \mathcal{O}(\varepsilon_x^{-1})) N^{5/3} m^{-7/6}).$$

Now, we show that $\widetilde{F}$ is close to the desired signal in the two claims below.

**Claim F.7.** *With probability at least* $1 - e^{-\Omega(\rho^2)}$,

$$\left| \widetilde{F}_s^{(L),\mathcal{K}}(\widetilde{\mathbf{h}}^{(L-1)}, \mathbf{x}^{(L)}) - \frac{d_{\text{out}}}{m} \sum_{i=1}^{L} \sum_{s' \in [d_{\text{out}}]} \sum_{r' \in [p]} \sum_{r \in \mathcal{K}} b_{r,s'} b_{r',s'}^\dagger \widetilde{\textbf{Back}}_{i \to L, r, s} \Phi_{r',s} \left( \left\langle \mathbf{w}_{r',s}^\dagger, [\overline{\mathbf{x}}^{(1)}, \cdots, \overline{\mathbf{x}}^{(L)}] \right\rangle \right) \right|$$

$$\leq \frac{d_{\text{out}}}{m} \cdot \mathcal{O}(\mathfrak{C}_\varepsilon(\Phi_{r's}, \mathcal{O}(\varepsilon_x^{-1})) \rho^2 \sqrt{d_{\text{out}} L p N})$$

$$+ \frac{d_{\text{out}} L p N}{m} \rho^2 (\varepsilon + \mathcal{O}(L_\Phi \rho^5 (N/m)^{1/6}) + \mathcal{O}(L_\Phi \varepsilon_x^{-1} L^4 \rho^{11} m^{-1/12} + L_\Phi \rho^2 L^{11/6} \varepsilon_x^{2/3})),$$

*for any* $\varepsilon \in (0, \min_{r,s} \frac{\sqrt{3}}{C_s(\Phi_{r,s}, \varepsilon_x^{-1})})$.

**Claim F.8.** *with probability at least* $1 - e^{-\Omega(\rho^2)}$,

$$\left| \frac{d_{\text{out}}}{m} \sum_{i=1}^{L} \sum_{s' \in [d_{\text{out}}]} \sum_{r' \in [p]} \sum_{r \in \mathcal{K}} b_{r,s'} b_{r',s'}^\dagger \widetilde{\textbf{Back}}_{i \to L, r, s} \Phi_{r',s} \left( \left\langle \mathbf{w}_{r',s}^\dagger, [\overline{\mathbf{x}}^{(1)}, \cdots, \overline{\mathbf{x}}^{(L)}] \right\rangle \right) \right.$$

$$\left. - \frac{d_{\text{out}}}{m} \sum_{i=1}^{L} \sum_{s' \in [d_{\text{out}}]} \sum_{r' \in [p]} \sum_{r \in \mathcal{K}} b_{r,s'} b_{r',s'}^\dagger \textbf{Back}_{i \to L, r, s} \Phi_{r',s} \left( \left\langle \mathbf{w}_{r',s}^\dagger, [\overline{\mathbf{x}}^{(1)}, \cdots, \overline{\mathbf{x}}^{(L)}] \right\rangle \right) \right|$$

$$\leq \mathcal{O}(\rho^8 C_\Phi d_{\text{out}} L p N^{7/6} m^{-7/6}).$$

The above two claims have been restated and proven in claim F.14 and F.15 respectively.

Thus, from Claim F.6, Claim F.7 and Claim F.8, we have with probability at least $1 - e^{-\Omega(\rho^2)}$, for any $\varepsilon \in (0, \min_{r,s} \frac{1}{C_s(\Phi_{r,s}, \mathcal{O}(\varepsilon_x^{-1}))})$,

$$\left| F_s^{(L),\mathcal{K}}(\mathbf{h}^{(L-1)}, \mathbf{x}^{(L)}) - \frac{d_{\text{out}}}{m} \sum_{i=1}^{L} \sum_{s' \in [d_{\text{out}}]} \sum_{r' \in [p]} \sum_{r \in \mathcal{K}} b_{r,s'} b_{r',s'}^\dagger \textbf{Back}_{i \to L, r, s} \Phi_{r',s} \left( \left\langle \mathbf{w}_{r',s}^\dagger, [\overline{\mathbf{x}}^{(2)}, \cdots, \overline{\mathbf{x}}^{(L-1)}] \right\rangle \right) \right|$$

$$\leq \left| F_s^{(L),\mathcal{K}}(\mathbf{h}^{(L-1)}, \mathbf{x}^{(L)}) - \widetilde{F}_s^{(L),\mathcal{K}}(\widetilde{\mathbf{h}}^{(L-1)}, \mathbf{x}^{(L)}) \right|$$

$$+ \left| \widetilde{F}_s^{(L),\mathcal{K}}(\widetilde{\mathbf{h}}^{(L-1)}, \mathbf{x}^{(L)}) - \frac{d_{\text{out}}}{m} \sum_{i=1}^{L} \sum_{s' \in [d_{\text{out}}]} \sum_{r' \in [p]} \sum_{r \in \mathcal{K}} b_{r,s'} b_{r',s'}^\dagger \widetilde{\textbf{Back}}_{i \to L, r, s} \Phi_{r',s} \left( \left\langle \mathbf{w}_{r',s}^\dagger, [\overline{\mathbf{x}}^{(2)}, \cdots, \overline{\mathbf{x}}^{(L-1)}] \right\rangle \right) \right|$$

$$+ \left| \frac{d_{\text{out}}}{m} \sum_{i=1}^{L} \sum_{s' \in [d_{\text{out}}]} \sum_{r' \in [p]} \sum_{r \in \mathcal{K}} b_{r,s'} b_{r',s'}^\dagger \widetilde{\textbf{Back}}_{i \to L, r, s} \Phi_{r',s} \left( \left\langle \mathbf{w}_{r',s}^\dagger, [\overline{\mathbf{x}}^{(2)}, \cdots, \overline{\mathbf{x}}^{(L-1)}] \right\rangle \right) \right.$$

$$\left. - \frac{d_{\text{out}}}{m} \sum_{i=1}^{L} \sum_{s' \in [d_{\text{out}}]} \sum_{r' \in [p]} \sum_{r \in \mathcal{K}} b_{r,s'} b_{r',s'}^\dagger \textbf{Back}_{i \to L, r, s} \Phi_{r',s} \left( \left\langle \mathbf{w}_{r',s}^\dagger, [\overline{\mathbf{x}}^{(2)}, \cdots, \overline{\mathbf{x}}^{(L-1)}] \right\rangle \right) \right|$$

$$\leq \mathcal{O}(d_{\text{out}} L p \rho^8 \mathfrak{C}_\varepsilon(\Phi, \mathcal{O}(\varepsilon_x^{-1})) N^{5/3} m^{-7/6}) + \frac{d_{\text{out}}}{m} \cdot \mathcal{O}(\mathfrak{C}_\varepsilon(\Phi_{r's}, \mathcal{O}(\varepsilon_x^{-1})) \rho^2 \sqrt{d_{\text{out}} L p N})$$

$$+ \frac{d_{\text{out}} L p N}{m} \rho^2 (\varepsilon + \mathcal{O}(L_\Phi \rho^5 (N/m)^{1/6}) + \mathcal{O}(\varepsilon_x^{-1} L_\Phi L^4 \rho^{11} m^{-1/12} + L_\Phi L^{11/6} \rho^2 \varepsilon_x^{2/3})) + \mathcal{O}(\rho^8 d_{\text{out}} L p N^{7/6} m^{-7/6}).$$

$$\square$$

## F.3 Helping lemmas

### F.3.1 Function approximation using hermite polynomials

The following theorem on approximating a smooth function using hermite polynomials has been taken from [15] and we will use this theorem to show that pseudo RNNs can approximate the target concept class.

**Theorem F.9** (Lemma 6.2 in [15]). *For every smooth function $\phi$, every $\varepsilon \in \left(0, \frac{1}{\mathfrak{C}_s(\phi,1)}\right)$ there exists a $H : \mathbb{R}^2 \to \left(-\mathfrak{C}_\varepsilon(\phi,1), \mathfrak{C}_\varepsilon(\phi,1)\right)$, satisfying $|H| \leq \mathfrak{C}_\varepsilon(\phi,1)$, and is $\mathfrak{C}_\varepsilon(\phi,1)$-lipschitz continuous in the first variable and for all $x_1 \in (-1,1)$*

$$\left| \mathbb{E}_{\alpha_1,\beta_1,b_0} \left[ \mathbb{I}_{\alpha_1 x_1 + \beta_1 \sqrt{1-x_1^2} + b_0 \geq 0} H(\alpha_1, b_0) \right] - \phi(x_1) \right| \leq \varepsilon$$

*where $\alpha_1, \beta_1$ and $b_0 \sim \mathcal{N}(0,1)$ are independent random variables.*

In [15], the function $H$ is shown to be lipschitz continuous in expectation w.r.t. the first variable $\alpha_1$ which follows a normal distribution. However, one can also show that the function $H$ is lipschitz continuous w.r.t. the first variable, even when the variable is perturbed by bounded noise to a variable that does not necessarily follow a gaussian distribution i.e. one can show that

$$\left| \mathbb{E}_{\alpha_1,\beta_1,b_0 \sim \mathcal{N}(0,1)} \mathbb{E}_{\theta:|\theta| \leq \gamma} \left[ H(\alpha_1, b_0) - H(\alpha_1 + \theta, b_0) \right] \right| \leq \gamma \mathfrak{C}_\varepsilon(\phi,1).$$

The proof will follow along the similar lines of Claim C.2 in [15]. We give a brief overview here. The function $H$ was shown to be a weighted combination of different hermite polynomials. Using the following property of hermite polynomials,

$$h_i(x+y) = \sum_{k=0}^{i} \binom{i}{k} x^{i-k} h_k(y),$$

we expand the function $H(\alpha_1 + \theta, b_0)$ and then, bound each term using the procedure in Claim C.2 of [15].

**Corollary F.9.1.** *For any $\sigma > 0$, $r_x > 0$ s.t. $\sigma \geq r_x/10$, $k_0 \geq 0$, and for every smooth function $\phi$, any $\varepsilon \in \left(0, \frac{r_x}{\sigma \mathfrak{C}_s(\phi, k_0 r_x)}\right)$ there exists a $H : \mathbb{R}^2 \to \left(-\frac{\sigma}{r_x} \mathfrak{C}_\varepsilon(\phi, k_0 r_x), \frac{\sigma}{r_x} \mathfrak{C}_\varepsilon(\phi, k_0 r_x)\right)$, which is $\frac{\sigma}{r_x} \mathfrak{C}_\varepsilon(\phi, k_0 r_x)$-lipschitz continuous and for all $x_1 \in (-r_x, r_x)$*

$$\left| \mathbb{E}_{\alpha_1,\beta_1,b_0} \left[ \mathbb{I}_{\alpha_1 x_1 + \beta_1 \sqrt{r_x^2 - x_1^2} + b_0 \geq 0} H(\alpha_1, b_0) \right] - \phi(k_0 x_1) \right| \leq \varepsilon$$

*where $\alpha_1, \beta_1 \sim \mathcal{N}(0,1)$ and $b_0 \sim \mathcal{N}(0,\sigma^2)$ are independent random variables.*

**Lemma F.10** (Function Approximators). *Let $r_x = \sqrt{2 + (L-2)\varepsilon_x^2}$. For each $\Phi_{r,s}$ and a constant $k_{0,r,s} = \Theta(\frac{1}{\varepsilon_x})$, there exists a function $H_{r,s}$ such that for any $\varepsilon \in (0, \min_{r,s} \frac{r_x}{C_s(\Phi_{r,s}, k_{0,r,s} r_x)})$, $H_{r,s} : \mathbb{R}^2 \to \left(-\frac{1}{r_x} \mathfrak{C}_\varepsilon(\Phi_{r,s}, k_{0,r,s} r_x), \frac{1}{r_x} \mathfrak{C}_\varepsilon(\Phi_{r,s}, k_{0,r,s} r_x)\right)$, is $\frac{1}{r_x} \mathfrak{C}_\varepsilon(\Phi_{r,s}, k_{0,r,s} r_x)$-lipschitz continuous, and for all $x_1 \in (-r_x, r_x)$*

$$\left| \mathbb{E}_{\alpha_1,\beta_1,b_0} \left[ \mathbb{I}_{\alpha_1 x_1 + \beta_1 \sqrt{r_x^2 - x_1^2} + b_0 \geq 0} H_{r,s}(\alpha_1, b_0) \right] - \Phi_{r,s}(k_{0,r,s} x_1) \right| \leq \varepsilon$$

*where $\alpha_1, \beta_1 \sim \mathcal{N}(0,1)$ and $b_0 \sim \mathcal{N}(0,1)$ are independent random variables.*

For any $\varepsilon_x \leq \frac{1}{L}$, we can see that for all $\Phi_{r,s}$, $|H_{r,s}| \leq \frac{1}{\sqrt{2}} \mathfrak{C}_\varepsilon\left(\Phi_{r,s}, \mathcal{O}(\varepsilon_x^{-1})\right)$ and $H_{r,s}$ is $\frac{1}{\sqrt{2}} \mathfrak{C}_\varepsilon\left(\Phi_{r,s}, \mathcal{O}(\varepsilon_x^{-1})\right)$ lipschitz, for any $\varepsilon \leq \frac{\sqrt{3}}{C_s(\Phi_{r,s}, \mathcal{O}(\varepsilon_x^{-1}))}$.

### F.3.2 Proofs of the helping lemmas

First, we mention one of the properties on correlations of $\mathbf{Back}_{i \to j}$ matrices, which will be heavily used later on.

**Lemma F.11** (Lemma C.1 in [37]). *For every $\varepsilon_x < 1/L$ and every normalized input sequence, $\mathbf{x}_1, \mathbf{x}_2, ..., \mathbf{x}_L$, with probability at least $1 - e^{-\Omega(\rho^2)}$ over $\mathbf{W}$, $\mathbf{A}$ and $\mathbf{B}$: for every $1 \leq i \leq j < j' \leq L$,*

$$\left| \langle \mathbf{u}^\top \mathbf{Back}_{i \to j}, \mathbf{v}^\top \mathbf{Back}_{i \to j'} \rangle \right| \leq \mathcal{O}\left(m^{0.75} \rho\right) \|\mathbf{u}\| \|\mathbf{v}\|,$$

*for any two vectors $\mathbf{u}$ and $\mathbf{v}$ in $\mathbb{R}^{d_{\text{out}}}$.*

**Claim F.12** (Restating claim F.5). *With probability at least $1 - e^{-\Omega(\rho^2)}$, for all $r' \in [p], s \in [d_{\text{out}}]$,*

$$\left| \widetilde{\theta}_{r',s} \theta_{r',s}^{-1} - 1 \right| \leq \mathcal{O}(\rho^5 (N/m)^{1/6}).$$

*Proof.* First of all,

$$\sqrt{2/m} \left| \widetilde{\theta}_{r',s}^{-1} - \theta_{r',s}^{-1} \right| = \left| \left\| \widetilde{\overline{\mathbf{W}}}^{[L]} \mathbf{w}_{r',s}^{\dagger} \right\| - \left\| \overline{\mathbf{W}}^{[L]} \mathbf{w}_{r',s}^{\dagger} \right\| \right|$$

$$\leq \left\| \widetilde{\overline{\mathbf{W}}}^{[L]} \mathbf{w}_{r',s}^{\dagger} - \overline{\mathbf{W}}^{[L]} \mathbf{w}_{r',s}^{\dagger} \right\|$$

$$\leq \max_{\ell \leq L} \left\| \left( \widetilde{\mathbf{W}}^{(L,\ell)} - \mathbf{W}^{(L,\ell)} \right) \mathbf{w}_{r',s}^{\dagger} \right\|_2$$

$$\leq \mathcal{O}(\rho^5 (N/m)^{1/6})$$

where in the pre-final step, we have used Lemma B.2 to have w.p. exceeding $1 - e^{-\Omega(\rho^2)}$ for any $\ell \leq L$,

$$\left\| \left( \widetilde{\mathbf{W}}^{(L,\ell)} - \mathbf{W}^{(L,\ell)} \right) \mathbf{w}_{r',s}^{\dagger} \right\|_2 = \left\| \left( \prod_{L \geq \ell' \geq \ell} \widetilde{\mathbf{D}}_{(0)}^{(\ell)} \widetilde{\mathbf{W}} - \prod_{L \geq \ell' \geq \ell} \mathbf{D}_{(0)}^{(\ell)} \mathbf{W} \right) \mathbf{w}_{r',s}^{\dagger} \right\|_2$$

$$\leq \mathcal{O}(\rho^5 (N/m)^{1/6}) \cdot \left\| \mathbf{w}_{r',s}^{\dagger} \right\| = \mathcal{O}(\rho^5 (N/m)^{1/6}).$$

Hence,

$$\left| \widetilde{\theta}_{r',s} \theta_{r',s}^{-1} - 1 \right| \leq \sqrt{m/2} \left| \widetilde{\theta}_{r',s} \right| \mathcal{O}(\rho^5 (N/m)^{1/6}) \leq \mathcal{O}(\rho^5 (N/m)^{1/6}),$$

where we have used the upper bound on $\sqrt{m/2} \left| \widetilde{\theta}_{r',s} \right|$ from eq. 41 in the final step. $\qquad\square$

**Claim F.13** (Restating claim F.6). *With probability at least $1 - e^{-\Omega(\rho^2)}$,*

$$\left| \widetilde{F}_s^{(L),\mathcal{K}} (\widetilde{\mathbf{h}}^{(L-1)}, \mathbf{x}^{(L)}) - F_s^{(L),\mathcal{K}} (\mathbf{h}^{(L-1)}, \mathbf{x}^{(L)}) \right| \leq \mathcal{O}(d_{\text{out}} L p \rho^8 \mathfrak{C}_\varepsilon(\Phi, \mathcal{O}(\varepsilon_x^{-1})) N^{5/3} m^{-7/6}).$$

*Proof.* We break the required term into three different terms.

$$\left| \widetilde{F}_s^{(L),\mathcal{K}} (\widetilde{\mathbf{h}}^{(L-1)}, \mathbf{x}^{(L)}) - F_s^{(L),\mathcal{K}} (\mathbf{h}^{(L-1)}, \mathbf{x}^{(L)}) \right|$$

$$= \left| \frac{d_{\text{out}}}{m} \sum_{i=1}^L \sum_{s' \in [d_{\text{out}}]} \sum_{r' \in [p]} \sum_{r \in \mathcal{K}} b_{r,s'} b_{r',s'}^{\dagger} \widetilde{\mathbf{Back}}_{i \to L,r,s} H_{r',s'} \left( \widetilde{\theta}_{r',s'} \langle \mathbf{w}_r, \widetilde{\overline{\mathbf{W}}}^{[L]} \mathbf{w}_{r',s'}^{\dagger} \rangle, \sqrt{m/2} a_{r,d} \right) \mathbb{I}_{\mathbf{w}_r^{\top} \widetilde{\mathbf{h}}^{(i-1)} + \mathbf{a}_r^{\top} \mathbf{x}^{(i)} \geq 0} \right.$$

$$\left. - \frac{d_{\text{out}}}{m} \sum_{i=1}^L \sum_{s' \in [d_{\text{out}}]} \sum_{r' \in [p]} \sum_{r \in \mathcal{K}} b_{r,s'} b_{r',s'}^{\dagger} \mathbf{Back}_{i \to L,r,s} H_{r',s'} \left( \theta_{r',s'} \langle \mathbf{w}_r, \overline{\mathbf{W}}^{[L]} \mathbf{w}_{r',s'}^{\dagger} \rangle, \sqrt{m/2} a_{r,d} \right) \mathbb{I}_{\mathbf{w}_r^{\top} \mathbf{h}^{(i-1)} + \mathbf{a}_r^{\top} \mathbf{x}^{(i)} \geq 0} \right|$$

$$\leq \left| \frac{d_{\text{out}}}{m} \sum_{i=1}^L \sum_{s' \in [d_{\text{out}}]} \sum_{r' \in [p]} \sum_{r \in \mathcal{K}} b_{r,s'} b_{r',s'}^{\dagger} \widetilde{\mathbf{Back}}_{i \to L,r,s} H_{r',s'} \left( \widetilde{\theta}_{r',s'} \langle \mathbf{w}_r, \widetilde{\overline{\mathbf{W}}}^{[L]} \mathbf{w}_{r',s'}^{\dagger} \rangle, \sqrt{m/2} a_{r,d} \right) \mathbb{I}_{\mathbf{w}_r^{\top} \widetilde{\mathbf{h}}^{(i-1)} + \mathbf{a}_r^{\top} \mathbf{x}^{(i)} \geq 0} \right.$$

$$\left. - \frac{d_{\text{out}}}{m} \sum_{i=1}^L \sum_{s' \in [d_{\text{out}}]} \sum_{r' \in [p]} \sum_{r \in \mathcal{K}} b_{r,s'} b_{r',s'}^{\dagger} \mathbf{Back}_{i \to L,r,s} H_{r',s'} \left( \widetilde{\theta}_{r',s'} \langle \mathbf{w}_r, \widetilde{\overline{\mathbf{W}}}^{[L]} \mathbf{w}_{r',s'}^{\dagger} \rangle, \sqrt{m/2} a_{r,d} \right) \mathbb{I}_{\mathbf{w}_r^{\top} \widetilde{\mathbf{h}}^{(i-1)} + \mathbf{a}_r^{\top} \mathbf{x}^{(i)} \geq 0} \right|$$

$$\tag{42}$$

$$+ \left| \frac{d_{\text{out}}}{m} \sum_{i=1}^L \sum_{s' \in [d_{\text{out}}]} \sum_{r' \in [p]} \sum_{r \in \mathcal{K}} b_{r,s'} b_{r',s'}^{\dagger} \mathbf{Back}_{i \to L,r,s} H_{r',s'} \left( \widetilde{\theta}_{r',s'} \langle \mathbf{w}_r, \widetilde{\overline{\mathbf{W}}}^{[L]} \mathbf{w}_{r',s'}^{\dagger} \rangle, \sqrt{m/2} a_{r,d} \right) \mathbb{I}_{\mathbf{w}_r^{\top} \widetilde{\mathbf{h}}^{(i-1)} + \mathbf{a}_r^{\top} \mathbf{x}^{(i)} \geq 0} \right.$$

$$\left. - \frac{d_{\text{out}}}{m} \sum_{i=1}^L \sum_{s' \in [d_{\text{out}}]} \sum_{r' \in [p]} \sum_{r \in \mathcal{K}} b_{r,s'} b_{r',s'}^{\dagger} \mathbf{Back}_{i \to L,r,s} H_{r',s'} \left( \widetilde{\theta}_{r',s'} \langle \mathbf{w}_r, \widetilde{\overline{\mathbf{W}}}^{[L]} \mathbf{w}_{r',s'}^{\dagger} \rangle, \sqrt{m/2} a_{r,d} \right) \mathbb{I}_{\mathbf{w}_r^{\top} \mathbf{h}^{(i-1)} + \mathbf{a}_r^{\top} \mathbf{x}^{(i)} \geq 0} \right|$$

$$\tag{43}$$

$$+\left|\frac{d_{\text{out}}}{m}\sum_{i=1}^{L}\sum_{s'\in[d_{\text{out}}]}\sum_{r'\in[p]}\sum_{r\in\mathcal{K}}b_{r,s'}b_{r',s'}^{\dagger}\mathbf{Back}_{i\to L,r,s}H_{r',s'}\left(\widetilde{\theta}_{r',s'}\langle\mathbf{w}_{r},\overline{\widetilde{\widetilde{\mathbf{W}}}}^{[L]}\mathbf{w}_{r',s'}^{\dagger}\rangle,\sqrt{m/2}a_{r,d}\right)\mathbb{I}_{\mathbf{w}_{r}^{\top}\mathbf{h}^{(i-1)}+\mathbf{a}_{r}^{\top}\mathbf{x}^{(i)}\geq0}\right.$$

$$\left.-\frac{d_{\text{out}}}{m}\sum_{i=1}^{L}\sum_{s'\in[d_{\text{out}}]}\sum_{r'\in[p]}\sum_{r\in\mathcal{K}}b_{r,s'}b_{r',s'}^{\dagger}\mathbf{Back}_{i\to L,r,s}H_{r',s'}\left(\theta_{r',s'}\langle\mathbf{w}_{r},\overline{\mathbf{W}}^{[L]}\mathbf{w}_{r',s'}^{\dagger}\rangle,\sqrt{m/2}a_{r,d}\right)\mathbb{I}_{\mathbf{w}_{r}^{\top}\mathbf{h}^{(i-1)}+\mathbf{a}_{r}^{\top}\mathbf{x}^{(i)}\geq0}\right|.$$

(44)

We now show that each of the three equations, eq. 42, eq. 43 and eq. 44 are small. First, we will need a couple of bounds on the terms that appear in the equations.

- Since $b_{r,s'}\sim\mathcal{N}(0,\frac{1}{d_{\text{out}}})$, using the fact A.4, we can show that with probability $1-e^{-\Omega(\rho^{2})}$, $\max_{r,s'}|b_{r,s'}|\leq\frac{\rho}{\sqrt{d_{\text{out}}}}$.

- From the definition of concept class, $\max_{r',s'}|b_{r',s'}^{\dagger}|\leq1$.

- By the definition of $H$ from def F.10, we have $\max_{r',s'}|H_{r',s'}\left(\widetilde{\theta}_{r',s'}\langle\mathbf{w}_{r},\overline{\widetilde{\widetilde{\mathbf{W}}}}^{[L]}\mathbf{w}_{r',s'}^{\dagger}\rangle,\sqrt{m/2}a_{r,d}\right)|\leq\mathfrak{C}_{\varepsilon}(\Phi,\mathcal{O}(\varepsilon_{x}^{-1}))$.

- Since $\mathbf{b}_{s}\sim\mathcal{N}(0,\frac{1}{d_{\text{out}}}\mathbf{I})$, using fact A.3, we can show that $\|\mathbf{b}_{s}\|\leq\mathcal{O}(\frac{\rho}{\sqrt{d_{\text{out}}}})$, w.p. $1-e^{-\rho^{2}}$. Hence, from lemma B.1, we have w.p. atleast $1-e^{-\Omega(\rho^{2})}$, for any $1\leq i\leq j\leq L,s\in[d_{\text{out}}],r\in[m],\left|\mathbf{e}_{s}^{\top}\mathbf{Back}_{i\to j}\mathbf{e}_{r}\right|=\left|\mathbf{b}_{s}^{\top}\mathbf{D}^{(\ell)}\mathbf{W}\cdots\mathbf{D}^{(i+1)}\mathbf{W}\mathbf{e}_{r}\right|\leq\left\|\mathbf{b}_{s}\right\|\left\|\mathbf{D}^{(\ell)}\mathbf{W}\cdots\mathbf{D}^{(i+1)}\mathbf{W}\mathbf{e}_{r}\right\|\leq\mathcal{O}(\frac{\rho}{\sqrt{d_{\text{out}}}})$.

First, let's focus on eq. 42. From Lemma B.2, we have with probability at least $1-e^{-\Omega(\rho^{2})}$,

$$\left|\mathbf{e}_{s}^{\top}\left(\mathbf{Back}_{i\to j}-\widetilde{\mathbf{Back}_{i\to j}}\right)\mathbf{e}_{r}\right|=\left|\mathbf{b}_{s}^{\top}\left(\mathbf{D}^{(j)}\mathbf{W}\cdots\mathbf{D}^{(i)}\mathbf{W}-\widetilde{\mathbf{D}}^{(j)}\widetilde{\mathbf{W}}\cdots\widetilde{\mathbf{D}}^{(i)}\widetilde{\mathbf{W}}\right)\mathbf{e}_{r}\right|$$
$$\leq\left\|\mathbf{b}_{s}\right\|\left\|\left(\mathbf{D}^{(j)}\mathbf{W}\cdots\mathbf{D}^{(i)}\mathbf{W}-\widetilde{\mathbf{D}}^{(j)}\widetilde{\mathbf{W}}\cdots\widetilde{\mathbf{D}}^{(i)}\widetilde{\mathbf{W}}\right)\mathbf{e}_{r}\right\|$$
$$\leq\mathcal{O}(\rho^{7}d_{\text{out}}^{-1/2}N^{1/6}m^{-1/6}),\text{ for all }r\in[m],s\in[d_{\text{out}}]\text{ and }1\leq i\leq j\leq L.$$

Thus,

$$\left|\frac{d_{\text{out}}}{m}\sum_{i=1}^{L}\sum_{s'\in[d_{\text{out}}]}\sum_{r'\in[p]}\sum_{r\in\mathcal{K}}b_{r,s'}b_{r',s'}^{\dagger}\widetilde{\mathbf{Back}}_{i\to L,r,s}H_{r',s'}\left(\widetilde{\theta}_{r',s'}\langle\mathbf{w}_{r},\overline{\widetilde{\widetilde{\mathbf{W}}}}^{[L]}\mathbf{w}_{r',s'}^{\dagger}\rangle,\sqrt{m/2}a_{r,d}\right)\mathbb{I}_{\mathbf{w}_{r}^{\top}\widetilde{\mathbf{h}}^{(i-1)}+\mathbf{a}_{r}^{\top}\mathbf{x}^{(i)}\geq0}\right.$$

$$\left.-\frac{d_{\text{out}}}{m}\sum_{i=1}^{L}\sum_{s'\in[d_{\text{out}}]}\sum_{r'\in[p]}\sum_{r\in\mathcal{K}}b_{r,s'}b_{r',s'}^{\dagger}\mathbf{Back}_{i\to L,r,s}H_{r',s'}\left(\widetilde{\theta}_{r',s'}\langle\mathbf{w}_{r},\overline{\widetilde{\widetilde{\mathbf{W}}}}^{[L]}\mathbf{w}_{r',s'}^{\dagger}\rangle,\sqrt{m/2}a_{r,d}\right)\mathbb{I}_{\mathbf{w}_{r}^{\top}\widetilde{\mathbf{h}}^{(i-1)}+\mathbf{a}_{r}^{\top}\mathbf{x}^{(i)}\geq0}\right|$$

$$\leq\sum_{i=1}^{L}\sum_{r'\in[p]}\sum_{s'\in[d_{\text{out}}]}\sum_{r\in\mathcal{K}}\frac{d_{\text{out}}}{m}\left|\mathbf{e}_{s}^{\top}\left(\mathbf{Back}_{i\to j}-\widetilde{\mathbf{Back}_{i\to j}}\right)\mathbf{e}_{r}\right|$$
$$\cdot\left|b_{r,s'}b_{r',s'}^{\dagger}H_{r',s'}\left(\widetilde{\theta}_{r',s'}\langle\mathbf{w}_{r},\overline{\widetilde{\widetilde{\mathbf{W}}}}^{[L]}\mathbf{w}_{r',s'}^{\dagger}\rangle,\sqrt{m/2}a_{r,d}\right)\mathbb{I}_{\mathbf{w}_{r}^{\top}\widetilde{\mathbf{h}}^{(i-1)}+\mathbf{a}_{r}^{\top}\mathbf{x}^{(i)}\geq0}\right|$$

$$\leq\sum_{i=1}^{L}\sum_{r'\in[p]}\sum_{s'\in[d_{\text{out}}]}\sum_{r\in\mathcal{K}}\frac{d_{\text{out}}}{m}\left|\mathbf{e}_{s}^{\top}\left(\mathbf{Back}_{i\to j}-\widetilde{\mathbf{Back}_{i\to j}}\right)\mathbf{e}_{r}\right|$$
$$\cdot|b_{r,s'}|\left|b_{r',s'}^{\dagger}\right|\left|H_{r',s'}\left(\widetilde{\theta}_{r',s'}\langle\mathbf{w}_{r},\overline{\widetilde{\widetilde{\mathbf{W}}}}^{[L]}\mathbf{w}_{r',s'}^{\dagger}\rangle,\sqrt{m/2}a_{r,d}\right)\right|\left|\mathbb{I}_{\mathbf{w}_{r}^{\top}\widetilde{\mathbf{h}}^{(i-1)}+\mathbf{a}_{r}^{\top}\mathbf{x}^{(i)}\geq0}\right|$$

$$\leq\sum_{i=1}^{L}\sum_{r'\in[p]}\sum_{s'\in[d_{\text{out}}]}\sum_{r\in\mathcal{K}}\frac{d_{\text{out}}}{m}\cdot\mathcal{O}(d_{\text{out}}^{-1/2}\rho^{7}(N/m)^{1/6})\cdot\frac{\rho}{\sqrt{d_{\text{out}}}}\cdot1\cdot\mathfrak{C}_{\varepsilon}(\Phi,\mathcal{O}(\varepsilon_{x}^{-1}))\cdot1 \qquad(45)$$

$$\leq \mathcal{O}(d_{\text{out}} p L \rho^8 \mathfrak{C}_\varepsilon(\Phi_{r',s'}, \mathcal{O}(\varepsilon_x^{-1})) (N/m)^{7/6}). \tag{46}$$

Now, we focus on eq. 43. Lemma B.2 shows that with probability at least $1 - e^{-\Omega(\rho^2)}$,

$$|\mathbf{w}_r^\top (\widetilde{\mathbf{h}}^{(L-1)} - \mathbf{h}^{(L-1)})| \leq \mathcal{O}\left(\rho^5 N^{2/3} m^{-2/3}\right) \quad \text{for every } r \in [m], \ell \in [L].$$

From lemma B.1, we have w.p. at least $1 - e^{-\Omega(\rho^2)}$ for any $s \leq \frac{\rho^2}{m}$,

$$\left|\left\{r \in [m] \Big| \left|\mathbf{w}_r^\top \mathbf{h}^{(L-1)} + \mathbf{a}_r^\top \mathbf{x}^{(L-1)}\right| \leq \frac{s}{\sqrt{m}}\right\}\right| \leq \mathcal{O}(sm).$$

This can be modified for the subset $\mathcal{K}$, w.p. at least $1 - e^{-\Omega(\rho^2)}$ for any $s \leq \frac{\rho^2}{m}$,

$$\left|\left\{r \in \mathcal{K} \Big| \left|\mathbf{w}_r^\top \mathbf{h}^{(L-1)} + \mathbf{a}_r^\top \mathbf{x}^{(L-1)}\right| \leq \frac{s}{\sqrt{m}}\right\}\right| \leq \mathcal{O}(sN).$$

Thus, w.p. at least $1 - e^{-\Omega(\rho^2)}$,

$$\sum_{r \in \mathcal{K}} \mathbb{I}\left[\left|\mathbf{w}_r^\top \mathbf{h}^{(L-1)} + \mathbf{a}_r^\top \mathbf{x}^{(L-1)}\right| \leq \rho^5 N^{2/3} m^{-2/3}\right] \leq \mathcal{O}(\rho^5 N^{5/3} m^{-1/6}).$$

Hence, that implies w.p. at least $1 - e^{-\Omega(\rho^2)}$,

$$\sum_{r \in \mathcal{K}} \left|\mathbb{I}\left[\mathbf{w}_r^\top \mathbf{h}^{(L-1)} + \mathbf{a}_r^\top \mathbf{x}^{(L-1)}\right] - \mathbb{I}\left[\mathbf{w}_r^\top \widetilde{\mathbf{h}}^{(L-1)} + \mathbf{a}_r^\top \mathbf{x}^{(L-1)}\right]\right|$$

$$\leq \sum_{r \in \mathcal{K}} \mathbb{I}\left[|\mathbf{w}_r^\top \mathbf{h}^{(L-1)} + \mathbf{a}_r^\top \mathbf{x}^{(L-1)}| \leq |\mathbf{w}_r^\top \widetilde{\mathbf{h}}^{(L-1)} + \mathbf{a}_r^\top \mathbf{x}^{(L-1)} - \mathbf{w}_r^\top \mathbf{h}^{(L-1)} - \mathbf{a}_r^\top \mathbf{x}^{(L-1)}|\right]$$

$$= \sum_{r \in \mathcal{K}} \mathbb{I}\left[|\mathbf{w}_r^\top \mathbf{h}^{(L-1)} + \mathbf{a}_r^\top \mathbf{x}^{(L-1)}| \leq |\mathbf{w}_r^\top (\widetilde{\mathbf{h}}^{(L-1)} - \mathbf{h}^{(L-1)}|)\right]$$

$$\leq \sum_{r \in \mathcal{K}} \mathbb{I}\left[|\mathbf{w}_r^\top \mathbf{h}^{(L-1)} + \mathbf{a}_r^\top \mathbf{x}^{(L-1)}| \leq \mathcal{O}\left(\rho^5 N^{2/3} m^{-2/3}\right)\right]$$

$$\leq \mathcal{O}(\rho^5 N^{5/3} m^{-1/6}).$$

Thus, we have in eq. 43,

$$\left|\frac{d_{\text{out}}}{m} \sum_{i=1}^{L} \sum_{s' \in [d_{\text{out}}]} \sum_{r' \in [p]} \sum_{r \in \mathcal{K}} b_{r,s'} b_{r',s'}^\dagger \mathbf{Back}_{i \to L, r, s} H_{r',s'}\left(\widetilde{\theta}_{r',s'} \langle \mathbf{w}_r, \overline{\widetilde{\mathbf{W}}}^{[L]} \mathbf{w}_{r',s'}^\dagger \rangle, \sqrt{m/2} a_{r,d}\right) \mathbb{I}_{\mathbf{w}_r^\top \widetilde{\mathbf{h}}^{(i-1)} + \mathbf{a}_r^\top \mathbf{x}^{(i)} \geq 0}\right.$$

$$\left.- \frac{d_{\text{out}}}{m} \sum_{i=1}^{L} \sum_{s' \in [d_{\text{out}}]} \sum_{r' \in [p]} \sum_{r \in \mathcal{K}} b_{r,s'} b_{r',s'}^\dagger \mathbf{Back}_{i \to L, r, s} H_{r',s'}\left(\widetilde{\theta}_{r',s'} \langle \mathbf{w}_r, \overline{\widetilde{\mathbf{W}}}^{[L]} \mathbf{w}_{r',s'}^\dagger \rangle, \sqrt{m/2} a_{r,d}\right) \mathbb{I}_{\mathbf{w}_r^\top \mathbf{h}^{(i-1)} + \mathbf{a}_r^\top \mathbf{x}^{(i)} \geq 0}\right|$$

$$\leq \max_{r,s}|b_{r,s}| \cdot \max_{r',s'}|b_{r',s'}^\dagger| \cdot \max_{i,r,s}|\mathbf{Back}_{i \to L, r, s}| \cdot \max_{r',s',r}\left|H_{r',s'}\left(\widetilde{\theta}_{r',s'} \langle \mathbf{w}_r, \overline{\widetilde{\mathbf{W}}}^{[L]} \mathbf{w}_{r',s'}^\dagger \rangle, \sqrt{m/2} a_{r,d}\right)\right|$$

$$\cdot \frac{d_{\text{out}}}{m} \sum_{i=1}^{L} \sum_{s' \in [d_{\text{out}}]} \sum_{r' \in [p]} \sum_{r \in \mathcal{K}} \left|\mathbb{I}_{\mathbf{w}_r^\top \mathbf{h}^{(i-1)} + \mathbf{a}_r^\top \mathbf{x}^{(i)} \geq 0} - \mathbb{I}_{\mathbf{w}_r^\top \widetilde{\mathbf{h}}^{(i-1)} + \mathbf{a}_r^\top \mathbf{x}^{(i)} \geq 0}\right|$$

$$\leq \frac{\rho}{\sqrt{d_{\text{out}}}} \cdot 1 \cdot \mathcal{O}(\frac{\rho}{\sqrt{d_{\text{out}}}}) \cdot \mathfrak{C}_\varepsilon(\Phi, \mathcal{O}(\varepsilon_x^{-1})) \cdot \frac{d_{\text{out}}^2 L p}{m} \cdot \mathcal{O}(\rho^4 N^{5/3} m^{1/6}) \leq \mathcal{O}(d_{\text{out}} L p \rho^6 N^{5/3} m^{-7/6} \mathfrak{C}_\varepsilon(\Phi, \mathcal{O}(\varepsilon_x^{-1}))). \tag{47}$$

Now, we focus on eq. 44. We have

$$\left|\frac{d_{\text{out}}}{m} \sum_{i=1}^{L} \sum_{s' \in [d_{\text{out}}]} \sum_{r' \in [p]} \sum_{r \in \mathcal{K}} b_{r,s'} b_{r',s'}^\dagger \mathbf{Back}_{i \to L, r, s} H_{r',s'}\left(\widetilde{\theta}_{r',s'} \langle \mathbf{w}_r, \overline{\widetilde{\mathbf{W}}}^{[L]} \mathbf{w}_{r',s'}^\dagger \rangle, \sqrt{m/2} a_{r,d}\right) \mathbb{I}_{\mathbf{w}_r^\top \mathbf{h}^{(i-1)} + \mathbf{a}_r^\top \mathbf{x}^{(i)} \geq 0}\right.$$

$$- \frac{d_{\text{out}}}{m} \sum_{i=1}^{L} \sum_{s' \in [d_{\text{out}}]} \sum_{r' \in [p]} \sum_{r \in \mathcal{K}} b_{r,s'} b_{r',s'}^{\dagger} \mathbf{Back}_{i \to L, r, s} H_{r',s'} \left( \theta_{r',s'} \langle \mathbf{w}_r, \overline{\mathbf{W}}^{[L]} \mathbf{w}_{r',s'}^{\dagger} \rangle, \sqrt{m/2} a_{r,d} \right) \mathbb{I}_{\mathbf{w}_r^{\top} \mathbf{h}^{(i-1)} + \mathbf{a}_r^{\top} \mathbf{x}^{(i)} \geq 0} \Big|$$

$$\leq \max_{r,s'} |b_{r,s'}| \cdot \max_{r',s'} |b_{r',s'}^{\dagger}| \cdot \max_{i,r,s} |\mathbf{Back}_{i \to L, r, s}| \cdot \max_{i,r} |\mathbb{I}_{\mathbf{w}_r^{\top} \mathbf{h}^{(i-1)} + \mathbf{a}_r^{\top} \mathbf{x}^{(i)} \geq 0}|$$

$$\cdot \frac{d_{\text{out}}}{m} \sum_{i=1}^{L} \sum_{s' \in [d_{\text{out}}]} \sum_{r' \in [p]} \sum_{r \in \mathcal{K}} \left| H_{r',s'} \left( \theta_{r',s'} \langle \mathbf{w}_r, \overline{\mathbf{W}}^{[L]} \mathbf{w}_{r',s'}^{\dagger} \rangle, \sqrt{m/2} a_{r,d} \right) - H_{r',s'} \left( \widetilde{\theta}_{r',s'} \langle \mathbf{w}_r, \widetilde{\overline{\mathbf{W}}}^{[L]} \mathbf{w}_{r',s'}^{\dagger} \rangle, \sqrt{m/2} a_{r,d} \right) \right|$$

$$\leq \frac{\rho}{\sqrt{d_{\text{out}}}} \cdot 1 \cdot \mathcal{O}(\frac{\rho}{\sqrt{d_{\text{out}}}}) \cdot 1 \tag{48}$$

$$\cdot \frac{d_{\text{out}}}{m} \sum_{i=1}^{L} \sum_{s' \in [d_{\text{out}}]} \sum_{r' \in [p]} \sum_{r \in \mathcal{K}} \cdot \mathfrak{C}_{\varepsilon}(\Phi_{r',s'}, \mathcal{O}(\varepsilon_x^{-1})) \cdot \left| \theta_{r',s'} \langle \mathbf{w}_r, \overline{\mathbf{W}}^{[L]} \mathbf{w}_{r',s'}^{\dagger} \rangle - \widetilde{\theta}_{r',s'} \langle \mathbf{w}_r, \widetilde{\overline{\mathbf{W}}}^{[L]} \mathbf{w}_{r',s'}^{\dagger} \rangle \right|$$

$$\tag{49}$$

$$\leq \frac{\rho}{\sqrt{d_{\text{out}}}} \cdot 1 \cdot \mathcal{O}(\frac{\rho}{\sqrt{d_{\text{out}}}}) \cdot 1 \cdot \frac{d_{\text{out}}^2 pLN}{m} \cdot \mathfrak{C}_{\varepsilon}(\Phi, \mathcal{O}(\varepsilon_x^{-1})) \cdot \mathcal{O}(\rho^6 N^{2/3} m^{-2/3}) \tag{50}$$

$$\leq \mathcal{O}(d_{\text{out}} Lp\rho^8 \mathfrak{C}_{\varepsilon}(\Phi, \mathcal{O}(\varepsilon_x^{-1})) N^{5/3} m^{-5/3}),$$

where we get eq. 49 by using the lipschitz continuity of the function $H_{r',s'}$ from def. F.10. We get eq. 50 by bounding the following term:

$$\left| \theta_{r',s'} \langle \mathbf{w}_r, \overline{\mathbf{W}}^{[L]} \mathbf{w}_{r',s'}^{\dagger} \rangle - \widetilde{\theta}_{r',s'} \langle \mathbf{w}_r, \widetilde{\overline{\mathbf{W}}}^{[L]} \mathbf{w}_{r',s'}^{\dagger} \rangle \right|$$

$$\leq \left| \theta_{r',s'} \left( \langle \mathbf{w}_r, \overline{\mathbf{W}}^{[L]} \mathbf{w}_{r',s'}^{\dagger} \rangle - \langle \mathbf{w}_r, \widetilde{\overline{\mathbf{W}}}^{[L]} \mathbf{w}_{r',s'}^{\dagger} \rangle \right) \right| + \left| \left( \theta_{r',s'} - \widetilde{\theta}_{r',s'} \right) \langle \mathbf{w}_r, \widetilde{\overline{\mathbf{W}}}^{[L]} \mathbf{w}_{r',s'}^{\dagger} \rangle \right|$$

$$\leq |\theta_{r',s'}| \cdot \left| \left\langle \mathbf{w}_r, \left( \overline{\mathbf{W}}^{[L]} - \widetilde{\overline{\mathbf{W}}}^{[L]} \right) \mathbf{w}_{r',s'}^{\dagger} \right\rangle \right| + \left| \theta_{r',s'} - \widetilde{\theta}_{r',s'} \right| \cdot \left| \langle \mathbf{w}_r, \widetilde{\overline{\mathbf{W}}}^{[L]} \mathbf{w}_{r',s'}^{\dagger} \rangle \right|$$

$$\leq \mathcal{O}(\rho^6 (N/m)^{2/3}),$$

where we use the following bounds that are true for all $r \in \mathcal{K}, r' \in [p], s' \in [d_{\text{out}}]$ w.p. at least $1 - e^{-\Omega(\rho^2)}$:

- Eq. 37 gives an upper bound of $O(1)$ on $|\theta_{r',s'}|$.

- Eq. 36 can be easily modified to get a similar upper bound on $|\langle \mathbf{w}_r, (\overline{\mathbf{W}}^{[L]} - \widetilde{\overline{\mathbf{W}}}^{[L]}) \mathbf{w}_{r',s'}^{\dagger} \rangle|$.

- Cor. F.5.1 gives an upper bound on $|\theta_{r',s'} - \widetilde{\theta}_{r',s'}|$.

- Since $\left\| \widetilde{\overline{\mathbf{W}}}^{[L]} \mathbf{w}_{r',s'}^{\dagger} \right\| := \sqrt{m/2} \widetilde{\theta}_{r',s'}^{-1}$, we can use Eq. 41 to give an upper bound on the norm.
  Then, we can use Fact A.4 to bound $\max_{r \in \mathcal{K}} |\langle \mathbf{w}_r, \widetilde{\overline{\mathbf{W}}}^{[L]} \mathbf{w}_{r',s'}^{\dagger} \rangle| = \frac{\rho}{\sqrt{m}} \cdot \left\| \widetilde{\overline{\mathbf{W}}}^{[L]} \mathbf{w}_{r',s'}^{\dagger} \right\|$.

$$\square$$

**Claim F.14** (Restating claim F.7). *With probability at least $1 - e^{-\Omega(\rho^2)}$,*

$$\left| \widetilde{F}_s^{(L), \mathcal{K}}(\widetilde{\mathbf{h}}^{(L-1)}, \mathbf{x}^{(L)}) - \frac{d_{\text{out}}}{m} \sum_{i=1}^{L} \sum_{s' \in [d_{\text{out}}]} \sum_{r' \in [p]} \sum_{r \in \mathcal{K}} b_{r,s'} b_{r',s'}^{\dagger} \widetilde{\mathbf{Back}}_{i \to L, r, s} \Phi_{r',s} \left( \left\langle \mathbf{w}_{r',s}^{\dagger}, [\overline{\mathbf{x}}^{(1)}, \cdots, \overline{\mathbf{x}}^{(L)}] \right\rangle \right) \right|$$

$$\leq \frac{d_{\text{out}}}{m} \cdot \mathcal{O}(\mathfrak{C}_{\varepsilon}(\Phi_{r's}, \mathcal{O}(\varepsilon_x^{-1})) \rho^2 \sqrt{d_{\text{out}} LpN})$$

$$+ \frac{d_{\text{out}} LpN}{m} \rho^2 (\varepsilon + \mathcal{O}(L_\Phi \rho^5 (N/m)^{1/6}) + \mathcal{O}(L_\Phi \varepsilon_x^{-1} L^4 \rho^{11} m^{-1/12} + L_\Phi \rho^2 L^{11/6} \varepsilon_x^{2/3})),$$

*for any* $\varepsilon \in (0, \min_{r,s} \frac{\sqrt{3}}{C_s(\Phi_{r,s}, \varepsilon_x^{-1})})$.

*Proof.* We will take the expectation w.r.t. the weights $\{\mathbf{w}_r, \mathbf{a}_r\}_{r \in \mathcal{K}}$. The difference between $\widetilde{F}$ and the expected value is given by

$$\left| \widetilde{F}_s^{(L),\mathcal{K}}(\widetilde{\mathbf{h}}^{(L-1)}, \mathbf{x}^{(L)}) - \mathbb{E}_{\{\mathbf{w}_r, \mathbf{a}_r\}_{r \in \mathcal{K}}} \widetilde{F}_s^{(L),\mathcal{K}}(\widetilde{\mathbf{h}}^{(L-1)}, \mathbf{x}^{(L)}) \right|$$

$$= \left| \frac{d_{\text{out}}}{m} \sum_{i=1}^{L} \sum_{s' \in [d_{\text{out}}]} \sum_{r' \in [p]} \sum_{r \in \mathcal{K}} b_{r,s'} b_{r',s'}^{\dagger} \widetilde{\mathbf{Back}}_{i \to L, r, s} H_{r',s}\left( \widetilde{\theta}_{r',s} \langle \mathbf{w}_r, \overline{\overline{\widetilde{\mathbf{W}}}}^{[L]} \mathbf{w}_{r',s}^{\dagger} \rangle, \sqrt{m/2} a_{r,d} \right) \mathbb{I}_{\mathbf{w}_r^{\top} \widetilde{\mathbf{h}}^{(L-1)} + \mathbf{a}_r^{\top} \mathbf{x}^{(L)} \geq 0} \right.$$

$$- \mathbb{E}_{[\mathbf{w},\mathbf{a}] \sim \mathcal{N}(0, \frac{2}{m}\mathbf{I})} \frac{d_{\text{out}}}{m} \sum_{i=1}^{L} \sum_{s' \in [d_{\text{out}}]} \sum_{r' \in [p]} \sum_{r \in \mathcal{K}} b_{r,s'} b_{r',s'}^{\dagger} \widetilde{\mathbf{Back}}_{i \to L, r, s}$$

$$\left. \cdot H_{r',s}\left( \widetilde{\theta}_{r',s} \langle \mathbf{w}, \overline{\overline{\widetilde{\mathbf{W}}}}^{[L]} \mathbf{w}_{r',s}^{\dagger} \rangle, \sqrt{m/2} a_d \right) \mathbb{I}_{\mathbf{w}^{\top} \widetilde{\mathbf{h}}^{(L-1)} + \mathbf{a}^{\top} \mathbf{x}^{(L)} \geq 0} \right|$$

$$= \left| \frac{d_{\text{out}}}{m} \sum_{i=1}^{L} \sum_{s' \in [d_{\text{out}}]} \sum_{r' \in [p]} \sum_{r \in \mathcal{K}} b_{r,s'} b_{r',s'}^{\dagger} \widetilde{\mathbf{Back}}_{i \to L, r, s} H_{r',s}\left( \widetilde{\theta}_{r',s} \langle \mathbf{w}_r, \overline{\overline{\widetilde{\mathbf{W}}}}^{[L]} \mathbf{w}_{r',s}^{\dagger} \rangle, \sqrt{m/2} a_{r,d} \right) \mathbb{I}_{\mathbf{w}_r^{\top} \widetilde{\mathbf{h}}^{(L-1)} + \mathbf{a}_r^{\top} \mathbf{x}^{(L)} \geq 0} \right.$$

$$- \frac{d_{\text{out}}}{m} \sum_{i=1}^{L} \sum_{s' \in [d_{\text{out}}]} \sum_{r' \in [p]} \sum_{r \in \mathcal{K}} b_{r,s'} b_{r',s'}^{\dagger} \widetilde{\mathbf{Back}}_{i \to L, r, s}$$

$$\left. \cdot \mathbb{E}_{\mathbf{w},\mathbf{a} \sim \mathcal{N}(0, \frac{2}{m}\mathbf{I})} H_{r',s}\left( \widetilde{\theta}_{r',s} \langle \mathbf{w}, \overline{\overline{\widetilde{\mathbf{W}}}}^{[L]} \mathbf{w}_{r',s}^{\dagger} \rangle, \sqrt{m/2} a_d \right) \mathbb{I}_{\mathbf{w}^{\top} \widetilde{\mathbf{h}}^{(L-1)} + \mathbf{a}^{\top} \mathbf{x}^{(L)} \geq 0} \right|, \tag{51}$$

where in the final step, we have used the fact that $\widetilde{\mathbf{Back}}$ and $\mathbf{B}$ are independent of the variables $\{\mathbf{w}_r, \mathbf{a}_r\}_{r \in \mathcal{K}}$ w.r.t. which we are taking the expectation.

Note that, the random variable under consideration is a bounded random variable, because: using the fact that $b_{r,s'} \sim \mathcal{N}(0,1)$, it is bounded by $\rho$ with high probability, $\widetilde{\mathbf{Back}}_{i \to L, r, s} = \mathbf{b}_s^{\top}(\prod_{i \leq \ell \leq L} \mathbf{D}^{(\ell)} \mathbf{W}) \mathbf{e}_r$ is bounded by $\mathcal{O}(\rho)$ using bound on norm of $\mathbf{b}_s$ and Claim B.1, and the function $H$ is bounded by def. F.10. Denoting the inequality in eq. 51 as $P(\{\mathbf{w}_r, \mathbf{a}_r\}_{r \in \mathcal{K}})$, we get using hoeffding's inequality for bounded variables (fact A.5)

$$\Pr\left[ P(\{\mathbf{w}_r, \mathbf{a}_r\}_{r \in \mathcal{K}}) > \frac{d_{\text{out}}}{m} \cdot \mathcal{O}(\mathfrak{C}_\varepsilon(\Phi_{r's}, \mathcal{O}(\varepsilon_x^{-1})) \rho^2 \sqrt{d_{\text{out}} L p N}) \right] \leq e^{-\rho^2/8}.$$

Now, we focus on the expected value in eq. 51. For typographical simplicity in the next few steps, we denote the vector $\mathbf{v} = \widetilde{\mathbf{h}}^{(L-1)}$, vector $\mathbf{q} = \sqrt{m/2} \cdot \mathbf{w}$ and vector $\mathbf{t} = \widetilde{\theta}_{r',s} \cdot \sqrt{2/m} \cdot \overline{\overline{\widetilde{\mathbf{W}}}}^{[L]} \mathbf{w}_{r',s}^{\dagger}$. Also, let $\mathbf{t}^{\perp}$ denote a vector in the subspace orthogonal to $\mathbf{t}$ that is closest to the vector $\mathbf{v}$.

By the definition of the function $H_{r',s}$ from Def. F.10, where we use $k_{0,r,s} = \varepsilon_x^{-1} \sqrt{m/2} \theta_{r',s}^{-1}$ for each $r' \in [p], s \in [d_{\text{out}}]$ in def. F.10 ($\theta_{r',s}$ is defined in def. D.2), we have

$$\mathbb{E}_{\mathbf{w},\mathbf{a} \sim \mathcal{N}(0, \frac{2}{m}\mathbf{I})} H_{r',s}\left( \widetilde{\theta}_{r',s} \langle \mathbf{w}, \overline{\overline{\widetilde{\mathbf{W}}}}^{[L]} \mathbf{w}_{r',s}^{\dagger} \rangle, \sqrt{m/2} a_d \right) \cdot \mathbb{I}\left[ \langle \mathbf{w}, \widetilde{\mathbf{h}}^{(L-1)} \rangle + a_d \geq 0 \right]$$

$$= \mathbb{E}_{\mathbf{w},\mathbf{a} \sim \mathcal{N}(0, \frac{2}{m}\mathbf{I})} H_{r',s}\left( \widetilde{\theta}_{r',s} \langle \mathbf{w}, \overline{\overline{\widetilde{\mathbf{W}}}}^{[L]} \mathbf{w}_{r',s}^{\dagger} \rangle, \sqrt{m/2} a_d \right) \cdot \mathbb{I}\left[ \sqrt{m/2} \langle \mathbf{w}, \widetilde{\mathbf{h}}^{(L-1)} \rangle + \sqrt{m/2} a_d \geq 0 \right]$$

$$= \mathbb{E}_{\mathbf{w},\mathbf{a} \sim \mathcal{N}(0, \frac{2}{m}\mathbf{I})} H_{r',s}\left( \langle \mathbf{q}, \mathbf{t} \rangle, \sqrt{m/2} a_d \right) \cdot \mathbb{I}\left[ \langle \mathbf{v}, \mathbf{q} \rangle + \sqrt{m/2} a_d \geq 0 \right]$$

$$= \mathbb{E}_{\mathbf{w},\mathbf{a} \sim \mathcal{N}(0, \frac{2}{m}\mathbf{I})} H_{r',s}\left( \langle \mathbf{q}, \mathbf{t} \rangle, \sqrt{m/2} a_d \right) \cdot \mathbb{I}\left[ \langle \mathbf{v}, \mathbf{t} \rangle \langle \mathbf{q}, \mathbf{t} \rangle + \sqrt{\|\mathbf{v}\|^2 - \langle \mathbf{v}, \mathbf{t} \rangle^2} \langle \mathbf{q}, \mathbf{t}^{\perp} \rangle + \sqrt{m/2} a_d \geq 0 \right]$$

$$= \Phi_{r',s}\left( \varepsilon_x^{-1}(\sqrt{m/2} \theta_{r',s}^{-1}) \langle \mathbf{t}, \mathbf{v} \rangle \right) \pm \varepsilon$$

$$= \Phi_{r',s}\left( \varepsilon_x^{-1} \theta_{r',s}^{-1} \widetilde{\theta}_{r',s} \langle \overline{\overline{\widetilde{\mathbf{W}}}}^{[L]} \mathbf{w}_{r',s}^{\dagger}, \widetilde{\mathbf{h}}^{(L-1)} \rangle \right) \pm \varepsilon$$

$$= \Phi_{r',s}\left(\varepsilon_x^{-1}\langle\overline{\overline{\widetilde{\mathbf{W}}}}^{[L]}\mathbf{w}_{r',s}^\dagger, \widetilde{\mathbf{h}}^{(L-1)}\rangle \pm \mathcal{O}(\rho^5(N/m)^{1/6})\right) \pm \varepsilon$$

$$= \Phi_{r',s}\left(\varepsilon_x^{-1}\langle\overline{\overline{\widetilde{\mathbf{W}}}}^{[L]}\mathbf{w}_{r',s}^\dagger, \widetilde{\mathbf{h}}^{(L-1)}\rangle\right) \pm \varepsilon \pm \mathcal{O}(L_{\Phi_{r',s}}\rho^5(N/m)^{1/6}),$$

where in the pre-final step, we have used claim F.5 to bound the value of $\theta_{r',s}^{-1}\widetilde{\theta}_{r',s}$ and in the final step, we have used the lipschitz constant of $\Phi_{r',s}$ in the desired range. Corollary D.1.1 shows that with probability at least $1 - e^{-\Omega(\rho^2)}$ w.r.t. the weights $\widetilde{\mathbf{W}}$ and $\widetilde{\mathbf{A}}$,

$$\left|\overline{\overline{\widetilde{\mathbf{W}}}}^{[L]\top}\widetilde{h}^{(L-1)} - \varepsilon_x[\overline{\mathbf{x}}^{(2)}, \cdots, \overline{\mathbf{x}}^{(L-1)}]\right|$$

$$\leq \mathcal{O}\left(L^4 \cdot (\rho^{11}m^{-1/12} + \rho^7 m^{-1/12} + \rho^7 m^{-1/4} + \rho^{11}m^{-1/4})\right) + \mathcal{O}(\rho^2 L^{11/6}\varepsilon_x^{2/3})$$

$$\leq \mathcal{O}(L^4\rho^{11}m^{-1/12} + \rho^2 L^{11/6}\varepsilon_x^{5/3}).$$

Thus,

$$\mathbb{E}_{\mathbf{w},\mathbf{a}\sim\mathcal{N}(0,\frac{2}{m}\mathbf{I})}H_{r',s}\left(\widetilde{\theta}_{r',s}\langle\mathbf{w}, \overline{\overline{\widetilde{\mathbf{W}}}}^{[L]}\mathbf{w}_{r',s}^\dagger\rangle, \sqrt{m/2}a_d\right) \cdot \mathbb{I}\left[\langle\mathbf{w}, \widetilde{\mathbf{h}}^{(L-1)}\rangle + a_d \geq 0\right]$$

$$= \Phi_{r',s}\left(\varepsilon_x^{-1}\langle\overline{\overline{\widetilde{\mathbf{W}}}}^{[L]}\mathbf{w}_{r',s}^\dagger, \widetilde{\mathbf{h}}^{(L-1)}\rangle\right) \pm \varepsilon \pm \mathcal{O}(L_{\Phi_{r',s}}\rho^5(N/m)^{1/6})$$

$$= \Phi_{r',s}\left(\varepsilon_x^{-1}\langle\mathbf{w}_{r',s}^\dagger, \overline{\overline{\widetilde{\mathbf{W}}}}^{[L]\top}\widetilde{\mathbf{h}}^{(L-1)}\rangle\right) \pm \varepsilon \pm \mathcal{O}(L_{\Phi_{r',s}}\rho^5(N/m)^{1/6})$$

$$= \Phi_{r',s}\left(\varepsilon_x^{-1}\langle\mathbf{w}_{r',s}^\dagger, \varepsilon_x[\overline{\mathbf{x}}^{(2)}, \cdots, \overline{\mathbf{x}}^{(L-1)}]\rangle \pm \varepsilon'\right) \pm \varepsilon \pm \mathcal{O}(L_{\Phi_{r',s}}\rho^5(N/m)^{1/6})$$

$$= \Phi_{r',s}\left(\langle\mathbf{w}_{r',s}^\dagger, [\overline{\mathbf{x}}^{(2)}, \cdots, \overline{\mathbf{x}}^{(L-1)}]\rangle\right) \pm \varepsilon \pm \mathcal{O}(L_{\Phi_{r',s}}\rho^5(N/m)^{1/6}) \pm L_{\Phi_{r',s}}\varepsilon',$$

where $\varepsilon \in (0, \min_{r,s}\frac{\sqrt{3}}{C_s(\Phi_{r,s},\varepsilon_x^{-1})})$ and $\varepsilon' = \mathcal{O}(\varepsilon_x^{-1}L^4\rho^{11}m^{-1/12} + \rho^2 L^{11/6}\varepsilon_x^{2/3})$. Thus, we have

$$\left|\frac{d_{\text{out}}}{m}\sum_{i=1}^{L}\sum_{s'\in[d_{\text{out}}]}\sum_{r'\in[p]}\sum_{r\in\mathcal{K}}b_{r,s'}b_{r',s'}^\dagger\widetilde{\mathbf{Back}}_{i\to L,r,s}\right.$$

$$\cdot\mathbb{E}_{\mathbf{w},\mathbf{a}\sim\mathcal{N}(0,\frac{2}{m}\mathbf{I})}H_{r',s}\left(\widetilde{\theta}_{r',s}\langle\mathbf{w}, \overline{\overline{\widetilde{\mathbf{W}}}}^{[L]}\mathbf{w}_{r',s}^\dagger\rangle, \sqrt{m/2}a_d\right)\mathbb{I}_{\mathbf{w}^\top\widetilde{\mathbf{h}}^{(L-1)}+\mathbf{a}^\top\mathbf{x}^{(L)}\geq 0}$$

$$\left.-\frac{d_{\text{out}}}{m}\sum_{i=1}^{L}\sum_{s'\in[d_{\text{out}}]}\sum_{r'\in[p]}\sum_{r\in\mathcal{K}}b_{r,s'}b_{r',s'}^\dagger\widetilde{\mathbf{Back}}_{i\to L,r,s}\Phi_{r',s}\left(\langle\mathbf{w}_{r',s}^\dagger, [\overline{\mathbf{x}}^{(2)}, \cdots, \overline{\mathbf{x}}^{(L-1)}]\rangle\right)\right| \qquad (52)$$

$$\leq \frac{d_{\text{out}}}{m}\sum_{i=1}^{L}\sum_{s'\in[d_{\text{out}}]}\sum_{r'\in[p]}\sum_{r\in\mathcal{K}}b_{r,s'}b_{r',s'}^\dagger\widetilde{\mathbf{Back}}_{i\to L,r,s}\cdot(\varepsilon + \mathcal{O}(L_{\Phi_{r',s}}\rho^5(N/m)^{1/6}) + L_{\Phi_{r',s}}\varepsilon')$$

$$\leq \frac{d_{\text{out}}}{m}\sum_{i=1}^{L}\sum_{s'\in[d_{\text{out}}]}\sum_{r'\in[p]}\sum_{r\in\mathcal{K}}\left|b_{r,s'}\right|\left|b_{r',s'}^\dagger\right|\left|\widetilde{\mathbf{Back}}_{i\to L,r,s}\right|\cdot(\varepsilon + \max_{r',s}L_{\Phi_{r',s}}\varepsilon' + \mathcal{O}(L_{\Phi_{r',s}}\rho^5(N/m)^{1/6}))$$

$$\leq \frac{d_{\text{out}}LpN}{m}\rho^2(\varepsilon + \mathcal{O}(L_\Phi\rho^5(N/m)^{1/6}) + L_\Phi\varepsilon'), \qquad (53)$$

where $\varepsilon \in (0, \min_{r,s}\frac{\sqrt{3}}{C_s(\Phi_{r,s},\varepsilon_x^{-1})})$ and $\varepsilon' = \mathcal{O}(\varepsilon_x^{-1}L^4\rho^{11}m^{-1/12} + \rho^2 L^{11/6}\varepsilon_x^{2/3})$. Hence, using eq. 53 and eq. 51, we have w.p. at least $1 - e^{-\Omega(\rho^2)}$,

$$\left|\widetilde{F}_s^{(L),\mathcal{K}}(\widetilde{\mathbf{h}}^{(L-1)}, \mathbf{x}^{(L)}) - \frac{d_{\text{out}}}{m}\sum_{i=1}^{L}\sum_{s'\in[d_{\text{out}}]}\sum_{r'\in[p]}\sum_{r\in\mathcal{K}}b_{r,s'}b_{r',s'}^\dagger\widetilde{\mathbf{Back}}_{i\to L,r,s}\Phi_{r',s}\left(\left\langle\mathbf{w}_{r',s}^\dagger, [\overline{\mathbf{x}}^{(1)}, \cdots, \overline{\mathbf{x}}^{(L)}]\right\rangle\right)\right|$$

$$\leq \left| \widetilde{F}_s^{(L),\mathcal{K}}(\widetilde{\mathbf{h}}^{(L-1)}, \mathbf{x}^{(L)}) - \mathbb{E}_{\{\mathbf{w}_r, \mathbf{a}_r\}_{r\in\mathcal{K}}} \widetilde{F}_s^{(L),\mathcal{K}}(\widetilde{\mathbf{h}}^{(L-1)}, \mathbf{x}^{(L)}) \right|$$

$$+ \left| \mathbb{E}_{\{\mathbf{w}_r, \mathbf{a}_r\}_{r\in\mathcal{K}}} \widetilde{F}_s^{(L),\mathcal{K}}(\widetilde{\mathbf{h}}^{(L-1)}, \mathbf{x}^{(L)}) \right.$$

$$\left. - \frac{d_{\text{out}}}{m} \sum_{i=1}^{L} \sum_{s'\in[d_{\text{out}}]} \sum_{r'\in[p]} \sum_{r\in\mathcal{K}} b_{r,s'} b_{r',s'}^\dagger \widetilde{\mathbf{Back}}_{i\to L,r,s} \Phi_{r',s} \left( \left\langle \mathbf{w}_{r',s}^\dagger, [\overline{\mathbf{x}}^{(1)}, \cdots, \overline{\mathbf{x}}^{(L)}] \right\rangle \right) \right|$$

$$\leq \frac{d_{\text{out}} L p N}{m} \rho^2 (\varepsilon + \mathcal{O}(L_\Phi \rho^5 (N/m)^{1/6}) + L_\Phi \varepsilon') + \frac{d_{\text{out}}}{m} \cdot \mathcal{O}(\mathfrak{C}_\varepsilon(\Phi_{r's}, \mathcal{O}(\varepsilon_x^{-1}))\rho^2 \sqrt{d_{\text{out}} L p N}),$$

where $\varepsilon \in (0, \min_{r,s} \frac{\sqrt{3}}{C_s(\Phi_{r,s,\varepsilon_x^{-1}})})$ and $\varepsilon' = \mathcal{O}(\varepsilon_x^{-1} L^4 \rho^{11} m^{-1/12} + \rho^2 L^{11/6} \varepsilon_x^{2/3})$.

$\square$

**Claim F.15** (Restating claim F.8). *With probability at least* $1 - e^{-\Omega(\rho^2)}$,

$$\left| \frac{d_{\text{out}}}{m} \sum_{i=1}^{L} \sum_{s'\in[d_{\text{out}}]} \sum_{r'\in[p]} \sum_{r\in\mathcal{K}} b_{r,s'} b_{r',s'}^\dagger \widetilde{\mathbf{Back}}_{i\to L,r,s} \Phi_{r',s} \left( \left\langle \mathbf{w}_{r',s}^\dagger, [\overline{\mathbf{x}}^{(1)}, \cdots, \overline{\mathbf{x}}^{(L)}] \right\rangle \right) \right.$$

$$\left. - \frac{d_{\text{out}}}{m} \sum_{i=1}^{L} \sum_{s'\in[d_{\text{out}}]} \sum_{r'\in[p]} \sum_{r\in\mathcal{K}} b_{r,s'} b_{r',s'}^\dagger \mathbf{Back}_{i\to L,r,s} \Phi_{r',s} \left( \left\langle \mathbf{w}_{r',s}^\dagger, [\overline{\mathbf{x}}^{(1)}, \cdots, \overline{\mathbf{x}}^{(L)}] \right\rangle \right) \right|$$

$$\leq \mathcal{O}(\rho^8 C_\Phi d_{\text{out}} L p N^{7/6} m^{-7/6}).$$

*Proof.* From Lemma B.2, we have with probability at least $1 - e^{-\Omega(\rho^2)}$,

$$\left| \mathbf{e}_s^\top \left( \mathbf{Back}_{i\to j} - \widetilde{\mathbf{Back}}_{i\to j} \right) \mathbf{e}_r \right| = \left| \mathbf{b}_s^\top \left( \mathbf{D}^{(j)} \mathbf{W} \cdots \mathbf{D}^{(i)} \mathbf{W} - \widetilde{\mathbf{D}}^{(j)} \widetilde{\mathbf{W}} \cdots \widetilde{\mathbf{D}}^{(i)} \widetilde{\mathbf{W}} \right) \mathbf{e}_r \right|$$

$$\leq \left\| \mathbf{b}_s \right\| \left\| \left( \mathbf{D}^{(j)} \mathbf{W} \cdots \mathbf{D}^{(i)} \mathbf{W} - \widetilde{\mathbf{D}}^{(j)} \widetilde{\mathbf{W}} \cdots \widetilde{\mathbf{D}}^{(i)} \widetilde{\mathbf{W}} \right) \mathbf{e}_r \right\|$$

$$\leq \mathcal{O}(d_{\text{out}}^{-1/2} \rho^7 N^{1/6} m^{-1/6}), \text{ for all } r \in [m], s \in [d_{\text{out}}] \text{ and } 1 \leq i \leq j \leq L.$$

Hence,

$$\left| \frac{d_{\text{out}}}{m} \sum_{i=1}^{L} \sum_{s'\in[d_{\text{out}}]} \sum_{r'\in[p]} \sum_{r\in\mathcal{K}} b_{r,s'} b_{r',s'}^\dagger \widetilde{\mathbf{Back}}_{i\to L,r,s} \Phi_{r',s} \left( \left\langle \mathbf{w}_{r',s}^\dagger, [\overline{\mathbf{x}}^{(1)}, \cdots, \overline{\mathbf{x}}^{(L)}] \right\rangle \right) \right.$$

$$\left. - \frac{d_{\text{out}}}{m} \sum_{i=1}^{L} \sum_{s'\in[d_{\text{out}}]} \sum_{r'\in[p]} \sum_{r\in\mathcal{K}} b_{r,s'} b_{r',s'}^\dagger \mathbf{Back}_{i\to L,r,s} \Phi_{r',s} \left( \left\langle \mathbf{w}_{r',s}^\dagger, [\overline{\mathbf{x}}^{(1)}, \cdots, \overline{\mathbf{x}}^{(L)}] \right\rangle \right) \right|$$

$$= \left| \frac{d_{\text{out}}}{m} \sum_{i=1}^{L} \sum_{s'\in[d_{\text{out}}]} \sum_{r'\in[p]} \sum_{r\in\mathcal{K}} b_{r,s'} b_{r',s'}^\dagger \mathbf{e}_r^\top \left( \widetilde{\mathbf{Back}}_{i\to L} \Phi_{r',s} - \mathbf{Back}_{i\to L} \right) \mathbf{e}_s \Phi_{r',s} \left( \left\langle \mathbf{w}_{r',s}^\dagger, [\overline{\mathbf{x}}^{(1)}, \cdots, \overline{\mathbf{x}}^{(L)}] \right\rangle \right) \right|$$

$$\leq \frac{d_{\text{out}}}{m} \sum_{i=1}^{L} \sum_{s'\in[d_{\text{out}}]} \sum_{r'\in[p]} \sum_{r\in\mathcal{K}} |b_{r,s'}| \cdot |b_{r',s'}^\dagger| \cdot \left| \Phi_{r',s} \left( \left\langle \mathbf{w}_{r',s}^\dagger, [\overline{\mathbf{x}}^{(1)}, \cdots, \overline{\mathbf{x}}^{(L)}] \right\rangle \right) \right| \cdot \left| \mathbf{e}_r^\top \left( \widetilde{\mathbf{Back}}_{i\to L} \Phi_{r',s} - \mathbf{Back}_{i\to L} \right) \mathbf{e}_s \right|$$

$$\leq \frac{d_{\text{out}}^2 L p N}{m} \cdot \frac{\rho}{\sqrt{d_{\text{out}}}} \cdot 1 \cdot C_\Phi \cdot \mathcal{O}(d_{\text{out}}^{-1/2} \rho^7 N^{1/6} m^{-1/6})$$

$$\leq \mathcal{O}(\rho^8 C_\Phi d_{\text{out}} L p N^{7/6} m^{-7/6}).$$

In the final step, we have used the bounds of different terms as follows. we will need a couple of bounds on the terms that appear in the equations.

- Using the fact A.4, we can show that with probability $1 - e^{-\Omega(\rho^2)}$, $\max_{r,s'} |b_{r,s'}| \leq \frac{\rho}{\sqrt{d_{\text{out}}}}$.

- From the definition of concept class, $\max_{r',s'} |b_{r',s'}^\dagger| \leq 1$ and $\max_{r',s} |\Phi_{r',s}| \leq C_\Phi$ in the desired range.

$\square$

**Claim F.16** (Restating claim F.3). *With probability exceeding* $1 - e^{-\Omega(\rho^2)}$,

$$\left| b_{r',s}^\dagger \Phi_{r',s}\left(\left\langle \mathbf{w}_{r',s}^\dagger, [\overline{\mathbf{x}}^{(2)}, \cdots, \overline{\mathbf{x}}^{(L-1)}]\right\rangle\right)\right.$$

$$\left. - \frac{d_{\text{out}}}{m}\sum_{i=1}^{L}\sum_{s'\in[d_{\text{out}}]}\sum_{r\in[m]} b_{r,s'} b_{r',s'}^\dagger \mathbf{Back}_{i\to L,r,s}\Phi_{r',s}\left(\left\langle \mathbf{w}_{r',s}^\dagger, [\overline{\mathbf{x}}^{(2)}, \cdots, \overline{\mathbf{x}}^{(L-1)}]\right\rangle\right)\right|$$

$$\leq \mathcal{O}(L d_{\text{out}} \rho C_\Phi m^{-0.25}).$$

*Proof.*

$$\left|\Phi_{r',s}\left(\left\langle \mathbf{w}_{r',s}^\dagger, [\overline{\mathbf{x}}^{(2)}, \cdots, \overline{\mathbf{x}}^{(L-1)}]\right\rangle\right)\right.$$

$$\left. - \frac{d_{\text{out}}}{m}\sum_{i=1}^{L}\sum_{s'\in[d_{\text{out}}]}\sum_{r\in[m]} b_{r,s'} b_{r',s'}^\dagger \mathbf{Back}_{i\to L,r,s}\Phi_{r',s}\left(\left\langle \mathbf{w}_{r',s}^\dagger, [\overline{\mathbf{x}}^{(2)}, \cdots, \overline{\mathbf{x}}^{(L-1)}]\right\rangle\right)\right|$$

$$\leq \left|\Phi_{r',s}\left(\left\langle \mathbf{w}_{r',s}^\dagger, [\overline{\mathbf{x}}^{(2)}, \cdots, \overline{\mathbf{x}}^{(L-1)}]\right\rangle\right) - \frac{d_{\text{out}}}{m}\sum_{r\in[m]} b_{r,s} b_{r',s}^\dagger \mathbf{Back}_{L\to L,r,s}\Phi_{r',s}\left(\left\langle \mathbf{w}_{r',s}^\dagger, [\overline{\mathbf{x}}^{(2)}, \cdots, \overline{\mathbf{x}}^{(L-1)}]\right\rangle\right)\right|$$

$$+ \left|\frac{d_{\text{out}}}{m}\sum_{s'\in[d_{\text{out}}]:s'\neq s}\sum_{r\in[m]} b_{r,s'} b_{r',s'}^\dagger \mathbf{Back}_{L\to L,r,s}\Phi_{r',s}\left(\left\langle \mathbf{w}_{r',s}^\dagger, [\overline{\mathbf{x}}^{(2)}, \cdots, \overline{\mathbf{x}}^{(L-1)}]\right\rangle\right)\right|$$

$$+ \left|\frac{d_{\text{out}}}{m}\sum_{i=1}^{L-1}\sum_{s'\in[d_{\text{out}}]:s'\neq s}\sum_{r\in[m]} b_{r,s'} b_{r',s'}^\dagger \mathbf{Back}_{i\to L,r,s}\Phi_{r',s}\left(\left\langle \mathbf{w}_{r',s}^\dagger, [\overline{\mathbf{x}}^{(2)}, \cdots, \overline{\mathbf{x}}^{(L-1)}]\right\rangle\right)\right|$$

$$+ \left|\frac{d_{\text{out}}}{m}\sum_{i=1}^{L-1}\sum_{r\in[m]} b_{r,s} b_{r',s}^\dagger \mathbf{Back}_{i\to L,r,s}\Phi_{r',s}\left(\left\langle \mathbf{w}_{r',s}^\dagger, [\overline{\mathbf{x}}^{(2)}, \cdots, \overline{\mathbf{x}}^{(L-1)}]\right\rangle\right)\right|.$$

Since, $\mathbf{B} = \mathbf{Back}_{L\to L}$ by definition, we can simplify the above 4 terms as

$$\left|\Phi_{r',s}\left(\left\langle \mathbf{w}_{r',s}^\dagger, [\overline{\mathbf{x}}^{(2)}, \cdots, \overline{\mathbf{x}}^{(L-1)}]\right\rangle\right)\right. \tag{54}$$

$$\left. - \frac{d_{\text{out}}}{m}\sum_{i=1}^{L}\sum_{s'\in[d_{\text{out}}]}\sum_{r\in[m]} b_{r,s'} b_{r',s'}^\dagger \mathbf{Back}_{i\to L,r,s}\Phi_{r',s}\left(\left\langle \mathbf{w}_{r',s}^\dagger, [\overline{\mathbf{x}}^{(2)}, \cdots, \overline{\mathbf{x}}^{(L-1)}]\right\rangle\right)\right|$$

$$\leq \left|\Phi_{r',s}\left(\left\langle \mathbf{w}_{r',s}^\dagger, [\overline{\mathbf{x}}^{(2)}, \cdots, \overline{\mathbf{x}}^{(L-1)}]\right\rangle\right) - \frac{d_{\text{out}}}{m}\sum_{r\in[m]} b_{r,s}^2 b_{r',s}^\dagger \Phi_{r',s}\left(\left\langle \mathbf{w}_{r',s}^\dagger, [\overline{\mathbf{x}}^{(2)}, \cdots, \overline{\mathbf{x}}^{(L-1)}]\right\rangle\right)\right|$$

$$+ \left|\frac{d_{\text{out}}}{m}\sum_{s'\in[d_{\text{out}}]:s'\neq s}\sum_{r\in[m]} b_{r,s'} b_{r',s'}^\dagger b_{r,s}\Phi_{r',s}\left(\left\langle \mathbf{w}_{r',s}^\dagger, [\overline{\mathbf{x}}^{(2)}, \cdots, \overline{\mathbf{x}}^{(L-1)}]\right\rangle\right)\right|$$

$$+ \left|\frac{d_{\text{out}}}{m}\sum_{i=1}^{L}\sum_{s'\in[d_{\text{out}}]:s'\neq s}\sum_{r\in[m]} \mathbf{Back}_{L\to L,r,s'} b_{r',s'}^\dagger \mathbf{Back}_{i\to L,r,s}\Phi_{r',s}\left(\left\langle \mathbf{w}_{r',s}^\dagger, [\overline{\mathbf{x}}^{(2)}, \cdots, \overline{\mathbf{x}}^{(L-1)}]\right\rangle\right)\right|$$

$$+ \left|\frac{d_{\text{out}}}{m}\sum_{i=1}^{L-1}\sum_{r\in[m]} \mathbf{Back}_{L\to L,r,s} b_{r',s}^\dagger \mathbf{Back}_{i\to L,r,s}\Phi_{r',s}\left(\left\langle \mathbf{w}_{r',s}^\dagger, [\overline{\mathbf{x}}^{(2)}, \cdots, \overline{\mathbf{x}}^{(L-1)}]\right\rangle\right)\right|$$

$$= \left|\Phi_{r',s}\left(\left\langle \mathbf{w}_{r',s}^\dagger, [\overline{\mathbf{x}}^{(2)}, \cdots, \overline{\mathbf{x}}^{(L-1)}]\right\rangle\right) - \frac{d_{\text{out}}}{m}\sum_{r\in[m]} b_{r,s}^2 b_{r',s}^\dagger \Phi_{r',s}\left(\left\langle \mathbf{w}_{r',s}^\dagger, [\overline{\mathbf{x}}^{(2)}, \cdots, \overline{\mathbf{x}}^{(L-1)}]\right\rangle\right)\right|$$

$$\tag{55}$$

$$+ \left|\frac{d_{\text{out}}}{m}\sum_{s'\in[d_{\text{out}}]:s'\neq s}\sum_{r\in[m]} b_{r,s'} b_{r',s'}^\dagger b_{r,s}\Phi_{r',s}\left(\left\langle \mathbf{w}_{r',s}^\dagger, [\overline{\mathbf{x}}^{(2)}, \cdots, \overline{\mathbf{x}}^{(L-1)}]\right\rangle\right)\right| \tag{56}$$

$$+ \left| \frac{d_{\text{out}}}{m} \sum_{i=1}^{L-1} \sum_{s' \in [d_{\text{out}}]: s' \neq s} b_{r',s'}^{\dagger} \Phi_{r',s} \left( \left\langle \mathbf{w}_{r',s}^{\dagger}, [\overline{\mathbf{x}}^{(2)}, \cdots, \overline{\mathbf{x}}^{(L-1)}] \right\rangle \right) \left\langle \mathbf{e}_s^{\top} \mathbf{Back}_{L \to L}, \mathbf{e}_{s'}^{\top} \mathbf{Back}_{i \to L} \right\rangle \right|$$

$$\tag{57}$$

$$+ \left| \frac{d_{\text{out}}}{m} \sum_{i=1}^{L-1} b_{r',s}^{\dagger} \Phi_{r',s} \left( \left\langle \mathbf{w}_{r',s}^{\dagger}, [\overline{\mathbf{x}}^{(2)}, \cdots, \overline{\mathbf{x}}^{(L-1)}] \right\rangle \right) \left\langle \mathbf{e}_s^{\top} \mathbf{Back}_{L \to L}, \mathbf{e}_s^{\top} \mathbf{Back}_{i \to L} \right\rangle \right|. \tag{58}$$

First, we can use Lemma F.11 to show that both eq. 57 and eq. 58 are small.

$$\left| \frac{d_{\text{out}}}{m} \sum_{i=1}^{L-1} \sum_{s' \in [d_{\text{out}}]: s' \neq s} b_{r',s'}^{\dagger} \Phi_{r',s} \left( \left\langle \mathbf{w}_{r',s}^{\dagger}, [\overline{\mathbf{x}}^{(2)}, \cdots, \overline{\mathbf{x}}^{(L-1)}] \right\rangle \right) \left\langle \mathbf{e}_s^{\top} \mathbf{Back}_{L \to L}, \mathbf{e}_{s'}^{\top} \mathbf{Back}_{i \to L} \right\rangle \right|$$

$$\leq \sum_{i=1}^{L-1} \sum_{s' \in [d_{\text{out}}]: s' \neq s} \frac{d_{\text{out}}}{m} \cdot \left| b_{r',s'}^{\dagger} \right| \cdot \left| \Phi_{r',s} \left( \left\langle \mathbf{w}_{r',s}^{\dagger}, [\overline{\mathbf{x}}^{(2)}, \cdots, \overline{\mathbf{x}}^{(L-1)}] \right\rangle \right) \right| \cdot \left| \left\langle \mathbf{e}_s^{\top} \mathbf{Back}_{L \to L}, \mathbf{e}_{s'}^{\top} \mathbf{Back}_{i \to L} \right\rangle \right|$$

$$\leq \mathcal{O}(L d_{\text{out}} \rho C_\Phi m^{-0.25}).$$

Also,

$$\left| \frac{d_{\text{out}}}{m} \sum_{i=1}^{L-1} b_{r',s}^{\dagger} \Phi_{r',s} \left( \left\langle \mathbf{w}_{r',s}^{\dagger}, [\overline{\mathbf{x}}^{(2)}, \cdots, \overline{\mathbf{x}}^{(L-1)}] \right\rangle \right) \left\langle \mathbf{e}_s^{\top} \mathbf{Back}_{L \to L}, \mathbf{e}_s^{\top} \mathbf{Back}_{i \to L} \right\rangle \right|$$

$$\leq \sum_{i=1}^{L-1} \frac{d_{\text{out}}}{m} \cdot \left| b_{r',s}^{\dagger} \right| \cdot \left| \Phi_{r',s} \left( \left\langle \mathbf{w}_{r',s}^{\dagger}, [\overline{\mathbf{x}}^{(2)}, \cdots, \overline{\mathbf{x}}^{(L-1)}] \right\rangle \right) \right| \cdot \left| \left\langle \mathbf{e}_s^{\top} \mathbf{Back}_{L \to L}, \mathbf{e}_s^{\top} \mathbf{Back}_{i \to L} \right\rangle \right|$$

$$\leq \mathcal{O}(L \rho C_\Phi m^{-0.25}).$$

Since, $\mathbf{b}_s \sim \mathcal{O}(0, \frac{1}{d_{\text{out}}} \mathbb{I})$, we can show using using fact A.3 that with probability at least $1 - e^{-\Omega(\rho^2)}$,

$$\left| \frac{d_{\text{out}}}{m} \sum_{r \in [m]} b_{r,s}^2 - 1 \right| \leq \mathcal{O}(\frac{\rho}{\sqrt{m}}).$$

Hence, eq. 55 can be simplified as

$$\left| \Phi_{r',s} \left( \left\langle \mathbf{w}_{r',s}^{\dagger}, [\overline{\mathbf{x}}^{(2)}, \cdots, \overline{\mathbf{x}}^{(L-1)}] \right\rangle \right) - \frac{d_{\text{out}}}{m} \sum_{r \in [m]} b_{r,s}^2 b_{r',s}^{\dagger} \Phi_{r',s} \left( \left\langle \mathbf{w}_{r',s}^{\dagger}, [\overline{\mathbf{x}}^{(2)}, \cdots, \overline{\mathbf{x}}^{(L-1)}] \right\rangle \right) \right|$$

$$\leq \mathcal{O}(\frac{\rho}{\sqrt{m}}) \cdot \left| b_{r',s}^{\dagger} \right| \left| \Phi_{r',s} \left( \left\langle \mathbf{w}_{r',s}^{\dagger}, [\overline{\mathbf{x}}^{(2)}, \cdots, \overline{\mathbf{x}}^{(L-1)}] \right\rangle \right) \right| \leq \mathcal{O}(C_\Phi \frac{\rho}{\sqrt{m}}).$$

Also,

$$\frac{d_{\text{out}}}{m} \sum_{r \in [m]} b_{r,s} b_{r,s'} = \frac{1}{2m} \left( \left\| b_{r,s} + b_{r,s'} \right\|^2 - \left\| b_{r,s} - b_{r,s'} \right\|^2 \right).$$

Since, both $\mathbf{b}_s$ and $\mathbf{b}_{s'}$ are independent gaussian vectors, $\mathbf{b}_s + \mathbf{b}_{s'} \sim \mathcal{N}(0, \frac{2}{d_{\text{out}}} \mathbb{I})$ and $\mathbf{b}_s - \mathbf{b}_{s'} \sim \mathcal{N}(0, \frac{2}{d_{\text{out}}} \mathbb{I})$. Hence, using fact A.3 we have with probability at least $1 - e^{-\Omega(\rho^2)}$, for all $s' \in [d_{\text{out}}]$,

$$\left| \frac{d_{\text{out}}}{m} \sum_{r \in [m]} (b_{r,s'} + b_{r,s})^2 - 2 \right| \leq \mathcal{O}(\frac{\rho}{\sqrt{m}})$$

$$\left| \frac{d_{\text{out}}}{m} \sum_{r \in [m]} (b_{r,s'} - b_{r,s})^2 - 2 \right| \leq \mathcal{O}(\frac{\rho}{\sqrt{m}}),$$

and thus
$$\left| \frac{d_{\text{out}}}{m} \sum_{r \in [m]} b_{r,s} b_{r,s'} \right| \le \mathcal{O}\left(\frac{\rho}{\sqrt{m}}\right).$$

This can be used to simplify eq. 58.

$$\left| \frac{d_{\text{out}}}{m} \sum_{s' \in [d_{\text{out}}]: s' \ne s} \sum_{r \in [m]} b_{r,s'} b_{r',s'}^{\dagger} b_{r,s} \Phi_{r',s} \left( \left\langle \mathbf{w}_{r',s}^{\dagger}, [\overline{\mathbf{x}}^{(2)}, \cdots, \overline{\mathbf{x}}^{(L-1)}] \right\rangle \right) \right|$$

$$\le \sum_{s' \in [d_{\text{out}}]: s' \ne s} \left| \frac{d_{\text{out}}}{m} \sum_{r \in [m]} b_{r,s'} b_{r',s'}^{\dagger} b_{r,s} \Phi_{r',s} \left( \left\langle \mathbf{w}_{r',s}^{\dagger}, [\overline{\mathbf{x}}^{(2)}, \cdots, \overline{\mathbf{x}}^{(L-1)}] \right\rangle \right) \right|$$

$$\le \mathcal{O}\left(\frac{\rho}{\sqrt{m}}\right) \sum_{s' \in [d_{\text{out}}]: s' \ne s} \cdot \left| b_{r',s'}^{\dagger} \right| \left| \Phi_{r',s} \left( \left\langle \mathbf{w}_{r',s}^{\dagger}, [\overline{\mathbf{x}}^{(2)}, \cdots, \overline{\mathbf{x}}^{(L-1)}] \right\rangle \right) \right|$$

$$\le \mathcal{O}\left(C_{\Phi} d_{\text{out}} \frac{\rho}{\sqrt{m}}\right).$$

Hence, adding everything up, we have with probability exceeding $1 - e^{-\Omega(\rho^2)}$,

$$\left| \Phi_{r',s} \left( \left\langle \mathbf{w}_{r',s}^{\dagger}, [\overline{\mathbf{x}}^{(2)}, \cdots, \overline{\mathbf{x}}^{(L-1)}] \right\rangle \right) \right.$$

$$\left. - \frac{d_{\text{out}}}{m} \sum_{i=1}^{L} \sum_{s' \in [d_{\text{out}}]} \sum_{r \in [m]} b_{r,s'} b_{r',s'}^{\dagger} \mathbf{Back}_{i \to L, r, s} \Phi_{r',s} \left( \left\langle \mathbf{w}_{r',s}^{\dagger}, [\overline{\mathbf{x}}^{(2)}, \cdots, \overline{\mathbf{x}}^{(L-1)}] \right\rangle \right) \right|$$

$$\le \mathcal{O}\left(C_{\Phi} \frac{\rho}{\sqrt{m}}\right) + \mathcal{O}\left(C_{\Phi} d_{\text{out}} \frac{\rho}{\sqrt{m}}\right) + \mathcal{O}(L d_{\text{out}} \rho C_{\Phi} m^{-0.25}) + \mathcal{O}(L \rho C_{\Phi} m^{-0.25})$$

$$\le \mathcal{O}(L d_{\text{out}} \rho C_{\Phi} m^{-0.25}).$$

$\square$

Thus, introducing claim F.3 in eq. 40, we have

$$\left| F_s^{(L)}(\mathbf{h}^{(\ell-1)}, \mathbf{x}^{(\ell)}) - \sum_{r' \in [p]} b_{r',s}^{\dagger} \Phi_{r',s} \left( \left\langle \mathbf{w}_{r',s}^{\dagger}, [\overline{\mathbf{x}}^{(2)}, \cdots, \overline{\mathbf{x}}^{(L-1)}] \right\rangle \right) \right|$$

$$\le \mathcal{O}(d_{\text{out}} L p \rho^8 \mathfrak{C}_{\varepsilon}(\Phi, \mathcal{O}(\varepsilon_x^{-1})) m^{-1/30}) + \mathcal{O}(\mathfrak{C}_{\varepsilon}(\Phi_{r's}, \mathcal{O}(\varepsilon_x^{-1})) \rho^2 \sqrt{d_{\text{out}}^3 L p} m^{-0.1})$$

$$+ d_{\text{out}} L p \rho^2 (\varepsilon + \mathcal{O}(L_{\Phi} \rho^5 m^{-2/15}) + \mathcal{O}(\varepsilon_x^{-1} L_{\Phi} L^4 \rho^{11} m^{-1/12} + L_{\Phi} L^{11/6} \rho^2 \varepsilon_x^{2/3}))$$

$$+ \mathcal{O}(\rho^8 d_{\text{out}} L p m^{-2/15}) + \mathcal{O}(L p d_{\text{out}} \rho C_{\Phi} m^{-0.25})$$

$$\le \mathcal{O}(d_{\text{out}} L p \rho^2 \varepsilon + d_{\text{out}} L^{17/6} p \rho^4 L_{\Phi} \varepsilon_x^{2/3} + d_{\text{out}}^{3/2} L^5 p \rho^{11} L_{\Phi} C_{\Phi} \mathfrak{C}_{\varepsilon}(\Phi, \mathcal{O}(\varepsilon_x^{-1})) m^{-1/30}).$$

**Lemma F.17.** *With probability at least $1 - e^{-\Omega(\rho^2)}$,*
$$\|\mathbf{W}^*\|_F = 0,$$
$$\|\mathbf{A}^*\|_F \le \mathcal{O}\left( \rho d_{\text{out}}^{1/2} \frac{\mathfrak{C}_{\varepsilon}(\Phi, \mathcal{O}(\varepsilon_x^{-1}))}{\sqrt{m}} \right).$$

*Proof.* The norm of $\mathbf{W}^*$ follows from the fact that it is a zero matrix. From def. D.2, we have that

$$\mathbf{a}_r^* = \frac{d_{\text{out}}}{m} \sum_{s \in [d_{\text{out}}]} \sum_{r' \in [p]} b_{r,s} b_{r',s}^{\dagger} H_{r',s} \left( \theta_{r',s} \left( \langle \mathbf{w}_r, \overline{\mathbf{W}}^{[L]} \mathbf{w}_{r',s}^{\dagger} \rangle \right), \sqrt{m/2} a_{r,d} \right) \mathbf{e}_d, \quad \forall r \in [m],$$

where

$$\theta_{r',s} = \frac{\sqrt{m/2}}{\|\overline{\mathbf{W}}^{[L]} \mathbf{w}_{r',s}^{\dagger}\|}.$$

Since, there is a dependence between $\mathbf{w}_r$ and $\overline{\mathbf{W}}^{[L]}$, we again need to re-randomize some rows of $\overline{\mathbf{W}}^{[L]}$ as has been done in thm. D.2. Following the steps as has been done to bound eq. 44, we can get

$$\mathbf{A}^* = \widetilde{\mathbf{A}}^* + \overline{\mathbf{A}}^*,$$

where $\left\|\overline{\mathbf{A}}^*\right\|_F \leq \mathcal{O}(\mathfrak{C}_\varepsilon(\Phi, \mathcal{O}(\varepsilon_x^{-1}))\rho^6 m^{-5/6})$ with probability at least $1 - e^{-\Omega(\rho^2)}$ and for each $r \in [m]$,

$$\widetilde{\mathbf{a}}_r^* = \frac{d_{\text{out}}}{m} \sum_{s \in [d_{\text{out}}]} \sum_{r' \in [p]} b_{r,s} b_{r',s}^\dagger H_{r',s}\left(\theta_{r',s}\left(\langle \mathbf{w}_r, \widetilde{\overline{\mathbf{W}}}^{[L]} \mathbf{w}_{r',s}^\dagger\rangle\right), \sqrt{m/2} a_{r,d}\right) \mathbf{e}_d,$$

where $\widetilde{\overline{\mathbf{W}}}^{[L]}$ doesn't depend on the weight vector $\mathbf{w}_r$. Using the properties of the function $H_{r',s}$ from def. F.10, we can show that with probability at least $1 - e^{-\Omega(\rho^2)}$,

$$\left\|\widetilde{\mathbf{A}}^*\right\|_F \leq \mathcal{O}(d_{\text{out}}^{1/2}\mathfrak{C}_\varepsilon(\Phi, \mathcal{O}(\varepsilon_x^{-1}))\rho m^{-1/2}).$$

$\square$

# G  Optimization and Generalization: proofs

## G.1  Proof of lemma D.3

**Lemma G.1** (Restating lemma D.3). *For a constant $\varepsilon_x = \frac{1}{\text{poly}(\rho)}$ and for every constant $\varepsilon \in \left(0, \frac{1}{p \cdot \text{poly}(\rho) \cdot \mathfrak{C}_s(\Phi, \mathcal{O}(\varepsilon_x^{-1}))}\right)$, there exists $C' = \mathfrak{C}_\varepsilon(\Phi, \mathcal{O}(\varepsilon_x^{-1}))$ and a parameter $\lambda = \Theta\left(\frac{\varepsilon}{L\rho}\right)$ so that, as long as $m \geq \text{poly}\left(C', p, L, d_{\text{out}}, \varepsilon^{-1}\right)$ and $N \geq \Omega\left(\frac{\rho^3 p C_\Phi^2}{\varepsilon^2}\right)$, setting learning rate $\eta = \Theta\left(\frac{1}{\varepsilon\rho^2 m}\right)$ and $T = \Theta\left(\frac{p^2 C'^2 \text{poly}(\rho)}{\varepsilon^2}\right)$, we have*

$$\mathbb{E}_{\text{sgd}}\left[\frac{1}{T}\sum_{t=0}^{T-1} \mathbb{E}_{(\overline{\mathbf{x}}, \mathbf{y}^*) \sim \mathcal{Z}} \text{Obj}(\overline{\mathbf{x}}, \mathbf{y}^*; \mathbf{W} + \mathbf{W}_t, \mathbf{A} + \mathbf{A}_t)\right] \leq \text{OPT} + \frac{\varepsilon}{2} + \frac{1}{\text{poly}(\rho)},$$

*and $\|W_t\|_F \leq \frac{\Delta}{\sqrt{m}}$ for $\Delta = \frac{C'^2 p^2 \text{poly}(\rho)}{\varepsilon^2}$.*

*Proof.* The proof will follow exactly the same routine as lemma 7.1 in [37]. We allow $\mathbf{A}$ to change, which leads to changes in the proof. We outline the major differences here for completeness. For simplicity, we outline the proof for Gradient Descent.

The training objective is given by

$$\text{Obj}(\mathbf{W}_t, \mathbf{A}_t) = \mathbb{E}_{(\overline{\mathbf{x}}, y^*) \sim \mathcal{Z}} \text{Obj}(\overline{\mathbf{x}}, y^*; \mathbf{W}_t, \mathbf{A}_t), \text{ where}$$

$$\text{Obj}(\overline{\mathbf{x}}, y^*; \mathbf{W}_t, \mathbf{A}_t) = G(\lambda F_{\text{rnn}}^{(\ell)}(\mathbf{x}, y^*; \mathbf{W} + \mathbf{W}_t, \mathbf{A} + \mathbf{A}_t)).$$

Let $\overline{\mathbf{x}}$ be a true sequence and $\mathbf{x}$ be its normalized version. Let's consider the matrices $\mathbf{W} + \mathbf{W}_t$, $\mathbf{A} + \mathbf{A}_t$ after SGD iteration $t$. Let

- at RNN cell $i$, $\mathbf{h}^{(i)}$, $\text{Back}_{i \to L}$ and $\mathbf{D}^{(i)}$ are defined w.r.t. $\mathbf{A}, \mathbf{W}, \mathbf{B}, \mathbf{x}$.

- at RNN cell $i$, $\mathbf{h}^{(i)} + \mathbf{h}_t^{(i)}$, $\text{Back}_{i \to L} + \text{Back}_{i \to L, t}$ and $\mathbf{D}^{(i)} + \mathbf{D}_t^{(i)}$ are defined w.r.t. $\mathbf{A} + \mathbf{A}_t, \mathbf{W} + \mathbf{W}_t, \mathbf{B}, \mathbf{x}$.

Define the following regularization term:

$$R(\mathbf{x}; \mathbf{W}', \mathbf{A}') = \sum_{i=2}^{L}(\text{Back}_{i \to L} + \text{Back}_{i \to L, t})(\mathbf{D}^{(i)} + \mathbf{D}_t^{(i)})(\mathbf{W}'(\mathbf{h}^{(i-1)} + \mathbf{h}_t^{(i-1)}) + \mathbf{A}'\mathbf{x}^{(i)}),$$

$$= \sum_{i=2}^{L}(\text{Back}_{i \to L} + \text{Back}_{i \to L, t})(\mathbf{D}^{(i)} + \mathbf{D}_t^{(i)})\left([\mathbf{W}', \mathbf{A}']_r \left[\mathbf{h}^{(i-1)} + \mathbf{h}_t^{(i-1)}, \mathbf{x}^{(i)}\right]\right)$$

which is a linear function over $[\mathbf{W}', \mathbf{A}']_r$. Define the following regularized loss function:

$$\widetilde{G}(\mathbf{W}', \mathbf{A}') = \mathop{\mathbb{E}}_{(\overline{\mathbf{x}}, y^*) \sim \mathcal{Z}} \widetilde{G}(\overline{\mathbf{x}}, y^*; \mathbf{W}', \mathbf{A}'), \text{ where}$$

$$\widetilde{G}(\overline{\mathbf{x}}, y^*; \mathbf{W}', \mathbf{A}') = G(\lambda F_{\mathrm{rnn}}^{(L)}(\overline{\mathbf{x}}; \mathbf{W} + \mathbf{W}_t, \mathbf{A} + \mathbf{A}_t) + \lambda R(\mathbf{x}; \mathbf{W}', \mathbf{A}'))$$

Note that, $\widetilde{G}(\mathbf{0}, \mathbf{0}) = \mathrm{Obj}(\mathbf{W}_t, \mathbf{A}_t)$ and $\nabla_{[\mathbf{W}', \mathbf{A}']}\widetilde{G}(\mathbf{0}, \mathbf{0}) = \nabla_{[\mathbf{W}_t, \mathbf{A}_t]}\mathrm{Obj}(\mathbf{W}_t, \mathbf{A}_t)$. First of all, we have

$$
\begin{aligned}
&F_{\mathrm{rnn}}^{(\ell)}(\mathbf{x}; \mathbf{W} + \mathbf{W}_t, \mathbf{A} + \mathbf{A}_t) - F_{\mathrm{rnn}}^{(\ell)}(\mathbf{x}; \mathbf{W}, \mathbf{A}) && \text{(difference between RNN output)} \\
&= \sum_{\ell \in [L]} \mathbf{Back}_{\ell \to L} \mathbf{D}^{(\ell)}\left(\mathbf{W}_t \mathbf{h}^{(\ell-1)} + \mathbf{A}_t \mathbf{x}^{(\ell)}\right) + \varepsilon' && \text{(using lemma G.3)} \\
&= \sum_{\ell \in [L]} \left(\mathbf{Back}_{\ell \to L} + \mathbf{Back}_{\ell \to L, t}\right)\left(\mathbf{D}^{(\ell)} + \mathbf{D}_t^{(\ell)}\right)\left(\mathbf{W}_t\left(\mathbf{h}^{(\ell-1)} + h_t^{(\ell-1)}\right) + \mathbf{A}_t \mathbf{x}^{(\ell)}\right) + \varepsilon' + \varepsilon'' \\
& && \text{(using lemma G.2)} \\
&= R(\mathbf{x}; \mathbf{W}_t, \mathbf{A}_t) + \varepsilon' + \varepsilon'', && \text{(using the definition of } R) 
\end{aligned}
$$

where $0 \le \varepsilon', \varepsilon'' \le \mathcal{O}(\rho^7 \Delta^{4/3} m^{-1/6})$. Also, from lemma G.4, we have

$$R(\mathbf{x}; \mathbf{W}^*, \mathbf{A}^*) = F^*(\overline{\mathbf{x}}) \pm \varepsilon''',$$

with $\varepsilon''' \le \varepsilon/2 + poly(\rho)^{-1}$, when $\varepsilon_x \le poly(\rho)^{-1}$, $\varepsilon < (p \cdot poly(\rho) \cdot \mathfrak{C}_s(\Phi, \mathcal{O}(\varepsilon_x^{-1})))^{-1}$ and $m \ge poly(\varrho)$. Hence,

$$
\begin{aligned}
&\widetilde{G}\left(\frac{1}{\lambda}\mathbf{W}^* - \mathbf{W}_t, \frac{1}{\lambda}\mathbf{A}^* - \mathbf{A}_t\right) \\
&= G\left(\lambda F_{\mathrm{rnn}}^{(L)}(\mathbf{x}; \mathbf{W} + \mathbf{W}_t, \mathbf{A} + \mathbf{A}_t) + \lambda R\left(\mathbf{x}; \frac{1}{\lambda}\mathbf{W}^* - \mathbf{W}_t, \frac{1}{\lambda}\mathbf{A}^* - \mathbf{A}_t\right)\right) \\
&= G\left(\lambda F_{\mathrm{rnn}}^{(L)}(\mathbf{x}; \mathbf{W}, \mathbf{A}) + \lambda R(\mathbf{x}; \mathbf{W}_t, \mathbf{A}_t) + \lambda R\left(\mathbf{x}; \frac{1}{\lambda}\mathbf{W}^* - \mathbf{W}_t, \frac{1}{\lambda}\mathbf{A}^* - \mathbf{A}_t\right)\right) \pm \varepsilon' \pm \varepsilon'' \\
& \qquad\qquad\qquad\qquad\qquad\qquad\qquad \text{(using the difference of } F_{\mathrm{rnn}}^{(L)} \text{ derived above)} \\
&= G\left(\lambda F_{\mathrm{rnn}}^{(L)}(\overline{\mathbf{x}}; \mathbf{W}, \mathbf{A}) + R(\mathbf{x}; \mathbf{W}^*, \mathbf{A}^*)\right) \pm \varepsilon' \pm \varepsilon'' \\
&= G\left(R(\mathbf{x}; \mathbf{W}^*, \mathbf{A}^*)\right) \pm \varepsilon' \pm \varepsilon'' \pm \varepsilon && (59) \\
&= G(F^*(\mathbf{x})) \pm \varepsilon' \pm \varepsilon'' \pm \varepsilon \pm \varepsilon''' && \text{(using eq. using the difference of } F_{\mathrm{rnn}}^{(L)} \text{ derived above)} \\
&= OPT + \mathcal{O}(\varepsilon) + \frac{1}{poly(\rho)}, && (60)
\end{aligned}
$$

after setting everything properly. Here $\lambda$ is chosen above such that with high probability $\left|\lambda F_{\mathrm{rnn}}^{(L)}(\overline{\mathbf{x}}; \mathbf{W}, \mathbf{A})\right| \le \varepsilon$ to get eq. 59.

Now, at each step of gradient descent, we have

$$
\begin{aligned}
[\mathbf{W}_{t+1}, \mathbf{A}_{t+1}] &= [\mathbf{W}_t, \mathbf{A}_t] - \eta \nabla_{[\mathbf{W}_t, \mathbf{A}_t]}\mathrm{Obj}(\mathbf{W}_t, \mathbf{A}_t) \\
&= [\mathbf{W}_t, \mathbf{A}_t] - \eta \nabla_{[\mathbf{W}', \mathbf{A}']}\widetilde{G}(\mathbf{0}, \mathbf{0}). \\
& \qquad \text{(using the equivalence between gradient derived above)}
\end{aligned}
$$

Hence, we have

$$\left\| [\mathbf{W}_{t+1}, \mathbf{A}_{t+1}] - \frac{1}{\lambda}[\mathbf{W}^*, \mathbf{A}^*] \right\|^2 = \left\| [\mathbf{W}_t, \mathbf{A}_t] - \frac{1}{\lambda}[\mathbf{W}^*, \mathbf{A}^*] \right\|^2 + \left\| [\mathbf{W}_{t+1}, \mathbf{A}_{t+1}] - [\mathbf{W}_t, \mathbf{A}_t] \right\|^2$$

$$+ 2 \left\langle [\mathbf{W}_{t+1}, \mathbf{A}_{t+1}] - [\mathbf{W}_t, \mathbf{A}_t], [\mathbf{W}_t, \mathbf{A}_t] - \frac{1}{\lambda}[\mathbf{W}^*, \mathbf{A}^*] \right\rangle$$

$$= \left\| [\mathbf{W}_t, \mathbf{A}_t] - \frac{1}{\lambda}[\mathbf{W}^*, \mathbf{A}^*] \right\|^2 + \eta^2 \left\| \nabla_{[\mathbf{W}', \mathbf{A}']} \widetilde{G}(\mathbf{0}, \mathbf{0}) \right\|^2$$

(from descent update)

$$- 2\eta \left\langle \nabla_{[\mathbf{W}', \mathbf{A}']} \widetilde{G}(\mathbf{0}, \mathbf{0}), [\mathbf{W}_t, \mathbf{A}_t] - \frac{1}{\lambda}[\mathbf{W}^*, \mathbf{A}^*] \right\rangle$$

$$\geq \left\| [\mathbf{W}_t, \mathbf{A}_t] - \frac{1}{\lambda}[\mathbf{W}^*, \mathbf{A}^*] \right\|^2 + \eta^2 \left\| \nabla_{[\mathbf{W}', \mathbf{A}']} \widetilde{G}(\mathbf{0}, \mathbf{0}) \right\|^2$$

$$- 2\eta \left( \widetilde{G}(\mathbf{W}_t - \frac{1}{\lambda}\mathbf{W}^*, \mathbf{A}_t - \frac{1}{\lambda}\mathbf{A}^*) - \widetilde{G}(0, 0) \right)$$

(using the convexity of $\widetilde{G}$)

$$= \left\| [\mathbf{W}_t, \mathbf{A}_t] - \frac{1}{\lambda}[\mathbf{W}^*, \mathbf{A}^*] \right\|^2 + \eta^2 \left\| \nabla_{[\mathbf{W}_t, \mathbf{A}_t]} \mathrm{Obj}(\mathbf{W}_t, \mathbf{A}_t) \right\|^2$$

$$- 2\eta \left( OPT + \mathcal{O}(\varepsilon) + \frac{1}{poly(\rho)} - \mathrm{Obj}(\mathbf{W}_t, \mathbf{A}_t) \right).$$

(using eq. 60)

Thus,

$$\frac{1}{T} \sum_{t \in [T]} \mathrm{Obj}(\mathbf{W}_t, \mathbf{A}_t) \leq \frac{1}{2\eta T} \left( \left\| [\mathbf{W}_T, \mathbf{A}_T] - \frac{1}{\lambda}[\mathbf{W}^*, \mathbf{A}^*] \right\|^2 - \left\| \frac{1}{\lambda}[\mathbf{W}^*, \mathbf{A}^*] \right\|^2 \right)$$

$$+ \frac{\eta}{2T} \sum_{t \in [T]} \left\| \nabla_{[\mathbf{W}_t, \mathbf{A}_t]} \mathrm{Obj}(\mathbf{W}_t, \mathbf{A}_t) \right\|^2 + OPT + \frac{1}{poly(\rho)} + \mathcal{O}(\varepsilon).$$

We can then finish the proof by bounding $\left\| \nabla_{[\mathbf{W}_t, \mathbf{A}_t]} \mathrm{Obj}(\mathbf{W}_t, \mathbf{A}_t) \right\| \approx \left\| \nabla \mathrm{Obj}(\mathbf{0}, \mathbf{0}) \right\| \leq \mathcal{O}(\lambda \rho^2 \sqrt{m})$ for $\mathbf{W}_t, \mathbf{A}_t \leq \frac{\Delta}{\sqrt{m}}$. We then show that $\Delta = C'^2 poly(\rho) \varepsilon^{-2}$, since it can be bounded by the term $\eta T \sqrt{m} \sup_{t \in [T]} \cdot \left\| \nabla_{[\mathbf{W}_t, \mathbf{A}_t]} \mathrm{Obj}(\mathbf{W}_t, \mathbf{A}_t) \right\|.$ $\square$

### G.2 Helping lemmas

**Lemma G.2.** *[first order coupling] Let* $\mathbf{W}, \mathbf{A}, \mathbf{B}$ *be at random initialization,* $\mathbf{x}^{(1)}, \cdots, \mathbf{x}^{(L)}$ *be a fixed normalized input sequence, and* $\Delta \in [\varrho^{-100}, \varrho^{100}]$ *. With probability at least* $1 - e^{-\Omega(\rho)}$ *over* $\mathbf{W}, \mathbf{A}, \mathbf{B}$ *the following holds. Given any matrices* $W'$ *with* $\|\mathbf{W}'\|_2 \leq \frac{\Delta}{\sqrt{m}}$, $\mathbf{A}'$ *with* $\|\mathbf{A}'\|_2 \leq \frac{\Delta}{\sqrt{m}}$, *and any* $\widetilde{\mathbf{W}}$ *with* $\|\widetilde{\mathbf{W}}\|_2 \leq \frac{\omega}{\sqrt{m}}$, $\widetilde{\mathbf{A}}$ *with* $\|\widetilde{\mathbf{A}}\|_2 \leq \frac{\omega}{\sqrt{m}}$, *letting* $\mathbf{h}^{(\ell)}, \mathbf{D}^{(\ell)}, \mathbf{Back}_{i \to L}$ *be defined with respect to* $\mathbf{W}, \mathbf{A}, \mathbf{B}, \overline{\mathbf{x}}$, *and* $\mathbf{h}^{(\ell)} + \mathbf{h}^{(\ell)'}, \mathbf{D}^{(\ell)} + \mathbf{D}^{(\ell)'}, \mathbf{Back}_{i \to L} + \mathbf{Back}'_{\ell \to j}$ *be defined with respect to* $\mathbf{W} + \mathbf{W}', \mathbf{A} + \mathbf{A}', \mathbf{B}, \overline{\mathbf{x}}$, *then*

$$\| \sum_{\ell \in [L]} \left( \mathbf{Back}_{\ell \to L} + \mathbf{Back}'_{\ell \to L} \right) \left( \mathbf{D}^{(\ell)} + \mathbf{D}^{(\ell)'} \right) \left( \widetilde{\mathbf{W}} \left( \mathbf{h}^{(\ell-1)} + h^{(\ell-1)'} \right) + \widetilde{\mathbf{A}} \mathbf{x}^{(\ell)} \right)$$

$$- \sum_{\ell \in [L]} \mathbf{Back}_{\ell \to L} \mathbf{D}^{(\ell)} \left( \widetilde{\mathbf{W}} \mathbf{h}^{(\ell-1)} + \widetilde{\mathbf{A}} \mathbf{x}^{(\ell)} \right) \| \leq O \left( \frac{\omega \rho^6 \Delta^{1/3}}{m^{1/6}} \right).$$

*Proof.* The proof will follow the same technique as has been used in Lemma 6.2 in [37]. We give a brief overview here.

We allow a change in $\mathbf{A}$ by $\mathbf{A}'$, which wasn't allowed in their lemma. However, we show now that the primary 3 properties (specified in Lemma F.1 in [37]) used to prove the lemma change only by a constant factor, with the introduction of perturbation in $\mathbf{A}$. With probability at least $1 - e^{-\Omega(\rho^2)}$,

1. $\left\|\mathbf{h}^{(\ell)\prime}\right\|_2 \leq \mathcal{O}\left(\rho^6 \Delta / \sqrt{m}\right)$.

2. $\left\|\mathbf{D}^{(\ell)\prime}\right\|_0 \leq \mathcal{O}\left(\rho^4 \Delta^{2/3} m^{2/3}\right)$.

3. $\left\|\mathbf{Back}'_{\ell \to L}\right\|_2 \leq \mathcal{O}\left(\Delta^{1/3} \rho^6 m^{1/3}\right)$.

Property 1 and property 2 will follow from Claim C.2 and property 3 will follow from Claim C.9 in [19] with the following change. Due to the introduction of perturbation in $\mathbf{A}$, eq. C.2 in [19] changes to

$$\mathbf{g}^{(\ell)\prime} = \mathbf{W}'\mathbf{D}^{(\ell)}\mathbf{g}^{(\ell)} + (\mathbf{W} + \mathbf{W}') \cdot \mathbf{D}^{(\ell)\prime} \cdot \mathbf{g}^{(\ell)} + (\mathbf{W} + \mathbf{W}') \cdot (\mathbf{D}^{(\ell)} + \mathbf{D}^{(\ell)\prime}) \cdot \mathbf{g}^{(\ell)\prime} + \mathbf{A}'\mathbf{x}^{(\ell)},$$

where we introduce an extra last term. Thus, since $\left\|\mathbf{A}'\mathbf{x}^{(\ell)}\right\| \leq \left\|\mathbf{A}'\right\|\left\|\mathbf{x}^{(\ell)}\right\| \leq \mathcal{O}(\frac{\Delta}{\sqrt{m}})$, $\mathbf{g}^{(\ell)\prime}$ can be similarly written as $\mathbf{g}_1^{(\ell)\prime} + \mathbf{g}_2^{(\ell)\prime}$, where $\left\|\mathbf{g}_1^{(\ell)\prime}\right\| \leq \tau_1$ and $\left\|\mathbf{g}_2^{(\ell)\prime}\right\|_0 \leq \tau_2$, with $\tau_1$ just changing by a factor 2 in eq. C.1[19]. This minor change percolates to minor changes in the constant factors in $\left\|\mathbf{h}^{(\ell)\prime}\right\|$, $\left\|\mathbf{D}^{(\ell)\prime}\right\|_0$ and $\left\|\mathbf{Back}'_{\ell \to L}\right\|_2$.

Now, the proof follows from the following set of equations.

$$\| \sum_{\ell \in [L]} \left(\mathbf{Back}_{\ell \to L} + \mathbf{Back}'_{\ell \to L}\right) \left(\mathbf{D}^{(\ell)} + \mathbf{D}^{(\ell)\prime}\right) \left(\widetilde{\mathbf{W}}\left(\mathbf{h}^{(\ell-1)} + \mathbf{h}^{(\ell-1)\prime}\right) + \widetilde{\mathbf{A}}\mathbf{x}^{(\ell)}\right)$$

$$- \sum_{\ell \in [L]} \mathbf{Back}_{\ell \to L}\mathbf{D}^{(\ell)}\left(\widetilde{\mathbf{W}}\mathbf{h}^{(\ell-1)} + \widetilde{\mathbf{A}}\mathbf{x}^{(\ell)}\right)\|$$

$$\leq \| \sum_{\ell \in [L]} \mathbf{Back}'_{\ell \to L}\left(\mathbf{D}^{(\ell)} + \mathbf{D}^{(\ell)\prime}\right)\left(\widetilde{\mathbf{W}}\left(\mathbf{h}^{(\ell-1)} + \mathbf{h}^{(\ell-1)\prime}\right) + \widetilde{\mathbf{A}}\mathbf{x}^{(\ell)}\right)\|$$

$$\leq \underbrace{\| \sum_{\ell \in [L]} \mathbf{Back}'_{\ell \to L}\left(\mathbf{D}^{(\ell)} + \mathbf{D}^{(\ell)\prime}\right)\widetilde{\mathbf{W}}\left(\mathbf{h}^{(\ell-1)} + \mathbf{h}^{(\ell-1)\prime}\right)\|}_{\text{Term 1}}$$

$$+ \underbrace{\| \sum_{\ell \in [L]} \mathbf{Back}_{\ell \to L}\left(\mathbf{D}^{(\ell)} + \mathbf{D}^{(\ell)\prime}\right)\widetilde{\mathbf{W}}\mathbf{h}^{(\ell-1)\prime}\|}_{\text{Term 2}} + \underbrace{\| \sum_{\ell \in [L]} \mathbf{Back}_{\ell \to L}\mathbf{D}^{(\ell)\prime}\widetilde{\mathbf{W}}\mathbf{h}^{(\ell-1)}\|}_{\text{Term 3}}$$

$$+ \underbrace{\| \sum_{\ell \in [L]} \mathbf{Back}'_{\ell \to L}\left(\mathbf{D}^{(\ell)} + \mathbf{D}^{(\ell)\prime}\right)\widetilde{\mathbf{A}}\mathbf{x}^{(\ell)}\|}_{\text{Term 4}} + \underbrace{\| \sum_{\ell \in [L]} \mathbf{Back}_{\ell \to L}\mathbf{D}^{(\ell)\prime}\widetilde{\mathbf{A}}\mathbf{x}^{(\ell)}\|}_{\text{Term 5}}$$

Term 1, 2 and 3 appear in the proof of Claim 6.2[37]. Terms 4 and 5 can be bounded using similar technique by using the bound on $\left\|\mathbf{Back}'_{\ell \to L}\right\|$ and $\left\|\mathbf{D}^{(\ell)\prime}\right\|_0$ respectively. $\qquad \square$

**Lemma G.3.** *[first order approximation] Let $\mathbf{W}, \mathbf{A}, \mathbf{B}$ be at random initialization, $\mathbf{x}^{(1)}, \cdots, \mathbf{x}^{(L)}$ be a fixed normalized input sequence, and $\Delta \in \left[\varrho^{-100}, \varrho^{100}\right]$. With probability at least $1 - e^{-\Omega(\rho)}$ over $\mathbf{W}, \mathbf{A}, \mathbf{B}$ the following holds. Given any matrices $W'$ with $\left\|\mathbf{W}'\right\|_2 \leq \frac{\Delta}{\sqrt{m}}$, $\mathbf{A}'$ with $\left\|\mathbf{A}'\right\|_2 \leq \frac{\Delta}{\sqrt{m}}$, letting $\mathbf{h}^{(\ell)}, \mathbf{D}^{(\ell)}, \mathbf{Back}_{i \to L}$ be defined with respect to $\mathbf{W}, \mathbf{A}, \mathbf{B}, \overline{\mathbf{x}}$, and $\mathbf{h}^{(\ell)} + \mathbf{h}^{(\ell)\prime}, \mathbf{D}^{(\ell)} + \mathbf{D}^{(\ell)\prime}$, $\mathbf{Back}_{i \to L} + \mathbf{Back}'_{\ell \to j}$ be defined with respect to $\mathbf{W} + \mathbf{W}', \mathbf{A} + \mathbf{A}', \mathbf{B}, \overline{\mathbf{x}}$, then*

$$\left\|F_{\text{rnn}}^{(L)}(\mathbf{x}; \mathbf{W} + \mathbf{W}', \mathbf{A} + \mathbf{A}') - F_{\text{rnn}}^{(L)}(\mathbf{x}; \mathbf{W}, \mathbf{A}) - F^{(L)}(\mathbf{x}, \mathbf{W}', \mathbf{A}')\right\|$$

$$= \left\|\mathbf{B}\mathbf{h}^{(L)\prime} - \sum_{\ell \in [L]} \mathbf{Back}_{\ell \to L}\mathbf{D}^{(\ell)}\left(\mathbf{W}'\mathbf{h}^{(\ell-1)} + \mathbf{A}'\mathbf{x}^{(\ell)}\right)\right\| \leq \mathcal{O}(\frac{\rho^7 \Delta^{4/3}}{m^{1/6}}).$$

*Proof.* The proof will follow the same technique as has been used in Lemma 6.1 in [37]. We give a brief outline here.

We allow a change in $\mathbf{A}$ by $\mathbf{A}'$, which wasn't allowed in their lemma. This leads to an introduction of an additional term in eq. H.1 in [37]. That is, there exist diagonal matrices $\mathbf{D}^{(\ell)''}$, where $d_{rr}^{(\ell)''} \in [-1, 1]$ and is non zero only when $d_{rr}^{(\ell)'} \neq d_{rr}^{(\ell)}$,

$$\mathbf{B}(\mathbf{h}^{(L)} + \mathbf{h}^{(L)'}) - \mathbf{B}\mathbf{h}^{(L)} = \underbrace{\sum_{i=1}^{L-1} \mathbf{B}(\mathbf{D}^{(L)} + \mathbf{D}^{(L)''})\mathbf{W} \cdots \mathbf{W}(\mathbf{D}^{(i+1)} + \mathbf{D}^{(i+1)''})\mathbf{W}'(\mathbf{h}^{(i)} + \mathbf{h}^{(i)'})}_{\text{Term 1}}$$

$$+ \underbrace{\sum_{i=1}^{L-1} \mathbf{B}(\mathbf{D}^{(L)} + \mathbf{D}^{(L)''})\mathbf{W} \cdots \mathbf{W}(\mathbf{D}^{(i+1)} + \mathbf{D}^{(i+1)''})\mathbf{A}'\mathbf{x}^{(i)}}_{\text{Term 2}}.$$

In lemma 6.2 of [37], Term 1 was shown to be close to $\sum_{i=1}^{L-1} \mathbf{B}\mathbf{D}^{(L)}\mathbf{W} \cdots \mathbf{W}\mathbf{D}^{(i+1)}\mathbf{W}'\mathbf{h}^{(i)}$ by $\mathcal{O}(\frac{\varrho^7 \Delta^{4/3}}{m^{1/6}})$. The bound will stay the same, since we have shown similar bounds for $\left\| \mathbf{D}^{(\ell)'} \right\|_0$ and $\left\| \mathbf{h}^{(\ell)'} \right\|_2$ in the proof of lemma G.2.

Using the same technique, we can show that Term 2 is close to $\sum_{i=1}^{L-1} \mathbf{B}\mathbf{D}^{(L)}\mathbf{W} \cdots \mathbf{W}\mathbf{D}^{(i+1)}\mathbf{A}'\mathbf{x}^{(i)}$, since Term 2 can be similarly broken down into at most $2^L$ terms of the form

$$(\mathbf{B}\mathbf{D}\mathbf{W} \cdots \mathbf{D}\mathbf{W})\mathbf{D}''(\mathbf{W} \cdots \mathbf{D}\mathbf{W})\mathbf{D}'' \cdots \mathbf{D}''(\mathbf{W} \cdots \mathbf{D}\mathbf{W})\mathbf{A}'\mathbf{x}^{(i)}$$

and each term can then be similarly bounded to give an extra error bound $\mathcal{O}(\frac{\varrho^7 \Delta^{4/3}}{m^{1/6}})$. $\square$

**Lemma G.4.** *Let $\mathbf{W}^*$ and $\mathbf{A}^*$ be as defined in def. D.2. Let $\mathbf{W}, \mathbf{A}, \mathbf{B}$ be at random initialization, $\mathbf{x}^{(1)}, \cdots, \mathbf{x}^{(L)}$ be a fixed normalized input sequence, and $\Delta \in \left[\varrho^{-100}, \varrho^{100}\right]$. With probability at least $1 - e^{-\Omega(\rho)}$ over $\mathbf{W}, \mathbf{A}, \mathbf{B}$ the following holds. Given any matrices $W'$ with $\left\| \mathbf{W}' \right\|_2 \leq \frac{\Delta}{\sqrt{m}}$, $\mathbf{A}'$ with $\left\| \mathbf{A}' \right\|_2 \leq \frac{\Delta}{\sqrt{m}}$. Letting $\mathbf{h}^{(\ell)}, \mathbf{D}^{(\ell)}, \mathbf{Back}_{i \to L}$ be defined with respect to $\mathbf{W}, \mathbf{A}, \mathbf{B}, \overline{\mathbf{x}}$, and $\mathbf{h}^{(\ell)} + \mathbf{h}^{(\ell)'}, \mathbf{D}^{(\ell)} + \mathbf{D}^{(\ell)'}, \mathbf{Back}_{i \to L} + \mathbf{Back}'_{\ell \to j}$ be defined with respect to $\mathbf{W} + \mathbf{W}', \mathbf{A} + \mathbf{A}', \mathbf{B}, \overline{\mathbf{x}}$, then for all $s \in [k]$*

$$\sum_{\ell \in [L]} \mathbf{e}_s^\top \left( \mathbf{Back}_{\ell \to L} + \mathbf{Back}'_{\ell \to L} \right) \left( \mathbf{D}^{(\ell)} + \mathbf{D}^{(\ell)'} \right) \left( \mathbf{W}^* \left( \mathbf{h}^{(\ell-1)} + h^{(\ell-1)'} \right) + \mathbf{A}^*\mathbf{x}^{(\ell)} \right)$$

$$= \sum_{r' \in [p]} b_{r',s}^\dagger \Phi_{r',s} \left( \left\langle \mathbf{w}_{r',s}^\dagger, [\overline{\mathbf{x}}^{(2)}, \cdots, \overline{\mathbf{x}}^{(L-1)}] \right\rangle \right)$$

$$\pm \mathcal{O}(d_{\text{out}} L p \rho^2 \varepsilon + d_{\text{out}} L^{7/3} p \rho^2 L_\Phi \epsilon_x^{2/3} + d_{\text{out}} L^5 p \rho^{11} L_\Phi C_\Phi C_\varepsilon(\Phi, \mathcal{O}(\epsilon_x^{-1})) m^{-1/30})$$

$$\pm O \left( \frac{C_\varepsilon(\Phi, \mathcal{O}(\epsilon_x^{-1})) d_{\text{out}}^{1/2} \rho^8 \Delta^{1/3}}{m^{1/6}} \right).$$

*Proof.* The proof follows from Lemma G.2, using the bound on $\|\mathbf{W}^*\|$ and $\|\mathbf{A}^*\|$ from lemma F.17. $\square$

# H    On Concept Classes

The concept class in [37] matched the output to a true label at each step using loss function $G$, i.e. $F^*$ belongs to $\mathbb{R}^{L \times d} \to \mathbb{R}^{d_{\text{out}}}$, given by

$$F_s^{*(j)}(\mathbf{x}) = \sum_{i:i<j} \sum_{r \in [p]} \phi_{i \to j, r, s}(\mathbf{w}_{i \to j, r, s}^T \mathbf{x}^{(i)}), \tag{61}$$

for all $j \in [2, L]$ and $s \in [d_{\text{out}}]$. Here $\phi_{i \to j, r, s} : \mathbb{R} \to \mathbb{R}$ are smooth functions and $\mathbf{w}_{i \to j, r, s}$ unit vectors. We can rewrite (61) in a more compact vector form

$$F^{*(j)}(\mathbf{x}) = \sum_{i: i < j} \psi_{i \to j}(\mathbf{x}^{(i)}), \tag{62}$$

where $\psi_{i \to j}(\mathbf{x}^{(i)})$ is defined in the obvious way: it is the vector of inner sums in (61). From the previous equation it is clear that $F^{*(j)}(\mathbf{x}^{(i)})$ is a *sum* of functions of individual tokens $\mathbf{x}^{(i)}$. This suggests that this concept class can represent only a limited set of concepts. A clean framework for illustrating these issues is afforded by the task of recognizing membership in a given formal language. Fix a finite alphabet $\Sigma$ with each letter also encoded by a vector so that it can be processed by RNNs. We say that the RNN recognizes a language $\Lambda$ over $\Sigma$ if after processing the sequences $(\mathbf{x}^{(1)}, \ldots, \mathbf{x}^{(j)})$ encoding a string $w = (w_1, \ldots, w_j) \in \Sigma^j$, the output $\mathbf{y}^{(j)}$ satisfies $|\mathbf{y}^{(j)} - 1| < 1/3$, if $w \in \Lambda$ and $|\mathbf{y}^{(j)}| < 1/3$, otherwise. For simplicity, in the following, we will require the more stringent conditions $\mathbf{y}^{(j)} = 1$ and $\mathbf{y}^{(j)} = 0$; these can be easily relaxed with some extra work. Since our output is binary, the output dimension $d_{\text{out}}$ is set to 1; this is the setting in which our experiments are also done.

Below, we give examples of some simple regular languages that the above concept class can't recognize but can be recognized by functions in our concept class with small complexity.

We first consider a simple regular language $L_1$ over the alphabet $\{0, 1\}$ given by the regular expression $0^*10^*$. In words, a string is in $L_1$ iff it contains a single 1. This language can be thought of as modeling the occurrence of an event (a single blip) in a time series.

Consider the set $S$ of strings $\{0^q 000^{L-q-2}, 0^q 110^{L-q-2}, 0^q 010^{L-q-2}, 0^q 100^{L-q-2}\}$ where $0 \leq q \leq L - 2$. Clearly, $0^q 000^{L-q-2} \notin L_1$ and $0^q 110^{L-q-2} \notin L_1$ whereas $0^q 010^{L-q-2} \in L_1$ and $0^q 100^{L-q-2} \in L_1$. We choose uniform distribution on $S$ as the data distribution $D_{L_1}$.

**Theorem H.1.** *Any concept class of type* (61) *must err with probability at least* $1/4$ *on* $D_{L_1}$.

*Proof. (sketch)* Fix a $q$. Let $w = 0^q ab0^{L-q-2}$ where $a, b \in \{0, 1\}$, we can rewrite (62) as

$$F^{*(L)}(w) = \sum_{i: i \leq q} \alpha_i(0) + \alpha_{q+1}(a) + \alpha_{q+2}(b) + \sum_{i: q+3 \leq i \leq L} \alpha_i(0)$$
$$= A + \alpha_{q+1}(a) + \alpha_{q+2}(b),$$

where each $(\alpha_i(0), \alpha_i(1)) \in \mathbb{R}^2$ is any two-dimensional vector. Now, we must have $A + \alpha_{q+1}(1) + \alpha_{q+2}(0) = 1$ and $A + \alpha_{q+1}(0) + \alpha_{q+2}(1) = 1$. And also, $A + \alpha_{q+1}(0) + \alpha_{q+2}(0) = 0$ and $A + \alpha_{q+1}(1) + \alpha_{q+2}(1) = 0$. Summing the first two equations gives $2A + \alpha_{q+1}(0) + \alpha_{q+1}(1) + \alpha_{q+2}(0) + \alpha_{q+2}(1) = 2$ and summing the next two equations gives $2A + \alpha_{q+1}(0) + \alpha_{q+1}(1) + \alpha_{q+2}(0) + \alpha_{q+2}(1) = 0$. Thus at least one of the four equations above must fail. The concept class thus incurs an error with probability at least 1/4. $\qquad \square$

We can show that our concept class (Eq. (2)) can recognize the language $D_{L_1}$. Assume that we get an length-$L$ string as a length-$L$ input sequence $\mathbf{x}$, with '0' represented by one-dimensional vector 0 and '1' represented by one-dimensional vector 1. E.g. '0010' will be represented as a sequence $0, 0, 1, 0$. Then, one can count the number of 1's in the given input and claim that if the number of 1's is exactly 1, the string belongs to the language $D_{L_1}$. The required condition can be checked using a single neuron with activation $\phi(x) = 2x - x^2$, which is a quadratic activation, and weight vector containing all ones ($\mathbf{1}$). Hence, one can show that acceptance condition is satisfied iff $\phi(\langle \mathbf{1}, \mathbf{x} \rangle + 1/2)$ is positive. Thus overall, we have shown that the language $D_{L_1}$ can be computed by a one-hidden layer neural network with a quadratic activation and 1 neuron, implying that our concept class can approximate the language $D_{L_1}$.

**Other pattern matching languages.** $D_{L_1}$ can be thought of as a very simple pattern matching problem. In fact, we can show a more general class of languages that can be learned by our concept class efficiently. Consider the following language: a string (of length at most $L$) belongs to the language iff it contains a particular substring (of some constant length $k$). We will denote this substring by $\bar{s}$. Assume that we get an $L$-length string as $L$-dimensional input $\mathbf{x}$, with '0' represented by a one-dimensional vector $-1$ and '1' represented by one-dimensional vector 1. E.g. '0010' will be

represented by the sequence $-1, -1, 1, -1$. Let $\mathbf{v}_{\bar{s}}$ denote the vector representation of the sequence for the substring $\bar{s}$. Then, we can enumerate all the consecutive substrings in the input and check if the required substring occurs in at least one of them. Mathematically, this translates to creating a one layer neural network with $(L - k + 1)$ neurons and activation function $\phi(t) = e^{ct}$, for some constant $c = \Omega(\log L)$. The $i$-th neuron will contain the weight vector $\mathbf{v}_i$, where the substring between position $i$ and $i + k - 1$ contains $\mathbf{v}_{\bar{s}}$ and the rest of the positions contain $0$. One can check that if the input string contains the desired substring $\bar{s}$, then $\sum_{i=1}^{L} \phi(\langle \mathbf{v}_i, \mathbf{x} \rangle - k) \geq 1$, otherwise it is less than $\frac{1}{L}$. Thus, overall we have shown that the language can be recognized by a one-layer network with exponential activations. Since, we have discussed before that exponential activations have $O(1)$ complexity (see Def. 2.1), we have shown that our concept class can efficiently solve the pattern matching problem.

We can generalize the above ideas to address some other related problems where we need to find multiple substrings, problems where we need to make sure that the number of times a particular substring occurs is at most a certain limit, etc.

**General regular languages.** More generally, our concept class can express all regular languages. However, the complexity of the concept class can be super-polynomial in the sequence length $L$ depending on the regular language. Here is a sketch of a general construction. As previously mentioned, RNNs with ReLU activations and finite precision are known to be equivalent to deterministic finite automata (DFA) and thus capture regular languages [11]. The ReLU can be approximated by polynomials [47] so that the resulting RNN still approximates the DFA up to some required length (the larger the length, the better the approximation needs to be—and the higher the degree of the approximating polynomial). In turn, such an RNN using polynomial activations can be easily represented by our concept class. The complexity (Def. 2.1) of the concept class is small as polynomials have small complexity. We omit the routine but technical details of this construction.

Many regular languages allow special treatment though. For example, consider the language PARITY. PARITY is the language over alphabet $\{0, 1\}$ with a string $w = (w_1, \ldots, w_j) \in$ PARITY iff $w_1 + \ldots + w_j = 1 \bmod 2$, for $j \geq 1$. We can show that PARITY is hard for the above concept class for the uniform distribution on $\{0, 1\}^L$. A simple proof of this can be obtained via Boolean Fourier analysis (e.g., [48]) which we now sketch. In this setting, we note that PARITY of $L$ bits corresponds to a degree-$L$ polynomial $(2w_1 - 1)(2w_2 - 1) \ldots (2w_L - 1)$; the output now takes values in $\{-1, 1\}$ instead of $\{0, 1\}$. On the other hand, the functions in (61) with $d_{\text{out}} = 1$ correspond to linear functions of the form $\sum_i \alpha_i w_i + \beta_i$ for some constants $\alpha_i, \beta_i \in \mathbb{R}$ for all $i$. Using these facts, the correlation between the two can be easily shown to be $0$ via the Plancherel–Parseval theorem, which implies that all functions of type (61) make significant error on PARITY.

However, we can show that PARITY is easily expressible by our concept class with small complexity. Assume that we get an length-$L$ string as a length-$L$ input sequence $\mathbf{x}$, with '0' represented by one-dimensional vector $0$ and '1' represented by one-dimensional vector $1$. E.g. '0010' will be represented as a sequence $0, 0, 1, 0$. Then, one can count the number of 1's in the given input and claim that if the number of 1's is even, the string belongs to the language PARITY. The required condition can be checked using a single neuron with activation $\phi(x) = \cos(\pi x)$ and weight vector containing all ones ($\mathbf{1}$). Hence, one can show that acceptance condition is satisfied iff $\phi(\langle \mathbf{1}, \mathbf{x} \rangle - 1)$ is positive. Thus overall, we have shown that the language PARITY can be computed by a one-hidden layer neural network with a $\cos$ activation and 1 neuron. Since, we have discussed before that $\cos$ activations have $O(1)$ complexity (see Def. 2.1), we have shown that our concept class can efficiently recognize PARITY.

We performed experiments on the ability of RNNs to learn various regular languages (see sec. I for details). In almost all of the regular languages that we tested on, RNNs can achieve near perfect test accuracies (table 1).

# I Experiments

**RNN inversion at random initialization.** We consider a randomly initialized RNN, with the entries of the weights $\mathbf{W}$ and $\mathbf{A}$ randomly picked from the distribution $\mathcal{N}(0, 1)$. Sequences are generated i.i.d. from normal distribution i.e. for each sequence, $\mathbf{x}^{(i)} \sim N(0, \mathbf{I})$ for each $i \in [L]$. We use

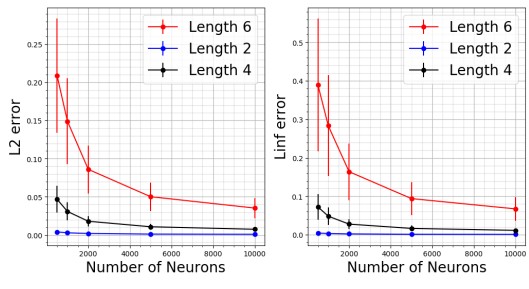

Figure 1: Data dimension: 2

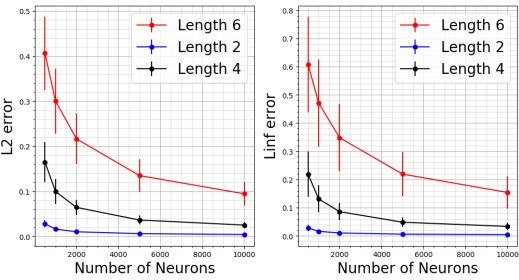

Figure 2: Data dimension: 4

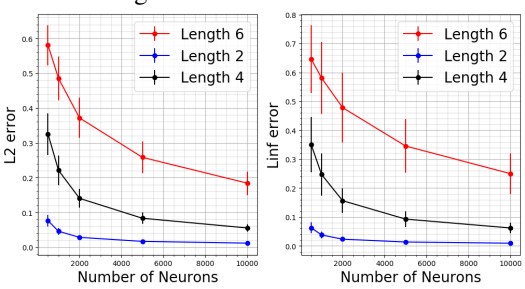

Figure 3: Data dimension: 8

Figure 4: Invertibitiliy of RNNs at random initialization: Checking behavior of inversion error with number of neurons and the sequence length at different data dimensions.

SGD with batch size 128, momentum 0.9 and learning rate 0.1 to compute the linear matrix $\overline{\mathbf{W}}^{[L]}$ so that $\|\overline{\mathbf{W}}^{[L]}\mathbf{h}^{(L)} - [\mathbf{x}^{(1)}, \ldots, \mathbf{x}^{(L)}]\|^2$ is minimized. We compute the following two quantities on the test dataset, containing 1000 sequences: average $L_2$ error given by $\mathbb{E}_{\mathbf{x}} \frac{\|\overline{\mathbf{W}}^{[L]}\mathbf{h}^{(L)} - [\mathbf{x}^{(1)},\ldots,\mathbf{x}^{(L)}]\|}{\|[\mathbf{x}^{(1)},\ldots,\mathbf{x}^{(L)}]\|}$ and average $L_\infty$ error given by $\mathbb{E}_{\mathbf{x}} \|\overline{\mathbf{W}}^{[L]}\mathbf{h}^{(L)} - [\mathbf{x}^{(1)}, \ldots, \mathbf{x}^{(L)}]\|_\infty$. We plot both the quantities for different settings of data dimension $d$, sequence length $L$ and the number of neurons $m$. $L$ takes values from the set $\{2, 4, 6\}$, $d$ takes from $\{2, 4, 8\}$ and $m$ takes from $\{500, 1000, 2000, 5000, 10000\}$ (Figure 4). The trends support our bounds in Theorem 4.5, i.e. the error increases with increasing $L$ and decreases with increasing $m$. Note that the data distribution is different from the one assumed in normalized sequence Def. 3.1. It was easier to conduct experiments in the current data setting and a similar statement as Thm. 4.5 can be given.

**Performance of RNNs on different regular languages.** We check the performance of RNNs on the formal language recognition task for a wide variety of regular languages. We follow the set-up in [10] who conducted experiments on LSTMs etc. but not on RNNs.

We consider the regular languages as considered in [10]. Tomita grammars [49] contain 7 regular languages representable by DFAs of small sizes, a popular benchmark for evaluating recurrent models (see references in [10]). We reproduce the definitions of the Tomita grammars from there verbatim: Tomita Grammars are 7 regular langauges defined on the alphabet $\Sigma = \{0, 1\}$. Tomita-1 has the

regular expression $1^*$. Tomita-2 is defined by the regular expression $(10)^*$. Tomita-3 accepts the strings where odd number of consecutive 1s are always followed by an even number of 0's. Tomita-4 accepts the strings that do not contain three consecutive 0's. In Tomita-5 only the strings containing an even number of 0's and even number of 1's are allowed. In Tomita-6 the difference in the number of 1's and 0's should be divisible by 3 and finally, Tomita-7 has the regular expression $0^*1^*0^*1^*$.

We also check the performance of RNNs on $\text{Parity}$, which contains all languages with strings of the form $(w_1, \ldots, w_L)$ s.t. $w_1 + \ldots + w_L = 1 \mod 2$. Languages $\mathcal{D}_n$ are recursively defined as the set of all strings of the form $(0w1)^*$, where $w \in \mathcal{D}_{n-1}$, with $\mathcal{D}_0$ containing only $\epsilon$, the empty word. Other languages considered are $(00)^*$, $(0101)^*$ and $(00)^*(11)^*$. Table 1 shows the number of examples in train and test data, the range of the length of the strings in the language, and the test accuracy of the RNNs with activation functions ReLU and tanh on the regular languages mentioned above.

| Task | No. of Training/Test examples | Range of length of strings | RNN(Relu) | RNN(Tanh) |
|---|---|---|---|---|
| Tomita 1 | 50/100 | [2, 50] | 1.0 | 1.0 |
| Tomita 2 | 25/50 | [2, 50] | 1.0 | 1.0 |
| Tomita 3 | 10000/2000 | [2, 50] | 1.0 | 1.0 |
| Tomita 4 | 10000/2000 | [2, 50] | 1.0 | 1.0 |
| Tomita 5 | 10000/2000 | [2, 50] | 1.0 | 1.0 |
| Tomita 6 | 10000/2000 | [2, 50] | 1.0 | 1.0 |
| Tomita 7 | 10000/2000 | [2, 50] | 0.259 | 0.99 |
| Parity | 10000/2000 | [2, 50] | 1.0 | 1.0 |
| $\mathcal{D}_2$ | 10000/2000 | [2, 100] | 1.0 | 1.0 |
| $\mathcal{D}_3$ | 10000/2000 | [2, 100] | 0.99 | 1.0 |
| $\mathcal{D}_4$ | 10000/2000 | [2, 100] | 1.0 | 0.99 |
| $(00)^*$ | 250/50 | [2, 500] | 1.0 | 1.0 |
| $(0101)^*$ | 125/25 | [4, 500] | 0.99 | 1.0 |
| $(00)^*(11)^*$ | 10000/2000 | [2, 200] | 0.99 | 1.0 |

Table 1: Performance of RNNs on different regular languages.

We vary $m$, the dimension of the hidden state, in the range $[3, 32]$, used RMSProp optimizer [50] with the smoothing constant $\alpha = 0.99$ and varied the learning rate in the range $[10^{-2}, 10^{-3}]$. For each language we train models corresponding to each language for 100 epochs and a batch size of 32. We experimented with two different activations ReLU and tanh. In all but one case (Tomita 7 with ReLU) the test accuracies with near-perfect. This was the case across runs. Tomita 7 results could perhaps be improved by more extensive hyperparameter tuning. We train and test on strings of length up to 50, and in a few cases strings of larger lengths (when the number of strings in the language is small).