# OpenReview forum: "Learning and Generalization in RNNs"
_NeurIPS.cc/2021/Conference — NeurIPS 2021 Poster_

### Official Review · Reviewer_pm6R · 2021-07-12

**Rating:** 7
**Confidence:** 3

**Summary:**

The authors provide a proof that sufficiently wide recurrent neural networks can accurately learn functions of sequences of data, providing a much needed extension of known results about standard feed-forward neural network to recurrent neural networks.

**Limitations And Societal Impact:**

I see not negative societal impact from this work.

**Main Review:**

The authors first extend the result shown in (https://arxiv.org/pdf/1811.04918.pdf) to recurrent neural networks, showing that a wide enough RNN can learn *any* function from sequence input space (i.e. R^(d*L), where d is the input dimension and L is the sequence length) to output space, so long as it can parametrized by a neural network with one hidden layer and smooth activation, which is a very large concept class.

This is not trivial, as RNNs are 'limited' to a constant function between input space and hidden layer, hidden layer at one time-point and the next, and the hidden layer to output, and they authors showed SGD is in fact able to learn any function from the concept class described above.

While the proof is obviously very long, the authors do provide a straight forward 'sketch' of the proof, which makes it clear what are the key innovations and ideas that went into it. The further make note of their contributions beyond other works such as https://arxiv.org/pdf/1902.01028.pdf and https://papers.nips.cc/paper/2019/file/62dad6e273d32235ae02b7d321578ee8-Paper.pdf, which follow a similar formalism.

Overall, this work is a critical step in the theoretical analysis of sequential neural networks and I recommend acceptance.

**Time Spent Reviewing:**

2 hours

---

> ### Author Response · Authors · 2021-08-09
> **Response to Reviewer pm6R**
>
> Thank you very much for taking the time to read our paper and for your comments.

---

### Official Review · Reviewer_Gmdk · 2021-07-16

**Rating:** 5
**Confidence:** 3

**Summary:**

In this paper, the authors try to prove that overparametrized RNNs can efﬁciently learn concept classes consisting of one-hidden-layer neural networks that take the entire sequence of tokens as input. The training algorithm used is SGD with sufﬁciently small step size. Conceptually and technically, the introduced new ideas enable us to extract information from the hidden state of the RNN, which addresses a crucial weakness in previous work. In the end, the authors illustrate their results on some regular language recognition problems.

**Limitations And Societal Impact:**


1. The concept class (definition 3.1), as the main difference from related work [37], should be discussed in more detail. It would be better to add a remark (e.g., below Definition 3.3) to illustrate the relation between the concept classes in this paper and [37].

2. There’s some notational confusion here and there, e.g.,  Back_{i->j} (line 217, Equation 4) is used before being introduced,  parameter p (Equation 2) is not formally introduced, etc.

3. (line 142) Why is the output coefficient b_{r,s} set to |b_{r,s}|\leq 1?

4.  In Theorem 3.1, we see that the step size \eta=\frac{1}{\epsilon \rho^2 m}. If I am not wrong, the step size \eta tends to be large when \epsilon moves towards 0, which seems to contradict the statement in line 69  (“The training algorithm used is SGD with sufﬁciently small step size.”).

Some typos, e.g.,
Line 160: sgd  SGD
Line 162: traintrained

**Main Review:**

Originality:
The motivation of this paper comes from the fact that, compared to the previous work [37], this paper investigates the learnability of RNN for more general concept class. The motivation is ok and the proofs seem non-trivial.

Quality:
I have checked  details and did not find technical ﬂaws.

Clarity:
The paper is in general well-written.

**Time Spent Reviewing:**

12

---

> ### Author Response · Authors · 2021-08-09
> **Response to Reviewer Gmdk**
>
> Thank you very much for taking the time to read our paper and for your comments. Your score doesn't seem to be in line with your positive comments about the paper. We hope you will revisit.
>
> 1. _"The concept class (definition 3.1), as the main difference from related work [37], should be discussed in more detail. It would be better to add a remark (e.g., below Definition 3.3) to illustrate the relation between the concept classes in this paper and [37]."_
>
> This is done in Section 5 in detail and briefly in the intro. We will add a pointer to Section 5 after Definition 3.3 and in the intro with a remark.
>
> 2. _"There’s some notational confusion here and there, e.g., Back{i->j} (line 217, Equation 4) is used before being introduced, parameter p (Equation 2) is not formally introduced, etc."_
>
> $Back_{i->j}$ is defined just a few lines below Eq. 4. In notation-heavy situations, it can be clearer (and not uncommon) to first write an equation and then define the terms in it. We will explicitly mention this right before Eq. 4. For parameter $p$, you are right and we will formally introduce it. We will also further check any other instances of confusion.
>
> 3. _"(line 142) Why is the output coefficient b{r,s} set to |b{r,s}|\leq 1?"_
>
> The output coefficient $ b_{r,s} $ is set to $ |b_{r,s}| \leq 1 $ just for convenience. Without this, we will require the number of neurons $ m $ to depend polynomially on $ |b_{r,s}| $.
>
>
> 4. _"In Theorem 3.1, we see that the step size \eta=\frac{1}{\epsilon \rho^2 m}. If I am not wrong, the step size \eta tends to be large when \epsilon moves towards 0, which seems to contradict the statement in line 69 (“The training algorithm used is SGD with sufﬁciently small step size.”)."_
>
> $m$ (which can depend on $\epsilon$ as $1/poly(\epsilon)$) is sufficiently large to counteract the effect of $\epsilon$ so as to make $\eta$ small.

---

### Official Review · Reviewer_N1qr · 2021-07-17

**Rating:** 8
**Confidence:** 3

**Summary:**

Reference [37] demonstrated that that SGD learns the recurrent weight matrix of an RNN with provable generalization if the number of training sequences is at least polynomial in the logarithm of the number of hidden nodes, and if the target function computes, at each time step, a sum of differentiable functions of linear transforms of previous  time steps.  This paper uses the same notation, and many of the same methods, but significantly extends the previous work  in the following way: the function class that can be computed is significantly expanded, to include sums of differentiable functions of linear transforms of the __concatenated sequence__ of inputs, not just transforms of individual inputs. In order to do so, this paper cosiders bounds on the input matrix (A) and recurrent matrix (W) sufficient to encode the entire sequence of inputs into the state vector.

**Limitations And Societal Impact:**

The key contribution of the paper is a result demonstrating generalization error bounds for very wide recurrent neural networks trained with SGD.

**Main Review:**

Strengths

In the spirit of many recent papers, the generalization error bound considered here is not a monotonically increasing function of the number of hidden nodes.  Instead, as in  reference [37], the generalization bound is actually a monotonically decreasing function of the number of hidden nodes, for any fixed dataset size.

An interesting  and surprising result in this derivation is that the number of hidden nodes is required to be very large: at least polynomial in the sequence length, function class complexity, and output dimension.  This result is fundamental to the proof method, because the proof method encodes the entire input sequence into the state vector.  I'm actually not sure if this is a strength or a weakness of the paper: it obviously limits the applicability of this result  in practical settings, but the purpose of that limitation (the state vector needs to encode the input) is quite interesting, and might cause practitioners to think about the designs of their architectures.

Weaknesses

The convergence proof depends on similarity between the pseudo-network and the learned network.  The pseudo-network and learned network differ if many of the ReLUs have switched on or off during  training.  Although each step of SGD might switch very few  ReLUs, the optimum weights (the end of SGD) might  have a completely  different set of ReLU activations than the initial set, and in that case, the pseudo-network will be an arbitrarily poor approximation of the true network.  In that case, I believe, the convergence proof fails.

The requirement that the length is fixed at L is proposed in the text as being "w.l.o.g."  It is more accurate to say, as the conclusion says, that because of this requirement, the results in this article are ony proven for test sequences that are shorter than the longest training sequence.

"parameterization" is sometimes spelled "parameterization", sometimes "paramterization", sometimes "paramtrization".  OK, we know what you mean, but this  seems sloppy.  There are  other places where spelling or grammar are sloppy.





**Time Spent Reviewing:**

4

---

> ### Author Response · Authors · 2021-08-10
> **Response to Reviewer N1qr**
>
> Thank you very much for taking the time to read our paper and for your comments.
>
> 1. _“The convergence proof depends on the similarity between the pseudo-network and the learned network. The pseudo-network and learned network differ if many of the ReLUs have switched on or off during training. Although each step of SGD might switch very few ReLUs, the optimum weights (the end of SGD) might have a completely different set of ReLU activations than the initial set, and in that case, the pseudo-network will be an arbitrarily poor approximation of the true network. In that case, I believe, the convergence proof fails.”_
>
> A part of our proof is to show that the pseudo network and the learned network stay close to each other during training with high probability. Hence, with high probability, the relu patterns of the pseudo network and the learned network remain close throughout the SGD training. Please refer to Lemma G.3 in the appendix for more details.
>
> 2. _“The requirement that the length is fixed at L is proposed in the text as being "w.l.o.g." It is more accurate to say, as the conclusion says, that because of this requirement, the results in this article are only proven for test sequences that are shorter than the longest training sequence.”_
>
> Let us clarify this point as "w.l.o.g." refers to a different aspect of sequences than bounded length. First, the relevant text from the intro:
> "As in previous work, we work with sequences of bounded length $L$. Without loss of generality, we work with token sequences $x^{(1)}, \ldots, x^{(L)}$ of fixed length as opposed to sequences of length up to $L$."
>
> As we clearly say here and repeat in conclusion, the sequence length $L$ is bounded. What "w.l.o.g." refers to is the restriction that _all_ sequences have length _exactly_ $L$ instead of _up to_ $L$.
>
> If one were to drop the above w.l.o.g. assumption and work with sequences of length up to $L$, our results can be adapted to work for test sequences that are of length _up to_ the length of the longest training sequence (not "shorter" as you write above).

---

### Official Review · Reviewer_vs66 · 2021-07-22

**Rating:** 6
**Confidence:** 2

**Summary:**

This paper shows that overparamterized RNNs trained with SGD can learn a concept class defined over the entire input sequence. The paper provides convergence and generalization guarantees.

**Ethical Concerns:**

No ethical concerns arise from this submission

**Limitations And Societal Impact:**

The main limitation is the practicality of the obtained guarantees, it is not clear if the sample complexity and network width obtained by theorem 3.1 are meaningful.
There is no concern for negative social impact by this submission.

**Main Review:**

**Originality** - The work is a natural extension of previous work, the main difference between this work and prior work [1] is the the analysis under a more general concept class and instead of fixing A and B at initialization only B is fixed in this work. The contribution seems marginal as the majority of technical tools were introduced in prior work.

**Quality** - The submission seems technically sound and well supported although verifying correctness of all claims will require hundreds of hours to do rigorously and is beyond the capacity of a standard review.
I would appreciate a discussion on the practical bounds achieved, i.e. are the bounds meaningful in practice or do they require huge widths that practically make this result no different than those obtained via NTK?

**Clarity** - The paper is clear and well written. The authors do a good job in interpreting their technical claims.
The implications of theorem 3.1 are not clear enough, an example would be very informative, for example in lemma D.4 the bound contains $\rho^7$ where in theorem 3.1 $\rho$ is defined to be $\rho=100Ld_{out}$, for sequence length $10$ and output dimension $1$, the value is huge: $\rho^7=10^{21}$. It seems a bit misleading to present this as analysis in the finite case.

**Significance** - I think this paper’s significance is incremental as it is mostly technical changes to prior work and the technical tools that constitute a significant contribution were mostly established in [1].



Additional points:
1. The introduction provides background and then shifts to be related work in line 52.
2. Line 97 - why is C_s of sin z O(1) ?
3. Lemma 4.2 - if you set W*=0, doesn’t this imply you find a feed-forward NN and not an RNN?
4. Re-randomization 295-305 - what are the assumptions implied by this process? Is it equivalent to having random weights?

[1] Can SGD Learn Recurrent Neural Networks with Provable Generalization? Zeyuan Allen-Zhu, Yuanzhi Li, 2019

**Time Spent Reviewing:**

7

---

> ### Author Response · Authors · 2021-08-10
> **Response to Reviewer vs66**
>
> Thank you for taking the time to read our paper and for your comments.
>
> 1) _"The contribution seems marginal as the majority of technical tools were introduced in prior work."
> "Significance - I think this paper’s significance is incremental as it is mostly technical changes to prior work and the technical tools that constitute a significant contribution were mostly established in [1]."_
>
> We respectfully disagree: Our paper does build upon the work of [1]---but that by no means implies that our work is "incremental" or "marginal". We clearly state the results from [1] that we use. We also clearly state the limitations of [1] (e.g., lines 56, 60 in the intro and later in more detail, e.g. lines 281, 341). These are fundamental limitations: [1]'s techniques couldn't take the order of tokens into account and this stemmed from the fact that they could not make use of the information in the hidden state. We also enunciate the new contributions that allow us to overcome those limitations---conceptual and technical (Sections 4 and 5, e.g. Theorem 4.5 among others).
>
> We would be happy to discuss with the reviewer about a potentially simpler proof strategy. But currently, we believe our technique is necessary for proving the strong generalization bounds for RNN, which isn’t a direct extension of [1].
>
> 2) _"I would appreciate a discussion on the practical bounds achieved, i.e. are the bounds meaningful in practice or do they require huge widths that practically make this result no different than those obtained via NTK?"_
>
> Our results require large widths like most other results in the theory of deep learning. The requirement for larger width is due to the limitations of the current tools used in the theory of deep learning.
>
> 3) _"The implications of theorem 3.1 are not clear enough, an example would be very informative, …It seems a bit misleading to present this as analysis in the finite case."_
>
> We give concrete examples of the application of Theorem 3.1 in Section 5 on concept classes. We respectfully disagree with the second statement. Finite but large (still polynomial) bounds are far more informative than results that use infinite limits and may serve as stepping-stones for future work for tighter bounds. It is also a common practice to report results in this way.
>
>
> 4) _“Line 97 - why is C_s of sin z O(1) ?”_
>
> The coefficients in the Taylor expansion of sin function are of the form $c_i = \frac{ 1} { i! } $, if $i$ is odd, $0$ otherwise. By the definition of $C_s$, we have $C_s$ of $sin$ as $ \sum_i (i+1)^{1.75} \frac{R^i }{ i! } $. Since the growth of the denominator is faster than that of the numerator, the value of $C_s$ will be some constant depending on $R$.
>
> 5) _“Lemma 4.2 - if you set W*=0, doesn’t this imply you find a feed-forward NN and not an RNN?”_
>
> No. The diagonal matrices $ D^{(i)} $ contain the activation pattern of the RNN at cell $ i $. The activation pattern depends on both the hidden state and the input. Hence, setting $ W^{\star} $ to $0$ doesn't necessarily mean we find a feed-forward NN and not an RNN.
>
> 6) _“Re-randomization 295-305 - what are the assumptions implied by this process? Is it equivalent to having random weights?”_
>
> A small fraction of the rows are chosen randomly and these are “re-randomized”, that is they are sampled afresh.

---

> > ### Comment · Reviewer_vs66 · 2021-09-19
> > **Following the rebuttal**
> >
> > Thank you for the detailed response.
> > I have read other reviews and the authors' responses and have decides to raise my score to an accepting one.

---

### Author Response · Authors · 2021-08-09
**Learning and Generalization in RNNs**

We thank all the reviewers for their thoughtful feedback. There is a general appreciation for the clarity and quality of the overall presentation. Two of the reviewers appreciated our contributions. Other reviewers had some concerns that we have answered below. We will carefully incorporate the reviewers’ suggestions (regarding typos, omissions, unclear writing) in the revision of the paper. In our replies below, we will focus on other points.

Our work is an “end-to-end” analysis of RNNs: We show that (overparametrized) RNNs, when trained using SGD, can learn very general concept classes. Such a result requires a combination of analysis of RNNs with respect to representation power, optimization, and generalization. All of the previous work, with the exception of [37], analyzed at most two of the above three. As noted in the paper, the concept class in [37] is very limited.

We reiterate that the analysis of RNNs poses major challenges compared to feedforward networks because of the time component which leads to reuse of weights and effectively deeper networks. The hidden state of the RNN is where the network stores information about the past tokens. Our work provides tools for utilizing the information in the hidden state using a linear transformation---to our knowledge, for the first time. This allows us to prove that using SGD, RNNs can learn far more general concept classes than previously known. There are other contributions in the paper, in particular, the study of RNNs as a recognizer of regular languages.

---

### Decision · Program_Chairs · 2021-09-27

**Decision:**

Accept (Poster)

**Comment:**

This paper proves that sufficiently overparameterized RNNs can learn functions of their input that can be computed by a hidden single layer MLP operating on the entire sequence concatenated together (the size of which is related to the size of the required RNN). If I understand correctly, the main idea is to show that a carefully initialized RNN basically encodes the entire sequence up to the current time step within its hidden state vector, similar to an echo-state network. From there, an NTK-style argument is given that this information can be decoded by the output layer through gradient descent training.

While probably still an oversimplification of how RNNs actually work in practice, this paper represents an improvement on our state of formal understanding compared to previous theoretical works. The reviewers and myself found the paper to be well written and relatively easy to read, despite the difficult subject matter. The reviewers weren't able to find any serious technical issues with the proofs or problems with the plausibility of the claims, although they weren't able to carefully check the proofs thoroughly due to the very long length of the appendix. (I suppose it is questionable whether a paper of this kind of suitable for NeurIPS.) Thus, while I cannot 100% certify the correctness of this paper, I think it's most likely to be correct, and thus a solid contribution to the theory of overparameterized neural networks.